# Robustness May be More Brittle than We Think under Different Degrees of Distribution Shifts

## Abstract

Out-of-distribution (OOD) generalization is a complicated problem due to the idiosyncrasies of possible distribution shifts between training and test domains. Most benchmarks employ diverse datasets to address this issue; however, the degree of the distribution shift between the training domains and the test domains of each dataset remains largely fixed. This may lead to biased conclusions that either underestimate or overestimate the actual OOD performance of a model. Our study delves into a more nuanced evaluation setting that covers a broad range of shift degrees. We show that the robustness of models can be quite brittle and inconsistent under different degrees of distribution shifts, and therefore one should be more cautious when drawing conclusions from evaluations under a limited range of degrees. In addition, we observe that large-scale pre-trained models, such as CLIP, are sensitive to even minute distribution shifts of novel downstream tasks. This indicates that while pre-training may improve downstream in-distribution performance, it could have minimal or even adverse effects on generalization in certain OOD scenarios of the downstream task. In light of these findings, we encourage future research to conduct evaluations across a broader range of shift degrees whenever possible.

## 1 Introduction

Out-of-distribution (OOD) generalization is vital to the safety and reliability of machine learning applications in the real world. However, the complexities of distribution shifts between the training domains and the real test domains make OOD generalization a challenging problem. Numerous empirical studies (Gulrajani & Lopez-Paz, 2021; Wiles et al., 2022) have suggested that most algorithms only offer very little improvement in OOD performance over empirical risk minimization (ERM) (Vapnik, 1998). Furthermore, algorithms performing better than ERM against one type of distribution shift often perform poorly against another (Ye et al., 2022). The inconsistency suggests that it is important to consider various possible types of distribution shifts of a task when evaluating the OOD performance of a model; otherwise, the evaluation might lead to biased conclusions.

To address the issue, most OOD benchmarks (Koh et al., 2021; Hendrycks et al., 2021; Gulrajani & Lopez-Paz, 2021; Ye et al., 2022) incorporate multiple datasets exhibiting a diverse range of distribution shifts. However, another potential source of evaluation bias is often overlooked: the test domains only capture a largely fixed degree of each distribution shift. For example, in (Li et al., 2017; Koh et al., 2021; He et al., 2021; Zhao et al., 2022), each test domain represents a different "direction" of the potential distribution shifts of a task but there is no distinction between different degrees of shift on the same direction. Similar problems can also arise when only the aggregate performance across multiple degrees is examined (Hendrycks & Dietterich, 2019). Such kind of evaluation can result in misconceptions about model performance on the same grounds as those of the evaluation based on limited types (or "directions") of distribution shifts.

Consider the situation (that we observed in this work) illustrated in Figure 1, where the performance of a model is evaluated in only two domains, one for in-distribution (ID) performance in the training domain $\mathcal{D}_{\text{train}}$, and the other for OOD performance in the test domain $\mathcal{D}_{\text{test}}$. In this case, the observed

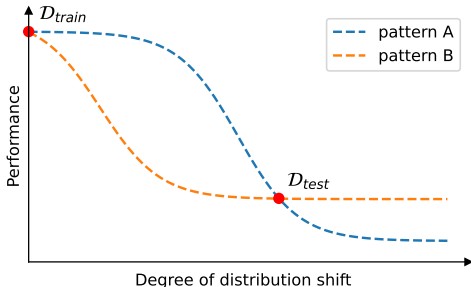

Figure 1: A typical situation where an evaluation under a limited number of degrees of a distribution shift cannot tell any difference between two distinct OOD generalization patterns (labeled as A and B) that can be realized by a model.

performance, which can be explained by at least two distinct generalization patterns as shown in the figure, presents an oversimplified summary of the OOD generalization ability of the model. This simplification may lead to incorrect assumptions about model robustness under various degrees. For example, when a model outperforms another model under a distribution shift of certain severity, it could leave the wrong impression that the first model is more robust in general, i.e., outperforming the other model under almost every possible degree of the concerned distribution shift, while in fact, the first model has much poorer worst-case performance.

In this study, we take a closer look at OOD generalization under distribution shifts of varying degrees. We are interested in the behavior of different models under a broad range of shift degrees and also the relation between the performance of a model at different degrees. Through extensive experiments, we make several observations about the generalization behavior of models under the considered evaluation setting. First, we highlight that the advantage of a model under a mild shift may not apply to stronger shifts of the same type, even if the shift is just slightly stronger[1]. Therefore, caution should be taken when interpreting evaluation results obtained under a limited range of shift degrees. Second, we find that training a model with strongly shifted data can sometimes guarantee robustness to all milder shifts, while at other times it only has a limited impact on robustness and may even harm the OOD performance under milder shifts.

Lastly, the brittleness of robustness to different degrees of distribution shift is also observed in large-scale pre-trained models. We find that while CLIP (Radford et al., 2021) models are able to adapt to many novel tasks, achieving great (sometimes near-perfect) downstream ID performance, they can be extremely sensitive to downstream distribution shifts. In the presence of a distribution shift that is rarely seen during pre-training, even a very mild degree of the shift can cause a disproportionate performance drop in CLIP models in comparison to models trained from scratch. Interestingly, further adapting to the shift to which the models are sensitive significantly improves their general robustness. We believe that such characterizations of the "growth" of the OOD generalization ability of a model is a generally good practice. We encourage future research to adopt this kind of evaluation to generate more valuable insights into OOD generalization.

## 2 RELATED WORK

**Out-of-distribution (OOD) generalization.** Deep neural networks have demonstrated incredible generalization on a variety of complicated tasks, sometimes exceeding human performance (Russakovsky et al., 2015; Silver et al., 2017; OpenAI, 2023), but they are shown to generalize very differently as we do and are very sensitive to all kinds of distribution shifts (Szegedy et al., 2014; Geirhos et al., 2020; Wang et al., 2023). Such brittleness severely undermines the reliability of neural networks and hence limits their applications in the real world where the stakes can be very high. For this reason, OOD generalization and related areas such as domain generalization (Blan-

---

[1]We use mild/strong and low/high-degree interchangeably when describing a distribution shift.

chard et al., 2011; Zhou et al., 2021; Wang et al., 2022) has gained much attention rapidly in recent years (Shen et al., 2021).

**Distribution shifts and OOD benchmarks.** A considerable amount of the research efforts on OOD generalization has been dedicated to the evaluation of models and methods. Hendrycks & Dietterich (2019) provided benchmarks for evaluating the robustness of deep models against common image corruptions and perturbations. One of the benchmarks, ImageNet-C, consists of images under different severity levels of corruptions. In this benchmark, the authors examined the average accuracy of a number of models over all severity levels and showed that the models were all vulnerable to the considered corruptions. They did not, however, provide any analysis at the level of each individual severity level of corruption. Hendrycks et al. (2021) further proposed OOD benchmarks under natural distribution shifts, but this time like DG benchmarks such as (Gulrajani & Lopez-Paz, 2021), they do not involve any evaluation or discussion with regard to different degrees of distribution shifts. Meanwhile, Koh et al. (2021) proposed a diverse set of OOD benchmarks derived from real-world tasks but the degrees of distribution shifts are still largely fixed in each task. Similar examples include (Peng et al., 2019; He et al., 2021; Liang & Zou, 2022; Zhao et al., 2022) which focus on incorporating as many diverse types of distribution shifts without considering different degrees of distribution shifts. Lynch et al. (2023) considered three levels of spurious correlation but did not discuss the connection between the model performance at each level.

**Distribution shifts in model learning.** Schott et al. (2022) showed that models regardless of supervision signal and architectural bias could not learn the underlying mechanism that causes the distribution shifts on several datasets with controllable factors. In comparison, our finding suggests that learning the shifting-inducing mechanism of certain task is possible if the model can be made to be robust to the highest possible degree of the distribution shift. In a different context, Shi et al. (2022) also studied OOD generalization under multiple degrees of distribution shift. Their main focus is whether unsupervised methods can learn more robust representations than supervised learning. They conducted evaluation against three different degrees of spurious correlation and found that unsupervised methods are generally more robust than supervised learning and the advantage grows as the degree of the distribution shift increases.

**Robustness of foundation models.** Pre-training usually has a great impact on generalization. Foundation models such as CLIP (Radford et al., 2021) leverage massive scale of training data to generalize to a great variety of downstream tasks. At the same level of ID accuracy, zero-shot CLIP models are able to attain much higher OOD accuracy on several ImageNet variants than other models trained with a much smaller scale of data. Despite the breakthrough, however, the authors of CLIP have cautioned that the zero-shot performance of CLIP is significantly worse than existing models on data that are hardly present in the training data, e.g., MNIST (LeCun et al., 2010). Later, it is further shown that the main source of the remarkable robustness of CLIP is the diversity of its training data distribution (Fang et al., 2022). While CLIP can be made even more robust in some tasks after proper adaptation (Wortsman et al., 2022), what remains unclear in the literature is to what extent the robustness of CLIP and other foundation models can transfer to downstream tasks and how the models would behave as the degree of the downstream distribution shifts increases.

## 3 DIFFERENT DEGREES OF DISTRIBUTION SHIFTS

The degree of a distribution shift can be quantified in many ways. In this paper, we do not restrict our study to a particular way of quantification as different types of distribution shift may favor different ways of quantification. Instead, we consider the degrees of only one type of distribution shift at a time, so the degrees can take arbitrary values as long as they preserve a certain ordering of a set of domains under the same type of distribution shift.

For simplicity, we use natural numbers to represent the order of a given set of domains under the same type of distribution shift, with smaller numbers indicating lower degrees of distribution shift. We use $\mathcal{D}_d$ to denote a domain where $d$ is its degree and refer to $\mathcal{D}_0$ as a clean domain or a domain under no distribution shift. A model is said to be more robust to a degree $d$ of distribution shift than another model if it attains better performance in $\mathcal{D}_d$.

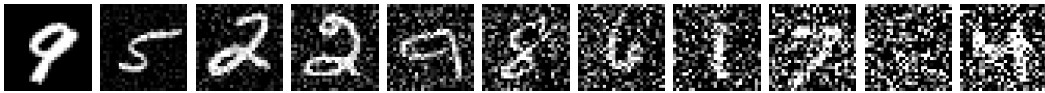

Figure 2: Examples of the NOISYMNIST dataset which consists of 11 subsets of MNIST, one of which is clean, while the other 10 subsets are affected by different degrees of Gaussian noise.

A simple example of problems under different degrees of distribution shift is shown in Figure 2. In this example, the degree of distribution shift in a domain corresponds to the intensity of pixel-level Gaussian noise in the images. In practice, we may not have access to data under all possible degrees of a distribution shift and must rely on data under a limited set of shift degrees to train and evaluate our models. For this context, out-of-distribution generalization refers to the generalization from data under a given set of shift degrees to the rest of possible degrees.

This setting allows for a nuanced understanding of how different degrees of distribution shifts impact the learning and generalization capabilities of models. Although we focus on image classification problems and use accuracy as the metric for measuring model performance, we believe that the conclusions drawn from our experiment results also hold for more general problems.

## 4 ROBUSTNESS MAY BE MORE BRITTLE THAN WE THINK

This section explores the complexities and nuances of model robustness under varying degrees of distribution shifts in neural networks. We first demonstrate that models exhibiting robustness under a certain degree of shift can experience substantial performance degradation under slightly higher degrees of shift. Then we explore whether models robust to high degrees of shift maintain this robustness under lower ones, revealing contrasting results dependent on the specific task and dataset, thereby highlighting the inherent brittleness in neural network robustness under different degrees of distribution shift.

### 4.1 EXPERIMENT SETUP

**Datasets.** Our study employs two altered versions of the MNIST dataset (LeCun et al., 2010), herein referred to as NOISYMNIST and ROTATEDMNIST. To introduce varying degrees of distribution shifts, the NOISYMNIST dataset is generated by introducing Gaussian noise to the original images, resulting in 10 shifted domains under different degrees. More specifically, The standard deviation of the noise is linearly spaced between 0 and 0.8, in increments of 0.08, at the pixel level, normalized to the pixel value range of 0 to 1. Any pixel value beyond this range is clipped to fit within the 0-1 boundary. The ROTATEDMNIST dataset is created by rotating the original images, with degrees linearly spaced from 0 to 80, at intervals of 10 degrees, resulting in 8 shifted domains. Note that our ROTATEDMNIST is different from the ones in other papers, e.g., (Gulrajani & Lopez-Paz, 2021) which covers a smaller set of rotation degrees.

Our study also employs an altered version of CIFAR10 (Krizhevsky et al., 2009). The dataset is called LOWLIGHTCIFAR10. The distribution shift in this dataset is a combination of two primitive types of distribution shifts which are shifts in brightness and shot-noise intensity. Since photos captured in darker environments tend to exhibit more intense shot noises, this dataset simulates realistic photographic effects in photos captured under low-light conditions and hence is much more realistic than the MNIST variants.

**Algorithms.** We experiment with Empirical Risk Minimization (**ERM**, Vapnik (1998)) and over 20 Domain Generalization (DG) algorithms, including but not limited to Invariant Risk Minimization (**IRM**, Arjovsky et al. (2019)), Variance Risk Extrapolation (**VREx**, Krueger et al. (2020)), Spectral Decoupling (**SD**, Pezeshki et al. (2021)), Deep Correlation Alignment (**CORAL**, Sun et al. (2017)), Group Distributionally Robust Optimization (**GroupDRO**, Sagawa et al. (2020)), Representation Self Challenging (**RSC**, Huang et al. (2020)), Domain-Adversarial Neural Networks (**DANN**, Ganin et al. (2016)), Inter-domain Mixup (**Mixup**, Yan et al. (2020)), Adaptive Risk Minimization (**ARM**, Zhang et al. (2020)), and many others representing various OOD research areas (see the full list in Appendix A.2).

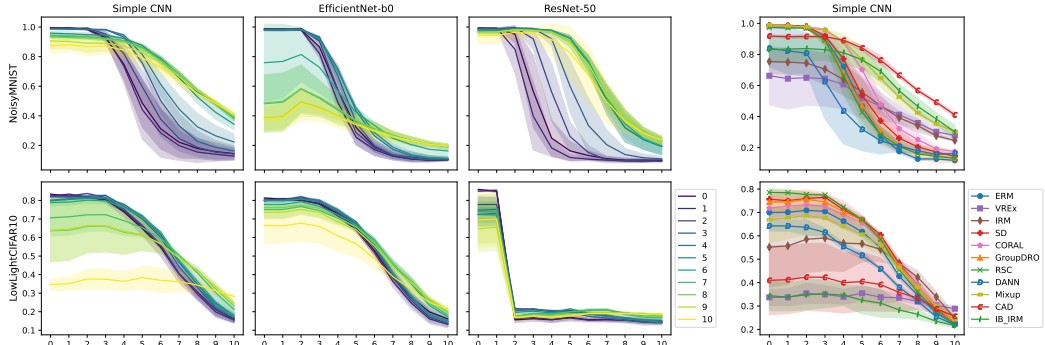

Figure 3: **(Left)** Performance of the best-performing models at each degree of NOISYMNIST and LOWLIGHTCIFAR10. The label of the curves denotes the domain on which the models perform best. The results are averaged over the top 5 models of all algorithms at each degree. **(Right)** Performance of ERM and representative domain generalization algorithms on the two datasets. The results are averaged over the top 3 models of each algorithm selected by worst-domain accuracy.

**Implementation details.** To conduct our experiments, we employed two neural network architectures: a simple 4-layer Convolutional Neural Network (CNN) and a more complex ResNet-50 (He et al., 2016) model. Both models were implemented without any form of pretraining. For optimization purposes, we utilized the Adam optimizer with a static learning rate of 0.001. The total batch size was fixed at 64, and was evenly divided across each training domain, and no weight decay was applied during the training process. Training iterations were set to a maximum of 5,000 for the 4-layer CNN and 10,000 for the ResNet-50 to ensure convergence. No form of data augmentation was used throughout the training process, preserving the inherent distribution and characteristics of the datasets. To ensure the reliability of our results, we conducted a thorough random search for hyperparameters, repeated 20 times.

## 4.2 ROBUSTNESS MAY NOT EVEN EXTRAPOLATE TO SLIGHTLY HIGHER DEGREES

In the real world, data under severe distribution shifts are usually very rare. We often face situations where we only have access to a reasonable amount of data under relatively mild distribution shifts. With these data, we can be fairly certain about the performance of a model in mild situations, but this is hardly satisfactory for any application that demands a certain level of reliability also in worse situations. Therefore, an important question is: how much can the performance of a model under some distribution shift tell us about its performance under stronger shifts?

To approach the question, we constructed a dataset, NOISYMNIST, by gradually adding Gaussian noise to MNIST (LeCun et al., 2010). As illustrated in Figure 2, NOISYMNIST consists of a clean subset $\mathcal{D}_0$ of MNIST and 10 subsets $\{\mathcal{D}_i\}_{i=1}^{10}$ under different degrees of noise. While the construction process of NOISYMNIST is simple, the dataset is nonetheless representative of a wide range of distribution shifts that gradually corrupt predictive features in an image. Intuitively, models that are more robust to relatively mild noises should also be more robust to stronger noises, at least to some extent. If this is true, then in the case of NOISYMNIST, we should be able to rely on domains under only mild shifts such as $\mathcal{D}_4$, to pick the best-performing models in slightly worse domains such as $\mathcal{D}_5$ and $\mathcal{D}_6$ or even much worse domains such as $\mathcal{D}_{10}$.

For this investigation, we trained a pool of models on $\mathcal{D}_0$ and $\mathcal{D}_1$ with ERM and more than 20 domain generalization (DG) algorithms. The models share the same architecture (a 4-layer CNN) but are trained with different initializations and hyperparameters in addition to the different learning algorithms. The performance of the best-performing models in each domain is shown in Figure 3 (left, ERM+DG). The result indicates that models that are better under milder shifts are often significantly *worse* than the other models under stronger shifts. In particular, the average accuracy of the best-performing models in $\mathcal{D}_4$ has dropped by more than 10% in $\mathcal{D}_5$ which is only under a slightly more intense noise than $\mathcal{D}_4$.

| Algorithm | CNN | | | ResNet-50 | | |
|---|---|---|---|---|---|---|
| | $\mathcal{D}_4$ | $\mathcal{D}_5$ | $\mathcal{D}_6$ | $\mathcal{D}_4$ | $\mathcal{D}_5$ | $\mathcal{D}_6$ |
| ERM | 77.8±2.8 (0.0) | 47.7±5.2 (38.7) | 26.5±5.0 (66.0) | 97.4±0.3 (0.0) | 84.0±5.5 (13.8) | 54.8±14.6 (43.8) |
| VREx | 90.1±1.7 (0.0) | 74.3±5.6 (17.6) | 53.4±6.4 (40.8) | 64.6±1.7 (0.0) | 32.1±2.3 (50.3) | 17.4±0.9 (73.1) |
| IRM | 78.7±2.4 (0.0) | 57.6±8.2 (26.8) | 38.0±11.3 (51.7) | 95.7±0.8 (0.0) | 82.4±1.3 (13.9) | 56.6±6.8 (40.9) |
| SD | 81.7±1.2 (0.0) | 57.7±2.3 (29.4) | 35.7±2.1 (56.4) | 97.8±0.4 (0.0) | 92.4±2.1 (5.5) | 76.5±6.0 (21.7) |
| GroupDRO | 74.0±1.7 (0.0) | 50.3±5.7 (32.1) | 29.9±8.4 (59.6) | 82.4±9.0 (0.0) | 53.1±17.7 (35.5) | 30.5±10.4 (63.0) |
| RSC | 84.3±4.6 (0.0) | 61.4±7.2 (27.2) | 39.6±7.1 (53.0) | 88.4±4.0 (0.0) | 64.6±8.6 (26.9) | 39.2±8.0 (55.6) |
| Mixup | 93.2±0.4 (0.0) | 84.1±2.3 (9.7) | 69.2±1.9 (25.7) | 85.4±3.7 (0.0) | 49.2±16.2 (42.4) | 26.6±13.2 (68.8) |
| CAD | 94.1±1.0 (0.0) | 78.7±3.1 (16.3) | 58.6±4.0 (37.7) | 78.8±19.6 (0.0) | 50.6±22.0 (35.8) | 30.5±13.1 (61.4) |
| IB-IRM | 86.1±5.6 (0.0) | 68.9±9.7 (19.9) | 54.6±13.3 (36.5) | 91.0±2.9 (0.0) | 59.5±12.0 (34.6) | 30.1±12.3 (66.9) |

Table 1: Performance of the best models in $\mathcal{D}_4$ of ERM and representative DG algorithms. The relative performance drops (%) with respect to the performance in $\mathcal{D}_4$ are shown in the parentheses. All results are averaged over the top 3 models among 20 models with different initialization and hyperparameters for training.

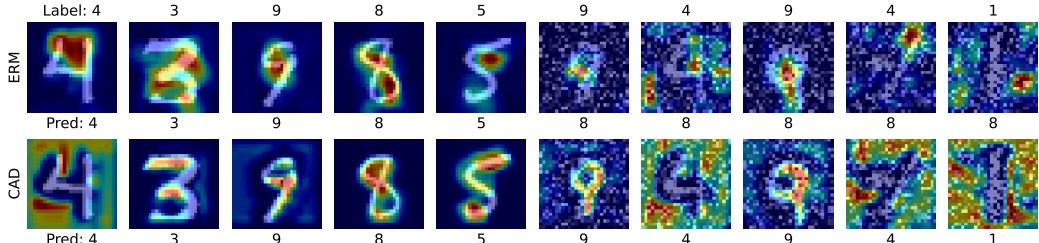

Figure 4: GradCAM visualization of model attention on random examples from $\mathcal{D}_0$ (left) and $\mathcal{D}_7$ (right) of NOISYMNIST. The two models (ERM and CAD) demonstrate distinctive generalization patterns, one relying on the local features while the other more on the global structures. The local features become unreliable as the noise becomes intense.

We further experimented with ResNet-50 to see if networks with much greater capacities can learn a representation that is generally more robust under all the considered levels of noise. As shown in Figure 3 (left, ERM+DG (ResNet-50)), the overall pattern still remains, although the difference among the best-performing models under the stronger end of noises has become less significant. While this shows that larger networks helps, the gap may never be able to be closed by increasing the capacity of the network. More importantly, *the robustness of a model may be more brittle than we think: even under the same type of distribution shift, a slight increase in the degree of the shift may severely harm the performance of the model.*

Besides individual models, the brittleness of robustness under different shift degrees also has implications in evaluating different learning algorithms. The performance of ERM and representative DG algorithms on NOISYMNIST are shown in Figure 3 (right), where the algorithms exhibit very different generalization patterns that cannot be accurately captured by evaluations under only a limited set of shift degrees. A number of DG algorithms, like ERM, are highly robust to low degrees of distribution shift but are quite brittle in the presence of higher degrees of shift. In contrast, there are also algorithms that are significantly better than ERM under high shift degrees but are worse than ERM in other cases. Moreover, when looking at the best-performing models in $\mathcal{D}_4$, the same brittleness can be generally observed for all the algorithms in Table 1, where the performance drop can be even more drastic than that is shown in Figure 3. Astoundingly, the relative performance drop can go up to 50.3% from $\mathcal{D}_4$ to $\mathcal{D}_5$ and 73.1% from $\mathcal{D}_4$ to $\mathcal{D}_6$.

To better understand the observed brittleness, we visualized the attention of ERM and CAD models on NOISYMNIST using GradCAM (Selvaraju et al., 2017). As shown in Figure 4, while both ERM and CAD models can make accurate predictions in the clean domain $\mathcal{D}_0$, they rely on radically different patterns to do so. ERM prefers the most predictive features regardless of whether they are robust or not. In the case of NOISYMNIST, these features turn out to be local features, which are easily corrupted by the noise, and thus no longer predictive when the noise becomes intense.

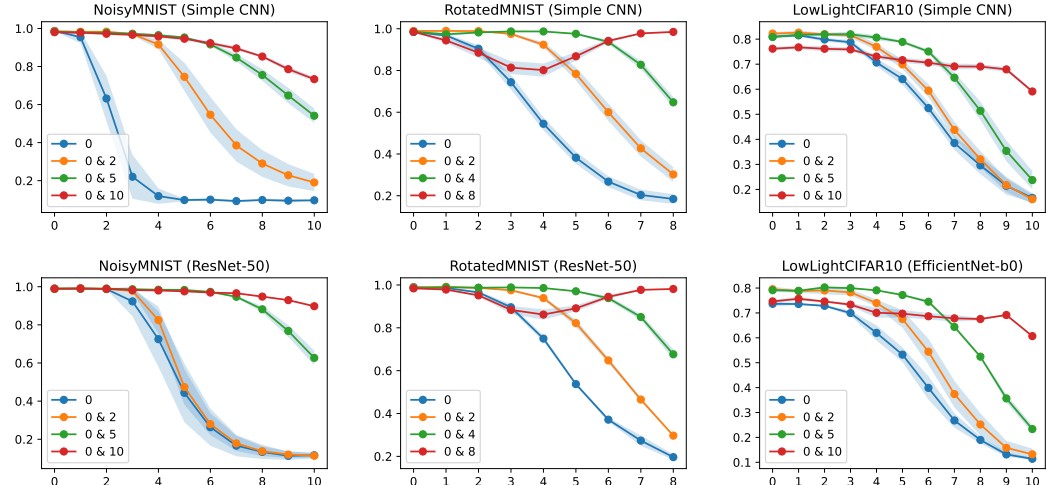

Figure 5: Performance of ERM models trained on domains under different shift degrees. The label of the curves denotes the indices of the training domains, e.g., "0 & 2" means that the models are trained on $\mathcal{D}_0$ and $\mathcal{D}_2$ of the corresponding dataset. The results are averaged over 20 models with different initialization.

From this perspective, we can see that the brittleness manifests when the spurious correlation between the local features and the target labels reaches a breaking point. However, where this breaking point is and how rapidly the correlation breaks seem to be totally dependent on the nature of the distribution shift and the task itself. While NOISYMNIST demonstrates a simple case where the breaking point is at a moderate degree of distribution shift, there can be scenarios where the break happens at a much lower or higher degree of distribution shift and happens much more rapidly. As a consequence, evaluations that only consider a narrow range of possible shift degrees would be highly unreliable in those scenarios.

### 4.3 ROBUSTNESS AT HIGHER DEGREES DOES NOT ALWAYS GUARANTEE ROBUSTNESS AT LOWER DEGREES

We have demonstrated in the previous section that models being more robust to milder distribution shifts does not imply that they are also more robust to stronger distribution shifts, even when the shift is just slightly stronger. In this section, we shed some light on the reverse question: does models being more robust to stronger distribution shifts imply them being more robust to milder distribution shifts?

To start with, we obtained models that are robust to strong distribution shifts by training the models on strongly shifted data together with clean data. In Figure 5, we compare these models with (i) models trained on much more mildly shifted data (also in addition to clean data) and (ii) models trained on clean data alone. For NOISYMNIST, the answer to our question is affirmative. The models that are more robust to stronger shifts are indeed more robust to milder shifts in general. However, this is not the whole story as the pattern seems to depend on the specific task in consideration.

We experimented with another dataset, ROTATEDMNIST, which is constructed in a similar fashion to NOISYMNIST while replacing noise with rotation (see Figure 7 for some examples). On the contrary to NOISYMNIST, robustness against higher shift degrees does *not* guarantee robustness to lower degrees in the case of ROTATEDMNIST when the shift degree of the additional training data is high. In comparison with models that trained on only clean data without any rotation, being more robust to the strongest shift may even harm generalization at the mildest degrees. Notably, the results are largely consistent between the two model architectures which have a great difference in complexity; however, the improvement brought by model complexity is still far from closing the gap. Again, this demonstrates the brittleness of the robustness of neural networks.

An important practical implication of the above finding is that, even for the same type of distribution shift, training on strongly shifted data may not be sufficient to obtain a model that is robust to milder shifts. Combined with our finding in Section 4, we arrive at the conclusion that the corresponding training data may be necessary to guarantee robustness at a certain degree of distribution shift for some tasks. Meanwhile, we should also note that there are scenarios where obtaining a dataset under a sufficiently strong shift is able to guarantee robustness to all milder shifts as in the case of NOISYMNIST. For these kinds of distribution shifts, it may require much less data to guarantee general robustness.

## 5 PRE-TRAINED REPRESENTATIONS ARE SENSITIVE TO NOVEL DOWNSTREAM DISTRIBUTION SHIFTS

Pre-training on large-scale datasets is one of the most effective ways that are known to consistently improve the generalization of neural networks across a wide range of tasks (Taori et al., 2020; Miller et al., 2021). In particular, foundation models like CLIP (Radford et al., 2021) have demonstrated remarkable zero-shot capability on a number of datasets that models trained on much smaller datasets fail to generalize to. In this section, we further investigate how pre-training would influence the OOD generalization behavior of models on downstream tasks under multiple shift degrees.

### 5.1 EXPERIMENT SETUP

**Datasets.** In addition to NOISYMNIST and ROTATEDMNIST, we consider two more complicated datasets, NOISYIMAGENET15 and LR-IMAGENET15, which are modifications of a 15-category subset of ImageNet on bird species. NOISYIMAGENET15 follows a similar construction to NOISYMNIST, introducing Gaussian noise on the pixel level, linearly spaced between 0 and 0.8, with values clipped to the 0-1 range. Meanwhile, LR-IMAGENET15 involves altering image resolution, first downsampling via bilinear interpolation and subsequently upsampling to $256 \times 256$, with the downsampled resolution in each domain corresponding to a factor of $0.8^d \cdot 256$, where $d$ represents the degree of distribution shift. We extend ROTATEDMNIST to span from 0 to 100 degrees in the experiments of this section.

**Pre-trained models.** ImageNet pre-trained ResNet-50 and ViT-B/32 from torchvision, along with CLIP checkpoints of these models released by OpenAI, serve as our primary models. These are adapted to downstream tasks through linear probing aligned with (Radford et al., 2021).

**Implementation details.** We use training-domain validation to select the best models among different iterations. All MNIST-based datasets were resized to $224 \times 224$ for uniformity across models. Pre-trained models normalized all datasets based on the statistics of their respective pre-training datasets, while randomly initialized models normalized based on MNIST statistics. Specifically, for the randomly initialized ResNet-50, no data augmentation was implemented during training on MNIST-based datasets. For the ViT-B/32 model, random affine transformations were applied, with rotation and shearing disabled for ROTATEDMNIST. We do not use any data augmentation for the experiments on NOISYIMAGENET15 and LR-IMAGENET15.

### 5.2 RESULTS

If large-scale pre-trained models like CLIP have learned generally more robust visual representations, they should be able to support a classifier that has better performance across a broad range of shift degrees than models trained from scratch as well as models trained on much smaller datasets, e.g., ImageNet (Deng et al., 2009). In Figure 6, we compare CLIP models with ImageNet pre-trained models and randomly initialized models that are trained from scratch on the downstream tasks to investigate the robustness of pre-trained representations.

On NOISYMNIST, although the pre-trained models perform equally well on clean domains as the randomly initialized models, they are surprisingly much more brittle to the distribution shift induced by the noise. Notably, the gap of accuracy between $CLIP_0$ and $RI_0$ increased by more than 40% from $\mathcal{D}_0$ to $\mathcal{D}_1$ on ResNet-50. This gap continued to increase until the shift reached a moderate degree. Moreover, on ViT-B/32, a similar pattern is observed albeit slightly improved. We hypothesize

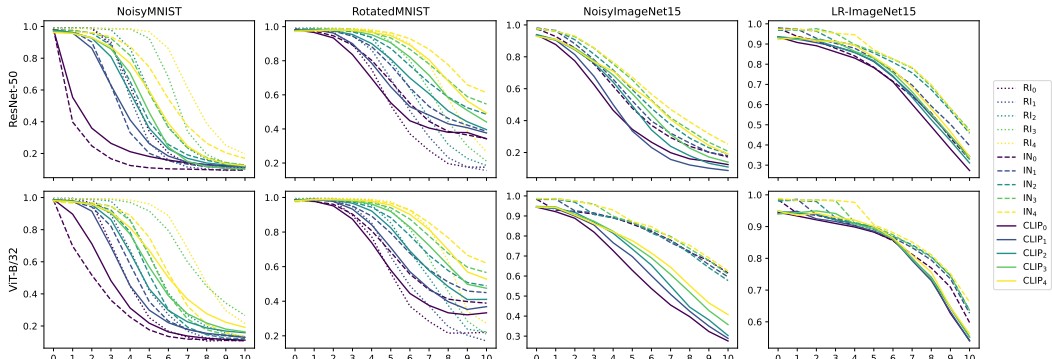

Figure 6: Performance of randomly initialized (**RI**) models, ImageNet (**IN**) pre-trained models, and **CLIP** models on different downstream tasks, evaluated over a broad range of shift degrees. The color of the curves indicates the domains used to train/adapt the models, e.g., $\text{RI}_d$ stands for models trained on $\{\mathcal{D}_0, \ldots, \mathcal{D}_d\}$ from scratch. The pre-trained models are adapted to the downstream tasks through linear probing. The results are averaged over three runs. Error bars are omitted for clarity (see Appendix B.5 for more details).

that the sensitiveness is largely because Gaussian noise is very rare in the training data of CLIP and also in ImageNet. Evaluation under a more common type of distribution shift, rotation, has provided some evidence to support our hypothesis. On ROTATEDMNIST, the pre-trained models are only slightly worse than the randomly initialized models under mild to moderate shifts while being much better under strong shifts.

In Figure 6, we also compare CLIP pre-trained models with ImageNet pre-trained models on harder problems: NOISYIMAGENET15 and LR-IMAGENET15. On these two datasets, we observe that ImageNet pre-trained models are generally more robust than CLIP models under both distribution shifts. Furthermore, the gap between the two model families starts out being small but gradually enlarges as the shift gets stronger. This suggests that not only the property of the downstream distribution shift (e.g., noise) but also the difference between the pre-training task and the downstream task itself plays a role in determining the robustness of the pre-trained models against downstream distribution shifts. This implies that even if the pre-training data encompass a diverse set of distribution shifts, the robustness against those shifts may not transfer or only transfer very little to a different task under similar kinds of shifts.

Last but not least, we note that further adapting the pre-trained models to downstream distribution shifts can sometimes significantly improve their robustness as shown in Figure 6. On one hand, this corroborates existing findings that large-scale pre-trained representations are highly versatile. On the other hand, this also suggests that unleashing the power of pre-training may still require sufficiently diverse downstream task data that covers the potential distribution shifts. Nevertheless, there are still inherent limits to the power of pre-training under novelty downstream distribution shifts as demonstrated in the case of NOISYMNIST where further adaptations help but only to a limited degree compared with training from scratch.

## 6 CONCLUSION

In this work, we have shown that even when a model is robust to some degree of distribution shift, a slight increase in the degree of the shift can still cause a significant performance drop. In addition, we also find that training with data under a high degree of distribution shift sometimes guarantees robustness to all lower degrees, but not always. Furthermore, we observe that large-scale pre-trained models like CLIP are sensitive to downstream distribution shifts, especially unseen or rarely seen ones. These findings all suggest that the robustness of neural networks under certain degrees of distribution shift can be quite brittle. For this reason, we should be more careful when interpreting evaluations based on data with a limited range of shift degrees. We also encourage future research to adopt a more comprehensive evaluation of OOD generalization such as the one in this work that considers multiple degrees of distribution shifts.

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

## A    ADDITIONAL INFORMATION ABOUT EXPERIMENT SETUP

### A.1    DATASETS

Random examples drawn from each domain of the datasets we used (except NOISYMNIST which is shown in Figure 2) are shown in Figure 7-10. The order of the examples is arranged according to the degree of the distribution shift from low to high.

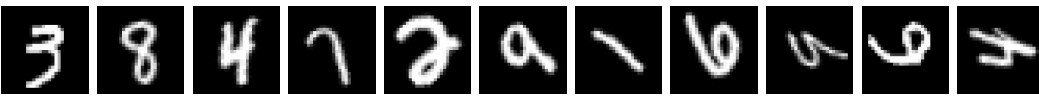

Figure 7: Examples of ROTATEDMNIST

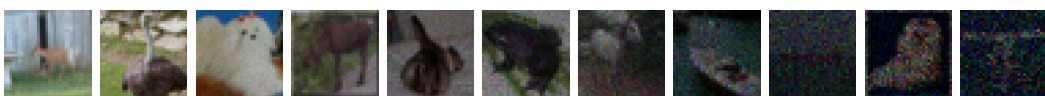

Figure 8: Examples of LOWLIGHTCIFAR10

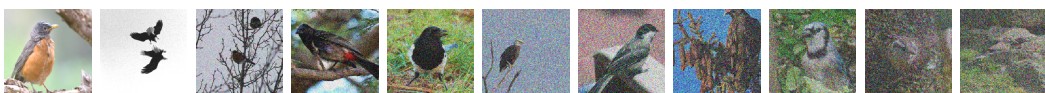

Figure 9: Examples of NOISYIMAGENET15

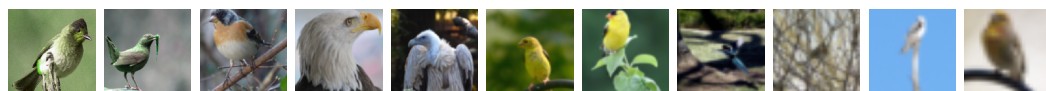

Figure 10: Examples of LR-IMAGENET15

For NOISYMNIST and ROTATEDMNIST, 60,000 images were divided into distinct training domains. For instance, in scenarios involving two training domains, each domain would encompass 30,000 images. Within the training domains, 20% of the data is allocated for in-distribution validation, aiding model calibration and selection. Every test domain of each altered dataset consists of 10,000 images, constructed using the same set of original images.

For NOISYIMAGENET15 and LR-IMAGENET15, we use the images in the training split of ImageNet to construct the training domains and the images in the validation split of ImageNet to construct the test domains. Similarly, the training domains divide the total 15,000 images in the training split. The test domains are constructed using the same set of original images, which consist of 750 images in total.

The 15 categories of birds we used in NOISYIMAGENET15 and LR-IMAGENET15, which correspond to indices 10 to 24 of the 1,000 categories of ImageNet, are "brambling, Fringilla montifringilla", "goldfinch, Carduelis carduelis", "house finch, linnet, Carpodacus mexicanus", "junco, snowbird", "indigo bunting, indigo finch, indigo bird, Passerina cyanea", "robin, American robin, Turdus migratorius", "bulbul", "jay", "magpie", "chickadee", "water ouzel, dipper", "kite", "bald eagle, American eagle, Haliaeetus leucocephalus", "vulture", and "great grey owl, great gray owl, Strix nebulosa".

In addition to the above datasets studied in the paper, we have also conducted preliminary experiments on another dataset called IMPULSENOISEMNIST. The dataset is constructed by gradually adding impulse noise to MNIST. Below are some of the examples of this dataset. The experiment results on this dataset are given in Appendix B.2.

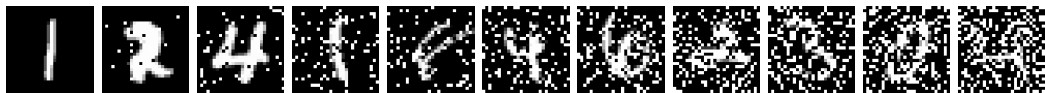

Figure 11: Examples of IMPULSENOISEMNIST

## A.2 ALGORITHMS

Here is the full list of domain generalization algorithms we used in this study:

- Invariant Risk Minimization (**IRM**, Arjovsky et al. (2019))
- Group Distributionally Robust Optimization (**GroupDRO**, Sagawa et al. (2020))
- Interdomain Mixup (**Mixup**, Yan et al. (2020))
- Marginal Transfer Learning (**MTL**, Blanchard et al. (2011))
- Maximum Mean Discrepancy (**MMD**, Li et al. (2018a))
- Deep CORAL (**CORAL**, Sun et al. (2017))
- Domain Adversarial Neural Network (**DANN**, Ganin et al. (2016))
- Conditional Domain Adversarial Neural Network (**CDANN**, Li et al. (2018b))
- Style Agnostic Networks (**SagNet**, Nam et al. (2021))
- Adaptive Risk Minimization (**ARM**, Zhang et al. (2021b))
- Variance Risk Extrapolation (**VREx**, Krueger et al. (2020))
- Representation Self-Challenging (**RSC**, Huang et al. (2020))
- Spectral Decoupling (**SD**, Pezeshki et al. (2021))
- Learning Explanations that are Hard to Vary (**AND-Mask**, Parascandolo et al. (2021))
- Smoothed-AND mask (**SAND-mask**, Shahtalebi et al. (2021))
- Out-of-Distribution Generalization with Maximal Invariant Predictor (**IGA**, Koyama & Yamaguchi (2020))
- Gradient Matching for Domain Generalization (**Fish**, Shi et al. (2021))
- Self-supervised Contrastive Regularization (**SelfReg**, Kim et al. (2021))
- Learning Representations that Support Robust Transfer of Predictors (**TRM**, Xu & Jaakkola (2021))
- Invariance Principle Meets Information Bottleneck for Out-of-Distribution Generalization (**IB-ERM** & **IB-IRM**, Ahuja et al. (2021))
- Optimal Representations for Covariate Shift (**CAD** & **CondCAD**, Ruan et al. (2021))
- Quantifying and Improving Transferability in Domain Generalization (**Transfer**, Zhang et al. (2021a))
- Invariant Causal Mechanisms through Distribution Matching (**CausIRL** with CORAL or MMD, Chevalley et al. (2022))
- Empirical Quantile Risk Minimization (**EQRM**, Eastwood et al. (2022))

We use the DomainBed (Gulrajani & Lopez-Paz, 2021) implementation for all the above algorithms.

## A.3 ADDITIONAL IMPLEMENTATION DETAILS

Except for learning rate, batch size, weight decay, and dropout, the search of other hyperparameters follows that of DomainBed (Gulrajani & Lopez-Paz, 2021). For experiments utilizing the ResNet-50 architecture, the original MNIST digits were resized to a resolution of 224x224 pixels. Subsequent normalization was performed using the mean and standard deviation inherent to the MNIST dataset to preserve data integrity and distribution.

# B ADDITIONAL EXPERIMENT RESULTS

## B.1 EFFECTS OF INCREASING THE NUMBER OF TRAINING DOMAINS

In Figure 3, we have shown the best-performing models (trained on $\mathcal{D}_0$ and $\mathcal{D}_1$) at each degree of NOISYMNIST. In Figure 12 and Figure 13 below, we show more results on this dataset, with models trained on wider ranges of domains. The results show that more training domains help. As more training domains are added, the gaps between the best-performing models gradually decrease. Nevertheless, the gaps seem to be closing at a slow rate. The discrepancy between the best-performing models is still largely present. In addition, the advantage of DG methods over ERM becomes less pronounced as the number of training domains increases.

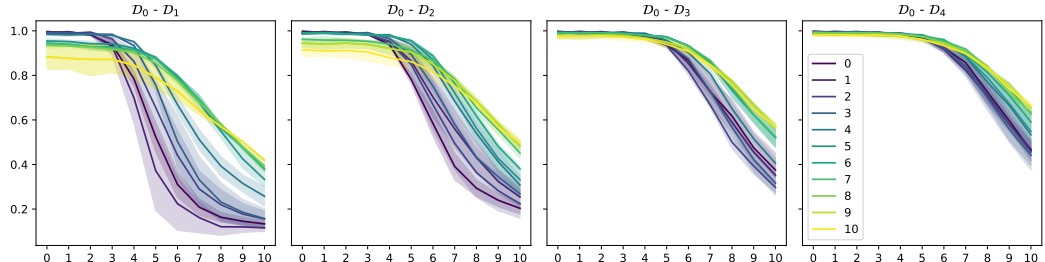

Figure 12: Performance of the best-performing models at each degree of NOISYMNIST. The label of the curves denotes the domain on which the models perform best. The title of each subplot denotes the training domains (e.g., "$\mathcal{D}_0$ - $\mathcal{D}_2$" means that the models are trained on $\mathcal{D}_0, \mathcal{D}_1, \mathcal{D}_2$). The results are averaged over the top 3 models of all algorithms at each degree.

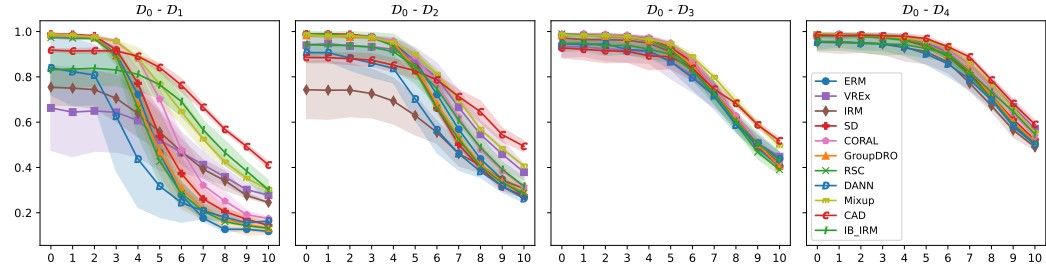

Figure 13: Performance of ERM and representative domain generalization algorithms on NOISYMNIST. The title of each subplot denotes the training domains (e.g., "$\mathcal{D}_0$ - $\mathcal{D}_2$" means that the models are trained on $\mathcal{D}_0, \mathcal{D}_1, \mathcal{D}_2$). The results are averaged over the top 3 models of each algorithm.

## B.2 EXPERIMENT RESULTS ON IMPULSENOISEMNIST

In Figure 14, we show the experiment results on the IMPULSENOISEMNIST dataset. The results on the generalization from milder to stronger shifts are largely consistent with the results in Figure 3. As for the results on the generalization from stronger to milder shifts, the pattern looks like that of LOWLIGHTCIFAR10. In particular, when the shift in the training data is not very strong (e.g., $\mathcal{D}_0$ and $\mathcal{D}_5$), the generalization pattern looks just like that of NoisyMNIST—robustness against milder shifts is not affected. On the other hand, when the shift in the training data is very strong (e.g., $\mathcal{D}_0$ and $\mathcal{D}_1 0$), the pattern looks like that of RotatedMNIST—robustness against milder shifts is significantly weakened.

## B.3 STRONG-TO-MILD GENERALIZATION PERFORMANCE OF DG ALGORITHMS

In this section, we show the performance of DG algorithms for the hardest generalization case in Figure. 5. From Figure 15 to Figure 17, we can see that most DG algorithms are helpful to the

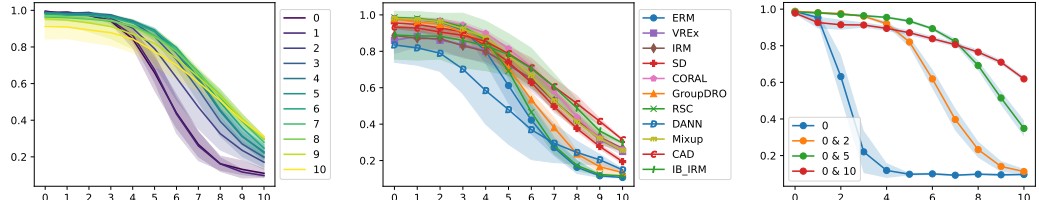

Figure 14: Results on IMPULSENOISEMNIST (Simple CNN). **(Left)** Performance of the best-performing models at each shift degree. The label of the curves denotes the domain on which the models perform best; **(Middle)** Performance of ERM and representative domain generalization algorithms on the two datasets; **(Right)** Performance of ERM models trained on domains under different shift degrees. The label of the curves denotes the indices of the training domains, e.g., "0 & 2" means that the models are trained on $\mathcal{D}_0$ and $\mathcal{D}_2$ of the corresponding dataset. The implementation details of these experiments follow those of NOISYMNIST.

generalization from stronger shifts to milder shifts in the case of ROTATEDMNIST, although only to a limited extent. On the other two datasets, only some of the DG algorithms are able to improve the generalization performance by a very small margin.

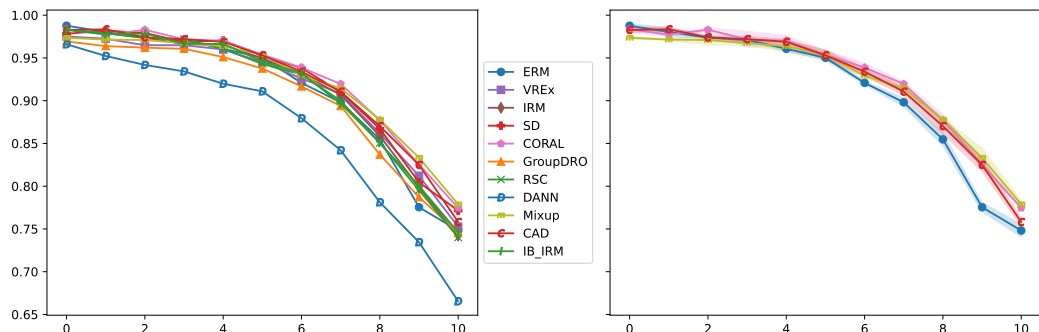

Figure 15: Average accuracy of the top-3 models of ERM and various DG algorithms trained on $\mathcal{D}_0$ and $\mathcal{D}_{10}$ of NOISYMNIST. The models are selected via training-domain validation. Error bars are omitted in the left sub-figure for clarity. The best-performing DG algorithms are compared with ERM in the right sub-figure. The results are averaged over 3 runs.

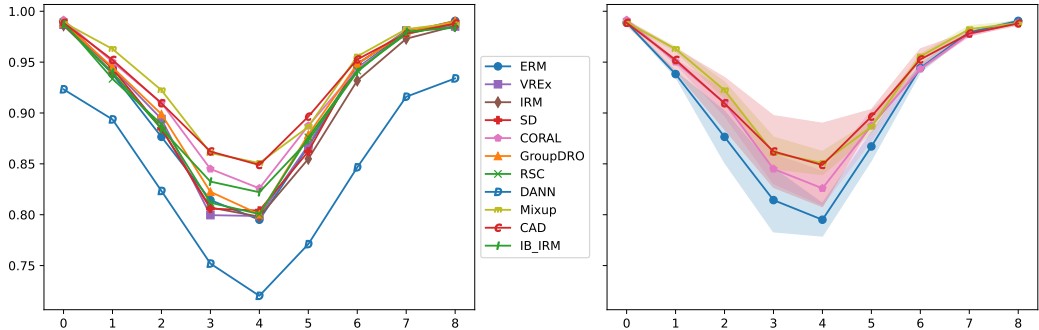

Figure 16: Average accuracy of the top-3 models of ERM and various DG algorithms trained on $\mathcal{D}_0$ and $\mathcal{D}_8$ of ROTATEDMNIST. The models are selected via training-domain validation. Error bars are omitted in the left sub-figure for clarity. The best-performing DG algorithms are compared with ERM in the right sub-figure. The results are averaged over 3 runs.

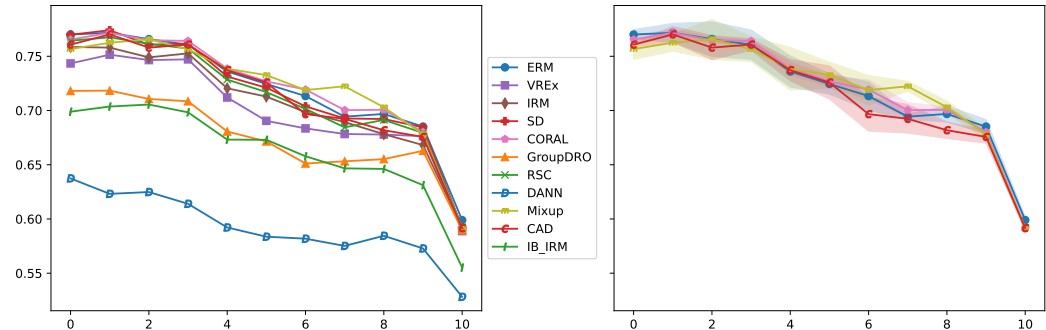

Figure 17: Average accuracy of the top-3 models of ERM and various DG algorithms trained on $\mathcal{D}_0$ and $\mathcal{D}_{10}$ of LowLightCIFAR10. The models are selected via training-domain validation. Error bars are omitted in the left sub-figure for clarity. The best-performing DG algorithms are compared with ERM in the right sub-figure. The results are averaged over 3 runs.

## B.4 Full numerical results of the experiments conducted in Section 4

From Table 2 to Table 19, we show the full numerical results of the experiments we conducted in Section 4. The first column of each table shows the domain on which the accuracy is used to select the top models.

| | Algorithm | $\mathcal{D}_0$ | $\mathcal{D}_1$ | $\mathcal{D}_2$ | $\mathcal{D}_3$ | $\mathcal{D}_4$ | $\mathcal{D}_5$ | $\mathcal{D}_6$ | $\mathcal{D}_7$ | $\mathcal{D}_8$ | $\mathcal{D}_9$ | $\mathcal{D}_{10}$ |
|---|---|---|---|---|---|---|---|---|---|---|---|---|
| $\mathcal{D}_0$ | ERM | 99.42±0.04 | 98.61±0.16 | 97.38±0.49 | 84.43±1.80 | 50.42±8.51 | 22.06±5.96 | 13.50±2.78 | 10.19±1.24 | 10.61±0.89 | 9.46±0.62 | 10.90±0.73 |
| | VREx | 99.29±0.13 | 98.74±0.29 | 97.46±0.21 | 84.25±4.04 | 50.37±4.10 | 23.95±5.34 | 13.68±3.39 | 11.40±1.37 | 9.51±1.12 | 10.04±0.21 | 10.14±0.78 |
| | IRM | 99.19±0.15 | 98.45±0.20 | 97.56±0.32 | 81.13±4.94 | 51.55±11.58 | 22.69±8.92 | 12.34±3.15 | 10.04±0.97 | 10.35±0.24 | 9.77±0.54 | 9.77±0.96 |
| | SD | 99.42±0.19 | 98.77±0.15 | 98.09±0.43 | 84.91±5.46 | 55.01±15.01 | 27.75±14.65 | 16.61±7.29 | 11.77±3.64 | 11.16±1.53 | 10.53±1.54 | 10.35±0.77 |
| | GroupDRO | 99.29±0.11 | 98.74±0.34 | 98.11±0.48 | 86.74±2.63 | 55.61±7.58 | 28.80±11.75 | 18.03±8.07 | 12.79±3.54 | 10.22±1.45 | 10.12±0.67 | 10.19±0.49 |
| | RSC | 99.29±0.11 | 98.79±0.48 | 97.80±0.45 | 87.74±5.85 | 54.85±11.72 | 24.11±6.79 | 15.67±2.45 | 11.37±1.23 | 10.85±0.91 | 10.85±0.33 | 9.72±0.43 |
| | Mixup | 99.55±0.04 | 98.64±0.23 | 98.06±0.51 | 94.65±1.13 | 84.46±4.24 | 68.37±5.30 | 53.49±8.36 | 40.96±7.53 | 32.26±6.57 | 27.88±5.68 | 23.11±4.44 |
| | CORAL | 99.32±0.04 | 98.93±0.10 | 98.53±0.10 | 93.87±2.97 | 72.09±12.96 | 40.75±18.42 | 21.62±12.85 | 16.38±6.60 | 12.50±3.12 | 12.16±2.52 | 11.14±1.69 |
| | MMD | 99.50±0.04 | 98.85±0.16 | 98.58±0.11 | 95.94±0.98 | 79.72±4.01 | 51.02±7.67 | 28.35±7.53 | 17.77±4.04 | 12.68±3.34 | 11.61±2.25 | 12.32±3.16 |
| | DANN | 97.82±0.71 | 97.17±0.63 | 95.91±0.22 | 82.86±6.73 | 49.37±13.61 | 27.07±10.24 | 18.11±5.83 | 14.31±4.69 | 11.37±3.75 | 12.40±2.04 | 10.93±1.89 |
| | CDANN | 98.01±0.20 | 97.12±0.20 | 96.93±0.40 | 88.73±1.80 | 64.52±6.08 | 35.93±6.59 | 21.28±5.24 | 15.12±2.83 | 12.58±1.41 | 11.53±0.74 | 10.56±0.89 |
| | MTL | 99.27±0.10 | 98.93±0.33 | 97.38±0.47 | 79.27±3.28 | 40.30±2.95 | 17.87±3.13 | 11.79±2.17 | 10.51±0.49 | 9.17±0.39 | 9.64±1.03 | 9.72±0.41 |
| | SagNet | 99.53±0.06 | 98.74±0.28 | 98.45±0.49 | 91.27±2.12 | 65.99±8.59 | 33.65±11.21 | 18.00±6.62 | 13.31±4.90 | 12.76±3.88 | 11.48±3.34 | 11.53±2.47 |
| | ARM | 99.40±0.04 | 98.87±0.23 | 98.01±0.07 | 82.86±0.64 | 51.70±3.69 | 23.27±3.42 | 13.44±1.63 | 9.83±0.48 | 9.75±0.82 | 9.20±0.23 | 9.51±0.57 |
| | ANDMask | 99.34±0.04 | 98.56±0.32 | 97.80±0.28 | 82.94±2.83 | 43.45±2.34 | 15.46±1.82 | 9.72±0.85 | 10.09±0.27 | 9.38±0.77 | 9.85±0.58 | 9.67±0.29 |
| | SANDMask | 99.37±0.06 | 98.51±0.06 | 98.14±0.32 | 84.72±1.91 | 49.61±3.78 | 20.07±5.03 | 11.95±2.57 | 12.08±1.78 | 10.25±2.30 | 10.46±0.56 | 10.14±0.72 |
| | IGA | 19.26±0.45 | 18.19±1.30 | 17.09±2.44 | 15.41±3.17 | 16.22±4.38 | 16.48±4.15 | 16.80±5.63 | 16.56±4.09 | 15.57±4.18 | 14.70±3.66 | 14.49±3.80 |
| | SelfReg | 99.48±0.07 | 99.14±0.13 | 98.14±0.71 | 93.79±0.84 | 72.22±2.31 | 41.51±5.86 | 20.89±4.89 | 13.13±2.62 | 11.45±1.80 | 11.32±0.78 | 9.49±0.49 |
| | Fish | 99.21±0.06 | 98.87±0.04 | 97.43±0.43 | 87.68±4.28 | 59.72±5.78 | 31.76±3.70 | 18.95±3.36 | 11.45±1.48 | 10.67±1.66 | 9.85±0.76 | 10.46±0.78 |
| | TRM | 99.45±0.06 | 98.85±0.23 | 97.54±0.48 | 83.36±2.57 | 49.87±5.40 | 21.67±5.79 | 13.78±3.04 | 10.80±2.49 | 10.32±2.06 | 9.80±0.55 | 9.56±0.30 |
| | IB-ERM | 99.45±0.06 | 98.98±0.23 | 98.61±0.23 | 90.96±4.96 | 64.07±16.06 | 37.13±21.05 | 24.69±16.93 | 19.10±11.65 | 15.88±7.53 | 13.97±4.93 | 11.79±3.07 |
| | IB-IRM | 99.29±0.17 | 99.08±0.13 | 98.24±0.21 | 95.31±2.04 | 77.83±10.02 | 51.52±16.77 | 33.94±20.96 | 24.08±16.44 | 19.99±13.82 | 17.82±10.34 | 14.33±6.35 |
| | CAD | 99.37±0.06 | 98.90±0.26 | 98.85±0.39 | 95.02±2.83 | 78.01±14.23 | 48.11±16.73 | 25.89±11.97 | 16.54±4.66 | 11.79±2.56 | 11.24±2.11 | 11.16±1.17 |
| | CondCAD | 99.45±0.11 | 98.85±0.10 | 98.85±0.27 | 96.33±1.29 | 82.47±8.03 | 60.19±15.98 | 42.16±17.25 | 28.43±12.13 | 21.80±9.15 | 18.32±5.24 | 14.86±4.36 |
| | Transfer | 43.00±39.47 | 42.64±39.99 | 40.12±41.18 | 38.60±39.14 | 30.16±28.98 | 20.41±14.04 | 13.26±4.75 | 10.80±1.04 | 9.96±0.39 | 9.30±0.82 | 10.48±0.50 |
| | CausIRL-CORAL | 99.55±0.15 | 99.19±0.35 | 98.19±0.45 | 95.02±3.35 | 79.80±8.56 | 54.01±14.17 | 32.97±10.68 | 20.39±6.04 | 14.88±3.74 | 12.55±2.28 | 11.84±1.78 |
| | CausIRL-MMD | 99.40±0.13 | 98.69±0.26 | 98.72±0.32 | 95.23±1.80 | 82.31±1.02 | 60.14±4.17 | 40.20±4.58 | 27.73±2.39 | 19.86±1.61 | 15.44±0.75 | 14.68±1.67 |
| | EQRM | 99.45±0.00 | 98.72±0.29 | 97.96±0.46 | 91.72±4.94 | 65.46±9.71 | 36.29±11.46 | 18.74±8.68 | 13.92±5.46 | 12.24±3.41 | 12.87±3.53 | 11.37±1.43 |
| $\mathcal{D}_1$ | ERM | 98.79±0.04 | 99.34±0.04 | 97.41±0.40 | 85.27±3.23 | 55.08±8.84 | 27.31±8.15 | 14.81±3.67 | 10.77±1.72 | 10.12±0.35 | 10.12±1.09 | 9.80±0.04 |
| | VREx | 98.93±0.16 | 99.16±0.07 | 97.72±0.22 | 87.68±8.03 | 62.47±13.32 | 32.34±9.61 | 18.11±5.18 | 13.21±2.81 | 10.67±2.28 | 10.67±1.35 | 10.14±0.59 |
| | IRM | 98.64±0.42 | 98.82±0.22 | 97.27±0.83 | 85.32±5.47 | 58.31±6.62 | 28.80±5.25 | 14.20±2.16 | 10.61±0.84 | 10.59±0.58 | 9.38±0.04 | 10.22±0.33 |
| | SD | 98.64±0.04 | 99.34±0.10 | 98.19±0.33 | 91.19±4.26 | 64.02±13.40 | 32.47±15.67 | 19.16±9.98 | 14.73±5.71 | 12.50±4.45 | 11.95±1.98 | 10.80±1.69 |
| | GroupDRO | 98.87±0.07 | 99.32±0.13 | 98.32±0.04 | 89.15±3.00 | 58.54±9.41 | 28.01±9.66 | 16.75±5.18 | 11.66±2.45 | 10.56±0.75 | 9.88±0.81 | 9.67±0.61 |
| | RSC | 98.95±0.26 | 99.27±0.04 | 98.09±0.16 | 89.05±2.78 | 59.09±3.58 | 25.00±2.38 | 13.71±1.94 | 9.98±1.19 | 10.32±0.74 | 10.27±0.87 | 9.51±0.06 |
| | Mixup | 99.16±0.10 | 99.50±0.07 | 98.24±0.32 | 94.37±1.52 | 83.28±3.82 | 68.16±3.85 | 51.47±3.22 | 39.05±1.59 | 29.95±1.73 | 24.50±1.77 | 20.89±2.86 |
| | CORAL | 99.00±0.13 | 99.45±0.06 | 98.24±0.21 | 93.00±1.72 | 65.49±7.42 | 33.60±7.54 | 15.59±2.12 | 10.64±0.16 | 9.88±0.42 | 9.38±1.09 | 10.27±0.49 |
| | MMD | 98.93±0.10 | 99.34±0.07 | 98.82±0.11 | 96.51±1.46 | 83.36±7.23 | 57.23±12.87 | 35.12±8.80 | 22.69±5.44 | 16.12±5.67 | 14.68±4.40 | 13.44±3.06 |
| | DANN | 97.54±0.94 | 97.56±0.36 | 95.94±0.43 | 83.31±7.15 | 54.85±17.49 | 33.10±14.30 | 21.04±7.04 | 13.99±3.39 | 11.45±1.71 | 11.24±0.68 | 9.96±0.58 |
| | CDANN | 97.75±0.20 | 97.30±0.24 | 96.49±0.21 | 88.05±2.49 | 65.30±7.81 | 38.71±7.22 | 22.64±4.48 | 14.31±3.41 | 11.37±1.75 | 10.56±0.77 | 10.04±0.64 |
| | MTL | 98.98±0.04 | 99.32±0.10 | 97.80±0.65 | 82.15±4.58 | 48.61±10.74 | 22.82±7.68 | 12.40±3.10 | 10.06±1.25 | 10.46±0.56 | 9.77±0.78 | 9.70±1.51 |
| | SagNet | 99.21±0.23 | 99.53±0.06 | 98.58±0.32 | 91.04±3.37 | 59.07±8.50 | 22.22±5.48 | 11.45±1.38 | 9.38±1.18 | 9.96±0.32 | 9.20±1.12 | 9.91±0.93 |
| | ARM | 98.82±0.51 | 99.29±0.11 | 97.48±0.42 | 79.48±6.45 | 40.83±11.38 | 17.32±4.26 | 10.72±1.03 | 10.82±0.52 | 9.96±0.53 | 9.54±0.69 | 9.36±0.78 |
| | ANDMask | 98.87±0.33 | 99.08±0.04 | 98.09±0.53 | 87.74±5.85 | 55.24±14.81 | 26.47±12.81 | 15.23±6.64 | 13.36±4.61 | 11.90±2.72 | 10.77±2.12 | 11.06±1.60 |
| | SANDMask | 98.56±0.29 | 99.00±0.04 | 97.90±0.35 | 83.05±1.25 | 49.90±2.65 | 24.82±4.62 | 13.60±0.74 | 11.32±1.58 | 9.46±1.04 | 10.30±1.28 | 9.75±0.74 |
| | IGA | 19.26±0.45 | 18.19±1.30 | 17.09±2.44 | 15.41±3.17 | 16.22±4.38 | 16.48±4.15 | 16.80±5.63 | 16.56±4.09 | 15.57±4.18 | 14.70±3.66 | 14.49±3.80 |
| | SelfReg | 99.08±0.20 | 99.42±0.07 | 98.74±0.06 | 92.32±2.26 | 62.79±12.83 | 31.42±11.51 | 17.53±4.90 | 13.10±2.70 | 11.24±2.12 | 10.88±1.03 | 10.46±0.58 |
| | Fish | 98.72±0.60 | 99.03±0.04 | 97.17±0.48 | 84.80±4.69 | 55.82±5.00 | 28.54±3.90 | 16.01±1.88 | 10.40±0.87 | 10.56±0.90 | 9.70±0.44 | 10.30±0.67 |
| | TRM | 99.03±0.30 | 99.34±0.20 | 97.93±0.23 | 88.42±1.81 | 57.18±4.82 | 23.72±4.12 | 13.76±0.68 | 10.43±0.85 | 8.88±0.29 | 9.46±1.04 | 9.07±0.89 |
| | IB-ERM | 98.72±0.13 | 99.40±0.15 | 98.82±0.17 | 95.02±1.01 | 77.18±5.52 | 48.61±8.87 | 27.59±5.74 | 19.44±3.63 | 16.33±2.63 | 13.94±1.63 | 13.13±1.33 |
| | IB-IRM | 98.29±0.20 | 99.21±0.19 | 98.48±0.19 | 94.60±2.76 | 80.40±11.85 | 61.24±19.24 | 46.49±20.79 | 34.62±16.23 | 27.96±12.94 | 23.17±8.71 | 19.16±6.21 |
| | CAD | 99.06±0.23 | 99.42±0.07 | 98.69±0.30 | 95.15±1.80 | 74.69±8.28 | 40.36±9.32 | 19.71±4.08 | 13.55±2.75 | 11.37±1.76 | 10.51±0.69 | 9.96±0.80 |
| | CondCAD | 98.98±0.33 | 99.41±0.07 | 98.85±0.76 | 81.87±2.98 | 55.39±11.55 | 33.28±13.80 | 21.57±8.14 | 17.43±6.86 | 13.92±5.51 | 13.44±3.20 | |
| | Transfer | 42.74±39.61 | 43.84±39.16 | 40.64±40.69 | 38.71±38.67 | 29.27±26.28 | 18.95±12.24 | 12.45±3.01 | 10.85±0.58 | 10.25±0.26 | 8.81±0.11 | 10.43±0.47 |
| | CausIRL-CORAL | 99.40±0.26 | 99.58±0.10 | 98.51±0.45 | 96.12±0.67 | 82.91±3.86 | 57.39±10.73 | 34.56±9.80 | 21.75±6.44 | 14.86±3.72 | 13.60±2.61 | 11.92±1.44 |
| | CausIRL-MMD | 99.06±0.28 | 99.24±0.10 | 98.98±0.17 | 96.86±1.11 | 82.73±7.45 | 57.65±12.07 | 37.79±13.10 | 24.69±8.92 | 18.92±6.72 | 14.78±5.17 | 12.87±2.80 |
| | EQRM | 98.56±0.20 | 99.27±0.10 | 97.90±0.43 | 89.52±2.21 | 61.43±3.20 | 30.48±3.83 | 14.94±4.13 | 11.69±2.65 | 10.14±1.61 | 10.53±0.48 | 10.48±0.30 |
| $\mathcal{D}_2$ | ERM | 98.93±0.19 | 98.82±0.13 | 98.74±0.06 | 90.64±1.73 | 65.38±4.72 | 33.94±6.71 | 17.48±5.12 | 11.64±2.36 | 9.72±1.08 | 9.96±0.77 | 10.77±0.61 |
| | VREx | 98.19±0.29 | 98.32±0.36 | 98.51±0.06 | 96.04±0.93 | 83.23±7.42 | 61.29±17.56 | 39.86±16.13 | 27.07±12.53 | 19.79±7.58 | 15.62±5.64 | 13.84±3.82 |
| | IRM | 98.90±0.17 | 98.53±0.38 | 98.45±0.16 | 89.83±3.63 | 59.77±5.81 | 31.18±5.70 | 13.29±1.89 | 10.40±0.78 | 10.95±0.37 | 9.56±0.74 | 8.99±0.56 |
| | SD | 98.87±0.58 | 98.79±0.26 | 98.87±0.10 | 91.19±2.50 | 69.31±10.48 | 41.43±16.79 | 24.48±9.56 | 16.59±5.33 | 12.60±3.02 | 11.90±2.21 | 10.74±1.46 |
| | GroupDRO | 98.74±0.11 | 98.85±0.20 | 98.66±0.06 | 89.91±3.50 | 60.12±9.76 | 30.66±11.49 | 18.55±6.36 | 13.55±2.26 | 11.48±1.87 | 11.03±2.44 | 9.83±1.19 |
| | RSC | 98.85±0.37 | 98.72±0.44 | 98.58±0.11 | 91.54±2.22 | 63.23±5.10 | 30.58±4.46 | 16.04±0.82 | 10.95±0.13 | 10.38±0.80 | 9.70±1.00 | 9.59±0.17 |
| | Mixup | 99.11±0.13 | 99.06±0.45 | 98.79±0.10 | 95.91±0.72 | 86.37±2.72 | 70.31±2.44 | 51.99±2.27 | 38.55±3.30 | 30.42±2.28 | 25.71±2.44 | 21.93±3.72 |
| | CORAL | 98.98±0.17 | 99.06±0.13 | 99.08±0.15 | 94.00±1.46 | 73.43±3.83 | 43.47±2.68 | 21.93±2.12 | 13.42±2.71 | 11.24±1.70 | 10.48±1.17 | 10.72±0.48 |
| | MMD | 99.08±0.10 | 98.90±0.28 | 99.37±0.06 | 97.33±0.06 | 88.84±2.05 | 69.65±4.64 | 49.87±4.82 | 33.12±4.19 | 24.97±3.91 | 19.99±3.20 | 18.21±0.24 |
| | DANN | 97.67±0.83 | 97.41±0.46 | 96.25±0.26 | 84.49±8.13 | 53.04±17.12 | 30.37±13.96 | 20.57±9.10 | 15.54±6.35 | 12.63±5.53 | 13.21±3.18 | 10.69±1.56 |
| | CDANN | 98.01±0.20 | 97.12±0.20 | 96.93±0.40 | 88.73±1.80 | 64.52±6.08 | 35.93±6.59 | 21.28±5.24 | 15.12±2.83 | 12.58±1.41 | 11.53±0.74 | 10.56±0.89 |
| | MTL | 98.82±0.23 | 98.74±0.06 | 98.66±0.06 | 90.07±1.48 | 64.07±7.72 | 35.48±9.43 | 19.60±5.64 | 14.47±3.50 | 12.03±2.33 | 11.58±2.20 | 11.03±1.60 |
| | SagNet | 99.19±0.04 | 98.93±0.07 | 99.21±0.06 | 93.19±2.07 | 69.34±3.33 | 34.75±4.83 | 17.37±3.40 | 12.21±0.55 | 10.48±0.78 | 9.56±1.31 | 10.22±1.29 |
| | ARM | 98.64±0.04 | 98.61±0.24 | 98.56±0.07 | 86.50±4.82 | 55.82±7.02 | 26.78±4.34 | 14.62±2.22 | 10.95±1.04 | 9.80±0.47 | 9.98±0.13 | 8.60±0.61 |
| | ANDMask | 98.95±0.10 | 98.95±0.16 | 98.58±0.17 | 88.97±5.08 | 61.11±11.96 | 29.85±11.91 | 17.40±5.46 | 13.36±4.61 | 11.90±2.73 | 10.90±2.03 | 11.14±1.55 |
| | SANDMask | 98.66±0.45 | 98.61±0.21 | 98.54±0.08 | 90.38±2.98 | 61.11±9.76 | 29.98±9.92 | 14.91±4.37 | 12.32±1.61 | 9.83±1.24 | 10.51±0.43 | 9.64±0.70 |
| | IGA | 19.26±0.45 | 18.19±1.30 | 17.09±2.44 | 15.41±3.17 | 16.22±4.38 | 16.48±4.15 | 16.80±5.63 | 16.56±4.09 | 15.57±4.18 | 14.70±3.66 | 14.49±3.80 |
| | SelfReg | 99.08±0.23 | 98.90±0.17 | 99.27±0.07 | 93.32±0.39 | 70.96±1.23 | 39.05±4.92 | 22.25±3.11 | 14.94±2.78 | 13.23±1.32 | 11.50±1.35 | 10.69±1.25 |
| | Fish | 98.66±0.28 | 98.74±0.22 | 98.45±0.10 | 88.42±6.36 | 60.69±8.57 | 31.21±6.20 | 16.04±2.51 | 11.24±1.48 | 10.48±1.87 | 9.70±0.77 | 9.70±0.47 |
| | TRM | 99.08±0.26 | 98.72±0.04 | 98.66±0.13 | 92.03±2.34 | 67.48±7.16 | 36.87±6.49 | 18.45±2.01 | 12.66±0.74 | 10.77±1.45 | 9.72±0.74 | 10.01±0.27 |
| | IB-ERM | 99.06±0.13 | 98.93±0.16 | 99.24±0.13 | 95.70±1.20 | 79.43±5.14 | 52.99±14.94 | 31.34±12.17 | 22.06±9.40 | 16.82±7.19 | 14.12±3.74 | 14.60±3.74 |
| | IB-IRM | 99.24±0.20 | 99.19±0.20 | 98.77±0.04 | 95.39±1.15 | 81.58±5.49 | 60.25±15.41 | 43.34±18.16 | 31.94±14.53 | 25.42±11.07 | 21.10±6.86 | 19.10±6.12 |
| | CAD | 98.98±0.23 | 98.85±0.39 | 99.11±0.16 | 95.83±0.87 | 81.73±3.61 | 50.42±4.01 | 28.59±2.41 | 18.45±3.76 | 13.97±3.10 | 13.16±1.99 | 11.71±1.01 |
| | CondCAD | 98.98±0.33 | 98.95±0.04 | 99.18±0.10 | 95.94±0.83 | 84.62±3.63 | 63.52±8.68 | 44.86±10.89 | 31.47±8.64 | 22.22±6.40 | 18.40±5.33 | 16.14±3.06 |
| | Transfer | 41.38±40.63 | 42.09±40.38 | 40.93±40.60 | 38.76±39.02 | 31.08±28.33 | 20.73±13.80 | 14.20±4.08 | 11.50±0.58 | 10.27±0.33 | 9.75±1.28 | 10.80±0.35 |
| | CausIRL-CORAL | 99.27±0.04 | 99.24±0.19 | 99.29±0.11 | 94.79±1.00 | 77.88±6.68 | 50.05±10.97 | 26.94±7.73 | 16.93±3.77 | 12.71±1.87 | 11.56±1.50 | 10.17±0.23 |
| | CausIRL-MMD | 99.00±0.20 | 98.90±0.23 | 99.14±0.06 | 97.43±0.27 | 85.14±1.80 | 57.73±2.56 | 33.05±0.20 | 19.39±0.97 | 14.60±0.96 | 12.03±0.34 | 11.24±1.26 |
| | EQRM | 98.82±0.23 | 98.93±0.07 | 98.85±0.10 | 92.56±0.68 | 70.34±1.80 | 42.11±2.39 | 22.77±0.97 | 15.25±1.41 | 12.50±1.03 | 10.74±0.76 | 10.95±0.71 |
| $\mathcal{D}_3$ | ERM | 98.64±0.13 | 98.43±0.36 | 98.01±0.36 | 94.68±0.43 | 74.55±6.61 | 42.69±8.33 | 21.88±5.97 | 13.81±2.38 | 10.61±1.08 | 11.19±0.78 | 10.40±0.48 |
| | VREx | 97.88±0.74 | 98.24±0.58 | 97.93±0.52 | 96.65±0.58 | 86.77±6.45 | 67.56±15.06 | 46.17±15.66 | 30.87±15.57 | 21.78±11.10 | 18.21±9.03 | 15.38±5.66 |
| | IRM | 97.67±0.58 | 97.01±0.64 | 96.78±0.72 | 93.79±0.39 | 76.89±3.68 | 50.58±5.61 | 27.80±6.37 | 16.82±2.67 | 12.13±1.16 | 10.06±0.97 | 10.32±1.13 |
| | SD | 97.27±1.59 | 98.06±0.85 | 97.69±0.82 | 95.81±0.91 | 78.28±4.70 | 50.79±8.66 | 29.19±7.48 | 21.04±3.57 | 15.88±3.75 | 14.20±2.85 | 12.37±2.43 |
| | GroupDRO | 98.19±0.19 | 98.03±0.33 | 97.72±0.17 | 94.00±0.61 | 72.27±2.88 | 44.58±4.78 | 25.03±5.19 | 15.17±2.82 | 11.27±0.35 | 11.16±1.20 | 9.77±0.52 |
| | RSC | 97.38±0.23 | 96.80±0.66 | 95.89±0.52 | 94.58±0.53 | 84.25±4.63 | 61.37±7.21 | 39.60±7.10 | 25.13±4.38 | 18.29±2.29 | 15.41±1.31 | 12.66±0.80 |
| | Mixup | 98.61±0.27 | 98.64±0.48 | 98.03±0.19 | 97.27±0.32 | 91.72±2.38 | 79.66±3.98 | 64.23±5.17 | 48.51±3.95 | 37.29±3.31 | 30.06±2.90 | 25.73±2.75 |
| | CORAL | 99.06±0.26 | 99.08±0.32 | 98.48±0.20 | 97.43±0.26 | 85.06±2.85 | 57.08±6.17 | 33.25±5.23 | 21.36±3.70 | 15.41±2.17 | 14.02±1.24 | 12.79±0.73 |
| | MMD | 98.77±0.20 | 98.82±0.28 | 98.35±0.39 | 98.12±0.13 | 92.98±0.75 | 74.32±1.73 | 44.97±7.78 | 28.83±7.55 | 18.53±5.52 | 16.30±4.09 | 14.52±3.54 |
| | DANN | 96.20±0.52 | 95.94±0.93 | 94.92±1.35 | 91.17±1.16 | 70.96±2.33 | 43.76±7.20 | 29.14±5.71 | 20.13±3.47 | 16.09±3.26 | 13.60±2.96 | 11.40±1.09 |
| | CDANN | 96.96±0.73 | 96.04±1.00 | 96.31±0.34 | 92.22±1.39 | 72.90±4.50 | 44.76±2.77 | 27.02±0.64 | 17.69±2.04 | 13.26±1.16 | 10.77±1.28 | 11.01±0.84 |
| | MTL | 98.51±0.17 | 98.32±0.48 | 97.82±0.38 | 94.50±0.51 | 76.89±3.47 | 50.34±10.19 | 30.50±10.10 | 20.65±8.92 | 16.67±7.46 | 15.20±6.52 | 13.86±4.94 |
| | SagNet | 99.11±0.27 | 99.00±0.04 | 98.79±0.42 | 96.25±0.32 | 78.69±3.79 | 50.03±6.79 | 27.02±4.01 | 15.30±2.03 | 12.00±1.35 | 10.67±1.96 | 10.88±1.36 |
| | ARM | 98.11±0.63 | 98.11±0.68 | 97.75±0.36 | 94.84±1.20 | 77.23±7.30 | 51.26±13.87 | 28.56±9.32 | 18.32±6.27 | 12.87±2.82 | 11.90±3.06 | 10.80±1.33 |
| | ANDMask | 98.69±0.30 | 98.19±0.11 | 97.69±0.35 | 94.31±1.97 | 76.60±10.25 | 45.26±17.17 | 24.97±11.30 | 18.41±6.11 | 14.96±4.10 | 13.73±3.79 | 13.00±3.01 |
| | SANDMask | 97.20±0.92 | 96.75±0.81 | 96.75±0.32 | 93.68±0.27 | 76.97±6.89 | 50.73±14.03 | 27.86±13.44 | 19.84±8.01 | 13.63±3.87 | 11.95±1.05 | 10.64±0.23 |
| | IGA | 15.83±4.59 | 16.17±4.15 | 15.83±4.16 | 16.27±1.95 | 16.33±4.23 | 16.17±4.60 | 16.88±5.52 | 16.35±4.37 | 15.33±4.50 | 14.36±4.08 | 14.26±4.10 |
| | SelfReg | 99.08±0.32 | 98.82±0.19 | 98.24±0.58 | 96.67±0.16 | 83.10±2.82 | 60.04±9.14 | 37.45±12.21 | 22.69±10.08 | 17.24±8.21 | 15.25±5.34 | 12.47±2.08 |
| | Fish | 97.93±0.35 | 98.30±0.15 | 97.77±0.38 | 94.44±0.10 | 75.50±3.29 | 50.73±5.30 | 28.75±4.46 | 17.87±2.39 | 14.47±0.71 | 11.69±0.52 | 10.56±0.43 |
| | TRM | 98.51±0.78 | 98.77±0.10 | 98.27±0.42 | 94.86±0.39 | 74.50±2.39 | 42.16±2.61 | 19.47±1.83 | 11.95±1.43 | 10.43±1.05 | 9.77±1.44 | 9.72±1.33 |
| | IB-ERM | 98.43±0.56 | 98.40±0.71 | 98.27±0.29 | 97.25±0.50 | 89.94±3.52 | 73.45±10.12 | 54.56±15.50 | 41.17±12.87 | 32.44±10.58 | 26.10±7.81 | 22.56±6.63 |
| | IB-IRM | 99.27±0.17 | 99.06±0.11 | 98.53±0.16 | 95.96±1.43 | 84.41±6.41 | 64.26±14.98 | 47.64±19.17 | 34.88±15.86 | 28.33±12.42 | 23.35±8.47 | 19.23±6.10 |
| | CAD | 98.77±0.19 | 98.48±0.26 | 98.45±0.20 | 98.01±0.13 | 92.06±3.50 | 80.08±1.95 | 59.64±3.33 | 43.16±1.90 | 31.18±3.24 | 23.95±2.67 | 19.29±1.77 |
| | CondCAD | 98.79±0.35 | 98.53±0.26 | 98.45±0.24 | 98.19±0.13 | 92.82±0.67 | 78.43±3.51 | 60.25±7.64 | 42.14±5.61 | 30.45±5.64 | 25.34±4.94 | 19.44±3.95 |
| | Transfer | 41.09±40.93 | 41.38±40.98 | 39.99±41.16 | 39.91±39.60 | 31.16±29.39 | 20.41±14.92 | 13.02±4.16 | 10.59±0.86 | 10.27±0.16 | 9.41±0.96 | 10.40±0.35 |
| | CausIRL-CORAL | 99.27±0.44 | 99.03±0.36 | 98.38±0.32 | 98.19±0.19 | 91.56±2.54 | 72.14±4.58 | 48.38±6.07 | 31.29±6.00 | 21.86±3.32 | 16.77±3.29 | 14.75±2.42 |
| | CausIRL-MMD | 98.66±0.06 | 98.85±0.37 | 98.69±0.32 | 97.82±0.16 | 89.41±3.02 | 70.10±9.80 | 49.32±11.80 | 31.39±9.26 | 25.00±6.58 | 19.58±5.71 | 17.43±3.98 |
| | EQRM | 98.69±0.15 | 98.06±0.72 | 98.06±0.43 | 95.89±0.55 | 82.39±1.16 | 55.71±1.91 | 30.45±1.82 | 19.34±1.65 | 14.18±1.54 | 12.68±1.46 | 11.03±1.49 |

Table 2: Average accuracy of top-3 models of each algorithm at each shift degree of NOISYMNIST (Simple CNN). This table shows the results on shift degrees from $\mathcal{D}_0$ to $\mathcal{D}_3$.

| | Algorithm | $\mathcal{D}_0$ | $\mathcal{D}_1$ | $\mathcal{D}_2$ | $\mathcal{D}_3$ | $\mathcal{D}_4$ | $\mathcal{D}_5$ | $\mathcal{D}_6$ | $\mathcal{D}_7$ | $\mathcal{D}_8$ | $\mathcal{D}_9$ | $\mathcal{D}_{10}$ |
|---|---|---|---|---|---|---|---|---|---|---|---|---|
| $\mathcal{D}_4$ | ERM | 98.56±0.04 | 98.53±0.41 | 98.06±0.32 | 94.58±0.57 | 77.75±2.85 | 47.67±5.16 | 26.47±4.99 | 16.51±2.53 | 12.40±1.48 | 12.26±0.82 | 10.77±0.69 |
| | VREx | 97.67±0.49 | 97.77±0.30 | 97.75±0.55 | 96.65±0.58 | 90.15±1.72 | 74.27±5.62 | 53.38±6.37 | 37.34±9.45 | 25.21±2.70 | 19.76±7.71 | 16.33±4.77 |
| | IRM | 92.64±0.76 | 92.11±6.38 | 92.03±5.99 | 90.17±4.96 | 78.69±2.36 | 57.60±8.18 | 38.00±11.27 | 26.70±11.30 | 19.63±9.90 | 15.62±7.10 | 14.39±4.75 |
| | SD | 98.61±0.77 | 98.22±1.03 | 98.06±0.81 | 94.58±1.82 | 81.71±1.18 | 57.68±2.32 | 35.67±2.09 | 22.17±3.30 | 16.69±3.63 | 14.47±3.02 | 13.16±1.97 |
| | GroupDRO | 97.72±0.74 | 98.09±0.30 | 97.35±0.36 | 92.19±1.97 | 74.03±1.71 | 50.29±5.72 | 29.90±8.41 | 18.92±7.36 | 14.60±5.92 | 14.41±5.27 | 12.97±2.84 |
| | RSC | 97.38±0.23 | 96.80±0.66 | 95.89±0.52 | 94.58±0.53 | 84.25±4.63 | 61.37±7.21 | 39.60±7.10 | 25.13±4.38 | 18.29±2.29 | 15.41±1.31 | 12.66±0.80 |
| | Mixup | 98.32±0.55 | 98.56±0.58 | 97.77±0.55 | 96.86±0.78 | 93.16±0.36 | 84.09±2.32 | 69.21±1.87 | 53.80±4.09 | 41.48±2.64 | 33.60±2.21 | 29.09±2.01 |
| | CORAL | 98.14±1.14 | 97.56±1.26 | 97.56±0.71 | 95.26±0.95 | 88.23±1.24 | 68.71±5.42 | 46.41±7.95 | 30.73±6.31 | 23.87±5.71 | 18.37±2.53 | 16.12±1.91 |
| | MMD | 98.93±0.15 | 99.08±0.16 | 98.77±0.13 | 98.24±0.20 | 94.18±0.79 | 78.62±4.37 | 52.75±8.71 | 36.53±11.98 | 24.55±10.22 | 21.41±8.55 | 17.79±6.23 |
| | DANN | 96.20±0.52 | 95.94±0.93 | 94.92±1.35 | 91.17±1.16 | 70.96±2.33 | 43.76±7.20 | 29.14±5.71 | 20.13±4.37 | 16.09±3.26 | 13.60±2.96 | 11.40±1.09 |
| | CDANN | 92.19±4.52 | 91.98±4.12 | 93.08±3.73 | 91.69±1.81 | 77.54±3.79 | 48.85±7.60 | 27.70±8.06 | 16.14±3.74 | 11.56±2.22 | 10.38±1.48 | 10.51±0.32 |
| | MTL | 98.24±0.44 | 97.93±0.47 | 97.48±0.26 | 93.53±1.45 | 78.80±1.97 | 54.69±6.76 | 33.88±7.99 | 21.72±8.27 | 16.72±7.41 | 14.94±6.69 | 13.71±5.05 |
| | SagNet | 97.33±1.39 | 96.80±2.31 | 97.01±1.84 | 94.79±1.54 | 85.67±1.58 | 65.17±3.17 | 39.23±2.00 | 26.00±4.43 | 20.18±5.01 | 15.64±5.17 | 14.75±4.42 |
| | ARM | 97.80±0.45 | 97.93±0.49 | 97.38±0.36 | 94.37±1.63 | 79.27±5.63 | 55.21±11.17 | 32.97±5.77 | 21.04±4.36 | 14.52±2.33 | 12.37±2.65 | 11.01±1.17 |
| | ANDMask | 97.56±1.20 | 97.22±0.71 | 96.62±1.29 | 89.81±5.26 | 81.34±6.81 | 60.80±6.10 | 38.18±2.98 | 27.54±2.85 | 20.13±1.48 | 17.03±0.92 | 14.57±2.71 |
| | SANDMask | 96.78±0.40 | 96.41±0.35 | 96.72±0.29 | 92.69±1.67 | 78.83±6.00 | 57.08±11.28 | 33.20±11.76 | 20.60±2.75 | 13.10±3.87 | 11.56±1.29 | 10.93±0.62 |
| | IGA | 16.25±4.00 | 16.72±3.37 | 16.82±2.80 | 15.49±3.06 | 16.93±3.39 | 16.56±4.04 | 17.77±4.37 | 16.69±3.92 | 15.70±4.01 | 14.49±3.92 | 15.33±2.78 |
| | SelfReg | 98.64±0.36 | 98.30±0.63 | 97.93±0.35 | 96.25±0.30 | 85.38±0.61 | 67.11±8.72 | 44.97±15.83 | 29.56±12.58 | 21.51±8.87 | 17.87±5.58 | 13.05±2.79 |
| | Fish | 98.38±0.42 | 98.27±0.36 | 97.75±0.24 | 94.05±0.42 | 77.99±1.22 | 53.96±1.54 | 33.18±0.82 | 22.27±1.43 | 15.72±1.39 | 13.21±1.16 | 10.32±1.35 |
| | TRM | 97.04±1.77 | 96.04±1.97 | 96.65±1.36 | 92.90±0.52 | 78.83±3.82 | 51.60±0.78 | 28.49±5.00 | 16.43±3.61 | 12.40±1.80 | 10.30±0.32 | 9.54±0.89 |
| | IB-ERM | 97.80±0.40 | 97.54±0.45 | 98.03±0.46 | 96.80±0.85 | 92.69±1.31 | 79.43±5.08 | 61.77±8.12 | 44.13±8.84 | 33.28±9.08 | 26.02±6.83 | 20.81±7.51 |
| | IB-IRM | 97.12±2.51 | 96.96±2.50 | 96.12±2.37 | 94.81±2.48 | 86.06±5.65 | 68.92±9.72 | 54.64±13.29 | 41.09±10.53 | 33.62±8.82 | 27.36±6.62 | 21.41±3.62 |
| | CAD | 98.35±0.53 | 98.30±0.26 | 98.19±0.26 | 97.72±0.28 | 94.08±1.02 | 78.75±3.10 | 58.65±3.98 | 40.72±3.47 | 29.04±3.54 | 23.09±2.44 | 18.50±1.35 |
| | CondCAD | 98.03±1.31 | 97.69±1.18 | 97.43±1.37 | 97.22±1.38 | 93.06±0.35 | 82.86±3.95 | 69.13±6.84 | 51.76±8.80 | 39.78±7.95 | 31.42±5.49 | 24.29±3.88 |
| | Transfer | 40.88±41.00 | 42.27±40.24 | 40.70±40.66 | 38.84±40.37 | 32.26±28.62 | 20.86±14.61 | 14.39±3.25 | 10.69±0.97 | 10.74±0.43 | 9.85±1.43 | 10.82±0.32 |
| | CausIRL-CORAL | 99.08±0.44 | 99.11±0.35 | 98.43±0.39 | 98.06±0.36 | 91.59±2.50 | 72.77±3.76 | 50.10±4.43 | 33.81±3.01 | 24.06±1.42 | 19.31±1.27 | 16.09±1.78 |
| | CausIRL-MMD | 98.66±0.26 | 98.93±0.33 | 98.77±0.10 | 97.67±0.26 | 92.09±0.33 | 78.14±3.35 | 62.87±5.31 | 44.37±5.91 | 34.88±4.74 | 27.54±4.08 | 21.44±3.54 |
| | EQRM | 98.69±0.15 | 98.06±0.72 | 98.06±0.43 | 95.89±0.55 | 82.39±1.16 | 55.71±1.91 | 30.45±1.82 | 19.34±1.65 | 14.18±1.54 | 12.68±1.46 | 11.03±1.49 |
| $\mathcal{D}_5$ | ERM | 98.17±0.59 | 97.98±0.67 | 96.80±1.46 | 91.04±4.87 | 75.16±5.49 | 50.08±2.50 | 30.66±0.95 | 19.42±2.02 | 13.94±2.15 | 12.81±0.57 | 10.95±0.45 |
| | VREx | 96.88±0.81 | 97.54±0.55 | 97.14±1.16 | 96.04±1.43 | 90.04±1.86 | 75.58±3.80 | 55.69±4.35 | 39.28±8.73 | 26.73±6.79 | 20.83±7.05 | 16.75±4.45 |
| | IRM | 85.85±6.55 | 85.85±7.05 | 85.25±6.84 | 80.79±4.00 | 73.53±4.02 | 63.21±2.81 | 49.16±3.43 | 38.73±5.43 | 31.55±7.17 | 24.03±4.79 | 19.63±4.01 |
| | SD | 98.45±0.60 | 98.09±0.93 | 97.75±0.52 | 94.39±2.00 | 81.34±1.58 | 59.01±3.11 | 38.94±4.73 | 25.26±6.07 | 18.76±4.65 | 15.28±3.19 | 13.65±2.01 |
| | GroupDRO | 98.56±0.49 | 98.32±0.32 | 97.80±0.29 | 91.85±2.43 | 73.14±2.77 | 51.83±4.45 | 34.15±4.95 | 22.43±4.21 | 16.43±4.75 | 15.02±4.87 | 12.87±2.88 |
| | RSC | 97.38±0.23 | 96.80±0.66 | 95.89±0.52 | 94.58±0.53 | 84.25±4.63 | 61.37±7.21 | 39.60±7.10 | 25.13±4.38 | 18.29±2.29 | 15.41±1.31 | 12.66±0.80 |
| | Mixup | 98.32±0.55 | 98.56±0.58 | 97.77±0.55 | 96.86±0.78 | 93.16±0.36 | 84.09±2.32 | 69.21±1.87 | 53.80±4.09 | 41.48±2.64 | 33.60±2.21 | 29.09±2.01 |
| | CORAL | 98.27±1.18 | 97.69±1.32 | 97.82±0.81 | 95.52±0.39 | 86.98±2.49 | 71.28±3.71 | 48.27±6.58 | 32.29±5.18 | 25.26±4.24 | 18.55±2.49 | 16.82±1.41 |
| | MMD | 98.24±1.10 | 98.35±1.01 | 97.77±1.54 | 96.47±0.76 | 93.35±1.18 | 81.92±2.14 | 61.14±2.92 | 45.83±5.22 | 32.73±5.11 | 26.99±4.92 | 21.38±4.72 |
| | DANN | 92.66±2.60 | 92.79±2.37 | 92.22±2.05 | 86.77±3.15 | 67.74±3.85 | 49.69±4.09 | 34.12±3.84 | 22.41±1.99 | 15.99±1.88 | 13.26±1.90 | 11.08±1.31 |
| | CDANN | 92.06±4.48 | 90.70±3.52 | 92.14±3.35 | 89.83±3.60 | 74.84±7.47 | 50.71±5.64 | 32.73±4.93 | 20.99±4.91 | 15.80±4.19 | 12.29±2.67 | 11.37±1.38 |
| | MTL | 98.22±0.43 | 97.85±0.36 | 97.62±0.43 | 92.95±1.60 | 78.35±2.25 | 57.84±4.65 | 37.84±5.52 | 24.27±6.58 | 18.13±6.59 | 15.72±6.13 | 14.65±4.46 |
| | SagNet | 96.96±1.12 | 96.33±2.00 | 96.62±1.62 | 94.18±1.11 | 85.30±1.85 | 67.32±2.31 | 43.11±4.75 | 27.44±5.74 | 19.97±4.27 | 15.57±4.43 | 14.31±3.73 |
| | ARM | 98.03±0.23 | 98.32±0.10 | 97.54±0.38 | 93.27±2.45 | 77.59±7.27 | 57.52±9.25 | 35.12±4.05 | 23.64±2.78 | 18.45±3.35 | 14.78±3.19 | 12.89±2.22 |
| | ANDMask | 97.56±1.20 | 97.22±0.71 | 96.62±1.29 | 89.81±5.26 | 81.34±6.81 | 60.80±6.10 | 38.18±2.98 | 27.54±2.85 | 20.13±1.48 | 17.03±0.92 | 14.57±2.71 |
| | SANDMask | 96.78±0.40 | 96.41±0.35 | 96.72±0.29 | 92.69±1.67 | 78.83±6.00 | 57.08±11.28 | 33.20±11.76 | 20.60±2.75 | 13.10±3.87 | 11.56±1.29 | 10.93±0.62 |
| | IGA | 17.03±2.89 | 16.04±4.34 | 16.09±3.80 | 15.41±3.17 | 16.33±4.23 | 17.30±3.00 | 17.69±4.47 | 16.88±3.67 | 15.85±3.80 | 14.86±3.47 | 14.47±3.83 |
| | SelfReg | 98.69±0.35 | 98.82±0.13 | 98.11±0.29 | 95.65±0.75 | 84.38±1.82 | 70.44±4.02 | 52.88±4.74 | 36.79±2.63 | 26.21±2.36 | 20.15±2.44 | 15.28±0.42 |
| | Fish | 97.80±0.48 | 98.11±0.55 | 97.01±0.69 | 91.59±3.53 | 75.68±3.21 | 54.48±0.82 | 32.36±1.84 | 21.59±1.43 | 16.59±0.99 | 13.26±1.16 | 10.90±0.80 |
| | TRM | 97.12±1.64 | 96.54±2.17 | 96.65±1.36 | 93.37±0.61 | 77.67±4.63 | 55.24±7.28 | 34.04±4.00 | 21.83±4.60 | 16.67±3.34 | 14.28±4.17 | 12.47±2.98 |
| | IB-ERM | 97.80±0.40 | 97.54±0.45 | 98.03±0.46 | 96.80±0.85 | 92.69±1.31 | 79.43±5.08 | 61.77±8.12 | 44.13±8.84 | 33.28±9.08 | 26.02±6.83 | 20.81±7.51 |
| | IB-IRM | 83.15±8.42 | 83.44±8.38 | 83.78±7.74 | 83.02±6.95 | 81.13±7.23 | 76.55±6.02 | 69.10±6.48 | 56.60±4.25 | 46.54±5.43 | 38.39±4.04 | 30.08±4.37 |
| | CAD | 95.47±1.02 | 94.97±0.28 | 94.47±1.48 | 93.82±0.39 | 91.77±0.43 | 87.53±1.04 | 78.01±2.12 | 69.00±1.93 | 56.47±0.74 | 45.47±2.66 | 36.01±1.35 |
| | CondCAD | 95.26±0.77 | 94.60±1.04 | 94.71±0.58 | 93.82±1.55 | 91.25±1.36 | 86.08±1.47 | 78.49±1.33 | 65.20±1.22 | 52.59±1.37 | 44.13±4.37 | 35.46±4.59 |
| | Transfer | 40.59±41.24 | 39.36±42.32 | 39.88±41.23 | 39.20±40.10 | 30.95±29.54 | 21.28±14.31 | 13.23±3.99 | 11.01±0.76 | 10.38±0.80 | 9.83±0.78 | 10.98±0.44 |
| | CausIRL-CORAL | 99.08±0.44 | 99.11±0.35 | 98.43±0.39 | 98.06±0.36 | 91.59±2.50 | 72.77±3.76 | 50.10±4.43 | 33.81±3.01 | 24.06±1.42 | 19.31±1.27 | 16.09±1.78 |
| | CausIRL-MMD | 98.48±0.46 | 98.35±0.51 | 98.32±0.45 | 97.17±0.56 | 91.61±0.67 | 79.11±2.25 | 61.84±4.64 | 44.05±5.59 | 33.67±4.15 | 26.15±4.02 | 21.07±2.11 |
| | EQRM | 95.62±2.01 | 95.81±1.93 | 95.07±2.48 | 91.88±3.31 | 79.48±2.49 | 61.03±1.84 | 40.72±2.51 | 27.52±1.46 | 20.10±0.63 | 17.35±0.65 | 15.38±0.48 |
| $\mathcal{D}_6$ | ERM | 98.17±0.59 | 97.98±0.67 | 96.80±1.46 | 91.04±4.87 | 75.16±5.49 | 50.08±2.50 | 30.66±0.95 | 19.42±2.02 | 13.94±2.15 | 12.81±0.57 | 10.95±0.45 |
| | VREx | 94.31±3.02 | 94.81±3.40 | 93.82±3.80 | 92.35±3.82 | 86.90±4.35 | 73.06±4.44 | 57.86±3.49 | 45.62±5.59 | 33.39±6.91 | 27.44±7.27 | 22.06±6.55 |
| | IRM | 85.85±6.55 | 85.85±7.05 | 85.25±6.84 | 80.79±4.00 | 73.53±4.02 | 63.21±2.81 | 49.16±3.43 | 38.73±5.43 | 31.55±7.17 | 24.03±4.79 | 19.63±4.01 |
| | SD | 98.45±0.60 | 98.09±0.93 | 97.75±0.52 | 94.39±2.00 | 81.34±1.58 | 59.01±3.11 | 38.94±4.73 | 25.26±6.07 | 18.76±4.65 | 15.28±3.19 | 13.65±2.01 |
| | GroupDRO | 98.56±0.49 | 98.32±0.32 | 97.80±0.29 | 91.85±2.43 | 73.14±2.77 | 51.83±4.45 | 34.15±4.95 | 22.43±4.21 | 16.43±4.75 | 15.02±4.87 | 12.87±2.88 |
| | RSC | 97.46±0.13 | 97.20±0.69 | 95.96±0.63 | 93.21±2.43 | 82.15±7.54 | 59.98±9.14 | 41.19±4.84 | 26.86±2.01 | 18.29±2.29 | 15.02±1.57 | 12.40±0.54 |
| | Mixup | 98.32±0.55 | 98.56±0.58 | 97.77±0.55 | 96.86±0.78 | 93.16±0.36 | 84.09±2.32 | 69.21±1.87 | 53.80±4.09 | 41.48±2.64 | 33.60±2.21 | 29.09±2.01 |
| | CORAL | 98.27±1.18 | 97.69±1.32 | 97.82±0.81 | 95.52±0.39 | 86.98±2.49 | 71.28±3.71 | 48.27±6.58 | 32.29±5.18 | 25.26±4.24 | 18.55±2.49 | 16.82±1.41 |
| | MMD | 98.90±0.19 | 99.03±0.13 | 98.74±0.17 | 97.88±0.22 | 93.58±1.23 | 81.05±2.63 | 62.50±1.57 | 47.01±4.30 | 35.43±2.48 | 28.46±3.41 | 23.77±1.70 |
| | DANN | 94.31±1.87 | 93.47±3.02 | 92.66±2.85 | 85.38±5.16 | 66.93±3.63 | 48.98±3.21 | 34.80±3.98 | 23.74±2.84 | 16.21±3.43 | 14.68±2.73 | 11.66±1.69 |
| | CDANN | 91.51±3.94 | 90.80±3.64 | 91.48±2.66 | 87.00±2.08 | 67.64±10.48 | 46.93±9.00 | 33.12±4.42 | 22.64±3.06 | 16.77±3.37 | 13.34±1.52 | 12.05±1.03 |
| | MTL | 98.22±0.43 | 97.85±0.36 | 97.62±0.43 | 92.95±1.60 | 78.35±2.25 | 57.84±4.65 | 37.84±5.52 | 24.27±6.58 | 18.13±6.59 | 15.72±6.13 | 14.65±4.46 |
| | SagNet | 97.80±0.34 | 97.72±0.39 | 97.64±0.46 | 94.18±1.11 | 82.42±2.25 | 65.49±3.53 | 45.26±2.31 | 31.53±1.83 | 24.45±2.36 | 19.63±1.46 | 17.64±0.99 |
| | ARM | 97.59±0.81 | 97.25±1.50 | 96.54±1.78 | 89.44±6.35 | 72.72±10.76 | 56.05±10.49 | 35.51±3.70 | 22.93±3.71 | 18.61±3.21 | 15.12±2.72 | 12.60±2.57 |
| | ANDMask | 97.56±1.20 | 97.22±0.71 | 96.62±1.29 | 89.81±5.26 | 81.34±6.81 | 60.80±6.10 | 38.18±2.98 | 27.54±2.85 | 20.13±1.48 | 17.03±0.92 | 14.57±2.71 |
| | SANDMask | 97.04±0.62 | 96.54±0.39 | 96.57±0.38 | 90.46±2.79 | 76.44±8.90 | 56.47±12.05 | 35.27±9.19 | 21.38±6.94 | 13.97±2.96 | 11.19±1.56 | 10.85±0.67 |
| | IGA | 16.19±4.07 | 16.72±3.37 | 16.01±3.91 | 15.20±3.47 | 16.14±4.49 | 16.51±4.12 | 18.45±3.54 | 16.59±4.06 | 16.40±3.07 | 15.02±3.29 | 15.04±3.12 |
| | SelfReg | 98.69±0.35 | 98.82±0.13 | 98.11±0.29 | 95.65±0.75 | 84.38±1.82 | 70.44±4.02 | 52.88±4.74 | 36.79±2.63 | 26.21±2.36 | 20.15±2.44 | 15.28±0.42 |
| | Fish | 96.51±2.27 | 96.12±3.32 | 94.86±3.99 | 90.15±5.56 | 74.61±4.66 | 53.98±1.50 | 34.25±1.06 | 22.22±1.41 | 15.88±1.24 | 12.58±1.46 | 10.88±0.82 |
| | TRM | 97.35±1.80 | 96.75±2.32 | 96.86±1.50 | 92.16±1.37 | 76.44±5.61 | 55.01±7.53 | 35.38±2.25 | 23.74±2.80 | 18.27±2.82 | 15.72±3.88 | 13.36±3.08 |
| | IB-ERM | 93.66±5.47 | 92.95±6.08 | 93.37±6.15 | 91.46±7.79 | 89.20±5.90 | 78.85±5.73 | 64.75±4.98 | 49.55±5.16 | 38.73±5.17 | 30.92±4.64 | 25.76±5.36 |
| | IB-IRM | 83.15±8.42 | 83.44±8.38 | 83.78±7.74 | 83.02±6.95 | 81.13±7.23 | 76.55±6.02 | 69.10±6.48 | 56.60±4.25 | 46.54±5.43 | 38.39±4.04 | 30.08±4.37 |
| | CAD | 93.29±1.21 | 93.42±1.00 | 92.03±0.51 | 91.82±1.25 | 90.64±1.20 | 85.35±1.36 | 79.09±0.88 | 69.10±1.95 | 57.76±0.82 | 48.61±0.89 | 38.89±2.25 |
| | CondCAD | 95.26±0.77 | 94.60±1.04 | 94.71±0.58 | 93.82±1.55 | 91.25±1.36 | 86.08±1.47 | 78.49±1.33 | 65.20±1.22 | 52.59±1.37 | 44.13±4.37 | 35.46±4.59 |
| | Transfer | 39.07±42.26 | 40.36±41.57 | 39.31±41.77 | 37.45±39.95 | 30.95±28.46 | 20.23±14.16 | 13.50±3.00 | 11.95±1.94 | 11.69±1.25 | 11.06±1.67 | 11.32±0.99 |
| | CausIRL-CORAL | 99.08±0.44 | 99.11±0.35 | 98.43±0.39 | 98.06±0.36 | 91.59±2.50 | 72.77±3.76 | 50.10±4.43 | 33.81±3.01 | 24.06±1.42 | 19.31±1.27 | 16.09±1.78 |
| | CausIRL-MMD | 98.40±0.45 | 98.35±0.51 | 98.45±0.47 | 97.12±0.52 | 91.82±0.65 | 78.98±2.20 | 63.55±4.40 | 45.89±4.24 | 35.64±3.71 | 27.52±4.12 | 21.91±2.92 |
| | EQRM | 95.62±2.01 | 95.81±1.93 | 95.07±2.48 | 91.88±3.31 | 79.48±2.49 | 61.03±1.84 | 40.72±2.51 | 27.52±1.46 | 20.10±0.63 | 17.35±0.65 | 15.38±0.48 |
| $\mathcal{D}_7$ | ERM | 97.88±0.53 | 97.69±0.80 | 96.49±1.29 | 89.65±4.02 | 71.67±2.90 | 47.90±0.58 | 29.74±1.90 | 20.05±1.35 | 15.28±0.75 | 12.97±0.36 | 10.40±0.87 |
| | VREx | 94.10±2.93 | 94.44±3.22 | 93.32±3.61 | 92.35±3.82 | 85.95±4.43 | 72.43±4.89 | 57.57±3.87 | 46.96±3.74 | 35.80±3.70 | 30.42±3.25 | 24.87±3.34 |
| | IRM | 78.93±3.69 | 79.19±3.09 | 78.07±4.40 | 76.76±4.73 | 71.59±4.75 | 62.21±3.84 | 48.93±3.76 | 40.75±2.58 | 35.04±2.95 | 26.73±1.56 | 22.41±1.06 |
| | SD | 98.66±0.32 | 98.72±0.04 | 98.17±0.10 | 94.37±2.01 | 79.45±3.18 | 58.52±3.80 | 38.44±5.33 | 26.60±4.49 | 19.71±3.34 | 16.95±0.97 | 14.15±1.35 |
| | GroupDRO | 98.56±0.49 | 98.32±0.32 | 97.80±0.29 | 91.85±2.43 | 73.14±2.77 | 51.83±4.45 | 34.15±4.95 | 22.43±4.21 | 16.43±4.75 | 15.02±4.87 | 12.87±2.88 |
| | RSC | 97.46±0.13 | 97.20±0.69 | 95.96±0.63 | 93.21±2.43 | 82.15±7.54 | 59.98±9.14 | 41.19±4.84 | 26.86±2.01 | 18.29±2.29 | 15.02±1.57 | 12.40±0.54 |
| | Mixup | 98.51±0.78 | 98.45±0.58 | 97.72±0.50 | 96.59±0.81 | 92.03±1.33 | 81.97±4.79 | 68.06±3.02 | 54.22±3.65 | 41.88±2.47 | 34.51±2.22 | 29.56±1.73 |
| | CORAL | 98.22±1.15 | 97.72±1.34 | 97.80±0.80 | 94.63±1.09 | 84.67±5.74 | 66.95±9.14 | 47.12±7.44 | 32.99±4.58 | 24.79±4.40 | 20.44±1.62 | 16.69±1.59 |
| | MMD | 98.90±0.19 | 99.03±0.13 | 98.74±0.17 | 97.88±0.22 | 93.58±1.23 | 81.05±2.63 | 62.50±1.57 | 47.01±4.30 | 35.43±2.48 | 28.46±3.41 | 23.77±1.70 |
| | DANN | 92.87±4.43 | 92.03±5.74 | 89.86±6.06 | 80.63±10.98 | 61.43±7.39 | 45.10±1.61 | 31.79±3.44 | 24.97±1.85 | 19.73±1.29 | 16.01±1.32 | 14.02±0.80 |
| | CDANN | 90.07±6.74 | 89.78±5.80 | 89.02±6.87 | 81.18±8.01 | 59.62±6.25 | 41.33±3.25 | 30.97±3.40 | 24.66±2.61 | 19.58±2.20 | 15.07±1.13 | 13.57±0.26 |
| | MTL | 97.54±1.00 | 97.51±0.91 | 96.41±1.41 | 88.89±4.73 | 71.02±9.16 | 51.21±10.12 | 35.64±6.64 | 26.15±5.03 | 21.04±4.86 | 17.71±5.22 | 16.90±3.12 |
| | SagNet | 97.96±0.13 | 98.03±0.22 | 97.96±0.34 | 94.42±1.33 | 82.65±2.41 | 64.20±4.26 | 43.61±3.92 | 31.63±1.69 | 25.97±1.44 | 20.96±1.16 | 18.55±1.26 |
| | ARM | 98.11±0.17 | 98.06±0.36 | 97.14±0.93 | 89.60±6.14 | 71.62±11.96 | 53.64±13.00 | 33.60±5.72 | 23.95±2.38 | 19.23±2.77 | 16.40±1.08 | 13.42±1.66 |
| | ANDMask | 97.56±1.20 | 97.22±0.71 | 96.62±1.29 | 89.81±5.26 | 81.34±6.81 | 60.80±6.10 | 38.18±2.98 | 27.54±2.85 | 20.13±1.48 | 17.03±0.92 | 14.57±2.71 |
| | SANDMask | 96.99±0.56 | 97.09±1.01 | 97.04±0.51 | 91.06±2.12 | 76.15±9.27 | 54.53±14.58 | 34.25±10.43 | 22.27±6.13 | 14.60±1.62 | 12.47±1.43 | 11.32±0.67 |
| | IGA | 16.46±3.70 | 16.48±3.71 | 15.99±3.94 | 15.20±3.47 | 16.51±3.98 | 17.01±3.41 | 17.69±4.47 | 16.98±3.53 | 15.67±4.04 | 14.96±3.35 | 15.12±3.02 |
| | SelfReg | 98.69±0.35 | 98.82±0.13 | 98.11±0.29 | 95.65±0.75 | 84.38±1.82 | 70.44±4.02 | 52.88±4.74 | 36.79±2.63 | 26.21±2.36 | 20.15±2.44 | 15.28±0.42 |
| | Fish | 98.35±0.42 | 98.24±0.36 | 97.51±0.27 | 92.77±1.46 | 73.69±5.26 | 52.10±2.54 | 32.15±1.80 | 23.56±0.48 | 16.98±1.98 | 14.36±0.85 | 11.19±1.77 |
| | TRM | 98.38±0.36 | 97.93±0.67 | 97.56±0.51 | 91.04±1.90 | 70.60±3.01 | 47.09±4.72 | 33.07±3.96 | 23.95±2.63 | 19.23±1.63 | 16.77±2.58 | 14.28±1.85 |
| | IB-ERM | 94.10±5.80 | 93.45±6.44 | 93.63±6.34 | 91.77±7.98 | 88.29±5.57 | 77.04±6.45 | 63.44±6.03 | 49.82±4.82 | 39.12±4.69 | 31.84±3.36 | 26.94±3.75 |
| | IB-IRM | 83.15±8.42 | 83.44±8.38 | 83.78±7.74 | 83.02±6.95 | 81.13±7.23 | 76.55±6.02 | 69.10±6.48 | 56.60±4.25 | 46.54±5.43 | 38.39±4.04 | 30.08±4.37 |
| | CAD | 94.84±0.99 | 94.44±0.89 | 93.47±2.07 | 92.69±1.71 | 91.27±0.74 | 87.50±1.03 | 78.14±2.19 | 69.21±1.84 | 56.79±1.13 | 47.62±0.56 | 37.29±0.52 |
| | CondCAD | 94.60±0.43 | 93.89±0.04 | 94.34±0.11 | 93.03±1.16 | 90.64±0.97 | 85.40±0.97 | 77.91±1.93 | 66.56±1.13 | 55.01±1.28 | 46.88±0.58 | 38.52±0.88 |
| | Transfer | 39.65±41.87 | 40.41±41.54 | 39.07±41.93 | 37.60±39.84 | 30.19±28.97 | 19.79±14.48 | 14.47±4.01 | 12.97±0.84 | 12.19±1.30 | 10.93±1.62 | 11.03±1.15 |
| | CausIRL-CORAL | 99.08±0.44 | 99.11±0.35 | 98.43±0.39 | 98.06±0.36 | 91.59±2.50 | 72.77±3.76 | 50.10±4.43 | 33.81±3.01 | 24.06±1.42 | 19.31±1.27 | 16.09±1.78 |
| | CausIRL-MMD | 98.40±0.45 | 98.35±0.51 | 98.45±0.47 | 97.12±0.52 | 91.82±0.65 | 78.98±2.20 | 63.55±4.40 | 45.89±4.24 | 35.64±3.71 | 27.52±4.12 | 21.91±2.92 |
| | EQRM | 95.65±2.04 | 96.10±2.17 | 95.02±2.44 | 91.04±2.71 | 75.50±4.44 | 57.26±6.02 | 40.59±2.69 | 28.20±0.52 | 19.63±1.09 | 16.56±1.22 | 15.51±0.33 |

Table 3: Average accuracy of top-3 models of each algorithm at each shift degree of NOISYMNIST (Simple CNN). This table shows the results on shift degrees from $\mathcal{D}_4$ to $\mathcal{D}_7$.

| | Algorithm | $\mathcal{D}_0$ | $\mathcal{D}_1$ | $\mathcal{D}_2$ | $\mathcal{D}_3$ | $\mathcal{D}_4$ | $\mathcal{D}_5$ | $\mathcal{D}_6$ | $\mathcal{D}_7$ | $\mathcal{D}_8$ | $\mathcal{D}_9$ | $\mathcal{D}_{10}$ |
|---|---|---|---|---|---|---|---|---|---|---|---|---|
| $\mathcal{D}_8$ | ERM | 98.14±0.89 | 97.69±0.80 | 96.31±1.12 | 87.00±2.88 | 68.87±1.81 | 43.19±6.37 | 27.31±3.85 | 18.82±2.57 | 15.38±0.64 | 11.98±1.15 | 10.90±1.52 |
| | VREx | 83.83±14.04 | 82.94±15.65 | 82.21±15.52 | 81.71±15.40 | 77.31±13.83 | 66.43±11.98 | 54.87±7.57 | 46.62±4.21 | 37.42±1.95 | 31.32±2.20 | 26.47±2.67 |
| | IRM | 78.67±3.40 | 78.83±2.63 | 77.67±3.92 | 75.71±3.33 | 69.92±2.38 | 60.46±1.89 | 48.48±3.38 | 40.46±2.39 | 35.35±2.82 | 27.31±1.35 | 23.79±1.13 |
| | SD | 98.85±0.16 | 98.69±0.07 | 98.06±0.10 | 92.06±4.21 | 76.81±6.69 | 53.80±10.43 | 36.84±7.37 | 25.81±5.40 | 20.68±2.02 | 17.19±0.74 | 14.31±1.16 |
| | GroupDRO | 98.72±0.50 | 98.43±0.40 | 97.51±0.66 | 89.31±2.56 | 67.79±5.54 | 47.04±10.04 | 30.79±9.11 | 21.83±4.84 | 17.16±4.06 | 14.99±4.89 | 13.18±2.59 |
| | RSC | 97.72±0.29 | 97.33±0.79 | 96.41±1.26 | 93.84±1.55 | 82.26±7.39 | 58.41±11.33 | 39.10±7.80 | 25.55±3.80 | 19.05±1.54 | 15.17±1.45 | 12.40±0.54 |
| | Mixup | 98.77±0.81 | 98.40±0.51 | 97.59±0.45 | 95.73±0.46 | 88.26±4.60 | 76.73±8.28 | 64.70±5.75 | 52.49±4.88 | 42.43±1.97 | 35.25±1.23 | 30.24±1.25 |
| | CORAL | 98.09±1.04 | 97.64±1.28 | 97.67±0.69 | 94.65±1.11 | 83.07±5.14 | 66.14±8.94 | 47.48±7.25 | 32.15±5.24 | 25.97±3.53 | 19.63±2.77 | 15.80±1.62 |
| | MMD | 98.90±0.19 | 99.03±0.13 | 98.74±0.17 | 97.88±0.22 | 93.58±1.23 | 81.05±2.63 | 62.50±1.57 | 47.01±4.30 | 35.43±2.48 | 28.46±3.41 | 23.77±1.70 |
| | DANN | 83.81±12.08 | 80.95±14.73 | 79.35±13.78 | 63.16±25.24 | 49.50±19.80 | 38.02±11.16 | 29.43±6.70 | 23.72±3.32 | 19.94±1.00 | 16.38±0.94 | 14.88±1.78 |
| | CDANN | 83.91±6.88 | 82.42±7.03 | 79.95±10.12 | 63.42±21.12 | 45.26±18.36 | 34.88±11.83 | 28.69±6.62 | 23.58±4.13 | 20.05±1.54 | 15.30±0.81 | 14.07±0.71 |
| | MTL | 97.90±1.19 | 97.56±0.97 | 96.72±1.66 | 89.47±4.65 | 67.77±9.83 | 45.10±13.53 | 31.45±10.02 | 24.61±6.49 | 21.46±4.38 | 18.97±3.88 | 16.95±3.06 |
| | SagNet | 97.96±0.13 | 98.03±0.22 | 97.96±0.34 | 94.42±1.33 | 82.65±2.41 | 64.20±4.26 | 43.61±3.92 | 31.63±1.69 | 25.97±1.44 | 20.96±1.16 | 18.55±1.26 |
| | ARM | 98.11±0.17 | 98.06±0.36 | 97.14±0.93 | 89.60±6.14 | 71.62±11.96 | 53.64±13.00 | 33.60±5.72 | 23.95±2.38 | 19.23±2.77 | 16.40±1.08 | 13.42±1.66 |
| | ANDMask | 97.56±1.20 | 97.22±0.71 | 96.62±1.29 | 89.81±5.26 | 81.34±6.81 | 60.80±6.10 | 38.18±2.98 | 27.54±2.85 | 20.13±1.48 | 17.03±0.92 | 14.57±2.71 |
| | SANDMask | 97.41±0.91 | 97.43±1.04 | 97.06±0.52 | 92.06±2.24 | 74.29±9.35 | 48.19±15.80 | 28.90±12.74 | 21.51±6.64 | 15.12±2.14 | 12.87±0.88 | 11.03±0.58 |
| | IGA | 16.19±4.07 | 16.72±3.37 | 16.01±3.91 | 15.20±3.47 | 16.14±4.49 | 16.51±4.12 | 18.45±3.54 | 16.59±4.06 | 16.40±3.07 | 15.02±3.29 | 15.04±3.12 |
| | SelfReg | 98.69±0.35 | 98.82±0.13 | 98.11±0.29 | 95.65±0.75 | 84.38±1.82 | 70.44±4.02 | 52.88±4.74 | 36.79±2.63 | 26.21±2.36 | 20.15±2.44 | 15.28±0.42 |
| | Fish | 97.77±0.46 | 98.17±0.45 | 97.20±0.20 | 91.43±1.53 | 70.26±4.36 | 49.29±4.88 | 32.18±1.80 | 23.24±0.90 | 18.29±0.58 | 14.07±1.25 | 12.08±0.58 |
| | TRM | 98.38±0.36 | 97.93±0.67 | 97.56±0.51 | 91.04±1.90 | 70.60±3.01 | 47.09±4.72 | 33.07±3.96 | 23.95±2.63 | 19.23±1.63 | 16.77±2.58 | 14.28±1.85 |
| | IB-ERM | 94.10±5.80 | 93.45±6.44 | 93.63±6.34 | 91.77±7.98 | 88.29±5.57 | 77.04±6.45 | 63.44±6.03 | 49.82±4.82 | 39.12±4.69 | 31.84±3.36 | 26.94±3.75 |
| | IB-IRM | 83.15±8.42 | 83.44±8.38 | 83.78±7.74 | 83.02±6.95 | 81.13±7.23 | 76.55±6.02 | 69.10±6.48 | 56.60±4.25 | 46.54±5.43 | 38.39±4.04 | 30.08±4.37 |
| | CAD | 93.29±1.21 | 93.42±1.00 | 92.03±0.51 | 91.82±1.25 | 90.64±1.20 | 86.53±2.35 | 79.09±0.88 | 69.10±1.95 | 57.76±0.82 | 48.61±0.89 | 38.89±2.25 |
| | CondCAD | 89.70±6.50 | 89.54±6.17 | 88.55±8.30 | 87.79±6.39 | 85.46±6.59 | 80.45±6.27 | 74.03±3.86 | 64.83±2.57 | 55.63±2.10 | 47.80±1.85 | 39.12±1.71 |
| | Transfer | 10.46±1.58 | 10.72±1.04 | 9.49±1.16 | 9.70±0.49 | 9.72±0.55 | 9.85±0.74 | 11.61±0.97 | 12.84±0.96 | 12.71±0.58 | 12.03±0.45 | 11.61±1.07 |
| | CausIRL-CORAL | 99.11±0.41 | 99.08±0.39 | 98.43±0.39 | 97.38±0.94 | 88.78±3.97 | 70.23±5.03 | 48.45±5.92 | 33.33±3.28 | 24.82±0.64 | 20.28±0.17 | 16.77±0.96 |
| | CausIRL-MMD | 98.40±0.45 | 98.43±0.40 | 98.40±0.55 | 97.09±0.56 | 91.51±1.08 | 78.20±3.23 | 62.92±5.24 | 45.81±4.31 | 35.69±3.64 | 27.23±4.51 | 21.91±2.92 |
| | EQRM | 96.20±2.26 | 96.10±2.15 | 95.31±2.63 | 91.40±3.15 | 76.57±4.03 | 59.30±4.00 | 38.50±2.32 | 26.23±1.85 | 20.23±0.55 | 16.85±0.43 | 14.99±0.30 |
| $\mathcal{D}_9$ | ERM | 97.88±0.53 | 97.69±0.80 | 96.49±1.29 | 89.65±4.02 | 71.67±2.90 | 47.90±0.58 | 29.74±1.90 | 20.05±1.35 | 15.28±0.75 | 12.97±0.36 | 10.40±0.87 |
| | VREx | 83.83±14.04 | 82.94±15.65 | 82.21±15.52 | 81.71±15.40 | 77.31±13.83 | 66.43±11.98 | 54.87±7.57 | 46.62±4.21 | 37.42±1.95 | 31.32±2.20 | 26.47±2.67 |
| | IRM | 75.45±7.84 | 75.00±7.83 | 74.37±8.33 | 70.52±10.20 | 63.81±10.18 | 55.71±8.60 | 46.91±5.60 | 39.20±4.16 | 34.17±4.21 | 27.59±1.01 | 24.58±0.67 |
| | SD | 98.85±0.16 | 98.69±0.07 | 98.06±0.10 | 92.06±4.21 | 76.81±6.69 | 53.80±10.43 | 36.84±7.37 | 25.81±5.40 | 20.68±2.02 | 17.19±0.74 | 14.31±1.16 |
| | GroupDRO | 72.06±37.56 | 72.38±36.56 | 72.04±36.65 | 65.75±35.25 | 51.28±28.69 | 39.70±20.06 | 28.62±12.04 | 20.28±6.68 | 16.95±4.25 | 15.59±4.43 | 13.31±2.49 |
| | RSC | 97.43±0.24 | 97.14±0.52 | 96.41±0.60 | 91.06±4.88 | 71.99±16.71 | 49.69±15.90 | 31.97±9.53 | 22.35±4.76 | 17.92±2.39 | 15.46±1.26 | 13.05±0.39 |
| | Mixup | 98.77±0.81 | 98.40±0.51 | 97.59±0.45 | 95.73±0.46 | 88.26±4.60 | 76.73±8.28 | 64.70±5.75 | 52.49±4.88 | 42.43±1.97 | 35.25±1.23 | 30.24±1.25 |
| | CORAL | 98.22±1.15 | 97.72±1.34 | 97.80±0.80 | 94.63±1.09 | 84.67±5.74 | 66.95±9.14 | 47.12±7.44 | 32.99±4.58 | 24.79±4.40 | 20.44±1.62 | 16.69±1.59 |
| | MMD | 98.90±0.19 | 99.03±0.13 | 98.74±0.17 | 97.88±0.22 | 93.58±1.23 | 81.05±2.63 | 62.50±1.57 | 47.01±4.30 | 35.43±2.48 | 28.46±3.41 | 23.77±1.70 |
| | DANN | 85.30±8.24 | 84.91±8.74 | 79.72±12.45 | 62.13±25.68 | 39.94±22.59 | 28.83±13.33 | 22.88±7.12 | 18.66±3.97 | 17.40±2.17 | 17.11±0.41 | 13.16±0.27 |
| | CDANN | 84.83±3.31 | 86.01±3.78 | 83.10±2.72 | 65.67±3.04 | 40.25±7.54 | 26.65±9.93 | 22.33±7.56 | 20.28±4.12 | 17.22±2.45 | 16.35±0.22 | 13.47±0.69 |
| | MTL | 97.51±1.00 | 97.35±0.75 | 96.36±1.38 | 88.13±5.04 | 69.00±9.32 | 47.04±11.88 | 32.86±8.58 | 25.13±5.95 | 21.46±4.38 | 19.10±3.76 | 16.93±3.09 |
| | SagNet | 97.96±0.13 | 98.03±0.22 | 97.96±0.34 | 94.42±1.33 | 82.65±2.41 | 64.20±4.26 | 43.61±3.92 | 31.63±1.69 | 25.97±1.44 | 20.96±1.16 | 18.55±1.26 |
| | ARM | 98.45±0.46 | 98.30±0.10 | 97.01±1.11 | 88.84±7.14 | 70.41±13.36 | 53.20±13.51 | 33.12±6.29 | 23.53±2.91 | 18.79±3.06 | 16.59±0.90 | 13.84±1.35 |
| | ANDMask | 97.56±1.20 | 97.22±0.71 | 96.62±1.29 | 89.81±5.26 | 81.34±6.81 | 60.80±6.10 | 38.18±2.98 | 27.54±2.85 | 20.13±1.48 | 17.03±0.92 | 14.57±2.71 |
| | SANDMask | 68.34±40.38 | 68.95±39.98 | 68.45±40.40 | 64.78±37.75 | 54.74±30.67 | 39.73±23.45 | 27.59±13.89 | 20.75±7.28 | 14.70±2.39 | 13.39±0.16 | 11.37±0.71 |
| | IGA | 15.64±4.85 | 15.46±5.15 | 15.28±4.93 | 15.28±3.35 | 15.83±4.93 | 16.14±4.63 | 16.90±5.49 | 16.35±4.37 | 16.19±3.34 | 15.78±2.50 | 15.54±2.55 |
| | SelfReg | 98.69±0.35 | 98.82±0.13 | 98.11±0.29 | 95.65±0.75 | 84.38±1.82 | 70.44±4.02 | 52.88±4.74 | 36.79±2.63 | 26.21±2.36 | 20.15±2.44 | 15.28±0.42 |
| | Fish | 98.56±0.21 | 98.43±0.26 | 97.46±0.16 | 91.09±2.33 | 65.38±8.72 | 42.87±9.93 | 27.33±5.05 | 20.89±3.45 | 17.51±1.09 | 15.62±0.63 | 13.63±1.04 |
| | TRM | 98.38±0.36 | 97.93±0.67 | 97.56±0.51 | 91.04±1.90 | 70.60±3.01 | 47.09±4.72 | 33.07±3.96 | 23.95±2.63 | 19.23±1.63 | 16.77±2.58 | 14.28±1.85 |
| | IB-ERM | 93.32±5.25 | 92.51±5.81 | 92.35±5.60 | 90.09±7.23 | 87.13±5.59 | 76.65±6.73 | 62.50±7.00 | 49.21±5.60 | 39.12±4.69 | 32.29±2.75 | 26.68±4.11 |
| | IB-IRM | 83.15±8.42 | 83.44±8.38 | 83.78±7.74 | 83.02±6.95 | 81.13±7.23 | 76.55±6.02 | 69.10±6.48 | 56.60±4.25 | 46.54±5.43 | 38.39±4.04 | 30.08±4.37 |
| | CAD | 92.87±1.09 | 91.80±1.42 | 92.06±0.55 | 91.01±0.58 | 89.33±0.87 | 84.41±2.33 | 76.49±3.06 | 66.72±1.68 | 57.68±0.93 | 49.42±0.89 | 40.75±1.97 |
| | CondCAD | 89.70±6.50 | 89.54±6.17 | 88.55±8.30 | 87.79±6.39 | 85.46±6.59 | 80.45±6.27 | 74.03±3.86 | 64.83±2.57 | 55.63±2.10 | 47.80±1.85 | 39.12±1.71 |
| | Transfer | 11.03±2.19 | 9.83±0.23 | 8.65±0.68 | 9.98±0.69 | 9.85±0.71 | 10.19±0.27 | 11.58±1.00 | 12.63±1.07 | 12.21±0.85 | 12.24±0.24 | 11.14±1.72 |
| | CausIRL-CORAL | 99.11±0.41 | 99.08±0.39 | 98.43±0.39 | 97.38±0.94 | 88.78±3.97 | 70.23±5.03 | 48.45±5.92 | 33.33±3.28 | 24.82±0.64 | 20.28±0.17 | 16.77±0.96 |
| | CausIRL-MMD | 98.61±0.27 | 98.77±0.13 | 98.66±0.19 | 97.22±0.37 | 92.03±0.39 | 77.83±3.79 | 62.53±5.78 | 44.68±5.54 | 35.69±3.64 | 28.59±2.66 | 23.61±1.13 |
| | EQRM | 97.59±1.20 | 97.46±0.41 | 96.93±0.33 | 94.03±0.62 | 77.96±4.62 | 54.93±6.90 | 36.74±5.47 | 24.79±3.18 | 18.76±1.26 | 17.77±0.26 | 14.94±0.78 |
| $\mathcal{D}_{10}$ | ERM | 69.34±41.53 | 69.47±41.67 | 67.92±41.58 | 60.56±36.04 | 48.06±26.26 | 28.77±12.94 | 19.79±5.26 | 14.70±3.03 | 12.24±2.05 | 11.27±0.81 | 12.55±0.39 |
| | VREx | 66.27±18.73 | 64.44±19.71 | 64.99±17.77 | 64.36±17.63 | 60.72±16.05 | 52.18±13.50 | 46.57±8.97 | 41.19±5.75 | 35.82±3.63 | 30.19±2.98 | 27.78±1.26 |
| | IRM | 75.45±7.84 | 75.00±7.83 | 74.37±8.33 | 70.52±10.20 | 63.81±10.18 | 55.71±8.60 | 46.91±5.60 | 39.20±4.16 | 34.17±4.21 | 27.59±1.01 | 24.58±0.67 |
| | SD | 98.87±0.16 | 98.74±0.00 | 98.11±0.06 | 91.95±4.34 | 77.12±6.26 | 53.72±10.54 | 37.29±6.79 | 26.02±5.15 | 20.52±2.23 | 17.11±0.81 | 14.39±1.06 |
| | GroupDRO | 97.98±1.03 | 98.11±0.33 | 97.59±0.55 | 90.28±1.81 | 71.78±1.64 | 50.63±5.61 | 31.29±8.46 | 19.73±7.39 | 15.70±5.55 | 14.86±5.00 | 13.52±2.34 |
| | RSC | 97.33±0.19 | 97.12±0.49 | 96.83±0.16 | 89.28±3.96 | 66.25±12.32 | 42.69±9.55 | 28.54±5.22 | 20.73±2.46 | 16.06±0.59 | 14.28±0.48 | 13.08±0.37 |
| | Mixup | 98.77±0.81 | 98.40±0.51 | 97.59±0.45 | 95.73±0.46 | 88.26±4.60 | 76.73±8.28 | 64.70±5.75 | 52.49±4.88 | 42.43±1.97 | 35.25±1.23 | 30.24±1.25 |
| | CORAL | 98.11±1.11 | 97.54±1.24 | 97.62±0.77 | 95.75±0.64 | 87.97±1.11 | 70.18±4.94 | 47.59±7.05 | 32.15±5.30 | 25.13±4.27 | 19.18±1.85 | 17.45±0.55 |
| | MMD | 98.90±0.19 | 99.03±0.13 | 98.74±0.17 | 97.88±0.22 | 93.58±1.23 | 81.05±2.63 | 62.50±1.57 | 47.01±4.30 | 35.43±2.48 | 28.46±3.41 | 23.77±1.70 |
| | DANN | 80.56±11.92 | 78.07±14.31 | 76.02±13.40 | 54.64±25.40 | 41.38±22.50 | 32.39±13.30 | 25.71±7.73 | 21.38±3.00 | 18.37±1.64 | 14.96±0.69 | 16.33±0.66 |
| | CDANN | 82.26±4.42 | 82.57±4.95 | 80.77±3.94 | 57.70±2.31 | 26.89±2.54 | 20.49±2.04 | 17.90±0.15 | 15.57±1.42 | 16.46±0.20 | 14.57±1.10 | 15.57±0.28 |
| | MTL | 97.88±1.17 | 97.62±1.03 | 96.51±1.48 | 89.05±4.70 | 68.66±9.43 | 47.72±11.41 | 33.33±8.15 | 24.92±6.16 | 21.36±4.49 | 19.05±3.81 | 17.03±2.96 |
| | SagNet | 97.69±0.27 | 97.79±0.39 | 97.59±0.39 | 94.47±1.35 | 82.47±2.30 | 62.32±1.63 | 42.06±2.23 | 30.16±1.30 | 25.21±2.47 | 20.52±1.79 | 18.63±1.16 |
| | ARM | 98.48±0.49 | 98.32±0.10 | 97.14±0.93 | 88.71±7.32 | 70.23±13.58 | 52.99±13.75 | 33.05±6.39 | 23.38±3.12 | 18.74±3.10 | 16.40±1.08 | 13.86±1.33 |
| | ANDMask | 97.85±1.32 | 97.48±0.78 | 96.70±1.30 | 89.54±5.37 | 72.75±16.00 | 50.03±18.42 | 32.96±6.58 | 24.79±6.74 | 19.18±2.74 | 16.30±1.61 | 15.38±1.58 |
| | SANDMask | 69.16±40.96 | 69.13±40.10 | 68.66±40.54 | 62.55±36.16 | 49.24±26.76 | 34.43±18.23 | 23.03±8.45 | 16.01±1.95 | 12.81±0.17 | 11.87±1.12 | 12.00±0.20 |
| | IGA | 15.64±4.85 | 15.46±5.15 | 15.28±4.93 | 15.28±3.35 | 15.83±4.93 | 16.14±4.63 | 16.35±4.37 | 16.19±3.34 | 15.78±2.50 | 15.54±2.55 | |
| | SelfReg | 98.64±0.29 | 98.90±0.17 | 98.30±0.10 | 94.71±0.90 | 81.37±3.20 | 65.70±6.23 | 46.88±8.51 | 33.52±5.33 | 23.93±1.98 | 18.13±1.85 | 16.19±0.97 |
| | Fish | 98.53±0.19 | 98.38±0.20 | 97.30±0.13 | 91.19±2.21 | 67.14±6.57 | 44.47±7.68 | 27.46±4.87 | 20.58±3.86 | 16.85±1.65 | 15.36±0.90 | 13.65±1.02 |
| | TRM | 69.79±40.78 | 69.42±40.97 | 69.99±39.49 | 65.85±36.97 | 52.88±27.86 | 38.13±16.90 | 28.49±10.00 | 21.33±5.55 | 18.40±2.65 | 16.75±2.61 | 14.60±1.44 |
| | IB-ERM | 94.10±5.80 | 93.45±6.44 | 93.63±6.34 | 91.77±7.98 | 88.29±5.57 | 77.04±6.45 | 63.44±6.03 | 49.82±4.82 | 39.12±4.69 | 31.84±3.36 | 26.94±3.75 |
| | IB-IRM | 83.15±8.42 | 83.44±8.38 | 83.78±7.74 | 83.02±6.95 | 81.13±7.23 | 76.55±6.02 | 69.10±6.48 | 56.60±4.25 | 46.54±5.43 | 38.39±4.04 | 30.08±4.37 |
| | CAD | 91.82±0.69 | 91.40±1.10 | 91.54±1.23 | 91.46±0.35 | 89.10±0.60 | 84.12±1.93 | 76.28±2.87 | 66.56±1.51 | 56.79±1.19 | 49.19±1.22 | 41.09±1.49 |
| | CondCAD | 73.85±9.30 | 77.23±8.32 | 74.37±7.61 | 74.06±9.47 | 71.31±8.50 | 66.38±7.58 | 62.76±6.91 | 56.26±4.89 | 51.00±3.54 | 45.62±3.40 | 40.17±1.16 |
| | Transfer | 9.46±1.29 | 9.09±1.26 | 8.75±0.65 | 9.54±0.50 | 9.38±0.29 | 10.06±0.45 | 11.71±0.86 | 11.90±1.83 | 11.58±1.63 | 12.00±0.48 | 12.40±0.27 |
| | CausIRL-CORAL | 99.11±0.41 | 99.08±0.39 | 98.43±0.39 | 97.38±0.94 | 88.78±3.97 | 70.23±5.03 | 48.45±5.92 | 33.33±3.28 | 24.82±0.64 | 20.28±0.17 | 16.77±0.96 |
| | CausIRL-MMD | 98.61±0.27 | 98.77±0.13 | 98.66±0.19 | 97.22±0.37 | 92.03±0.39 | 77.83±3.79 | 62.53±5.78 | 44.68±5.54 | 35.69±3.64 | 28.59±2.66 | 23.61±1.13 |
| | EQRM | 95.86±2.28 | 96.12±2.20 | 95.39±2.77 | 89.02±3.17 | 75.29±4.71 | 58.31±4.63 | 39.15±4.71 | 27.02±2.16 | 19.99±0.71 | 17.09±0.71 | 15.75±0.24 |

Table 4: Average accuracy of top-3 models of each algorithm at each shift degree of NOISYMNIST (Simple CNN). This table shows the results on shift degrees from $\mathcal{D}_8$ to $\mathcal{D}_{10}$.

| | Algorithm | $\mathcal{D}_0$ | $\mathcal{D}_1$ | $\mathcal{D}_2$ | $\mathcal{D}_3$ | $\mathcal{D}_4$ | $\mathcal{D}_5$ | $\mathcal{D}_6$ | $\mathcal{D}_7$ | $\mathcal{D}_8$ | $\mathcal{D}_9$ | $\mathcal{D}_{10}$ |
|---|---|---|---|---|---|---|---|---|---|---|---|---|
| $\mathcal{D}_0$ | ERM | 98.64±0.04 | 98.24±0.23 | 98.33±0.09 | 84.00±6.16 | 49.13±14.90 | 26.13±14.19 | 18.77±9.39 | 13.78±5.04 | 11.05±2.62 | 10.17±1.45 | 10.32±0.80 |
| | VREx | 98.66±0.11 | 98.27±0.15 | 98.85±0.08 | 90.42±3.39 | 61.07±6.45 | 29.15±4.95 | 15.18±1.98 | 10.95±0.85 | 9.62±0.42 | 9.22±0.15 | 9.90±0.15 |
| | IRM | 98.50±0.21 | 98.53±0.14 | 98.35±0.60 | 84.62±5.92 | 58.97±7.66 | 36.03±5.22 | 23.15±1.31 | 15.22±0.19 | 11.73±0.31 | 10.42±0.37 | 10.52±0.16 |
| | SD | 98.63±0.03 | 98.41±0.14 | 98.45±0.63 | 83.92±5.91 | 48.75±7.87 | 20.72±3.48 | 12.88±1.69 | 10.90±0.92 | 9.57±0.34 | 9.28±0.16 | 9.87±0.06 |
| | CORAL | 98.62±0.07 | 98.34±0.11 | 96.22±0.77 | 69.63±5.19 | 33.73±3.03 | 16.38±4.07 | 12.75±3.19 | 11.48±2.77 | 10.37±1.90 | 10.25±1.80 | 10.83±1.60 |
| | GroupDRO | 98.68±0.09 | 98.33±0.13 | 96.37±2.14 | 70.15±8.73 | 36.23±8.62 | 17.02±5.31 | 12.65±2.31 | 10.50±0.83 | 9.67±0.52 | 9.47±0.37 | 9.77±0.06 |
| | RSC | 98.73±0.07 | 98.48±0.18 | 97.75±0.16 | 83.38±1.70 | 49.15±2.80 | 28.62±3.25 | 19.87±3.04 | 14.55±2.16 | 11.20±1.33 | 10.80±1.15 | 10.60±0.53 |
| | DANN | 95.22±0.88 | 94.79±1.20 | 95.42±0.12 | 75.80±6.00 | 45.97±7.75 | 27.75±7.42 | 20.57±4.98 | 15.67±3.47 | 14.93±2.86 | 13.28±2.18 | 13.58±1.95 |
| | Mixup | 98.30±0.07 | 97.93±0.24 | 83.75±12.92 | 40.97±17.46 | 20.35±7.57 | 12.32±2.77 | 10.42±0.52 | 9.47±0.06 | 8.95±0.00 | 9.00±0.07 | 9.73±0.05 |
| | CAD | 98.22±0.24 | 97.97±0.44 | 97.93±0.42 | 82.97±2.53 | 47.83±6.64 | 22.58±4.51 | 15.27±2.60 | 12.22±1.92 | 11.40±1.39 | 10.50±1.15 | 10.98±0.92 |
| | IB-IRM | 97.91±0.43 | 97.73±0.78 | 97.15±1.10 | 78.23±11.74 | 49.33±11.66 | 27.28±6.25 | 16.90±4.10 | 12.03±2.21 | 10.17±1.20 | 9.35±0.46 | 9.98±0.30 |
| $\mathcal{D}_1$ | ERM | 98.44±0.05 | 98.60±0.00 | 97.92±0.34 | 81.08±4.50 | 49.43±4.59 | 23.50±4.35 | 14.72±4.44 | 10.98±1.82 | 9.42±0.62 | 9.20±0.25 | 9.83±0.12 |
| | VREx | 98.38±0.06 | 98.49±0.08 | 98.43±0.45 | 83.53±8.46 | 46.90±11.96 | 19.18±2.76 | 11.77±0.65 | 9.77±0.32 | 9.02±0.09 | 9.02±0.06 | 9.72±0.02 |
| | IRM | 98.40±0.17 | 98.61±0.14 | 98.32±0.14 | 87.10±3.92 | 53.62±3.80 | 28.05±7.91 | 18.70±5.89 | 14.07±3.22 | 11.48±2.06 | 10.43±1.25 | 10.78±1.03 |
| | SD | 98.37±0.24 | 98.62±0.02 | 97.95±0.25 | 80.12±2.30 | 41.53±0.73 | 20.75±1.70 | 15.38±2.11 | 12.55±1.07 | 11.05±0.84 | 10.18±0.70 | 10.48±0.47 |
| | CORAL | 98.37±0.07 | 98.53±0.01 | 97.10±0.65 | 68.57±4.47 | 29.20±3.84 | 14.92±2.43 | 12.60±2.19 | 11.38±1.94 | 11.28±1.94 | 11.30±1.98 | 10.90±1.46 |
| | GroupDRO | 98.25±0.20 | 98.60±0.08 | 96.93±1.09 | 74.03±5.45 | 35.72±8.91 | 17.02±5.00 | 11.90±1.58 | 9.80±0.40 | 9.15±0.22 | 9.05±0.14 | 9.85±0.21 |
| | RSC | 98.56±0.22 | 98.68±0.04 | 98.13±0.42 | 86.47±2.52 | 52.22±2.36 | 25.02±3.27 | 16.68±2.99 | 12.50±2.01 | 10.48±1.02 | 10.07±0.70 | 10.40±0.32 |
| | DANN | 95.22±0.88 | 94.79±1.20 | 95.42±0.12 | 75.80±6.00 | 45.97±7.75 | 27.75±7.42 | 20.57±4.98 | 15.67±3.47 | 14.93±2.86 | 13.28±2.18 | 13.58±1.95 |
| | Mixup | 97.83±0.28 | 98.44±0.12 | 92.28±1.53 | 49.23±3.02 | 19.55±4.57 | 11.17±1.44 | 10.10±0.35 | 9.50±0.14 | 9.05±0.14 | 9.08±0.19 | 9.85±0.21 |
| | CAD | 98.18±0.29 | 98.32±0.14 | 98.20±0.59 | 83.90±5.68 | 50.13±4.04 | 27.37±3.94 | 19.85±4.85 | 15.22±3.65 | 13.83±3.96 | 12.08±2.33 | 11.88±1.80 |
| | IB-IRM | 97.71±0.60 | 97.78±0.71 | 96.30±1.03 | 74.13±10.21 | 45.57±8.52 | 26.80±5.12 | 17.97±3.19 | 12.95±2.20 | 10.87±1.72 | 10.08±1.23 | 10.42±0.87 |
| $\mathcal{D}_2$ | ERM | 98.11±0.25 | 98.21±0.12 | 98.88±0.06 | 90.00±1.22 | 55.78±2.63 | 27.28±5.05 | 17.97±5.08 | 14.12±3.27 | 12.12±2.25 | 11.15±1.58 | 11.10±1.02 |
| | VREx | 98.52±0.32 | 98.16±0.15 | 98.85±0.08 | 88.65±5.44 | 53.35±16.56 | 25.65±9.63 | 14.43±3.04 | 10.85±0.99 | 9.58±0.46 | 9.20±0.18 | 9.90±0.15 |
| | IRM | 98.39±0.29 | 98.22±0.35 | 98.85±0.12 | 91.38±0.92 | 64.30±5.58 | 33.45±8.08 | 18.28±4.88 | 12.83±2.11 | 10.68±1.30 | 9.98±0.78 | 10.20±0.41 |
| | SD | 98.11±0.42 | 98.31±0.12 | 99.03±0.13 | 90.62±0.94 | 56.68±3.32 | 27.88±5.26 | 17.57±3.03 | 13.22±2.04 | 11.00±1.18 | 10.67±1.28 | 10.68±0.91 |
| | CORAL | 97.84±0.29 | 97.86±0.10 | 98.42±0.22 | 79.15±3.26 | 41.90±7.26 | 19.50±2.61 | 13.98±0.79 | 11.03±0.46 | 9.90±0.43 | 9.58±0.35 | 10.37±0.52 |
| | GroupDRO | 98.30±0.07 | 98.23±0.10 | 98.48±0.16 | 85.37±0.47 | 51.78±3.02 | 26.95±4.37 | 17.45±2.13 | 12.72±1.06 | 10.68±1.44 | 10.45±1.46 | 9.97±0.24 |
| | RSC | 98.31±0.03 | 98.31±0.40 | 98.98±0.08 | 88.42±0.82 | 50.42±2.31 | 19.82±1.12 | 11.95±0.78 | 9.93±0.47 | 9.38±0.61 | 9.22±0.38 | 9.78±0.08 |
| | DANN | 94.90±0.45 | 94.58±0.91 | 95.47±0.16 | 78.18±2.74 | 48.30±6.08 | 29.28±7.27 | 21.17±5.18 | 15.90±3.61 | 15.08±2.93 | 13.30±2.19 | 13.80±2.09 |
| | Mixup | 97.07±0.54 | 97.03±0.69 | 97.30±0.20 | 72.68±1.79 | 27.87±2.06 | 11.42±0.93 | 9.90±0.07 | 9.40±0.00 | 8.95±0.00 | 8.95±0.00 | 9.70±0.00 |
| | CAD | 97.91±0.37 | 97.72±0.47 | 98.52±0.31 | 85.08±3.73 | 50.55±3.55 | 26.48±4.83 | 19.28±4.62 | 15.58±2.88 | 14.87±3.04 | 12.80±1.46 | 12.67±1.14 |
| | IB-IRM | 97.42±0.05 | 96.99±0.31 | 97.97±0.43 | 79.82±7.98 | 50.62±10.82 | 26.85±5.92 | 15.35±2.53 | 11.13±1.14 | 9.43±0.44 | 9.25±0.22 | 9.85±0.21 |
| $\mathcal{D}_3$ | ERM | 98.24±0.32 | 98.14±0.30 | 98.45±0.07 | 92.90±0.07 | 68.67±1.27 | 37.47±3.32 | 22.00±3.80 | 14.80±2.74 | 12.33±2.07 | 10.80±1.59 | 10.83±0.87 |
| | VREx | 97.90±0.82 | 97.64±0.84 | 98.10±0.63 | 93.13±1.40 | 61.53±5.72 | 25.87±6.46 | 13.87±2.17 | 10.60±0.82 | 9.45±0.45 | 9.23±0.21 | 10.03±0.27 |
| | IRM | 97.62±0.81 | 97.53±0.69 | 98.22±0.79 | 92.22±0.66 | 61.02±4.07 | 28.75±4.06 | 15.73±2.68 | 11.97±1.51 | 10.03±0.78 | 9.63±0.46 | 10.02±0.22 |
| | SD | 97.94±0.21 | 97.92±0.38 | 98.38±0.29 | 92.47±0.18 | 67.40±4.46 | 35.97±8.57 | 20.73±6.51 | 14.18±4.50 | 11.80±3.08 | 10.88±2.26 | 10.62±1.12 |
| | CORAL | 98.09±0.11 | 97.85±0.01 | 97.43±0.47 | 85.13±1.37 | 50.08±1.09 | 24.95±3.23 | 15.80±3.08 | 11.93±1.64 | 10.10±0.78 | 9.50±0.37 | 10.08±0.27 |
| | GroupDRO | 97.83±0.66 | 97.91±0.46 | 97.62±0.29 | 87.08±0.44 | 51.73±4.37 | 25.55±3.79 | 15.55±3.79 | 11.35±1.90 | 9.95±1.06 | 9.37±0.55 | 9.92±0.27 |
| | RSC | 97.83±0.02 | 97.82±0.22 | 98.10±0.72 | 92.20±0.39 | 68.62±4.88 | 38.80±5.94 | 23.58±4.84 | 16.10±3.31 | 12.02±1.82 | 10.73±1.04 | 10.65±0.60 |
| | DANN | 92.53±1.65 | 92.26±2.09 | 94.23±0.74 | 80.38±1.74 | 52.03±4.07 | 30.95±5.00 | 21.07±3.62 | 15.98±2.01 | 14.42±2.63 | 12.97±1.70 | 12.67±1.89 |
| | Mixup | 92.54±3.83 | 93.05±3.16 | 93.37±2.52 | 79.73±0.98 | 52.50±5.81 | 30.37±6.13 | 20.83±3.38 | 16.43±2.53 | 14.98±2.23 | 12.95±2.45 | 12.63±1.21 |
| | CAD | 97.71±0.66 | 97.82±0.42 | 98.23±0.54 | 88.03±1.67 | 55.88±3.99 | 29.77±3.97 | 20.43±4.31 | 15.45±3.45 | 14.90±3.86 | 12.82±2.37 | 12.47±1.92 |
| | IB-IRM | 96.08±2.46 | 96.02±1.91 | 96.83±1.06 | 86.47±2.02 | 56.13±4.08 | 30.60±3.38 | 18.93±3.55 | 13.92±2.55 | 11.60±1.86 | 10.75±1.27 | 10.92±0.91 |

Table 5: Average accuracy of top-3 models of each algorithm at each shift degree of NOISYMNIST (EfficientNet-b0). This table shows the results on shift degrees from $\mathcal{D}_0$ to $\mathcal{D}_3$.

| | Algorithm | $\mathcal{D}_0$ | $\mathcal{D}_1$ | $\mathcal{D}_2$ | $\mathcal{D}_3$ | $\mathcal{D}_4$ | $\mathcal{D}_5$ | $\mathcal{D}_6$ | $\mathcal{D}_7$ | $\mathcal{D}_8$ | $\mathcal{D}_9$ | $\mathcal{D}_{10}$ |
|---|---|---|---|---|---|---|---|---|---|---|---|---|
| $\mathcal{D}_4$ | ERM | 98.21±0.14 | 97.92±0.28 | 98.45±0.11 | 92.35±0.47 | 70.77±0.30 | 41.37±2.08 | 24.78±2.28 | 15.82±1.94 | 12.12±1.34 | 10.68±1.12 | 10.80±0.63 |
| | VREx | 98.28±0.35 | 98.19±0.15 | 98.53±0.46 | 92.12±2.26 | 65.13±4.30 | 31.42±7.25 | 16.47±3.70 | 11.20±1.18 | 9.73±0.56 | 9.18±0.16 | 9.85±0.15 |
| | IRM | 98.07±0.62 | 98.16±0.43 | 98.37±0.79 | 90.40±1.72 | 68.35±1.40 | 40.02±4.64 | 22.80±3.04 | 15.37±1.64 | 12.20±1.08 | 11.28±1.19 | 11.27±1.17 |
| | SD | 97.67±0.46 | 97.71±0.39 | 98.28±0.30 | 91.75±1.04 | 69.33±1.95 | 39.25±4.21 | 23.07±3.67 | 15.63±3.40 | 12.48±2.58 | 11.28±1.97 | 10.90±0.95 |
| | CORAL | 98.21±0.16 | 97.83±0.01 | 97.98±0.40 | 83.18±2.12 | 52.40±2.45 | 26.48±3.49 | 17.80±2.78 | 13.03±2.44 | 11.07±1.64 | 10.72±1.75 | 10.87±1.11 |
| | GroupDRO | 97.54±0.23 | 97.59±0.13 | 97.42±0.45 | 85.08±0.59 | 61.53±1.95 | 39.97±0.17 | 26.80±2.07 | 19.92±1.37 | 16.08±0.42 | 14.75±0.15 | 14.12±0.47 |
| | RSC | 97.64±0.22 | 97.81±0.49 | 97.88±0.41 | 89.20±2.21 | 70.10±3.90 | 39.98±5.20 | 23.77±4.27 | 15.55±2.67 | 11.62±1.11 | 10.33±0.84 | 10.62±0.58 |
| | DANN | 87.58±5.01 | 87.06±4.89 | 89.18±4.56 | 75.32±4.74 | 54.37±3.41 | 35.53±3.98 | 25.67±1.43 | 18.00±1.56 | 15.92±2.05 | 13.73±0.98 | 13.42±2.59 |
| | Mixup | 92.22±3.42 | 92.81±2.84 | 93.28±2.41 | 79.23±1.61 | 52.80±5.39 | 29.95±6.71 | 19.63±5.08 | 15.23±4.07 | 14.02±3.58 | 12.62±2.86 | 12.05±1.88 |
| | CAD | 92.71±6.13 | 93.06±5.84 | 93.28±5.75 | 80.57±6.29 | 57.93±2.65 | 35.55±2.80 | 24.38±4.58 | 18.52±4.77 | 15.67±4.02 | 14.32±3.33 | 13.42±2.49 |
| | IB-IRM | 94.24±2.22 | 94.22±1.99 | 95.97±1.88 | 83.88±3.18 | 56.48±4.22 | 32.78±2.04 | 20.37±1.11 | 15.32±1.85 | 12.60±1.81 | 11.43±1.35 | 11.40±0.89 |
| $\mathcal{D}_5$ | ERM | 96.87±2.20 | 96.76±1.72 | 97.95±0.78 | 90.42±2.17 | 67.65±3.35 | 43.28±1.83 | 28.77±2.58 | 18.25±2.63 | 13.93±2.18 | 11.60±1.49 | 11.38±0.98 |
| | VREx | 98.17±0.42 | 97.91±0.26 | 98.60±0.36 | 90.08±3.86 | 63.97±5.91 | 35.03±2.37 | 18.50±1.68 | 11.58±0.67 | 9.85±0.39 | 9.25±0.07 | 9.87±0.13 |
| | IRM | 83.16±21.05 | 84.08±19.87 | 87.73±14.71 | 78.30±15.61 | 63.05±8.85 | 43.27±0.06 | 29.45±6.47 | 21.07±6.89 | 17.72±6.91 | 15.60±5.30 | 14.88±4.45 |
| | SD | 97.86±0.15 | 97.92±0.38 | 98.43±0.33 | 91.22±1.22 | 68.10±3.02 | 43.58±0.61 | 29.82±1.49 | 20.55±1.64 | 16.32±1.21 | 14.37±1.63 | 12.63±1.18 |
| | CORAL | 97.57±0.59 | 97.29±0.38 | 96.02±1.08 | 76.85±3.20 | 51.55±2.99 | 31.58±2.94 | 22.85±3.32 | 17.13±2.45 | 14.37±2.48 | 13.22±2.04 | 12.17±1.52 |
| | GroupDRO | 97.68±0.15 | 97.79±0.20 | 97.60±0.41 | 84.95±0.54 | 61.02±1.54 | 40.40±0.59 | 27.33±1.92 | 20.08±1.39 | 16.20±0.58 | 15.15±0.62 | 14.82±0.64 |
| | RSC | 96.32±2.09 | 96.55±2.20 | 95.82±3.80 | 85.73±7.20 | 65.23±7.69 | 45.18±2.16 | 30.23±1.66 | 20.20±1.41 | 14.80±2.01 | 12.05±0.95 | 11.75±0.92 |
| | DANN | 91.91±4.32 | 91.54±4.72 | 92.98±3.49 | 76.97±3.03 | 53.08±5.22 | 35.58±3.91 | 25.57±1.57 | 18.80±0.52 | 17.08±0.47 | 14.47±0.55 | 15.27±0.02 |
| | Mixup | 91.54±2.58 | 92.81±2.85 | 89.30±3.51 | 72.50±11.05 | 51.42±7.33 | 32.60±3.05 | 23.25±0.11 | 18.23±1.41 | 16.82±0.62 | 14.38±1.35 | 13.75±1.15 |
| | CAD | 92.71±6.13 | 93.06±5.84 | 93.28±5.75 | 80.57±6.29 | 57.93±2.65 | 35.55±2.80 | 24.38±4.58 | 18.52±4.77 | 15.67±4.02 | 14.32±3.33 | 13.42±2.49 |
| | IB-IRM | 88.05±10.89 | 87.28±11.65 | 90.30±8.15 | 78.85±8.11 | 55.08±4.63 | 35.30±0.84 | 24.18±2.91 | 18.00±2.80 | 15.17±3.23 | 13.43±3.17 | 12.95±2.47 |
| $\mathcal{D}_6$ | ERM | 96.74±2.13 | 96.68±1.66 | 97.88±0.73 | 90.73±2.42 | 66.40±2.41 | 42.37±2.11 | 29.20±2.05 | 19.50±1.02 | 15.35±0.56 | 12.73±0.38 | 12.00±0.45 |
| | VREx | 97.89±0.40 | 97.58±0.51 | 97.25±1.12 | 84.82±4.97 | 55.87±9.85 | 34.38±2.61 | 22.07±1.39 | 15.68±2.63 | 12.93±2.15 | 11.50±1.66 | 11.30±1.10 |
| | IRM | 41.32±8.56 | 42.73±9.53 | 54.90±9.10 | 49.45±5.47 | 43.55±5.23 | 37.28±4.48 | 33.27±4.14 | 28.63±1.86 | 25.07±1.83 | 22.15±1.17 | 20.40±1.21 |
| | SD | 97.97±0.11 | 98.19±0.01 | 98.62±0.13 | 90.70±0.98 | 66.23±1.45 | 43.50±0.51 | 31.12±0.94 | 20.97±1.69 | 17.05±1.49 | 14.77±1.65 | 13.32±1.32 |
| | CORAL | 97.82±0.39 | 97.54±0.42 | 96.28±0.72 | 73.67±6.84 | 45.65±3.93 | 30.68±3.48 | 24.42±2.19 | 18.03±1.71 | 15.90±1.51 | 13.35±1.83 | 13.02±1.18 |
| | GroupDRO | 97.68±0.15 | 97.79±0.20 | 97.60±0.41 | 84.95±0.54 | 61.02±1.54 | 40.40±0.59 | 27.33±1.92 | 20.08±1.39 | 16.20±0.58 | 15.15±0.62 | 14.82±0.64 |
| | RSC | 96.32±2.09 | 96.55±2.20 | 95.82±3.80 | 85.73±7.20 | 65.23±7.69 | 45.18±2.16 | 30.23±1.66 | 20.20±1.41 | 14.80±2.01 | 12.05±0.95 | 11.75±0.92 |
| | DANN | 87.58±5.01 | 87.06±4.89 | 89.18±4.56 | 75.32±4.74 | 54.37±3.41 | 35.53±3.98 | 25.67±1.43 | 18.00±1.56 | 15.92±2.05 | 13.73±0.98 | 13.42±2.59 |
| | Mixup | 91.54±2.58 | 92.81±2.85 | 89.30±3.51 | 72.50±11.05 | 51.42±7.33 | 32.60±3.05 | 23.25±0.11 | 18.23±1.41 | 16.82±0.62 | 14.38±1.35 | 13.75±1.15 |
| | CAD | 92.86±6.26 | 93.09±5.93 | 93.32±5.88 | 81.17±7.09 | 53.80±2.62 | 32.82±4.93 | 26.25±3.39 | 20.63±3.34 | 18.08±2.56 | 15.67±2.06 | 14.23±1.86 |
| | IB-IRM | 89.44±11.64 | 88.78±12.46 | 90.82±8.41 | 76.98±8.41 | 54.15±5.10 | 34.53±1.83 | 24.25±2.85 | 17.62±2.97 | 14.90±3.34 | 13.15±3.27 | 12.78±2.54 |
| $\mathcal{D}_7$ | ERM | 95.26±2.40 | 95.07±2.13 | 96.53±1.67 | 85.27±6.52 | 62.37±5.58 | 41.05±3.45 | 29.13±2.12 | 20.32±0.75 | 16.08±1.08 | 13.77±1.65 | 13.15±1.69 |
| | VREx | 73.32±34.73 | 73.44±33.75 | 73.17±33.00 | 63.67±26.82 | 41.20±14.42 | 29.52±4.41 | 21.45±2.06 | 17.23±1.25 | 14.30±0.94 | 13.65±1.59 | 12.77±1.05 |
| | IRM | 41.32±8.56 | 42.73±9.53 | 54.90±9.10 | 49.45±5.47 | 43.55±5.23 | 37.28±4.48 | 33.27±4.14 | 28.63±1.86 | 25.07±1.83 | 22.15±1.17 | 20.40±1.21 |
| | SD | 96.60±1.88 | 96.93±1.78 | 97.25±1.80 | 87.52±3.69 | 64.23±4.15 | 41.47±3.34 | 30.33±2.04 | 21.28±1.27 | 18.17±0.19 | 15.78±0.53 | 14.30±0.07 |
| | CORAL | 97.61±0.28 | 97.48±0.40 | 96.07±0.41 | 69.67±4.21 | 43.52±4.48 | 29.72±4.37 | 23.72±2.76 | 18.20±1.59 | 16.12±1.53 | 13.55±1.82 | 13.35±1.46 |
| | GroupDRO | 97.69±0.14 | 97.86±0.14 | 97.20±0.36 | 82.75±2.71 | 58.05±3.52 | 38.75±2.79 | 26.97±2.38 | 20.18±1.25 | 16.48±0.43 | 15.22±0.60 | 14.82±0.64 |
| | RSC | 96.63±2.31 | 96.41±2.13 | 95.53±3.64 | 83.10±5.90 | 57.03±6.74 | 38.67±4.21 | 29.10±2.41 | 20.52±1.19 | 17.13±0.27 | 13.88±0.90 | 12.88±0.39 |
| | DANN | 88.05±2.07 | 87.28±2.03 | 88.37±0.95 | 72.30±2.50 | 50.53±4.71 | 33.87±4.30 | 24.30±2.07 | 19.22±0.44 | 16.82±0.81 | 13.60±0.14 | 13.27±1.41 |
| | Mixup | 92.95±3.36 | 93.69±2.94 | 73.43±20.12 | 51.97±24.82 | 38.92±16.14 | 26.87±8.04 | 21.30±2.86 | 18.58±0.93 | 17.05±0.31 | 15.65±0.94 | 14.92±0.56 |
| | CAD | 91.70±5.87 | 92.01±5.57 | 91.82±5.63 | 76.53±9.85 | 51.72±5.32 | 33.22±4.54 | 26.03±3.61 | 20.87±3.12 | 18.40±2.13 | 15.53±2.20 | 14.40±1.66 |
| | IB-IRM | 55.73±25.00 | 55.51±24.36 | 61.58±24.72 | 49.53±21.59 | 34.65±9.97 | 26.07±4.41 | 22.42±2.64 | 19.78±1.69 | 17.63±1.61 | 16.28±0.38 | 16.12±0.42 |

Table 6: Average accuracy of top-3 models of each algorithm at each shift degree of NOISYMNIST (EfficientNet-b0). This table shows the results on shift degrees from $\mathcal{D}_4$ to $\mathcal{D}_7$.

| | Algorithm | $\mathcal{D}_0$ | $\mathcal{D}_1$ | $\mathcal{D}_2$ | $\mathcal{D}_3$ | $\mathcal{D}_4$ | $\mathcal{D}_5$ | $\mathcal{D}_6$ | $\mathcal{D}_7$ | $\mathcal{D}_8$ | $\mathcal{D}_9$ | $\mathcal{D}_{10}$ |
|---|---|---|---|---|---|---|---|---|---|---|---|---|
| $\mathcal{D}_8$ | ERM | 93.96±0.57 | 93.89±0.69 | 94.55±1.80 | 76.12±9.31 | 51.92±10.73 | 34.90±5.89 | 25.72±3.10 | 19.13±1.45 | 16.65±0.55 | 14.53±1.22 | 14.08±1.19 |
| | VREx | 68.05±39.25 | 68.24±39.24 | 66.77±37.74 | 49.25±26.67 | 27.30±12.21 | 20.68±7.95 | 18.20±4.25 | 15.98±2.05 | 15.23±0.16 | 12.92±0.28 | 11.45±0.70 |
| | IRM | 41.32±8.56 | 42.73±9.53 | 54.90±9.10 | 49.45±5.47 | 43.55±5.23 | 37.28±4.48 | 33.27±4.14 | 28.63±1.86 | 25.07±1.83 | 22.15±1.17 | 20.40±1.21 |
| | SD | 96.73±1.97 | 96.94±1.79 | 96.97±1.64 | 85.37±3.78 | 55.72±9.91 | 36.37±6.48 | 27.62±3.43 | 20.22±0.98 | 18.20±0.14 | 15.47±0.15 | 14.15±0.25 |
| | CORAL | 97.69±0.39 | 97.43±0.34 | 96.57±0.64 | 75.20±4.90 | 45.62±3.97 | 29.60±4.40 | 23.87±2.71 | 18.08±1.69 | 16.68±0.83 | 14.12±1.18 | 14.17±0.45 |
| | GroupDRO | 96.16±2.12 | 96.34±2.15 | 94.83±2.94 | 70.40±10.43 | 45.82±11.71 | 31.45±6.90 | 24.10±2.27 | 18.45±1.46 | 16.55±0.37 | 14.87±1.09 | 13.80±1.33 |
| | RSC | 96.63±2.31 | 96.41±2.13 | 95.53±3.64 | 83.10±5.90 | 57.03±6.74 | 38.67±4.21 | 29.10±2.41 | 20.52±1.19 | 17.13±0.27 | 13.88±0.90 | 12.88±0.39 |
| | DANN | 90.71±3.94 | 90.24±4.45 | 91.12±3.27 | 74.17±0.95 | 50.58±4.66 | 34.43±3.59 | 24.62±1.83 | 19.10±0.60 | 17.08±0.47 | 13.95±0.58 | 14.23±1.47 |
| | Mixup | 92.65±3.11 | 93.67±2.93 | 73.47±20.07 | 54.52±21.52 | 40.12±14.61 | 27.37±7.39 | 21.68±2.32 | 18.58±0.93 | 17.43±0.29 | 15.97±1.29 | 15.70±1.64 |
| | CAD | 93.47±6.63 | 93.63±6.24 | 94.08±6.29 | 83.62±7.76 | 57.38±3.27 | 34.82±3.47 | 25.82±3.85 | 20.18±3.82 | 18.48±2.02 | 15.82±1.92 | 14.70±1.32 |
| | IB-IRM | 63.79±17.64 | 64.18±16.20 | 70.88±15.05 | 57.33±15.06 | 38.02±6.78 | 26.87±3.46 | 22.50±2.58 | 19.48±2.04 | 18.02±1.23 | 16.20±0.42 | 15.57±1.18 |
| $\mathcal{D}_9$ | ERM | 93.77±0.69 | 93.52±0.66 | 92.75±1.20 | 67.17±6.77 | 43.17±8.51 | 29.28±5.51 | 22.17±3.40 | 17.82±2.30 | 16.07±1.27 | 15.00±0.74 | 14.23±1.00 |
| | VREx | 51.79±32.78 | 52.77±31.84 | 53.92±31.61 | 47.82±25.99 | 31.95±11.56 | 22.70±2.55 | 18.27±0.93 | 16.28±0.63 | 14.02±0.25 | 14.47±0.91 | 13.33±0.66 |
| | IRM | 41.32±8.56 | 42.73±9.53 | 54.90±9.10 | 49.45±5.47 | 43.55±5.23 | 37.28±4.48 | 33.27±4.14 | 28.63±1.86 | 25.07±1.83 | 22.15±1.17 | 20.40±1.21 |
| | SD | 96.66±1.92 | 96.94±1.78 | 96.95±1.63 | 84.97±3.22 | 56.30±10.54 | 36.18±6.26 | 27.53±3.33 | 20.57±1.47 | 18.07±0.17 | 15.88±0.44 | 14.15±0.25 |
| | CORAL | 96.98±1.02 | 96.66±1.34 | 92.43±4.34 | 63.47±8.20 | 36.53±9.47 | 25.22±2.67 | 20.97±4.78 | 17.10±2.46 | 15.75±1.46 | 15.20±0.52 | 14.52±0.51 |
| | GroupDRO | 96.43±2.33 | 96.49±2.26 | 94.03±2.83 | 72.13±9.73 | 49.53±8.92 | 32.82±6.10 | 23.88±2.42 | 18.45±1.46 | 16.28±0.53 | 15.52±0.35 | 14.07±1.12 |
| | RSC | 98.04±0.23 | 97.53±0.39 | 97.93±0.45 | 86.55±2.30 | 53.83±3.57 | 32.85±0.82 | 23.63±3.04 | 18.20±1.94 | 16.02±1.18 | 14.42±0.53 | 13.35±0.98 |
| | DANN | 91.91±4.32 | 91.54±4.72 | 92.98±3.49 | 76.97±3.03 | 53.08±5.22 | 35.58±3.91 | 25.57±1.57 | 18.80±0.52 | 17.08±0.47 | 14.47±0.55 | 15.27±0.02 |
| | Mixup | 93.12±3.45 | 93.12±2.45 | 60.37±21.38 | 42.02±26.76 | 32.47±17.76 | 23.30±8.86 | 19.67±2.65 | 18.10±1.06 | 17.18±0.47 | 16.70±0.60 | 15.98±1.53 |
| | CAD | 91.11±5.83 | 91.44±5.54 | 89.23±6.90 | 72.55±14.35 | 48.60±9.63 | 31.42±6.51 | 25.45±4.27 | 20.60±3.37 | 18.25±2.33 | 16.08±1.71 | 14.22±1.88 |
| | IB-IRM | 50.49±18.28 | 50.21±17.76 | 57.20±19.54 | 46.30±17.34 | 35.55±11.15 | 27.73±6.54 | 23.12±3.63 | 19.72±1.60 | 17.57±1.52 | 16.57±0.77 | 16.03±0.34 |
| $\mathcal{D}_{10}$ | ERM | 95.04±1.51 | 94.83±1.74 | 94.77±2.08 | 73.92±6.85 | 45.35±7.38 | 29.13±5.65 | 22.23±3.32 | 18.08±2.02 | 16.23±1.05 | 14.92±0.81 | 14.70±0.53 |
| | VREx | 71.82±33.71 | 72.01±32.79 | 72.85±32.82 | 61.10±25.19 | 36.80±10.57 | 23.50±1.53 | 17.93±1.40 | 15.70±1.41 | 14.07±0.18 | 14.03±1.29 | 13.53±0.47 |
| | IRM | 41.32±8.56 | 42.73±9.53 | 54.90±9.10 | 49.45±5.47 | 43.55±5.23 | 37.28±4.48 | 33.27±4.14 | 28.63±1.86 | 25.07±1.83 | 22.15±1.17 | 20.40±1.21 |
| | SD | 96.84±2.06 | 96.91±1.76 | 96.73±1.58 | 84.48±4.47 | 54.75±11.21 | 35.23±7.93 | 26.00±5.51 | 19.43±1.84 | 17.13±1.65 | 15.33±0.19 | 14.38±0.10 |
| | CORAL | 97.08±1.05 | 96.87±1.41 | 92.73±4.58 | 61.57±5.54 | 33.35±5.02 | 21.73±3.28 | 18.80±1.98 | 16.00±1.06 | 15.30±0.83 | 14.48±0.71 | 14.82±0.27 |
| | GroupDRO | 97.69±0.14 | 97.86±0.14 | 97.20±0.36 | 82.75±2.71 | 58.05±3.52 | 38.75±2.79 | 26.97±2.38 | 20.18±1.25 | 16.48±0.43 | 15.22±0.60 | 14.82±0.64 |
| | RSC | 98.17±0.07 | 97.92±0.29 | 97.85±0.35 | 84.32±3.48 | 50.62±3.83 | 29.58±3.89 | 21.40±3.92 | 16.73±2.03 | 15.20±1.50 | 14.32±0.64 | 13.78±0.59 |
| | DANN | 91.91±4.32 | 91.54±4.72 | 92.98±3.49 | 76.97±3.03 | 53.08±5.22 | 35.58±3.91 | 25.57±1.57 | 18.80±0.52 | 17.08±0.47 | 14.47±0.55 | 15.27±0.02 |
| | Mixup | 96.03±0.77 | 95.80±1.34 | 54.08±12.49 | 29.55±9.47 | 22.45±3.83 | 18.52±2.16 | 18.45±1.00 | 17.50±0.21 | 17.00±0.57 | 16.42±0.94 | 16.50±1.06 |
| | CAD | 93.47±6.63 | 93.63±6.24 | 94.08±6.29 | 83.62±7.76 | 57.38±3.27 | 34.82±3.47 | 25.82±3.85 | 20.18±3.82 | 18.48±2.02 | 15.82±1.92 | 14.70±1.32 |
| | IB-IRM | 55.73±25.00 | 55.51±24.36 | 61.58±24.72 | 49.53±21.59 | 34.65±9.97 | 26.07±4.41 | 22.42±2.64 | 19.78±1.69 | 17.63±1.61 | 16.28±0.38 | 16.12±0.42 |

Table 7: Average accuracy of top-3 models of each algorithm at each shift degree of NOISYMNIST (EfficientNet-b0). This table shows the results on shift degrees from $\mathcal{D}_8$ to $\mathcal{D}_{10}$.

| | Algorithm | $\mathcal{D}_0$ | $\mathcal{D}_1$ | $\mathcal{D}_2$ | $\mathcal{D}_3$ | $\mathcal{D}_4$ | $\mathcal{D}_5$ | $\mathcal{D}_6$ | $\mathcal{D}_7$ | $\mathcal{D}_8$ | $\mathcal{D}_9$ | $\mathcal{D}_{10}$ |
|---|---|---|---|---|---|---|---|---|---|---|---|---|
| | ERM | 99.13±0.01 | 98.91±0.10 | 98.97±0.35 | 81.63±8.25 | 42.57±10.48 | 21.87±4.14 | 14.50±2.66 | 11.72±1.29 | 10.25±1.03 | 9.63±0.88 | 9.95±0.71 |
| | VREx | 99.12±0.05 | 98.91±0.15 | 97.50±1.38 | 66.30±6.76 | 32.75±7.59 | 18.10±2.58 | 13.23±0.21 | 10.80±0.27 | 10.27±0.25 | 9.12±1.19 | 9.42±0.35 |
| | IRM | 98.92±0.12 | 98.81±0.02 | 98.93±0.56 | 85.68±8.91 | 51.87±11.55 | 27.85±5.12 | 16.95±2.69 | 11.58±0.39 | 10.05±0.25 | 8.87±0.21 | 9.28±0.08 |
| | SD | 99.33±0.01 | 99.14±0.08 | 99.17±0.27 | 83.90±9.44 | 44.50±12.92 | 20.82±7.46 | 13.63±2.74 | 11.27±1.83 | 10.58±1.97 | 10.23±1.44 | 10.48±0.81 |
| | GroupDRO | 99.24±0.03 | 98.99±0.14 | 94.65±3.29 | 35.67±10.17 | 17.33±4.95 | 13.00±1.49 | 11.17±0.55 | 9.83±0.24 | 9.82±0.31 | 9.77±0.73 | 10.43±0.27 |
| | RSC | 99.19±0.03 | 98.99±0.17 | 96.08±2.45 | 52.77±15.85 | 26.20±9.72 | 14.78±5.71 | 11.52±3.46 | 9.73±2.24 | 9.05±1.01 | 8.80±1.13 | 9.72±0.97 |
| | Mixup | 99.33±0.03 | 99.04±0.08 | 98.47±0.60 | 77.77±9.61 | 31.47±4.98 | 16.05±1.73 | 11.48±0.96 | 9.65±0.25 | 8.83±0.50 | 8.82±0.25 | 9.27±0.65 |
| | CORAL | 99.26±0.03 | 99.09±0.09 | 96.40±2.63 | 51.12±11.53 | 23.93±5.58 | 14.07±3.12 | 11.45±1.35 | 10.50±1.14 | 9.58±0.39 | 9.57±0.29 | 10.28±0.65 |
| | MMD | 99.23±0.04 | 99.17±0.09 | 99.43±0.10 | 84.73±17.49 | 54.22±23.68 | 26.58±9.72 | 15.97±3.63 | 11.55±1.81 | 10.20±1.08 | 9.98±0.76 | 9.60±0.95 |
| | DANN | 99.02±0.09 | 98.67±0.13 | 83.62±3.68 | 26.32±3.29 | 16.87±2.90 | 12.52±2.54 | 11.50±1.62 | 10.58±1.57 | 9.93±1.32 | 9.63±0.93 | 9.98±0.40 |
| $\mathcal{D}_0$ | MTL | 98.97±0.01 | 98.75±0.08 | 97.80±0.82 | 70.78±11.97 | 37.83±11.64 | 21.12±4.96 | 13.63±1.48 | 11.17±0.86 | 10.17±0.68 | 9.85±0.39 | 9.87±0.54 |
| | SagNet | 99.22±0.04 | 99.03±0.06 | 97.53±1.07 | 61.53±18.81 | 30.97±8.93 | 18.00±3.70 | 12.87±2.05 | 11.02±1.06 | 9.58±0.69 | 9.35±0.40 | 10.05±0.33 |
| | ARM | 99.21±0.01 | 98.93±0.13 | 97.98±0.62 | 66.95±18.83 | 29.07±14.03 | 15.48±5.14 | 11.75±1.24 | 10.45±0.84 | 9.35±0.70 | 9.80±1.04 | 9.92±0.52 |
| | ANDMask | 98.94±0.05 | 98.68±0.21 | 95.68±2.73 | 53.00±10.52 | 24.70±3.69 | 15.75±2.93 | 11.55±2.23 | 10.25±1.27 | 9.07±1.00 | 8.58±0.78 | 9.45±0.23 |
| | SelfReg | 99.38±0.02 | 99.02±0.06 | 98.58±0.52 | 72.98±7.16 | 34.10±7.58 | 20.40±4.92 | 14.95±2.10 | 11.85±1.24 | 11.15±0.74 | 10.93±0.82 | 10.27±0.26 |
| | Fish | 99.41±0.01 | 98.96±0.17 | 92.73±6.40 | 42.05±21.44 | 16.35±5.73 | 12.35±3.78 | 11.45±2.26 | 10.17±1.08 | 9.17±0.31 | 9.23±0.40 | 9.80±0.14 |
| | IB-ERM | 99.23±0.02 | 99.07±0.21 | 95.48±5.01 | 68.73±26.69 | 30.65±15.53 | 16.42±3.17 | 11.85±0.92 | 10.48±0.25 | 9.42±1.19 | 9.92±0.80 | 9.95±1.10 |
| | IB-IRM | 99.06±0.11 | 98.88±0.21 | 98.43±1.01 | 87.67±14.03 | 66.00±23.24 | 36.30±13.70 | 18.98±4.58 | 12.33±1.75 | 10.92±0.53 | 9.38±0.62 | 9.72±0.66 |
| | CAD | 98.97±0.22 | 98.82±0.13 | 98.40±0.78 | 73.88±26.22 | 44.25±20.82 | 23.98±7.66 | 15.45±1.98 | 12.80±0.57 | 11.38±0.90 | 10.68±0.73 | 10.48±0.56 |
| | CondCAD | 99.24±0.01 | 99.02±0.22 | 94.30±5.89 | 48.55±18.28 | 20.67±5.20 | 13.28±2.17 | 11.45±0.99 | 9.98±0.90 | 9.48±0.55 | 9.50±0.54 | 9.90±0.43 |
| | EQRM | 99.17±0.03 | 98.89±0.14 | 96.17±1.48 | 54.80±11.46 | 25.35±7.32 | 14.92±3.14 | 10.62±0.59 | 9.83±0.41 | 8.65±0.88 | 8.65±0.36 | 9.43±0.22 |
| | ERM | 98.97±0.03 | 99.13±0.05 | 98.50±0.04 | 73.25±4.40 | 29.48±3.81 | 16.98±1.66 | 12.67±0.91 | 10.97±1.47 | 9.87±1.38 | 9.92±0.46 | 9.62±0.49 |
| | VREx | 98.99±0.13 | 99.11±0.02 | 98.12±1.12 | 70.73±16.33 | 39.83±13.67 | 21.25±4.72 | 13.43±1.35 | 10.68±0.58 | 10.07±0.47 | 9.38±1.14 | 9.73±0.55 |
| | IRM | 98.78±0.05 | 98.91±0.09 | 99.17±0.12 | 95.33±1.46 | 62.35±8.74 | 29.15±6.87 | 17.92±3.66 | 12.48±2.43 | 11.02±1.17 | 9.42±1.09 | 9.82±0.77 |
| | SD | 98.99±0.02 | 99.35±0.01 | 99.17±0.27 | 88.28±8.73 | 50.03±17.63 | 22.65±6.54 | 13.52±1.19 | 11.48±0.22 | 10.60±0.73 | 11.02±0.58 | 10.47±0.66 |
| | GroupDRO | 99.06±0.08 | 99.31±0.04 | 99.03±0.15 | 84.23±4.90 | 45.37±10.81 | 24.38±10.40 | 16.05±6.00 | 12.10±2.94 | 11.08±1.97 | 10.08±1.12 | 10.00±0.35 |
| | RSC | 99.03±0.15 | 99.25±0.07 | 94.02±5.34 | 57.10±16.06 | 31.87±9.29 | 18.73±3.64 | 13.93±1.86 | 11.43±1.06 | 9.93±0.41 | 9.73±0.48 | 10.50±0.43 |
| | Mixup | 98.98±0.04 | 99.33±0.05 | 98.32±0.44 | 61.98±18.40 | 26.47±13.21 | 14.15±5.37 | 11.85±2.83 | 10.52±1.58 | 9.70±1.06 | 9.23±0.40 | 9.83±0.19 |
| | CORAL | 99.16±0.10 | 99.26±0.04 | 96.85±1.70 | 49.00±10.56 | 23.28±4.31 | 14.08±1.72 | 10.12±1.44 | 9.57±0.45 | 9.10±0.36 | 9.18±0.62 | 9.90±0.88 |
| | MMD | 99.17±0.08 | 99.21±0.05 | 99.20±0.46 | 87.13±13.14 | 50.03±14.42 | 21.92±5.44 | 13.67±1.19 | 10.35±0.82 | 9.30±0.61 | 9.48±0.22 | 9.22±0.68 |
| | DANN | 98.84±0.05 | 99.01±0.09 | 94.22±2.50 | 39.18±11.31 | 22.28±6.18 | 13.95±2.83 | 11.20±1.38 | 9.83±0.37 | 9.28±0.21 | 9.02±0.21 | 9.77±0.13 |
| $\mathcal{D}_1$ | MTL | 98.76±0.14 | 98.93±0.03 | 90.80±5.96 | 50.92±20.84 | 26.73±10.98 | 15.58±3.94 | 11.37±1.33 | 9.62±0.31 | 9.35±0.65 | 9.17±0.37 | 9.10±0.66 |
| | SagNet | 98.48±0.16 | 99.24±0.05 | 98.77±0.54 | 74.23±13.24 | 34.50±10.69 | 18.48±5.26 | 12.85±1.82 | 10.72±1.05 | 9.50±0.40 | 9.43±0.22 | 9.82±0.02 |
| | ARM | 99.04±0.07 | 99.22±0.02 | 96.87±1.18 | 53.08±15.62 | 25.07±8.63 | 15.55±4.60 | 11.70±2.29 | 10.43±1.34 | 9.18±1.23 | 9.28±0.40 | 10.08±0.47 |
| | ANDMask | 98.82±0.05 | 98.98±0.02 | 94.90±4.82 | 54.38±22.48 | 25.03±10.41 | 13.35±4.30 | 10.27±1.05 | 9.32±0.64 | 9.07±0.24 | 8.82±0.15 | 9.30±0.46 |
| | SelfReg | 99.11±0.15 | 99.25±0.03 | 99.27±0.17 | 84.27±1.18 | 47.88±4.98 | 26.05±2.92 | 17.13±1.89 | 13.33±1.53 | 11.70±1.78 | 10.60±1.18 | 10.65±1.00 |
| | Fish | 99.25±0.18 | 99.42±0.04 | 64.22±3.43 | 16.62±2.93 | 12.43±0.72 | 9.62±0.22 | 9.85±0.08 | 9.38±0.10 | 9.00±0.00 | 8.97±0.02 | 9.70±0.00 |
| | IB-ERM | 99.04±0.16 | 99.27±0.05 | 93.37±4.28 | 43.78±11.41 | 19.45±1.35 | 13.33±1.54 | 11.47±0.93 | 10.10±0.39 | 9.87±0.88 | 9.90±0.82 | 10.45±0.63 |
| | IB-IRM | 98.93±0.19 | 98.96±0.16 | 99.20±0.08 | 95.18±2.25 | 67.47±7.87 | 33.05±2.76 | 19.32±1.45 | 13.07±0.84 | 11.42±1.12 | 10.47±1.00 | 10.37±0.29 |
| | CAD | 98.91±0.26 | 98.94±0.05 | 98.73±0.06 | 79.30±16.40 | 45.37±17.28 | 24.83±7.35 | 15.18±2.71 | 12.42±1.90 | 11.12±1.51 | 10.95±1.35 | 10.20±0.57 |
| | CondCAD | 99.09±0.20 | 99.28±0.15 | 93.63±5.54 | 45.82±18.35 | 21.52±5.13 | 13.65±2.15 | 11.47±0.99 | 10.40±0.72 | 9.62±0.51 | 9.25±0.60 | 9.90±0.43 |
| | EQRM | 98.83±0.08 | 98.00±0.12 | 95.00±1.16 | 40.72±2.60 | 20.78±1.21 | 14.38±0.68 | 12.03±1.17 | 10.88±0.24 | 9.42±0.46 | 9.62±0.65 | 10.00±0.43 |
| | ERM | 99.01±0.06 | 98.73±0.08 | 99.50±0.04 | 98.30±0.50 | 83.42±6.17 | 49.57±8.54 | 27.67±4.17 | 16.28±1.92 | 12.12±0.33 | 10.45±0.11 | 10.58±0.58 |
| | VREx | 98.88±0.05 | 99.00±0.08 | 99.17±0.08 | 86.00±4.30 | 44.58±9.62 | 22.68±4.17 | 14.50±1.00 | 11.27±0.29 | 10.12±0.08 | 10.10±0.48 | 9.97±0.14 |
| | IRM | 98.87±0.14 | 98.61±0.40 | 99.35±0.07 | 92.67±4.30 | 60.40±12.59 | 30.68±5.50 | 17.70±2.04 | 12.83±2.11 | 10.57±1.50 | 9.62±0.97 | 9.75±0.83 |
| | SD | 99.00±0.10 | 99.02±0.19 | 99.57±0.02 | 90.02±7.91 | 57.97±19.80 | 28.58±10.33 | 17.58±4.85 | 11.95±2.05 | 10.78±1.39 | 9.87±1.14 | 10.00±0.25 |
| | GroupDRO | 99.02±0.20 | 98.79±0.31 | 99.48±0.13 | 94.60±2.71 | 57.82±13.49 | 28.03±6.38 | 15.70±2.16 | 10.33±1.61 | 10.30±1.12 | 9.78±0.70 | 9.60±0.50 |
| | RSC | 98.99±0.10 | 99.00±0.11 | 99.33±0.06 | 95.47±1.32 | 68.25±8.14 | 33.32±7.40 | 17.63±2.84 | 11.67±1.26 | 9.50±0.70 | 9.88±0.66 | 10.13±0.98 |
| | Mixup | 99.01±0.11 | 98.78±0.04 | 99.42±0.20 | 86.22±6.67 | 32.37±8.58 | 14.12±3.47 | 10.73±1.28 | 9.18±0.81 | 8.68±0.38 | 8.67±0.55 | 9.38±0.45 |
| | CORAL | 98.84±0.10 | 98.75±0.15 | 99.38±0.06 | 95.07±3.09 | 68.82±16.78 | 34.43±12.48 | 18.78±4.67 | 12.52±1.74 | 11.03±1.25 | 9.70±0.33 | 10.07±0.45 |
| | MMD | 99.10±0.16 | 99.11±0.15 | 99.50±0.04 | 94.97±2.08 | 53.15±10.60 | 22.77±4.47 | 13.63±1.24 | 10.67±0.57 | 9.32±0.60 | 9.43±0.29 | 9.23±0.70 |
| | DANN | 97.97±0.76 | 98.17±0.76 | 99.17±0.17 | 88.37±6.14 | 56.65±13.71 | 29.27±5.16 | 17.70±1.06 | 12.53±1.64 | 10.58±2.03 | 9.80±2.37 | 10.20±1.83 |
| $\mathcal{D}_2$ | MTL | 98.48±0.15 | 98.44±0.15 | 99.05±0.04 | 84.62±8.83 | 41.18±9.18 | 22.77±1.90 | 11.53±0.76 | 10.08±0.62 | 10.55±0.62 | 10.40±0.15 | |
| | SagNet | 99.01±0.09 | 98.96±0.08 | 99.38±0.08 | 90.65±4.08 | 57.00±14.54 | 27.55±10.90 | 16.10±5.51 | 11.82±2.71 | 9.77±0.69 | 10.00±0.96 | 9.98±1.12 |
| | ARM | 98.76±0.09 | 98.62±0.25 | 99.33±0.02 | 93.03±5.30 | 60.60±20.22 | 31.40±17.68 | 17.83±2.50 | 12.47±3.20 | 10.38±1.26 | 9.53±0.42 | 10.05±0.36 |
| | ANDMask | 98.48±0.26 | 98.46±0.07 | 99.37±0.02 | 97.25±0.92 | 83.63±10.07 | 55.30±18.03 | 32.45±12.00 | 18.97±6.10 | 13.65±3.52 | 11.58±1.91 | 11.42±1.57 |
| | SelfReg | 98.89±0.17 | 99.01±0.07 | 99.50±0.04 | 90.15±4.09 | 55.75±13.07 | 29.82±8.57 | 18.23±3.06 | 12.98±1.76 | 11.43±1.84 | 10.85±1.28 | 10.40±1.10 |
| | Fish | 99.02±0.12 | 98.92±0.26 | 98.87±0.34 | 55.10±4.93 | 20.75±4.11 | 10.75±0.82 | 10.05±0.22 | 9.20±0.28 | 8.70±0.25 | 8.77±0.22 | 9.22±0.49 |
| | IB-ERM | 99.02±0.08 | 98.85±0.12 | 99.48±0.02 | 87.55±4.27 | 38.32±8.41 | 17.27±2.18 | 12.65±0.99 | 10.67±0.66 | 9.40±0.32 | 9.23±0.15 | 9.92±0.22 |
| | IB-IRM | 98.66±0.30 | 98.59±0.17 | 99.35±0.07 | 88.27±0.32 | 83.95±8.72 | 50.60±19.41 | 28.67±13.67 | 18.05±7.12 | 13.27±4.18 | 11.55±2.87 | 11.17±1.55 |
| | CAD | 98.62±0.03 | 98.48±0.24 | 99.17±0.26 | 95.85±1.74 | 61.85±17.78 | 26.47±10.46 | 15.58±4.25 | 12.17±2.01 | 10.38±0.97 | 9.82±0.84 | 10.40±0.49 |
| | CondCAD | 98.85±0.17 | 98.81±0.07 | 99.25±0.14 | 83.83±12.24 | 46.83±17.55 | 20.50±4.69 | 13.27±1.59 | 10.12±0.47 | 9.32±0.47 | 9.32±0.20 | 9.68±0.21 |
| | EQRM | 98.89±0.02 | 98.76±0.05 | 99.42±0.06 | 87.57±3.48 | 45.23±9.86 | 22.78±7.70 | 14.63±4.57 | 11.22±2.29 | 10.47±2.43 | 9.55±1.16 | 9.90±0.29 |
| | ERM | 98.81±0.21 | 98.63±0.25 | 99.30±0.23 | 99.02±0.06 | 93.68±2.92 | 67.93±16.83 | 43.30±22.04 | 27.33±14.08 | 17.33±6.94 | 14.62±5.31 | 12.37±3.10 |
| | VREx | 98.18±0.47 | 97.84±0.62 | 98.58±0.24 | 94.15±1.04 | 57.58±6.61 | 28.15±2.73 | 17.77±1.84 | 13.23±0.71 | 11.00±0.42 | 10.08±0.55 | 10.30±0.58 |
| | IRM | 98.54±0.26 | 98.42±0.31 | 99.10±0.11 | 98.65±0.12 | 91.73±2.28 | 70.30±8.89 | 42.47±8.03 | 24.42±4.15 | 16.92±2.43 | 13.42±2.15 | 12.67±1.41 |
| | SD | 98.81±0.16 | 98.72±0.04 | 99.27±0.17 | 99.05±0.07 | 92.28±5.97 | 62.48±22.33 | 37.13±22.97 | 22.48±12.60 | 14.75±5.07 | 12.22±3.26 | 11.42±1.93 |
| | GroupDRO | 98.47±0.18 | 98.37±0.27 | 98.87±0.08 | 97.58±0.65 | 78.58±12.19 | 45.70±22.64 | 24.68±13.67 | 15.43±5.06 | 11.33±1.74 | 10.47±1.34 | 10.32±0.61 |
| | RSC | 98.63±0.30 | 98.54±0.29 | 98.95±0.19 | 97.78±0.73 | 87.85±4.55 | 50.67±14.75 | 33.55±12.79 | 19.42±5.63 | 12.92±1.90 | 11.32±0.90 | 10.57±0.31 |
| | Mixup | 97.36±0.88 | 97.36±0.76 | 98.08±0.19 | 96.98±0.22 | 76.62±9.29 | 32.73±8.17 | 16.03±1.76 | 12.38±1.55 | 11.42±1.71 | 11.10±1.67 | 11.12±1.50 |
| | CORAL | 98.11±0.75 | 97.98±1.00 | 98.97±0.48 | 97.68±0.78 | 77.10±15.31 | 38.03±14.01 | 20.30±6.47 | 12.92±2.31 | 10.50±1.10 | 9.58±0.70 | 10.10±0.16 |
| | MMD | 98.54±0.14 | 98.52±0.28 | 99.15±0.25 | 98.78±0.05 | 85.13±4.17 | 44.35±4.20 | 21.98±1.41 | 14.45±1.31 | 11.55±0.82 | 10.50±0.76 | 9.87±0.33 |
| | DANN | 96.14±1.73 | 96.02±1.71 | 98.15±1.10 | 97.00±0.47 | 83.50±6.03 | 51.80±15.73 | 31.85±15.24 | 19.98±9.30 | 14.00±5.57 | 12.07±4.02 | 11.20±2.02 |
| $\mathcal{D}_3$ | MTL | 97.52±0.35 | 97.37±0.36 | 98.22±0.10 | 95.62±0.77 | 70.37±7.48 | 35.65±6.79 | 20.52±5.46 | 13.78±3.18 | 11.83±2.23 | 11.30±1.57 | 11.32±1.11 |
| | SagNet | 97.98±1.10 | 98.17±0.66 | 98.83±0.49 | 97.17±0.21 | 76.48±5.76 | 37.58±1.21 | 19.70±1.67 | 14.00±0.90 | 11.90±0.47 | 10.72±0.59 | 10.68±0.87 |
| | ARM | 98.52±0.20 | 98.28±0.35 | 99.02±0.20 | 98.10±0.25 | 87.32±3.29 | 54.25±2.86 | 27.23±0.72 | 17.27±0.90 | 13.20±0.90 | 11.23±0.95 | 11.08±0.59 |
| | ANDMask | 98.46±0.30 | 98.35±0.38 | 99.17±0.13 | 98.35±0.15 | 91.45±3.12 | 62.58±9.74 | 34.28±8.99 | 20.30±3.57 | 13.78±1.25 | 11.42±0.70 | 11.20±0.22 |
| | SelfReg | 98.69±0.22 | 98.73±0.30 | 99.20±0.25 | 98.13±0.44 | 83.92±5.86 | 50.67±14.75 | 28.00±11.71 | 17.20±4.76 | 12.35±2.75 | 11.00±1.23 | 11.42±1.05 |
| | Fish | 99.10±0.23 | 98.76±0.04 | 98.72±0.49 | 62.72±6.25 | 21.90±4.48 | 13.10±3.35 | 11.62±2.15 | 9.97±1.25 | 8.97±0.51 | 9.07±0.56 | 9.42±0.62 |
| | IB-ERM | 98.72±0.09 | 98.56±0.04 | 99.18±0.15 | 98.62±0.17 | 82.73±7.62 | 39.30±20.76 | 20.93±11.63 | 14.47±5.82 | 11.58±3.10 | 10.53±1.55 | 10.32±1.00 |
| | IB-IRM | 98.83±0.34 | 98.62±0.17 | 99.28±0.08 | 98.45±0.23 | 89.72±4.37 | 58.52±12.68 | 31.22±11.51 | 18.92±6.39 | 13.87±3.75 | 11.50±2.90 | 11.32±1.44 |
| | CAD | 98.15±0.32 | 98.28±0.24 | 98.95±0.28 | 97.50±1.78 | 77.15±18.43 | 43.25±17.80 | 22.73±7.54 | 15.22±1.70 | 11.47±1.06 | 10.48±1.11 | 10.70±0.86 |
| | CondCAD | 98.57±0.15 | 98.70±0.16 | 99.00±0.37 | 97.90±0.43 | 73.25±12.54 | 33.08±14.58 | 17.07±5.61 | 10.98±1.69 | 9.23±0.43 | 9.12±0.06 | 9.73±0.02 |
| | EQRM | 98.40±0.48 | 98.57±0.31 | 98.92±0.44 | 97.60±0.32 | 77.97±4.12 | 40.88±5.23 | 22.08±2.52 | 14.55±0.47 | 11.78±1.19 | 10.78±1.08 | 10.65±0.55 |

Table 8: Average accuracy of top-3 models of each algorithm at each shift degree of NOISYMNIST (ResNet-50). This table shows the results on shift degrees from $\mathcal{D}_0$ to $\mathcal{D}_3$.

| | Algorithm | $\mathcal{D}_0$ | $\mathcal{D}_1$ | $\mathcal{D}_2$ | $\mathcal{D}_3$ | $\mathcal{D}_4$ | $\mathcal{D}_5$ | $\mathcal{D}_6$ | $\mathcal{D}_7$ | $\mathcal{D}_8$ | $\mathcal{D}_9$ | $\mathcal{D}_{10}$ |
|---|---|---|---|---|---|---|---|---|---|---|---|---|
| | ERM | 98.37±0.16 | 98.14±0.11 | 99.07±0.17 | 98.68±0.21 | 97.40±0.29 | 84.00±5.48 | 54.77±14.62 | 31.57±11.76 | 19.48±6.04 | 15.30±4.91 | 12.73±2.86 |
| | VREx | 98.38±0.15 | 98.22±0.08 | 98.68±0.14 | 91.32±2.54 | 64.55±1.66 | 32.10±2.34 | 17.37±0.94 | 11.62±0.61 | 9.95±0.35 | 9.27±0.26 | 9.95±0.32 |
| | IRM | 97.27±1.13 | 97.12±0.76 | 98.70±0.47 | 98.05±0.47 | 95.73±0.84 | 82.43±1.30 | 56.62±6.82 | 34.08±7.33 | 25.03±4.69 | 19.55±4.42 | 16.12±2.37 |
| | SD | 98.29±0.51 | 98.41±0.23 | 99.03±0.39 | 98.67±0.35 | 97.78±0.36 | 92.42±2.05 | 76.53±6.04 | 53.52±10.72 | 35.67±11.15 | 26.28±7.54 | 21.60±5.93 |
| | GroupDRO | 98.71±0.15 | 98.69±0.25 | 98.88±0.10 | 96.93±1.40 | 82.40±8.96 | 53.15±17.74 | 30.50±10.43 | 18.50±3.10 | 13.57±1.53 | 11.85±1.19 | 11.02±0.70 |
| | RSC | 98.34±0.10 | 98.34±0.06 | 98.83±0.06 | 97.13±0.88 | 88.43±3.98 | 64.63±8.63 | 39.23±8.00 | 23.00±3.44 | 15.67±2.01 | 12.58±0.90 | 11.32±0.91 |
| | Mixup | 95.81±1.38 | 95.72±1.59 | 97.03±1.30 | 96.30±1.08 | 85.38±3.68 | 49.17±16.25 | 26.63±13.23 | 17.25±5.34 | 13.60±1.39 | 12.38±0.66 | 12.27±1.28 |
| | CORAL | 97.46±0.89 | 97.13±1.01 | 98.75±0.47 | 97.48±1.02 | 86.88±3.00 | 53.15±8.47 | 30.73±8.29 | 19.23±6.72 | 14.33±4.33 | 12.57±3.58 | 11.62±2.01 |
| | MMD | 97.41±0.69 | 97.16±0.57 | 98.70±0.29 | 98.48±0.18 | 97.05±0.40 | 84.88±8.91 | 53.12±14.55 | 28.33±11.32 | 17.10±6.14 | 13.00±4.14 | 11.53±2.24 |
| | DANN | 94.81±2.41 | 94.44±2.48 | 97.58±1.02 | 96.32±0.95 | 87.07±3.13 | 60.45±12.19 | 37.88±11.49 | 22.40±8.63 | 15.07±5.89 | 12.15±3.89 | 11.37±2.22 |
| $\mathcal{D}_4$ | MTL | 97.25±0.52 | 97.30±0.43 | 98.13±0.16 | 94.65±2.01 | 73.75±3.06 | 39.73±1.24 | 23.83±2.44 | 15.43±1.32 | 12.32±1.70 | 11.48±1.31 | 11.05±1.49 |
| | SagNet | 96.44±1.56 | 96.52±1.82 | 98.07±0.59 | 96.55±0.60 | 85.03±0.83 | 53.45±11.14 | 30.58±9.38 | 19.70±4.88 | 14.45±2.27 | 12.15±1.30 | 11.42±1.04 |
| | ARM | 98.52±0.20 | 98.28±0.35 | 99.02±0.20 | 98.10±0.25 | 87.32±3.29 | 54.25±2.86 | 27.23±0.72 | 17.27±0.90 | 13.20±0.90 | 11.23±0.95 | 11.08±0.59 |
| | ANDMask | 98.22±0.41 | 97.97±0.37 | 99.03±0.27 | 98.33±0.16 | 92.32±2.23 | 65.93±7.44 | 38.07±7.73 | 22.68±3.42 | 14.73±1.79 | 12.42±1.29 | 11.62±0.69 |
| | SelfReg | 98.23±0.58 | 98.37±0.41 | 98.77±0.43 | 97.35±0.60 | 89.33±1.55 | 67.37±7.34 | 40.58±11.82 | 23.10±8.18 | 16.27±4.34 | 13.82±2.82 | 12.88±1.49 |
| | Fish | 98.55±0.16 | 98.62±0.13 | 92.95±1.76 | 45.12±5.76 | 30.58±2.73 | 19.83±0.49 | 15.10±0.54 | 12.30±0.54 | 10.58±0.72 | 9.55±0.39 | 9.92±0.12 |
| | IB-ERM | 98.66±0.03 | 98.62±0.03 | 98.95±0.32 | 97.65±0.91 | 88.62±3.53 | 60.00±8.15 | 32.52±8.98 | 20.88±7.00 | 15.43±4.89 | 12.70±3.01 | 11.98±2.09 |
| | IB-IRM | 97.86±1.05 | 97.77±0.86 | 98.97±0.27 | 97.98±0.54 | 90.97±2.92 | 59.50±11.99 | 30.15±12.28 | 17.12±7.70 | 12.98±4.41 | 11.90±2.75 | 11.35±1.51 |
| | CAD | 98.53±0.59 | 98.44±0.33 | 98.93±0.29 | 97.38±1.71 | 78.80±19.61 | 50.58±22.00 | 30.45±13.14 | 19.60±6.37 | 14.48±3.64 | 12.05±2.04 | 10.60±0.46 |
| | CondCAD | 93.67±6.99 | 93.49±7.29 | 95.57±4.87 | 94.63±5.04 | 74.38±11.45 | 35.77±12.74 | 18.47±4.90 | 11.72±1.62 | 10.25±1.44 | 9.88±1.04 | 10.43±0.97 |
| | EQRM | 96.37±2.06 | 96.13±2.41 | 97.95±0.90 | 96.30±1.31 | 84.77±4.03 | 55.28±11.65 | 31.73±12.87 | 21.50±9.95 | 15.82±6.69 | 13.32±3.49 | 12.28±2.25 |
| | ERM | 97.12±1.11 | 96.79±1.11 | 98.48±0.48 | 97.98±0.68 | 96.47±0.96 | 89.32±1.27 | 68.78±4.04 | 43.47±5.49 | 28.07±4.32 | 22.75±3.80 | 18.62±3.56 |
| | VREx | 85.63±17.89 | 85.25±18.28 | 87.48±15.83 | 82.23±11.00 | 62.85±2.45 | 36.35±3.96 | 20.23±3.67 | 13.42±3.08 | 11.25±2.16 | 10.05±1.26 | 9.63±0.13 |
| | IRM | 97.27±1.13 | 97.12±0.76 | 98.70±0.47 | 98.05±0.47 | 95.73±0.84 | 82.43±1.30 | 56.62±6.82 | 34.08±7.33 | 25.03±4.69 | 19.55±4.42 | 16.12±2.37 |
| | SD | 98.29±0.51 | 98.41±0.23 | 99.03±0.39 | 98.67±0.35 | 97.78±0.36 | 92.42±2.05 | 76.53±6.04 | 53.52±10.72 | 35.67±11.15 | 26.28±7.54 | 21.60±5.93 |
| | GroupDRO | 98.59±0.27 | 98.49±0.37 | 98.65±0.43 | 94.53±3.49 | 80.80±10.45 | 57.02±14.28 | 34.85±6.29 | 20.90±1.39 | 15.42±1.84 | 12.65±1.80 | 11.33±0.88 |
| | RSC | 98.34±0.10 | 98.34±0.06 | 98.83±0.06 | 97.13±0.88 | 88.43±3.98 | 64.63±8.63 | 39.23±8.00 | 23.00±3.44 | 15.67±2.01 | 12.58±0.90 | 11.32±0.91 |
| | Mixup | 95.81±1.38 | 95.72±1.59 | 97.03±1.30 | 96.30±1.08 | 85.38±3.68 | 49.17±16.25 | 26.63±13.23 | 17.25±5.34 | 13.60±1.39 | 12.38±0.66 | 12.27±1.28 |
| | CORAL | 97.46±0.89 | 97.13±1.01 | 98.75±0.47 | 97.48±1.02 | 86.88±3.00 | 53.15±8.47 | 30.73±8.29 | 19.23±6.72 | 14.33±4.33 | 12.57±3.58 | 11.62±2.01 |
| | MMD | 97.41±0.69 | 97.16±0.57 | 98.70±0.29 | 98.48±0.18 | 97.05±0.40 | 84.88±8.91 | 53.12±14.55 | 28.33±11.32 | 17.10±6.14 | 13.00±4.14 | 11.53±2.24 |
| | DANN | 89.14±6.15 | 89.07±5.72 | 93.37±5.09 | 92.50±4.54 | 86.58±3.65 | 66.92±6.54 | 43.95±7.80 | 24.70±7.43 | 15.75±5.49 | 12.38±3.74 | 11.35±2.23 |
| $\mathcal{D}_5$ | MTL | 95.71±2.59 | 95.80±2.46 | 97.12±1.57 | 91.73±6.09 | 72.57±4.52 | 43.02±3.74 | 27.03±4.96 | 17.67±2.79 | 13.72±1.46 | 12.58±0.33 | 12.00±0.15 |
| | SagNet | 85.48±15.58 | 85.24±15.54 | 88.12±14.02 | 86.55±13.54 | 78.62±9.48 | 59.20±3.03 | 38.30±1.70 | 25.00±2.62 | 17.10±1.50 | 13.98±1.35 | 12.42±0.38 |
| | ARM | 98.52±0.20 | 98.28±0.35 | 99.02±0.20 | 98.10±0.25 | 87.32±3.29 | 54.25±2.86 | 27.23±0.72 | 17.27±0.90 | 13.20±0.90 | 11.23±0.95 | 11.08±0.59 |
| | ANDMask | 98.00±0.58 | 97.87±0.42 | 98.72±0.51 | 98.20±0.27 | 90.80±3.66 | 66.30±7.08 | 39.55±5.89 | 22.63±3.49 | 14.50±2.11 | 12.33±1.39 | 11.48±0.84 |
| | SelfReg | 98.38±0.39 | 98.35±0.44 | 98.95±0.22 | 96.77±1.43 | 88.23±3.10 | 69.12±4.89 | 45.02±6.07 | 26.43±4.52 | 17.88±2.73 | 14.20±2.51 | 13.18±1.14 |
| | Fish | 98.55±0.16 | 98.62±0.13 | 92.95±1.76 | 45.12±5.76 | 30.58±2.73 | 19.83±0.49 | 15.10±0.54 | 12.30±0.54 | 10.58±0.72 | 9.55±0.39 | 9.92±0.12 |
| | IB-ERM | 98.01±0.94 | 97.99±0.90 | 98.32±1.22 | 90.85±9.80 | 82.18±10.74 | 61.58±5.99 | 40.58±2.84 | 27.78±3.92 | 21.10±4.18 | 17.32±4.35 | 15.08±3.07 |
| | IB-IRM | 97.52±1.53 | 97.67±1.01 | 98.63±0.74 | 97.73±0.84 | 90.20±3.89 | 61.00±10.69 | 31.75±11.22 | 18.30±6.86 | 13.73±3.84 | 12.07±2.58 | 11.37±1.49 |
| | CAD | 98.53±0.59 | 98.44±0.33 | 98.93±0.29 | 97.38±1.71 | 78.80±19.61 | 50.58±22.00 | 30.45±13.14 | 19.60±6.37 | 14.48±3.64 | 12.05±2.04 | 10.60±0.46 |
| | CondCAD | 98.57±0.15 | 98.58±0.17 | 99.02±0.37 | 96.45±2.48 | 72.52±13.29 | 36.80±12.27 | 19.67±4.88 | 12.32±2.02 | 10.10±1.24 | 9.78±0.90 | 10.35±0.85 |
| | EQRM | 96.37±2.06 | 96.13±2.41 | 97.95±0.90 | 96.30±1.31 | 84.77±4.03 | 55.28±11.65 | 31.73±12.87 | 21.50±9.95 | 15.82±6.69 | 13.32±3.49 | 12.28±2.25 |
| | ERM | 95.94±0.76 | 95.59±0.73 | 97.83±0.63 | 97.23±0.38 | 96.02±0.37 | 89.30±1.24 | 70.32±6.07 | 46.25±8.15 | 32.00±6.51 | 24.93±4.60 | 20.15±3.41 |
| | VREx | 62.72±27.49 | 62.53±26.93 | 67.33±24.38 | 65.83±24.61 | 48.97±14.56 | 31.92±7.71 | 21.58±2.68 | 16.00±1.49 | 13.87±1.94 | 11.75±1.41 | 10.62±1.58 |
| | IRM | 97.27±1.13 | 97.12±0.76 | 98.70±0.47 | 98.05±0.47 | 95.73±0.84 | 82.43±1.30 | 56.62±6.82 | 34.08±7.33 | 25.03±4.69 | 19.55±4.42 | 16.12±2.37 |
| | SD | 98.15±0.33 | 98.30±0.08 | 98.93±0.28 | 98.47±0.08 | 97.63±0.25 | 92.28±2.12 | 78.00±4.47 | 54.68±9.31 | 36.23±10.45 | 25.15±9.00 | 20.62±7.19 |
| | GroupDRO | 98.36±0.17 | 98.21±0.12 | 98.45±0.35 | 93.18±3.78 | 78.07±11.98 | 56.45±14.68 | 35.57±5.75 | 22.13±0.39 | 16.27±2.29 | 13.43±2.28 | 11.72±1.19 |
| | RSC | 97.35±1.36 | 97.44±1.21 | 97.60±1.80 | 93.13±5.99 | 82.32±10.75 | 61.95±11.44 | 40.32±7.02 | 24.80±1.38 | 17.52±2.18 | 13.92±1.67 | 13.20±2.11 |
| | Mixup | 96.98±2.18 | 96.77±2.33 | 97.68±1.76 | 92.53±1.63 | 70.40±13.90 | 43.42±19.94 | 27.30±12.76 | 16.63±5.78 | 12.42±2.22 | 10.83±1.36 | 11.23±1.37 |
| | CORAL | 97.46±0.89 | 97.13±1.01 | 98.75±0.47 | 97.48±1.02 | 86.88±3.00 | 53.15±8.47 | 30.73±8.29 | 19.23±6.72 | 14.33±4.33 | 12.57±3.58 | 11.62±2.01 |
| | MMD | 97.41±0.69 | 97.16±0.57 | 98.70±0.29 | 98.48±0.18 | 97.05±0.40 | 84.88±8.91 | 53.12±14.55 | 28.33±11.32 | 17.10±6.14 | 13.00±4.14 | 11.53±2.24 |
| | DANN | 88.35±5.91 | 88.32±5.53 | 92.88±4.85 | 90.75±4.14 | 81.87±8.01 | 64.17±9.83 | 45.83±5.62 | 27.77±4.99 | 18.45±4.21 | 13.70±3.09 | 11.95±1.94 |
| $\mathcal{D}_6$ | MTL | 95.91±2.71 | 95.85±2.49 | 97.05±1.53 | 85.60±7.21 | 62.02±11.61 | 40.50±6.66 | 28.10±3.74 | 18.45±1.99 | 14.42±0.48 | 12.43±0.53 | 11.90±0.18 |
| | SagNet | 85.48±15.58 | 85.24±15.54 | 88.12±14.02 | 86.55±13.54 | 78.62±9.48 | 59.20±3.03 | 38.30±1.70 | 25.00±2.62 | 17.10±1.50 | 13.98±1.35 | 12.42±0.38 |
| | ARM | 98.52±0.20 | 98.28±0.35 | 99.02±0.20 | 98.10±0.25 | 87.32±3.29 | 54.25±2.86 | 27.23±0.72 | 17.27±0.90 | 13.20±0.90 | 11.23±0.95 | 11.08±0.59 |
| | ANDMask | 98.00±0.58 | 97.87±0.42 | 98.72±0.51 | 98.20±0.27 | 90.80±3.66 | 66.30±7.08 | 39.55±5.89 | 22.63±3.49 | 14.50±2.11 | 12.33±1.39 | 11.48±0.84 |
| | SelfReg | 98.38±0.39 | 98.35±0.44 | 98.95±0.22 | 96.77±1.43 | 88.23±3.10 | 69.12±4.89 | 45.02±6.07 | 26.43±4.52 | 17.88±2.73 | 14.20±2.51 | 13.18±1.14 |
| | Fish | 98.67±0.24 | 98.67±0.05 | 68.80±28.91 | 29.18±8.20 | 24.35±3.61 | 18.68±0.83 | 15.58±0.18 | 12.73±0.13 | 11.95±0.58 | 11.03±0.96 | 11.28±0.92 |
| | IB-ERM | 98.01±0.94 | 97.99±0.90 | 98.32±1.22 | 90.85±9.80 | 82.18±10.74 | 61.58±5.99 | 40.58±2.84 | 27.78±3.92 | 21.10±4.18 | 17.32±4.35 | 15.08±3.07 |
| | IB-IRM | 98.68±0.31 | 98.60±0.23 | 99.20±0.11 | 96.85±2.11 | 85.92±9.53 | 57.72±13.42 | 33.20±10.11 | 19.65±5.87 | 15.07±2.89 | 12.87±1.94 | 11.78±1.11 |
| | CAD | 98.11±1.19 | 97.85±1.17 | 98.73±0.56 | 95.28±4.66 | 72.60±28.35 | 50.03±22.75 | 31.02±12.43 | 19.05±6.98 | 14.38±3.75 | 11.93±2.17 | 10.53±0.55 |
| | CondCAD | 78.68±27.89 | 78.93±27.44 | 79.95±26.38 | 74.92±25.07 | 54.05±29.10 | 33.53±15.33 | 20.73±3.58 | 13.87±0.73 | 12.20±2.14 | 11.20±2.18 | 11.32±1.31 |
| | EQRM | 97.99±0.69 | 98.06±0.53 | 98.65±0.67 | 96.53±1.02 | 80.18±9.54 | 52.88±13.74 | 32.87±11.95 | 22.73±9.11 | 16.17±6.51 | 13.02±3.57 | 12.18±2.28 |
| | ERM | 95.94±0.76 | 95.59±0.73 | 97.83±0.63 | 97.23±0.38 | 96.02±0.37 | 89.30±1.24 | 70.32±6.07 | 46.25±8.15 | 32.00±6.51 | 24.93±4.60 | 20.15±3.41 |
| | VREx | 62.72±27.49 | 62.53±26.93 | 67.33±24.38 | 65.83±24.61 | 48.97±14.56 | 31.92±7.71 | 21.58±2.68 | 16.00±1.49 | 13.87±1.94 | 11.75±1.41 | 10.62±1.58 |
| | IRM | 90.46±9.04 | 90.41±8.86 | 94.28±5.94 | 93.35±6.23 | 91.32±6.83 | 77.79±4.87 | 55.58±7.52 | 35.78±6.45 | 26.83±3.35 | 20.57±3.87 | 16.68±2.02 |
| | SD | 98.15±0.33 | 98.30±0.08 | 98.93±0.28 | 98.47±0.08 | 97.63±0.25 | 92.28±2.12 | 78.00±4.47 | 54.68±9.31 | 36.23±10.45 | 25.15±9.00 | 20.62±7.19 |
| | GroupDRO | 98.36±0.17 | 98.21±0.12 | 98.45±0.35 | 93.18±3.78 | 78.07±11.98 | 56.45±14.68 | 35.57±5.75 | 22.13±0.39 | 16.27±2.29 | 13.43±2.28 | 11.72±1.19 |
| | RSC | 97.35±1.36 | 97.44±1.21 | 97.60±1.80 | 93.13±5.99 | 82.32±10.75 | 61.95±11.44 | 40.32±7.02 | 24.80±1.38 | 17.52±2.18 | 13.92±1.67 | 13.20±2.11 |
| | Mixup | 96.38±2.01 | 96.26±2.14 | 97.33±1.55 | 94.95±1.45 | 75.68±14.36 | 44.25±19.50 | 26.93±13.03 | 17.73±5.02 | 13.40±1.52 | 11.67±0.81 | 11.47±1.20 |
| | CORAL | 97.38±0.78 | 97.13±1.01 | 98.58±0.25 | 95.90±1.48 | 77.98±9.61 | 48.02±11.78 | 30.32±8.59 | 19.95±6.19 | 14.67±4.10 | 13.42±3.03 | 11.87±1.84 |
| | MMD | 97.66±0.91 | 97.53±0.86 | 98.88±0.44 | 98.52±0.19 | 96.53±1.05 | 82.18±12.57 | 52.22±15.53 | 29.02±10.62 | 17.55±5.72 | 13.35±3.86 | 11.65±2.16 |
| | DANN | 88.68±4.57 | 88.69±4.30 | 93.23±3.24 | 91.37±3.50 | 84.20±5.07 | 63.67±8.53 | 43.65±6.94 | 29.25±3.80 | 21.67±1.29 | 17.07±1.23 | 15.00±1.67 |
| $\mathcal{D}_7$ | MTL | 95.91±2.71 | 95.85±2.49 | 97.05±1.53 | 85.60±7.21 | 62.02±11.61 | 40.50±6.66 | 28.10±3.74 | 18.45±1.99 | 14.42±0.48 | 12.43±0.53 | 11.90±0.18 |
| | SagNet | 85.48±15.58 | 85.24±15.54 | 88.12±14.02 | 86.55±13.54 | 78.62±9.48 | 59.20±3.03 | 38.30±1.70 | 25.00±2.62 | 17.10±1.50 | 13.98±1.35 | 12.42±0.38 |
| | ARM | 89.21±13.16 | 88.94±13.07 | 90.05±12.76 | 87.97±14.26 | 76.70±13.07 | 48.85±6.66 | 26.87±1.05 | 17.67±0.65 | 13.37±0.95 | 11.22±0.94 | 11.17±0.56 |
| | ANDMask | 98.29±0.39 | 98.13±0.38 | 98.90±0.32 | 96.08±3.21 | 84.35±12.33 | 59.53±15.31 | 36.72±9.49 | 22.75±3.33 | 15.37±0.96 | 12.87±0.81 | 12.05±0.48 |
| | SelfReg | 98.38±0.39 | 98.35±0.44 | 98.95±0.22 | 96.77±1.43 | 88.23±3.10 | 69.12±4.89 | 45.02±6.07 | 26.43±4.52 | 17.88±2.73 | 14.20±2.51 | 13.18±1.14 |
| | Fish | 98.87±0.04 | 98.88±0.19 | 70.88±10.91 | 25.97±4.66 | 22.37±3.41 | 17.63±1.45 | 14.92±0.61 | 12.95±0.15 | 11.58±0.25 | 11.12±0.56 | 11.47±0.15 |
| | IB-ERM | 98.01±0.94 | 97.99±0.90 | 98.32±1.22 | 90.85±9.80 | 82.18±10.74 | 61.58±5.99 | 40.58±2.84 | 27.78±3.92 | 21.10±4.18 | 17.32±4.35 | 15.08±3.07 |
| | IB-IRM | 42.53±39.47 | 42.35±39.62 | 43.15±39.69 | 43.00±39.39 | 41.37±37.64 | 36.33±28.12 | 27.37±14.24 | 21.90±4.28 | 19.03±0.20 | 16.00±0.86 | 12.25±0.84 |
| | CAD | 98.53±0.59 | 98.44±0.33 | 98.93±0.29 | 97.38±1.71 | 78.80±19.61 | 50.58±22.00 | 30.45±13.14 | 19.60±6.37 | 14.48±3.64 | 12.05±2.04 | 10.60±0.46 |
| | CondCAD | 78.68±27.89 | 78.93±27.44 | 79.95±26.38 | 74.92±25.07 | 54.05±29.10 | 33.53±15.33 | 20.73±3.58 | 13.87±0.73 | 12.20±2.14 | 11.20±2.18 | 11.32±1.31 |
| | EQRM | 97.60±0.61 | 97.85±0.52 | 97.90±0.91 | 84.07±16.71 | 65.35±21.36 | 45.43±19.50 | 31.47±12.99 | 22.83±9.03 | 17.53±5.44 | 14.05±2.74 | 12.90±1.72 |

Table 9: Average accuracy of top-3 models of each algorithm at each shift degree of NOISYMNIST (ResNet-50). This table shows the results on shift degrees from $\mathcal{D}_4$ to $\mathcal{D}_7$.

| | Algorithm | $\mathcal{D}_0$ | $\mathcal{D}_1$ | $\mathcal{D}_2$ | $\mathcal{D}_3$ | $\mathcal{D}_4$ | $\mathcal{D}_5$ | $\mathcal{D}_6$ | $\mathcal{D}_7$ | $\mathcal{D}_8$ | $\mathcal{D}_9$ | $\mathcal{D}_{10}$ |
|---|---|---|---|---|---|---|---|---|---|---|---|---|
| $\mathcal{D}_8$ | ERM | 95.94±0.76 | 95.59±0.73 | 97.83±0.63 | 97.23±0.38 | 96.02±0.37 | 89.30±1.24 | 70.32±6.07 | 46.25±8.15 | 32.00±6.51 | 24.93±4.60 | 20.15±3.41 |
| | VREx | 40.12±14.30 | 40.11±13.66 | 46.35±13.33 | 45.43±15.15 | 40.27±14.18 | 29.22±8.90 | 20.10±3.90 | 16.00±1.49 | 15.07±0.70 | 12.37±0.77 | 11.82±1.68 |
| | IRM | 90.46±9.04 | 90.41±8.86 | 94.28±5.94 | 93.35±6.23 | 91.32±6.83 | 79.25±4.87 | 55.58±7.52 | 35.78±6.45 | 26.83±3.35 | 20.57±3.87 | 16.68±2.02 |
| | SD | 98.16±0.33 | 98.37±0.17 | 98.72±0.24 | 98.00±0.60 | 95.30±3.26 | 87.68±7.56 | 73.22±10.27 | 53.57±10.66 | 38.05±8.37 | 27.85±5.66 | 22.87±4.41 |
| | GroupDRO | 98.47±0.32 | 98.41±0.41 | 98.48±0.39 | 92.05±2.20 | 72.15±3.63 | 46.48±0.70 | 30.87±0.92 | 21.10±1.57 | 17.13±1.08 | 14.38±0.97 | 12.28±0.49 |
| | RSC | 96.03±1.58 | 96.21±1.58 | 96.77±1.57 | 89.77±4.84 | 70.20±11.23 | 49.10±9.13 | 33.02±3.39 | 23.08±1.05 | 18.58±0.76 | 14.98±0.88 | 14.08±1.46 |
| | Mixup | 95.81±1.38 | 95.72±1.59 | 97.03±1.30 | 96.30±1.08 | 85.38±3.68 | 49.17±16.25 | 26.63±13.23 | 17.25±5.34 | 13.60±1.39 | 12.38±0.66 | 12.27±1.28 |
| | CORAL | 97.54±0.99 | 97.28±1.08 | 97.82±1.71 | 84.77±15.57 | 60.82±21.65 | 37.47±19.23 | 25.58±11.93 | 18.82±6.99 | 15.33±3.62 | 13.38±3.12 | 12.38±1.58 |
| | MMD | 97.66±0.91 | 97.63±0.97 | 98.85±0.40 | 98.57±0.22 | 91.27±8.47 | 74.73±22.95 | 48.27±20.22 | 28.20±11.47 | 17.62±5.66 | 13.68±3.63 | 11.37±2.37 |
| | DANN | 87.61±5.24 | 87.91±4.81 | 92.32±3.70 | 90.42±4.06 | 81.03±8.43 | 60.75±11.47 | 41.30±9.30 | 28.47±4.53 | 22.77±1.04 | 19.42±2.12 | 18.07±3.30 |
| | MTL | 95.91±2.71 | 95.85±2.49 | 97.05±1.53 | 85.60±7.21 | 62.02±11.61 | 40.50±6.66 | 28.10±3.74 | 18.45±1.99 | 14.42±0.48 | 12.43±0.53 | 11.90±0.18 |
| | SagNet | 85.48±15.58 | 85.24±15.54 | 88.12±14.02 | 86.55±13.54 | 78.62±9.48 | 59.20±3.03 | 38.30±1.70 | 25.00±2.62 | 17.10±1.50 | 13.98±1.35 | 12.42±0.38 |
| | ARM | 89.13±13.11 | 88.92±13.05 | 89.92±12.67 | 87.92±14.23 | 77.23±13.58 | 48.87±6.68 | 26.38±0.49 | 17.50±0.86 | 13.78±0.39 | 11.75±0.32 | 11.25±0.47 |
| | ANDMask | 98.49±0.24 | 98.53±0.05 | 99.10±0.35 | 88.65±12.68 | 73.57±23.69 | 53.48±20.39 | 33.33±10.93 | 20.90±3.59 | 15.85±0.71 | 12.82±0.42 | 12.77±0.97 |
| | SelfReg | 98.38±0.39 | 98.35±0.44 | 98.95±0.22 | 96.77±1.43 | 88.23±3.10 | 69.12±4.89 | 45.02±6.07 | 26.43±4.52 | 17.88±2.73 | 14.20±2.51 | 13.18±1.14 |
| | Fish | 98.86±0.04 | 98.84±0.22 | 59.55±24.00 | 23.40±5.98 | 20.83±3.32 | 17.28±1.22 | 15.07±0.74 | 12.93±0.15 | 12.07±0.46 | 11.27±0.77 | 11.80±0.32 |
| | IB-ERM | 98.01±0.94 | 97.99±0.90 | 98.32±1.22 | 90.85±9.80 | 82.18±10.74 | 61.58±5.99 | 40.58±2.84 | 27.78±3.92 | 21.10±4.18 | 17.32±4.35 | 15.08±3.07 |
| | IB-IRM | 42.53±39.47 | 42.35±39.62 | 43.15±39.69 | 43.00±39.39 | 41.37±37.64 | 36.33±28.12 | 27.37±14.24 | 21.90±4.28 | 19.03±0.20 | 16.00±0.86 | 12.25±0.84 |
| | CAD | 69.72±41.34 | 69.34±41.48 | 69.80±41.47 | 69.58±41.00 | 65.20±38.79 | 47.50±26.26 | 29.43±14.42 | 19.13±6.88 | 14.98±3.13 | 12.43±1.64 | 11.18±0.37 |
| | CondCAD | 44.50±30.16 | 44.87±29.54 | 47.50±31.84 | 46.42±31.30 | 31.18±23.51 | 18.85±7.53 | 14.80±2.75 | 12.37±0.78 | 13.07±1.37 | 12.15±1.46 | 12.15±0.57 |
| | EQRM | 97.28±0.19 | 97.48±0.15 | 96.60±1.04 | 75.00±15.46 | 55.78±24.48 | 40.72±21.94 | 30.00±13.98 | 21.87±9.68 | 17.93±5.14 | 14.05±2.74 | 13.05±1.59 |
| $\mathcal{D}_9$ | ERM | 95.94±0.76 | 95.59±0.73 | 97.83±0.63 | 97.23±0.38 | 96.02±0.37 | 89.30±1.24 | 70.32±6.07 | 46.25±8.15 | 32.00±6.51 | 24.93±4.60 | 20.15±3.41 |
| | VREx | 17.78±8.87 | 18.41±9.06 | 20.73±12.67 | 19.22±11.31 | 16.95±8.11 | 15.13±5.56 | 14.40±3.34 | 13.25±2.06 | 13.87±1.51 | 13.27±0.19 | 13.38±1.13 |
| | IRM | 63.71±33.35 | 64.01±33.40 | 67.03±35.62 | 66.87±34.57 | 64.50±34.43 | 57.73±29.57 | 44.35±20.35 | 32.05±10.96 | 25.28±5.15 | 21.07±3.39 | 17.57±1.46 |
| | SD | 98.16±0.33 | 98.37±0.17 | 98.72±0.24 | 98.00±0.60 | 95.30±3.26 | 87.68±7.56 | 73.22±10.27 | 53.57±10.66 | 38.05±8.37 | 27.85±5.66 | 22.87±4.41 |
| | GroupDRO | 98.47±0.32 | 98.41±0.41 | 98.48±0.39 | 92.05±2.20 | 72.15±3.63 | 46.48±0.70 | 30.87±0.92 | 21.10±1.57 | 17.13±1.08 | 14.38±0.97 | 12.28±0.49 |
| | RSC | 95.48±0.85 | 95.74±0.98 | 96.53±1.27 | 89.38±4.32 | 65.28±5.47 | 43.60±4.35 | 30.32±3.03 | 21.03±2.56 | 17.67±1.55 | 15.38±0.44 | 14.42±1.16 |
| | Mixup | 95.81±1.38 | 95.72±1.59 | 97.03±1.30 | 96.30±1.08 | 85.38±3.68 | 49.17±16.25 | 26.63±13.23 | 17.25±5.34 | 13.60±1.39 | 12.38±0.66 | 12.27±1.28 |
| | CORAL | 97.39±0.78 | 97.21±0.98 | 97.63±1.55 | 84.30±15.27 | 60.70±21.63 | 40.02±18.39 | 27.70±10.81 | 19.57±6.50 | 15.22±3.72 | 14.05±2.51 | 12.47±1.49 |
| | MMD | 98.21±1.26 | 98.04±1.17 | 98.93±0.45 | 94.43±4.13 | 72.17±20.05 | 50.00±31.75 | 34.90±26.70 | 23.25±14.39 | 16.58±6.35 | 14.43±3.09 | 12.47±1.60 |
| | DANN | 81.75±3.08 | 81.72±3.96 | 88.27±2.03 | 85.95±2.28 | 74.35±4.74 | 51.52±5.19 | 35.62±3.67 | 25.53±1.32 | 22.00±1.13 | 19.70±1.91 | 18.75±2.63 |
| | MTL | 95.71±2.59 | 95.80±2.46 | 97.12±1.57 | 91.73±6.09 | 72.57±4.52 | 43.02±3.74 | 27.03±4.96 | 17.67±2.79 | 13.72±1.46 | 12.58±0.33 | 12.00±0.15 |
| | SagNet | 30.76±23.21 | 30.26±23.46 | 32.95±25.01 | 32.17±24.93 | 31.90±23.59 | 29.38±18.08 | 24.03±11.65 | 20.00±6.16 | 16.60±1.85 | 15.15±0.67 | 13.20±1.38 |
| | ARM | 98.74±0.36 | 98.52±0.52 | 98.53±0.45 | 73.40±17.33 | 45.97±26.03 | 27.72±15.91 | 18.08±6.07 | 13.95±3.22 | 12.20±1.45 | 11.80±0.29 | 11.33±0.43 |
| | ANDMask | 97.73±0.07 | 97.85±0.18 | 98.73±0.05 | 95.22±3.66 | 77.08±16.83 | 48.35±14.96 | 27.13±9.39 | 18.17±4.90 | 14.40±1.52 | 13.65±0.27 | 13.72±0.72 |
| | SelfReg | 98.91±0.04 | 98.87±0.20 | 97.90±1.87 | 89.95±10.16 | 73.88±15.35 | 50.02±18.87 | 33.67±14.19 | 22.92±7.53 | 16.92±3.51 | 15.23±1.85 | 13.72±0.72 |
| | Fish | 98.32±1.14 | 98.27±0.98 | 46.00±13.06 | 17.72±4.64 | 18.08±4.38 | 13.52±2.04 | 12.13±0.91 | 11.63±0.88 | 11.98±0.36 | 11.30±0.67 | |
| | IB-ERM | 98.14±1.04 | 98.08±0.97 | 98.33±1.23 | 90.13±9.25 | 76.00±7.20 | 53.28±8.39 | 36.40±8.45 | 25.77±6.64 | 20.80±4.56 | 17.58±4.05 | 15.70±2.40 |
| | IB-IRM | 14.42±2.39 | 14.11±2.63 | 15.03±2.77 | 14.82±2.32 | 13.62±2.27 | 14.71±2.66 | 16.33±2.60 | 17.08±1.53 | 16.78±0.55 | 14.02±1.48 | |
| | CAD | 69.71±41.34 | 69.46±41.56 | 69.65±41.37 | 67.73±39.77 | 54.53±34.60 | 37.27±25.62 | 24.68±15.38 | 17.87±7.35 | 14.33±3.51 | 12.57±1.59 | 11.20±0.36 |
| | CondCAD | 67.38±20.00 | 67.99±19.74 | 73.27±21.65 | 67.98±20.21 | 37.45±19.26 | 21.05±5.68 | 16.22±0.82 | 12.50±0.63 | 12.87±1.54 | 12.53±1.21 | 11.90±0.82 |
| | EQRM | 96.32±2.01 | 96.19±2.47 | 97.88±0.81 | 91.87±5.22 | 74.15±16.90 | 49.47±17.14 | 29.92±14.03 | 21.32±10.07 | 16.20±6.37 | 14.40±2.48 | 12.75±1.81 |
| $\mathcal{D}_{10}$ | ERM | 95.94±0.76 | 95.59±0.73 | 97.83±0.63 | 97.23±0.38 | 96.02±0.37 | 89.30±1.24 | 70.32±6.07 | 46.25±8.15 | 32.00±6.51 | 24.93±4.60 | 20.15±3.41 |
| | VREx | 17.57±8.58 | 17.53±8.64 | 18.88±11.63 | 18.72±11.00 | 17.20±10.64 | 14.88±5.65 | 12.70±2.31 | 12.28±1.38 | 12.35±2.31 | 12.58±0.52 | 14.27±0.80 |
| | IRM | 41.28±38.48 | 41.51±38.66 | 42.40±39.37 | 42.40±38.98 | 40.98±38.48 | 37.82±32.78 | 31.92±24.25 | 25.03±13.74 | 21.93±6.70 | 19.95±4.17 | 18.98±0.49 |
| | SD | 97.33±0.90 | 97.43±1.16 | 98.48±0.49 | 98.07±0.51 | 96.60±1.44 | 88.97±5.83 | 72.80±10.83 | 52.67±11.78 | 37.02±9.52 | 27.63±5.91 | 23.00±4.26 |
| | GroupDRO | 97.94±1.01 | 97.82±1.01 | 97.73±0.57 | 78.87±18.59 | 51.00±20.14 | 32.60±12.38 | 22.63±7.69 | 16.95±4.56 | 14.28±3.11 | 13.03±1.73 | 12.63±0.32 |
| | RSC | 95.48±0.85 | 95.74±0.98 | 96.53±1.27 | 89.38±4.32 | 65.28±5.47 | 43.60±4.35 | 30.32±3.03 | 21.03±2.56 | 17.67±1.55 | 15.38±0.44 | 14.42±1.16 |
| | Mixup | 96.61±2.04 | 96.41±2.17 | 97.33±1.55 | 95.12±1.67 | 77.07±14.99 | 47.12±18.33 | 26.92±13.04 | 17.62±5.11 | 13.52±1.46 | 12.17±0.97 | 12.38±1.12 |
| | CORAL | 97.45±0.86 | 97.18±0.94 | 94.57±3.67 | 67.10±22.20 | 44.80±29.26 | 31.97±22.95 | 23.50±13.60 | 18.07±7.59 | 15.22±3.72 | 13.60±2.90 | 12.78±1.18 |
| | MMD | 68.50±41.08 | 68.88±40.64 | 69.53±41.28 | 69.33±40.82 | 65.60±39.20 | 51.62±34.07 | 37.05±26.04 | 24.03±13.99 | 17.32±5.90 | 13.80±3.55 | 12.87±1.31 |
| | DANN | 74.98±12.63 | 74.43±14.26 | 82.37±10.37 | 80.25±10.33 | 69.67±9.74 | 43.98±13.93 | 29.73±8.61 | 22.52±4.28 | 21.30±1.88 | 19.17±2.35 | 18.85±2.55 |
| | MTL | 95.71±2.59 | 95.80±2.46 | 97.12±1.57 | 91.73±6.09 | 72.57±4.52 | 43.02±3.74 | 27.03±4.96 | 17.67±2.79 | 13.72±1.46 | 12.58±0.33 | 12.00±0.15 |
| | SagNet | 11.34±1.48 | 10.91±1.27 | 12.07±1.59 | 11.75±1.54 | 12.08±1.57 | 13.72±1.86 | 13.17±1.51 | 13.78±1.50 | 13.23±1.94 | 14.12±0.88 | 14.45±0.81 |
| | ARM | 98.59±0.23 | 98.51±0.52 | 97.68±0.86 | 69.87±19.72 | 43.53±27.75 | 27.90±15.82 | 18.18±6.03 | 13.92±3.25 | 11.97±1.64 | 11.28±0.85 | 11.53±0.26 |
| | ANDMask | 98.03±0.47 | 98.18±0.54 | 98.78±0.41 | 87.72±12.00 | 66.12±18.61 | 40.85±12.84 | 26.37±5.11 | 18.03±1.96 | 15.32±1.27 | 12.62±0.65 | 13.25±0.29 |
| | SelfReg | 98.91±0.04 | 98.87±0.20 | 97.90±1.87 | 89.95±10.16 | 73.88±15.35 | 50.02±18.87 | 33.67±14.19 | 22.92±7.53 | 16.92±3.51 | 15.23±1.85 | 13.72±0.72 |
| | Fish | 98.89±0.04 | 98.97±0.24 | 56.42±20.70 | 20.38±1.88 | 17.10±2.19 | 14.97±2.61 | 13.90±1.62 | 12.48±0.74 | 11.65±0.86 | 11.47±0.65 | 11.97±0.27 |
| | IB-ERM | 98.14±1.04 | 98.08±0.97 | 98.33±1.23 | 90.13±9.25 | 76.00±7.20 | 53.28±8.39 | 36.40±8.45 | 25.77±6.64 | 20.80±4.56 | 17.58±4.05 | 15.70±2.40 |
| | IB-IRM | 13.42±2.64 | 13.99±2.62 | 15.03±2.77 | 15.50±2.78 | 14.05±2.63 | 14.33±2.30 | 14.22±2.36 | 15.43±1.96 | 16.67±1.14 | 16.37±0.55 | 15.77±1.27 |
| | CAD | 40.18±41.10 | 40.00±41.42 | 40.48±41.48 | 40.58±40.99 | 36.28±36.60 | 22.62±16.29 | 14.57±5.11 | 12.98±2.17 | 12.32±0.45 | 11.28±0.54 | 11.77±0.09 |
| | CondCAD | 44.51±30.14 | 44.26±30.23 | 47.15±32.25 | 45.57±32.26 | 31.00±23.67 | 18.80±7.58 | 14.37±3.36 | 11.97±1.29 | 12.33±2.12 | 12.22±1.41 | 12.17±0.55 |
| | EQRM | 97.28±0.19 | 97.48±0.15 | 96.60±1.04 | 75.00±15.46 | 55.78±24.48 | 40.72±21.94 | 30.00±13.98 | 21.87±9.68 | 17.93±5.14 | 14.05±2.74 | 13.05±1.59 |

Table 10: Average accuracy of top-3 models of each algorithm at each shift degree of NOISYMNIST (ResNet-50). This table shows the results on shift degrees from $\mathcal{D}_8$ to $\mathcal{D}_{10}$.

| | Algorithm | $\mathcal{D}_0$ | $\mathcal{D}_1$ | $\mathcal{D}_2$ | $\mathcal{D}_3$ | $\mathcal{D}_4$ | $\mathcal{D}_5$ | $\mathcal{D}_6$ | $\mathcal{D}_7$ | $\mathcal{D}_8$ | $\mathcal{D}_9$ | $\mathcal{D}_{10}$ |
|---|---|---|---|---|---|---|---|---|---|---|---|---|
| | ERM | 83.20±0.24 | 82.25±0.61 | 81.53±0.42 | 79.30±1.17 | 73.20±2.50 | 62.88±3.14 | 52.55±4.19 | 37.13±3.41 | 27.58±2.17 | 18.87±1.37 | 14.77±0.78 |
| | VREx | 82.71±0.40 | 81.45±0.41 | 80.95±0.88 | 79.05±0.61 | 71.73±0.25 | 62.45±0.58 | 50.57±1.30 | 36.12±4.09 | 24.58±3.36 | 17.37±2.40 | 13.67±1.39 |
| | IRM | 82.09±0.33 | 81.97±0.17 | 81.38±0.59 | 79.38±0.25 | 73.17±1.02 | 64.62±1.58 | 53.22±1.47 | 39.03±2.45 | 27.48±3.47 | 18.42±4.57 | 14.00±2.93 |
| | SD | 83.21±0.15 | 82.37±0.35 | 81.52±0.43 | 80.05±0.97 | 72.40±0.83 | 64.92±1.73 | 53.93±2.70 | 38.82±2.15 | 28.35±1.71 | 19.48±1.56 | 15.50±1.10 |
| | CORAL | 83.19±0.06 | 82.11±1.04 | 81.42±0.87 | 80.45±1.19 | 73.98±1.57 | 67.00±3.13 | 55.23±4.30 | 40.17±5.88 | 29.37±7.35 | 20.22±7.00 | 14.93±3.80 |
| $\mathcal{D}_0$ | GroupDRO | 83.04±0.16 | 82.47±0.49 | 81.70±0.98 | 79.97±0.84 | 73.97±0.76 | 64.07±1.53 | 52.77±2.17 | 36.75±2.91 | 27.00±2.31 | 18.38±2.41 | 14.27±1.32 |
| | RSC | 82.61±0.12 | 82.00±0.51 | 81.58±1.43 | 79.40±0.91 | 72.77±2.31 | 63.15±2.27 | 51.28±2.89 | 35.78±4.13 | 25.68±3.65 | 17.47±2.98 | 13.98±2.21 |
| | DANN | 71.05±2.87 | 69.83±2.24 | 69.57±4.03 | 66.83±4.47 | 61.52±4.36 | 54.47±3.42 | 48.03±3.57 | 35.23±2.89 | 29.25±1.03 | 21.35±0.84 | 17.87±1.90 |
| | Mixup | 82.92±0.34 | 82.33±0.28 | 82.22±0.89 | 81.17±1.03 | 74.32±0.95 | 66.67±1.96 | 56.00±2.17 | 42.63±2.16 | 33.58±0.47 | 23.57±0.84 | 18.18±0.30 |
| | CAD | 83.17±0.09 | 82.00±0.32 | 82.25±0.36 | 81.03±0.52 | 74.43±1.41 | 66.50±1.63 | 56.32±3.33 | 42.62±2.98 | 31.23±3.68 | 22.30±2.99 | 16.27±1.76 |
| | IB-IRM | 77.26±5.47 | 76.06±5.52 | 77.37±4.59 | 75.97±3.97 | 68.80±4.07 | 61.38±2.30 | 50.67±0.81 | 36.40±1.68 | 25.88±2.40 | 17.87±2.14 | 13.68±1.47 |
| | ERM | 82.74±0.48 | 82.74±0.13 | 82.35±0.92 | 79.80±1.84 | 72.68±2.19 | 62.93±3.16 | 51.12±3.38 | 35.00±3.30 | 26.33±3.75 | 17.72±2.39 | 13.90±1.66 |
| | VREx | 82.43±0.68 | 81.74±0.09 | 81.85±0.88 | 79.82±0.84 | 72.62±1.06 | 64.62±2.50 | 52.90±3.00 | 38.80±2.53 | 26.90±2.18 | 18.42±1.57 | 14.17±0.68 |
| | IRM | 81.44±0.31 | 82.51±0.22 | 81.87±0.60 | 80.38±1.18 | 74.52±1.23 | 66.88±1.56 | 55.33±2.14 | 40.95±2.62 | 29.77±1.50 | 21.03±2.47 | 15.47±1.88 |
| | SD | 82.50±0.50 | 82.85±0.07 | 81.48±0.38 | 80.00±1.04 | 73.10±0.80 | 65.50±1.75 | 55.02±2.06 | 39.33±2.15 | 29.60±2.33 | 19.98±2.33 | 15.55±1.67 |
| | CORAL | 82.62±0.47 | 83.13±0.26 | 81.08±0.98 | 80.08±0.80 | 73.92±0.68 | 66.52±1.04 | 55.07±1.97 | 38.60±2.13 | 26.38±2.41 | 16.53±2.08 | 13.55±0.86 |
| $\mathcal{D}_1$ | GroupDRO | 82.03±0.28 | 83.13±0.18 | 82.25±1.21 | 80.20±1.15 | 73.67±0.90 | 65.17±1.00 | 54.17±0.96 | 39.77±1.00 | 29.68±1.01 | 20.48±1.05 | 15.82±1.29 |
| | RSC | 82.15±0.25 | 82.95±0.43 | 81.98±0.59 | 79.73±0.71 | 74.57±1.29 | 65.92±0.90 | 54.75±2.55 | 40.50±2.59 | 29.13±1.73 | 19.48±1.67 | 14.33±1.31 |
| | DANN | 70.89±3.00 | 69.92±2.15 | 70.57±2.95 | 67.35±4.01 | 62.23±3.70 | 54.92±3.03 | 48.48±3.15 | 36.97±0.60 | 30.35±0.63 | 22.55±1.69 | 17.78±1.89 |
| | Mixup | 82.49±0.63 | 82.99±0.33 | 82.28±0.31 | 81.65±0.48 | 74.75±0.68 | 66.38±1.35 | 56.50±1.35 | 41.50±2.16 | 31.22±2.69 | 21.32±1.80 | 16.32±1.19 |
| | CAD | 81.93±0.96 | 82.27±0.15 | 81.92±0.66 | 80.47±0.40 | 74.27±0.45 | 66.13±1.13 | 55.83±2.54 | 41.68±1.97 | 29.50±1.22 | 20.63±0.74 | 15.27±1.27 |
| | IB-IRM | 77.26±5.47 | 76.06±5.52 | 77.37±4.59 | 75.97±3.97 | 68.80±4.07 | 61.38±2.30 | 50.67±0.81 | 36.40±1.68 | 25.88±2.40 | 17.87±2.14 | 13.68±1.47 |
| | ERM | 81.92±0.57 | 82.09±0.56 | 83.58±0.25 | 81.92±0.37 | 76.00±0.80 | 67.87±1.18 | 56.67±0.88 | 40.67±0.99 | 29.10±1.45 | 19.37±1.04 | 14.37±1.19 |
| | VREx | 81.91±0.31 | 81.37±0.36 | 82.42±0.24 | 80.77±0.53 | 73.67±0.42 | 66.07±2.37 | 55.75±0.98 | 40.70±1.29 | 28.67±0.68 | 19.13±1.02 | 14.07±0.71 |
| | IRM | 81.56±0.56 | 82.03±0.29 | 82.40±0.35 | 80.18±0.72 | 74.60±1.56 | 66.90±2.37 | 56.08±4.16 | 42.73±2.79 | 31.70±3.57 | 22.20±3.55 | 16.62±2.23 |
| | SD | 81.91±0.57 | 81.45±0.51 | 83.23±0.13 | 81.47±0.48 | 75.92±2.08 | 67.57±2.47 | 57.97±2.79 | 42.05±2.01 | 31.22±1.57 | 21.03±1.19 | 15.93±0.57 |
| | CORAL | 81.89±0.38 | 81.85±0.32 | 82.93±0.25 | 81.50±0.57 | 76.17±0.36 | 67.50±0.80 | 57.25±2.64 | 42.15±3.83 | 30.22±3.27 | 20.23±2.63 | 14.50±1.52 |
| $\mathcal{D}_2$ | GroupDRO | 81.65±0.08 | 82.72±0.34 | 83.45±0.25 | 81.47±0.27 | 74.57±0.30 | 66.57±0.53 | 55.53±1.21 | 40.38±1.64 | 29.22±2.29 | 20.05±2.05 | 14.07±1.07 |
| | RSC | 82.19±0.45 | 82.00±0.13 | 83.55±0.29 | 80.72±0.85 | 75.30±1.06 | 66.00±2.53 | 57.32±2.60 | 42.73±3.05 | 32.67±2.92 | 23.27±2.90 | 16.67±1.46 |
| | DANN | 70.89±3.00 | 69.92±2.15 | 70.57±2.95 | 67.35±4.01 | 62.23±3.70 | 54.92±3.03 | 48.48±3.15 | 36.97±0.60 | 30.35±0.63 | 22.55±1.69 | 17.78±1.89 |
| | Mixup | 82.86±0.40 | 82.51±0.29 | 82.70±0.29 | 81.68±0.44 | 74.82±0.38 | 67.75±0.43 | 57.58±0.24 | 44.57±0.64 | 34.60±1.05 | 24.10±1.56 | 18.45±0.67 |
| | CAD | 82.01±0.92 | 81.51±0.56 | 82.85±0.36 | 80.62±0.40 | 74.28±0.51 | 65.65±0.92 | 55.87±0.43 | 41.87±2.24 | 30.60±3.10 | 20.58±1.25 | 14.95±1.51 |
| | IB-IRM | 77.12±5.33 | 75.87±5.32 | 77.52±4.75 | 76.58±4.72 | 70.03±5.60 | 63.28±4.70 | 52.90±3.96 | 38.32±3.45 | 27.05±3.60 | 18.35±2.72 | 14.37±2.15 |
| | ERM | 81.54±0.32 | 81.62±0.42 | 82.97±0.83 | 82.28±0.08 | 75.87±1.13 | 68.98±0.47 | 57.22±2.10 | 40.33±2.37 | 28.78±2.49 | 18.97±2.75 | 14.40±2.25 |
| | VREx | 81.30±0.33 | 80.29±0.49 | 81.78±0.39 | 81.12±0.22 | 74.62±0.59 | 67.15±0.97 | 56.45±0.33 | 40.03±0.27 | 28.65±0.41 | 19.03±0.09 | 14.18±0.82 |
| | IRM | 81.31±0.31 | 82.07±0.46 | 82.15±0.45 | 81.35±0.50 | 75.33±1.60 | 68.00±2.31 | 58.13±3.09 | 43.77±2.37 | 32.15±2.85 | 22.32±2.24 | 16.50±1.61 |
| | SD | 81.61±0.75 | 81.81±0.57 | 82.20±0.31 | 82.45±0.33 | 76.78±0.49 | 69.70±0.68 | 59.73±0.93 | 43.70±2.16 | 33.05±2.06 | 21.90±1.30 | 15.72±0.93 |
| | CORAL | 81.74±0.99 | 81.18±0.78 | 82.13±0.13 | 82.47±0.52 | 76.67±0.69 | 71.03±0.21 | 61.40±0.54 | 46.28±1.76 | 35.63±3.11 | 24.55±4.12 | 17.82±2.53 |
| $\mathcal{D}_3$ | GroupDRO | 81.63±0.41 | 82.05±0.73 | 82.93±0.71 | 82.08±0.24 | 75.50±0.97 | 67.57±1.18 | 57.58±2.49 | 42.75±2.91 | 32.48±3.29 | 22.20±2.22 | 16.13±1.73 |
| | RSC | 81.73±0.33 | 81.84±0.45 | 82.80±0.88 | 81.70±0.04 | 75.45±0.29 | 67.27±1.31 | 56.80±2.49 | 41.73±2.88 | 30.55±4.60 | 21.93±4.19 | 16.01±1.70 |
| | DANN | 69.35±4.05 | 69.29±2.57 | 70.25±3.23 | 68.17±3.42 | 62.82±3.32 | 56.87±1.66 | 50.30±1.86 | 38.65±2.50 | 30.83±2.30 | 21.93±2.39 | 16.60±1.49 |
| | Mixup | 82.71±0.48 | 82.50±0.31 | 81.78±0.56 | 82.12±0.14 | 76.27±0.70 | 69.45±1.13 | 59.63±1.79 | 46.27±2.33 | 36.23±2.91 | 25.57±2.42 | 20.00±1.76 |
| | CAD | 82.03±1.02 | 81.38±0.23 | 81.80±0.43 | 81.95±0.59 | 75.80±1.33 | 68.57±1.41 | 59.72±2.18 | 44.50±3.31 | 33.68±3.32 | 23.82±2.41 | 17.20±1.36 |
| | IB-IRM | 76.64±5.21 | 75.67±5.25 | 76.90±4.67 | 76.90±4.73 | 70.95±5.62 | 65.22±4.92 | 55.15±3.73 | 40.85±1.91 | 28.82±1.77 | 18.73±2.27 | 14.55±1.92 |

Table 11: Average accuracy of top-3 models of each algorithm at each shift degree of LOWLIGHT-CIFAR10 (Simple CNN). This table shows the results on shift degrees from $\mathcal{D}_0$ to $\mathcal{D}_3$.

| | Algorithm | $\mathcal{D}_0$ | $\mathcal{D}_1$ | $\mathcal{D}_2$ | $\mathcal{D}_3$ | $\mathcal{D}_4$ | $\mathcal{D}_5$ | $\mathcal{D}_6$ | $\mathcal{D}_7$ | $\mathcal{D}_8$ | $\mathcal{D}_9$ | $\mathcal{D}_{10}$ |
|---|---|---|---|---|---|---|---|---|---|---|---|---|
| | ERM | 80.29±0.85 | 80.99±0.66 | 81.83±1.43 | 81.13±0.88 | 77.22±0.70 | 70.77±1.78 | 60.00±2.88 | 44.22±2.63 | 33.03±3.47 | 22.62±3.47 | 16.10±2.11 |
| | VREx | 79.69±1.16 | 79.77±0.80 | 81.02±0.97 | 80.40±0.51 | 75.53±0.35 | 68.83±0.57 | 59.43±2.17 | 43.35±2.13 | 32.72±3.33 | 22.28±2.40 | 15.70±0.86 |
| | IRM | 80.71±0.92 | 81.02±1.81 | 81.72±1.03 | 81.13±0.74 | 76.23±0.42 | 69.25±0.67 | 59.45±1.66 | 45.38±0.78 | 34.17±2.36 | 23.52±2.39 | 16.90±1.63 |
| | SD | 80.37±0.72 | 80.73±0.38 | 82.48±0.62 | 81.82±0.16 | 77.50±0.74 | 70.78±0.16 | 61.92±0.74 | 46.77±1.73 | 35.42±1.93 | 24.48±2.04 | 17.18±1.04 |
| | CORAL | 79.99±0.51 | 81.07±0.28 | 81.53±0.69 | 80.65±0.50 | 77.68±0.37 | 70.40±0.14 | 60.48±0.48 | 47.28±2.69 | 35.47±5.32 | 25.45±6.04 | 18.77±4.19 |
| $\mathcal{D}_4$ | GroupDRO | 81.45±0.52 | 81.24±0.76 | 81.60±0.39 | 81.25±0.35 | 76.57±0.08 | 68.88±0.29 | 59.58±0.14 | 45.65±0.79 | 34.48±0.97 | 24.10±0.68 | 16.33±0.93 |
| | RSC | 81.50±0.16 | 81.42±0.38 | 82.28±0.72 | 81.25±0.33 | 76.18±0.14 | 67.15±1.26 | 56.78±0.86 | 42.68±1.23 | 30.37±1.64 | 20.97±0.69 | 15.43±0.53 |
| | DANN | 69.19±3.82 | 69.23±2.50 | 69.92±2.78 | 67.95±3.12 | 63.10±3.72 | 57.60±2.69 | 51.33±3.31 | 40.68±3.33 | 32.82±2.87 | 23.37±2.39 | 17.30±1.21 |
| | Mixup | 82.41±0.08 | 81.85±0.81 | 81.58±0.29 | 81.90±0.19 | 76.57±0.35 | 69.69±0.60 | 60.23±1.02 | 45.65±2.90 | 34.93±4.16 | 24.33±3.76 | 19.55±2.31 |
| | CAD | 80.60±1.77 | 80.35±0.88 | 81.62±0.57 | 81.33±1.07 | 76.95±0.37 | 69.73±0.41 | 61.42±1.64 | 45.57±2.21 | 33.30±2.88 | 22.15±3.44 | 15.77±2.07 |
| | IB-IRM | 76.64±5.21 | 75.67±5.25 | 76.90±4.67 | 76.90±4.73 | 70.95±5.62 | 65.22±4.92 | 55.15±3.73 | 40.85±1.91 | 28.82±1.77 | 18.73±2.27 | 14.55±1.92 |
| | ERM | 80.06±0.55 | 80.73±0.32 | 81.33±0.73 | 80.52±0.46 | 76.88±0.91 | 71.01±0.11 | 62.00±1.38 | 47.02±1.33 | 35.60±0.43 | 24.45±1.42 | 16.82±1.15 |
| | VREx | 78.23±1.51 | 78.65±0.78 | 79.27±1.64 | 80.02±0.68 | 74.95±0.36 | 69.58±0.18 | 60.38±1.73 | 44.57±0.61 | 33.20±1.24 | 23.37±1.11 | 16.57±0.69 |
| | IRM | 81.01±0.05 | 80.77±1.06 | 81.48±0.53 | 80.77±0.40 | 75.63±1.16 | 70.05±0.49 | 60.52±0.97 | 45.78±0.38 | 34.33±1.67 | 24.48±3.31 | 18.13±2.47 |
| | SD | 80.50±0.78 | 80.75±0.41 | 81.87±1.25 | 81.22±0.68 | 77.08±1.05 | 70.88±0.10 | 61.95±0.70 | 47.23±1.76 | 36.12±2.08 | 24.92±1.94 | 17.70±0.90 |
| | CORAL | 81.14±1.52 | 81.04±0.98 | 81.52±0.80 | 82.23±0.73 | 76.50±0.58 | 71.37±0.28 | 61.00±0.73 | 47.30±0.79 | 35.78±2.95 | 24.82±3.83 | 17.55±2.89 |
| $\mathcal{D}_5$ | GroupDRO | 80.55±0.41 | 80.34±0.20 | 81.53±0.22 | 80.72±0.49 | 75.97±0.51 | 69.40±0.11 | 60.00±1.14 | 45.92±1.91 | 34.93±1.15 | 25.10±0.63 | 18.17±0.74 |
| | RSC | 80.69±1.12 | 80.78±0.47 | 81.62±0.10 | 80.62±0.42 | 74.88±0.74 | 70.32±0.83 | 60.50±0.45 | 45.13±0.74 | 32.85±1.66 | 20.65±2.32 | 14.87±1.36 |
| | DANN | 66.48±3.54 | 66.81±4.01 | 66.97±4.47 | 65.97±3.53 | 62.23±3.37 | 58.47±2.98 | 52.25±2.26 | 43.20±1.88 | 34.08±1.67 | 24.33±1.23 | 18.28±0.55 |
| | Mixup | 79.60±2.05 | 80.22±1.87 | 80.13±0.90 | 80.37±1.83 | 75.72±0.65 | 71.07±0.38 | 62.17±0.70 | 48.88±0.98 | 37.80±2.08 | 25.75±2.31 | 20.00±1.82 |
| | CAD | 81.04±1.15 | 80.01±1.29 | 81.77±0.42 | 81.45±0.95 | 76.23±1.37 | 70.17±0.46 | 61.63±1.35 | 45.62±2.14 | 33.58±2.48 | 23.27±1.86 | 16.65±0.84 |
| | IB-IRM | 76.64±5.21 | 75.67±5.25 | 76.90±4.67 | 76.90±4.73 | 70.95±5.62 | 65.22±4.92 | 55.15±3.73 | 40.85±1.91 | 28.82±1.77 | 18.73±2.27 | 14.55±1.92 |
| | ERM | 79.88±0.60 | 80.32±0.66 | 81.20±0.82 | 80.53±0.45 | 76.50±1.21 | 71.28±1.35 | 62.50±0.74 | 46.10±2.31 | 34.63±1.17 | 22.93±0.74 | 15.95±0.71 |
| | VREx | 77.27±0.71 | 77.63±0.85 | 78.37±1.04 | 79.28±0.49 | 74.72±0.64 | 69.21±0.62 | 61.33±0.41 | 45.42±0.72 | 34.70±0.89 | 24.30±0.22 | 17.52±0.80 |
| | IRM | 80.11±1.25 | 80.35±1.64 | 80.70±1.58 | 80.23±1.16 | 75.48±1.37 | 69.35±1.46 | 61.13±0.62 | 46.78±1.17 | 35.75±2.85 | 25.33±4.29 | 18.90±3.40 |
| | SD | 78.53±1.82 | 78.94±1.45 | 79.06±1.93 | 80.23±2.08 | 74.83±1.48 | 70.57±0.13 | 62.62±0.30 | 47.65±1.00 | 37.30±0.59 | 26.62±0.49 | 19.48±0.59 |
| | CORAL | 81.07±0.14 | 80.82±0.65 | 81.73±0.53 | 82.03±1.00 | 76.97±0.27 | 70.93±0.22 | 61.80±0.22 | 45.72±1.18 | 33.83±1.14 | 22.43±1.69 | 16.62±1.82 |
| $\mathcal{D}_6$ | GroupDRO | 78.55±1.18 | 78.94±1.45 | 78.73±2.13 | 78.48±1.61 | 73.68±0.72 | 67.67±1.23 | 60.83±0.78 | 46.02±1.18 | 36.00±0.40 | 25.92±0.81 | 18.73±0.87 |
| | RSC | 79.57±0.56 | 79.81±0.90 | 81.23±0.45 | 80.43±0.29 | 74.87±0.72 | 69.48±0.97 | 60.62±0.34 | 45.32±0.83 | 34.53±1.00 | 23.33±2.36 | 17.12±2.00 |
| | DANN | 66.66±5.61 | 66.00±4.86 | 66.12±5.48 | 65.42±4.91 | 61.50±4.84 | 57.87±2.56 | 52.68±2.26 | 42.90±2.90 | 33.48±3.36 | 22.88±2.93 | 17.07±1.47 |
| | Mixup | 79.07±2.19 | 78.71±2.09 | 79.78±1.75 | 79.43±1.62 | 74.50±1.59 | 69.32±1.09 | 62.33±0.58 | 48.17±1.03 | 36.22±1.13 | 24.45±0.25 | 17.73±0.29 |
| | CAD | 81.71±0.23 | 81.18±0.74 | 82.20±0.48 | 81.27±1.15 | 76.62±0.83 | 68.67±1.81 | 62.17±0.74 | 47.32±0.31 | 35.73±0.65 | 24.88±0.49 | 17.20±0.23 |
| | IB-IRM | 72.85±10.34 | 72.29±9.92 | 72.97±9.88 | 72.80±10.25 | 66.66±8.79 | 64.53±5.87 | 56.73±1.56 | 44.03±2.63 | 32.87±4.90 | 22.37±5.33 | 16.92±3.92 |
| | ERM | 80.71±1.06 | 80.72±0.32 | 81.03±1.02 | 79.62±0.69 | 75.07±1.15 | 68.98±1.91 | 60.92±1.37 | 49.32±1.04 | 37.67±1.97 | 25.75±3.47 | 17.73±2.25 |
| | VREx | 74.54±5.93 | 73.77±5.71 | 74.53±5.43 | 75.07±4.32 | 70.88±3.39 | 65.57±2.00 | 59.75±1.12 | 47.32±1.70 | 37.92±1.77 | 27.07±1.84 | 19.78±1.41 |
| | IRM | 61.98±10.88 | 63.10±10.29 | 64.43±8.53 | 63.60±8.82 | 60.80±7.19 | 60.25±5.14 | 55.52±2.95 | 49.45±1.54 | 40.08±0.93 | 27.32±1.32 | 18.68±3.07 |
| | SD | 77.39±2.05 | 77.35±2.53 | 78.92±2.41 | 78.80±2.31 | 73.87±2.41 | 68.92±1.93 | 61.95±0.76 | 49.48±1.07 | 40.30±1.71 | 28.98±1.78 | 20.85±1.69 |
| | CORAL | 80.81±1.65 | 81.39±0.36 | 82.00±0.47 | 81.02±0.86 | 76.85±0.81 | 70.60±0.37 | 60.45±0.46 | 48.88±0.65 | 39.30±1.52 | 29.25±3.65 | 20.78±2.74 |
| $\mathcal{D}_7$ | GroupDRO | 75.13±5.86 | 75.71±5.30 | 76.48±4.88 | 75.28±4.87 | 71.13±4.12 | 67.38±2.00 | 60.22±1.02 | 48.83±0.31 | 38.58±1.86 | 28.65±2.12 | 20.60±1.21 |
| | RSC | 71.80±5.31 | 72.66±5.71 | 72.52±4.13 | 72.45±4.25 | 68.12±3.94 | 65.48±2.44 | 59.13±0.68 | 48.02±0.86 | 37.92±1.28 | 27.75±1.74 | 20.83±1.70 |
| | DANN | 67.70±5.13 | 67.31±4.59 | 67.83±5.34 | 66.65±4.40 | 62.73±4.06 | 58.05±2.39 | 52.63±2.72 | 43.93±1.55 | 35.25±0.88 | 25.07±0.21 | 18.27±0.58 |
| | Mixup | 74.14±8.61 | 74.88±8.13 | 74.23±8.42 | 73.90±8.62 | 70.15±7.77 | 66.55±6.01 | 59.80±3.36 | 49.15±0.60 | 39.47±1.18 | 27.52±1.99 | 20.92±2.21 |
| | CAD | 81.71±0.23 | 81.18±0.74 | 82.20±0.48 | 81.27±1.15 | 76.62±0.83 | 68.67±1.81 | 62.17±0.74 | 47.32±0.31 | 35.73±0.65 | 24.88±0.49 | 17.20±0.23 |
| | IB-IRM | 71.24±8.98 | 70.87±8.79 | 71.05±8.24 | 71.13±8.86 | 67.10±7.47 | 63.12±4.91 | 55.90±0.94 | 44.27±2.54 | 33.20±4.82 | 22.90±5.31 | 16.72±3.93 |

Table 12: Average accuracy of top-3 models of each algorithm at each shift degree of LOWLIGHT-CIFAR10 (Simple CNN). This table shows the results on shift degrees from $\mathcal{D}_4$ to $\mathcal{D}_7$.

| | Algorithm | $\mathcal{D}_0$ | $\mathcal{D}_1$ | $\mathcal{D}_2$ | $\mathcal{D}_3$ | $\mathcal{D}_4$ | $\mathcal{D}_5$ | $\mathcal{D}_6$ | $\mathcal{D}_7$ | $\mathcal{D}_8$ | $\mathcal{D}_9$ | $\mathcal{D}_{10}$ |
|---|---|---|---|---|---|---|---|---|---|---|---|---|
| $\mathcal{D}_8$ | ERM | 73.59±5.47 | 74.09±5.26 | 74.83±5.36 | 73.82±6.10 | 70.00±5.21 | 65.27±3.71 | 59.90±2.12 | 47.67±2.19 | 38.98±1.00 | 27.50±2.25 | 19.05±1.24 |
| | VREx | 74.87±6.24 | 74.25±6.16 | 75.12±5.89 | 75.12±4.35 | 71.18±3.56 | 65.73±1.90 | 59.63±1.28 | 47.03±1.96 | 38.17±1.76 | 26.83±1.80 | 19.65±1.49 |
| | IRM | 52.65±13.65 | 53.73±13.71 | 56.88±11.95 | 56.98±11.06 | 55.45±10.16 | 56.18±7.10 | 53.48±4.22 | 48.05±1.30 | 42.77±2.19 | 33.52±2.88 | 24.43±2.12 |
| | SD | 77.19±1.87 | 76.91±2.05 | 77.68±1.17 | 77.38±1.06 | 72.48±1.29 | 68.17±1.45 | 61.28±0.51 | 49.07±1.36 | 40.88±0.91 | 29.07±1.71 | 20.40±2.29 |
| | CORAL | 80.76±1.67 | 81.03±0.59 | 81.47±0.56 | 80.67±0.98 | 76.45±0.71 | 70.28±0.77 | 60.95±0.25 | 48.15±1.44 | 39.30±1.52 | 29.90±2.82 | 20.83±2.68 |
| | GroupDRO | 74.96±5.73 | 75.12±4.85 | 76.28±4.67 | 74.92±4.52 | 70.63±3.64 | 67.03±1.66 | 59.35±0.39 | 47.63±1.97 | 38.92±1.53 | 29.82±0.89 | 21.23±0.70 |
| | RSC | 71.80±5.31 | 72.66±5.71 | 72.52±4.13 | 72.45±4.25 | 68.12±3.94 | 65.48±2.44 | 59.13±0.68 | 48.02±0.86 | 37.92±1.28 | 27.75±1.74 | 20.83±1.70 |
| | DANN | 67.70±5.13 | 67.31±4.59 | 67.83±5.34 | 66.65±4.40 | 62.73±4.06 | 58.05±2.39 | 52.63±2.72 | 43.93±1.55 | 35.25±0.88 | 25.07±0.21 | 18.27±0.58 |
| | Mixup | 68.79±9.65 | 69.71±9.21 | 68.85±8.77 | 69.12±9.10 | 65.35±7.83 | 62.85±5.63 | 56.85±3.33 | 48.47±0.82 | 40.18±0.17 | 28.60±0.46 | 22.03±0.65 |
| | CAD | 66.77±10.77 | 66.54±10.41 | 66.67±10.73 | 66.37±9.28 | 62.12±9.17 | 58.68±5.90 | 54.38±3.14 | 45.55±1.13 | 37.82±1.08 | 28.23±1.46 | 22.15±1.97 |
| | IB-IRM | 53.94±21.57 | 53.55±21.33 | 54.22±20.96 | 54.48±20.64 | 52.07±18.98 | 50.00±15.75 | 45.67±13.70 | 39.48±8.74 | 33.43±4.56 | 25.13±2.88 | 19.75±2.48 |
| $\mathcal{D}_9$ | ERM | 77.48±3.24 | 77.34±3.64 | 78.10±2.47 | 77.65±2.10 | 73.72±2.23 | 67.60±2.06 | 59.42±1.51 | 46.78±2.82 | 37.73±1.91 | 29.23±1.11 | 21.65±0.88 |
| | VREx | 33.69±3.38 | 33.71±3.64 | 35.40±4.35 | 34.95±3.83 | 33.95±3.15 | 35.43±3.35 | 33.73±3.10 | 33.58±1.31 | 31.95±1.23 | 30.32±0.94 | 28.85±1.06 |
| | IRM | 55.16±17.03 | 55.69±16.37 | 58.53±14.26 | 59.07±13.97 | 56.87±12.11 | 56.62±7.70 | 54.17±5.16 | 48.30±1.27 | 42.27±2.74 | 33.88±2.36 | 24.70±1.88 |
| | SD | 76.51±2.50 | 76.39±2.79 | 76.97±2.32 | 76.37±2.64 | 71.77±2.32 | 66.65±1.90 | 60.02±1.15 | 48.15±2.11 | 39.55±1.59 | 29.97±1.20 | 21.77±0.94 |
| | CORAL | 81.43±1.53 | 81.38±0.36 | 81.37±0.67 | 80.52±1.10 | 75.38±1.79 | 69.50±1.85 | 59.28±2.11 | 47.68±2.06 | 38.27±2.95 | 30.60±2.00 | 21.68±1.89 |
| | GroupDRO | 74.87±5.67 | 75.05±4.80 | 76.15±4.63 | 74.82±4.47 | 70.35±3.51 | 67.08±1.69 | 59.30±0.32 | 47.22±1.71 | 38.28±0.66 | 30.22±1.45 | 22.58±1.42 |
| | RSC | 75.18±7.22 | 75.30±7.42 | 76.10±6.90 | 75.45±6.10 | 70.47±5.39 | 66.50±3.03 | 59.23±0.78 | 47.25±1.57 | 37.48±1.50 | 28.42±0.84 | 20.47±2.07 |
| | DANN | 62.99±0.94 | 63.13±0.80 | 62.40±1.79 | 61.03±1.31 | 55.78±1.86 | 53.13±2.47 | 47.12±1.65 | 39.80±1.88 | 33.85±1.31 | 25.93±0.34 | 21.28±1.72 |
| | Mixup | 63.73±5.91 | 64.69±4.76 | 65.42±4.17 | 64.37±4.09 | 60.78±3.83 | 59.02±3.20 | 54.33±2.35 | 45.87±0.84 | 38.47±0.95 | 30.05±0.32 | 24.37±1.37 |
| | CAD | 58.57±18.94 | 58.51±18.18 | 58.92±17.63 | 59.13±17.82 | 54.92±15.39 | 53.05±12.44 | 48.85±10.37 | 41.30±5.46 | 36.32±2.19 | 29.78±0.41 | 24.80±1.76 |
| | IB-IRM | 54.07±21.40 | 53.74±21.08 | 54.43±20.67 | 54.45±20.68 | 52.83±17.97 | 48.90±17.26 | 44.82±14.91 | 38.45±10.17 | 31.83±6.46 | 25.45±2.66 | 20.18±2.80 |
| $\mathcal{D}_{10}$ | ERM | 69.97±2.21 | 70.08±1.54 | 70.87±2.65 | 70.47±3.07 | 66.27±3.07 | 61.72±2.13 | 54.93±1.91 | 42.75±1.26 | 34.60±1.03 | 27.08±1.02 | 22.18±0.49 |
| | VREx | 33.69±3.38 | 33.71±3.64 | 35.40±4.35 | 34.95±3.83 | 33.95±3.15 | 35.43±3.35 | 33.73±3.10 | 33.58±1.31 | 31.95±1.23 | 30.32±0.94 | 28.85±1.06 |
| | IRM | 55.16±17.03 | 55.69±16.37 | 58.53±14.26 | 59.07±13.97 | 56.87±12.11 | 56.62±7.70 | 54.17±5.16 | 48.30±1.27 | 42.27±2.74 | 33.88±2.36 | 24.70±1.88 |
| | SD | 75.64±1.15 | 75.02±1.12 | 75.95±0.66 | 76.47±0.55 | 71.13±0.55 | 66.90±0.51 | 60.40±1.18 | 48.03±2.18 | 38.80±2.27 | 28.73±1.93 | 23.03±0.30 |
| | CORAL | 71.85±5.36 | 72.69±5.89 | 73.28±5.71 | 72.42±5.36 | 69.12±5.93 | 65.45±3.66 | 58.22±2.18 | 47.20±2.05 | 38.87±1.58 | 30.27±2.28 | 22.90±1.03 |
| | GroupDRO | 74.45±5.36 | 74.59±4.52 | 75.62±4.10 | 74.48±4.18 | 70.20±3.36 | 66.70±1.45 | 59.00±0.68 | 47.13±1.83 | 37.85±1.20 | 29.43±2.14 | 22.88±1.14 |
| | RSC | 78.54±1.52 | 78.44±1.43 | 77.62±0.89 | 77.48±0.17 | 72.28±0.31 | 66.88±1.33 | 58.48±1.55 | 45.53±1.31 | 35.93±0.54 | 27.62±1.33 | 22.32±0.78 |
| | DANN | 64.25±2.22 | 64.20±2.20 | 63.60±3.26 | 61.48±1.65 | 55.42±1.42 | 51.67±1.52 | 45.92±0.61 | 37.98±1.03 | 32.13±1.12 | 25.35±1.09 | 21.97±0.76 |
| | Mixup | 66.90±10.09 | 67.71±8.84 | 68.60±8.23 | 67.67±8.21 | 64.08±7.85 | 60.57±4.90 | 55.82±4.08 | 46.13±1.21 | 38.15±0.56 | 29.83±0.63 | 24.40±1.33 |
| | CAD | 40.97±14.78 | 41.24±14.51 | 42.37±14.34 | 42.27±14.48 | 39.97±13.05 | 40.45±12.49 | 39.17±11.38 | 35.78±8.01 | 33.45±4.98 | 28.77±1.82 | 25.62±1.00 |
| | IB-IRM | 34.38±6.42 | 33.81±5.87 | 34.83±6.37 | 35.18±5.99 | 34.65±4.47 | 32.48±5.83 | 31.22±5.98 | 28.32±3.39 | 26.43±2.11 | 23.43±1.00 | 21.67±1.01 |

Table 13: Average accuracy of top-3 models of each algorithm at each shift degree of LOWLIGHT-CIFAR10 (Simple CNN). This table shows the results on shift degrees from $\mathcal{D}_8$ to $\mathcal{D}_{10}$.

| | Algorithm | $\mathcal{D}_0$ | $\mathcal{D}_1$ | $\mathcal{D}_2$ | $\mathcal{D}_3$ | $\mathcal{D}_4$ | $\mathcal{D}_5$ | $\mathcal{D}_6$ | $\mathcal{D}_7$ | $\mathcal{D}_8$ | $\mathcal{D}_9$ | $\mathcal{D}_{10}$ |
|---|---|---|---|---|---|---|---|---|---|---|---|---|
| $\mathcal{D}_0$ | ERM | 81.01±0.10 | 80.37±0.42 | 80.40±0.83 | 77.58±0.50 | 71.12±2.23 | 63.72±3.68 | 50.48±3.66 | 36.22±3.19 | 26.90±2.36 | 19.20±2.20 | 14.98±1.29 |
| | VREx | 80.27±0.41 | 79.59±0.97 | 80.08±0.57 | 77.33±0.96 | 71.38±1.06 | 64.30±2.37 | 50.02±3.94 | 34.47±4.26 | 23.97±3.92 | 16.50±2.95 | 13.08±2.05 |
| | IRM | 78.99±1.38 | 78.46±1.52 | 78.32±1.06 | 76.15±1.32 | 68.65±3.01 | 56.18±7.10 | 55.15±5.68 | 24.18±5.14 | 15.75±3.97 | 13.58±3.84 | |
| | SD | 81.38±0.05 | 80.47±0.59 | 79.60±0.29 | 77.97±0.51 | 71.62±0.90 | 63.20±0.46 | 51.05±1.64 | 36.68±1.22 | 26.85±2.25 | 19.98±1.60 | 16.35±1.69 |
| | CORAL | 79.87±0.26 | 78.61±0.63 | 78.08±0.21 | 75.27±1.09 | 68.00±0.28 | 58.80±0.78 | 44.70±1.70 | 28.88±2.28 | 19.13±2.54 | 13.12±1.94 | 12.12±1.64 |
| | GroupDRO | 80.43±0.08 | 79.87±0.48 | 80.40±0.81 | 77.57±0.70 | 70.65±1.53 | 63.65±0.97 | 50.38±0.90 | 34.42±0.20 | 24.80±1.21 | 17.07±1.38 | 14.53±2.57 |
| | RSC | 80.10±0.44 | 79.11±0.99 | 80.43±1.26 | 77.40±1.06 | 71.68±1.68 | 64.53±1.75 | 52.48±3.75 | 37.27±3.82 | 27.12±2.74 | 18.15±2.27 | 14.23±0.95 |
| | DANN | 60.86±1.48 | 59.37±1.19 | 60.05±1.49 | 56.25±1.32 | 49.90±0.92 | 45.82±0.56 | 39.55±1.81 | 31.75±2.74 | 25.32±3.08 | 17.75±2.44 | 14.73±2.40 |
| | Mixup | 78.33±0.58 | 78.68±1.31 | 78.18±1.88 | 75.38±2.50 | 68.23±2.18 | 59.82±2.10 | 46.42±2.11 | 30.52±3.18 | 20.88±2.67 | 14.05±2.31 | 12.13±1.27 |
| | CAD | 77.74±0.68 | 77.37±0.73 | 78.37±0.74 | 75.83±0.80 | 69.27±0.98 | 61.17±1.62 | 49.10±1.21 | 33.58±1.44 | 23.73±1.70 | 14.57±1.99 | 12.68±1.68 |
| | IB-IRM | 61.19±9.57 | 59.69±9.14 | 61.05±8.80 | 58.75±8.99 | 52.30±9.78 | 45.73±10.46 | 36.57±8.70 | 25.93±6.55 | 19.00±4.13 | 14.07±2.38 | 12.45±1.67 |
| $\mathcal{D}_1$ | ERM | 80.41±0.61 | 80.76±0.11 | 80.42±0.35 | 77.97±0.27 | 72.42±1.26 | 65.48±2.43 | 53.47±1.60 | 39.08±1.16 | 28.48±1.35 | 19.17±2.27 | 14.85±1.30 |
| | VREx | 80.21±0.47 | 79.91±1.11 | 79.70±0.45 | 77.03±1.04 | 70.53±0.42 | 61.35±1.16 | 46.58±2.02 | 30.68±1.55 | 21.07±2.01 | 14.03±1.76 | 11.97±1.12 |
| | IRM | 78.34±1.78 | 79.25±1.54 | 78.87±1.82 | 76.52±0.91 | 69.03±2.39 | 60.77±3.15 | 46.82±5.10 | 32.08±6.47 | 21.78±4.44 | 14.63±2.93 | 12.68±1.57 |
| | SD | 79.56±0.43 | 80.50±0.10 | 80.17±0.41 | 78.05±0.51 | 71.63±1.40 | 61.88±1.46 | 46.35±1.00 | 36.35±1.06 | 24.97±2.74 | 16.53±2.56 | 13.58±2.01 |
| | CORAL | 78.99±0.49 | 79.53±0.18 | 78.42±0.08 | 76.60±0.84 | 69.28±1.81 | 60.30±1.45 | 48.42±3.41 | 32.88±3.39 | 23.10±2.47 | 15.55±2.25 | 13.08±1.36 |
| | GroupDRO | 80.19±0.32 | 80.59±0.28 | 80.62±0.27 | 77.43±0.77 | 70.72±1.57 | 62.80±1.93 | 48.27±2.36 | 31.55±2.75 | 21.68±2.79 | 14.25±2.61 | 12.32±1.67 |
| | RSC | 79.52±0.78 | 79.74±0.40 | 80.72±1.05 | 77.48±1.31 | 71.60±2.05 | 63.70±2.78 | 50.85±5.07 | 34.40±4.39 | 23.17±4.36 | 14.97±3.23 | 12.75±1.82 |
| | DANN | 60.09±1.33 | 59.80±1.37 | 60.95±1.82 | 58.28±2.08 | 51.97±1.59 | 47.68±1.15 | 40.52±0.94 | 33.72±1.69 | 26.12±2.56 | 17.92±2.26 | 14.78±1.36 |
| | Mixup | 78.29±0.60 | 79.05±0.80 | 79.00±0.76 | 76.68±0.68 | 68.80±1.66 | 59.52±2.39 | 45.03±2.50 | 28.03±2.38 | 18.03±2.19 | 12.00±1.82 | 11.03±1.26 |
| | CAD | 76.94±0.47 | 77.65±0.80 | 77.62±0.55 | 75.07±0.31 | 68.82±0.14 | 64.83±3.22 | 61.75±1.23 | 34.83±3.22 | 23.80±2.67 | 14.22±1.95 | 12.27±0.97 |
| | IB-IRM | 60.59±8.72 | 59.77±9.25 | 61.18±8.99 | 58.92±9.22 | 52.65±10.27 | 46.37±11.34 | 37.25±9.66 | 26.45±7.28 | 18.65±3.65 | 13.23±1.71 | 11.78±1.74 |
| $\mathcal{D}_2$ | ERM | 80.35±0.63 | 79.67±0.41 | 81.65±0.64 | 78.63±0.84 | 72.47±1.50 | 65.02±2.46 | 52.35±3.39 | 36.08±3.76 | 25.45±4.06 | 17.07±3.54 | 13.40±2.49 |
| | VREx | 80.25±0.41 | 79.85±1.18 | 80.18±0.44 | 77.98±0.41 | 71.45±1.03 | 64.33±2.33 | 48.92±3.50 | 32.28±3.38 | 21.50±2.58 | 14.30±2.13 | 11.45±0.40 |
| | IRM | 78.19±1.34 | 78.35±1.95 | 79.32±1.63 | 75.97±1.27 | 70.10±1.69 | 63.18±1.55 | 50.53±2.46 | 36.37±3.58 | 23.88±2.71 | 15.35±2.12 | 12.48±1.48 |
| | SD | 80.21±0.47 | 80.53±0.30 | 81.55±0.42 | 80.18±0.61 | 74.65±1.11 | 67.25±0.86 | 55.25±0.47 | 39.83±0.38 | 27.65±1.55 | 18.60±2.19 | 15.32±1.80 |
| | CORAL | 78.01±0.73 | 78.49±0.43 | 79.85±0.39 | 76.13±0.33 | 69.83±1.74 | 60.30±4.18 | 46.92±5.86 | 32.10±4.86 | 23.22±2.67 | 16.18±1.37 | 13.78±0.63 |
| | GroupDRO | 80.22±0.28 | 80.16±0.85 | 81.13±0.53 | 78.08±0.59 | 71.30±1.07 | 64.57±1.07 | 51.20±2.05 | 34.87±2.16 | 24.60±2.93 | 16.75±2.25 | 13.58±1.12 |
| | RSC | 79.55±0.76 | 79.33±0.80 | 80.88±0.94 | 77.67±1.37 | 72.43±2.43 | 64.75±3.22 | 52.38±5.18 | 36.43±4.50 | 25.50±4.29 | 16.75±3.21 | 14.02±1.61 |
| | DANN | 60.09±1.33 | 59.80±1.37 | 60.95±1.82 | 58.28±2.08 | 51.97±1.59 | 47.68±1.15 | 40.52±0.94 | 33.72±1.69 | 26.12±2.56 | 17.92±2.26 | 14.78±1.36 |
| | Mixup | 77.83±1.11 | 78.94±0.95 | 79.25±0.45 | 77.48±0.50 | 69.75±1.51 | 61.40±2.50 | 46.40±2.10 | 29.37±2.24 | 19.18±1.35 | 12.48±1.51 | 11.25±1.10 |
| | CAD | 77.71±0.70 | 77.41±0.66 | 78.53±0.54 | 76.00±0.82 | 70.13±1.06 | 62.27±1.50 | 50.32±2.01 | 34.00±1.66 | 22.93±1.01 | 13.70±0.90 | 11.57±0.19 |
| | IB-IRM | 60.86±9.09 | 59.72±9.18 | 61.82±9.88 | 59.47±9.99 | 53.52±11.48 | 46.48±11.50 | 37.85±10.50 | 27.45±8.70 | 19.85±5.30 | 14.60±2.98 | 13.12±2.08 |
| $\mathcal{D}_3$ | ERM | 79.95±0.26 | 80.13±0.29 | 80.52±0.36 | 79.73±0.05 | 73.63±1.05 | 65.35±2.62 | 52.82±3.35 | 37.20±4.99 | 26.10±4.23 | 16.90±2.22 | 12.80±1.26 |
| | VREx | 78.83±1.66 | 78.50±1.92 | 79.63±0.82 | 78.37±0.23 | 71.63±0.86 | 64.30±1.67 | 51.07±1.52 | 34.40±2.09 | 22.92±1.70 | 15.23±1.74 | 11.80±0.55 |
| | IRM | 78.47±1.91 | 78.97±1.23 | 78.35±1.13 | 76.55±0.95 | 69.05±2.41 | 60.72±3.09 | 47.10±5.50 | 32.37±6.87 | 22.95±6.02 | 15.78±4.33 | 14.30±3.53 |
| | SD | 80.21±0.47 | 80.53±0.30 | 81.55±0.42 | 80.18±0.61 | 74.65±1.11 | 67.25±0.86 | 55.25±0.47 | 39.83±0.38 | 27.65±1.55 | 18.60±2.19 | 15.32±1.80 |
| | CORAL | 77.92±0.81 | 78.43±0.64 | 78.68±0.30 | 77.37±0.22 | 70.92±0.74 | 62.93±1.07 | 51.52±1.86 | 37.12±1.49 | 27.02±0.56 | 18.08±0.60 | 14.67±0.69 |
| | GroupDRO | 79.66±0.77 | 79.77±0.64 | 80.35±1.53 | 78.45±0.08 | 72.17±1.00 | 66.03±1.07 | 53.45±2.05 | 37.38±2.38 | 26.73±0.91 | 17.82±0.84 | 13.92±0.66 |
| | RSC | 79.00±1.31 | 79.17±0.82 | 80.20±1.47 | 78.72±0.19 | 72.62±1.16 | 66.48±1.85 | 55.52±2.24 | 39.12±1.45 | 29.30±1.63 | 19.42±2.96 | 15.87±2.89 |
| | DANN | 60.09±1.33 | 59.80±1.37 | 60.95±1.82 | 58.28±2.08 | 51.97±1.59 | 47.68±1.15 | 40.52±0.94 | 33.72±1.69 | 26.12±2.56 | 17.92±2.26 | 14.78±1.36 |
| | Mixup | 77.57±0.84 | 78.59±0.70 | 78.78±0.29 | 77.82±0.34 | 69.87±1.62 | 61.80±2.55 | 48.00±4.05 | 32.18±5.69 | 22.00±5.23 | 14.83±4.80 | 12.68±3.12 |
| | CAD | 77.40±1.00 | 76.72±0.85 | 78.07±0.92 | 76.10±0.68 | 70.28±0.86 | 63.42±0.16 | 51.52±0.81 | 35.93±1.07 | 25.02±1.99 | 15.58±1.84 | 12.82±1.58 |
| | IB-IRM | 60.86±9.09 | 59.72±9.18 | 61.82±9.88 | 59.47±9.99 | 53.52±11.48 | 46.48±11.50 | 37.85±10.50 | 27.45±8.70 | 19.85±5.30 | 14.60±2.98 | 13.12±2.08 |

Table 14: Average accuracy of top-3 models of each algorithm at each shift degree of LOWLIGHT-CIFAR10 (EfficientNet-b0). This table shows the results on shift degrees from $\mathcal{D}_0$ to $\mathcal{D}_3$.

| | Algorithm | $\mathcal{D}_0$ | $\mathcal{D}_1$ | $\mathcal{D}_2$ | $\mathcal{D}_3$ | $\mathcal{D}_4$ | $\mathcal{D}_5$ | $\mathcal{D}_6$ | $\mathcal{D}_7$ | $\mathcal{D}_8$ | $\mathcal{D}_9$ | $\mathcal{D}_{10}$ |
|---|---|---|---|---|---|---|---|---|---|---|---|---|
| $\mathcal{D}_4$ | ERM | 80.36±0.58 | 80.27±0.46 | 80.65±0.36 | 79.25±0.71 | 74.27±0.40 | 67.38±2.18 | 54.92±2.34 | 39.60±4.32 | 28.73±3.43 | 19.53±2.38 | 14.52±1.74 |
| | VREx | 77.97±1.35 | 77.29±0.79 | 79.17±1.06 | 77.85±0.96 | 72.10±0.83 | 65.63±1.62 | 54.00±3.02 | 39.68±5.58 | 28.28±6.14 | 18.70±3.45 | 13.93±2.49 |
| | IRM | 77.75±2.06 | 77.84±1.80 | 78.73±1.14 | 76.23±1.03 | 70.78±1.23 | 63.90±0.72 | 51.53±2.45 | 36.58±3.99 | 25.72±4.03 | 17.63±2.64 | 14.83±2.95 |
| | SD | 80.53±0.52 | 80.62±0.33 | 81.03±0.80 | 80.00±0.76 | 75.20±0.75 | 67.50±0.92 | 55.02±1.07 | 39.25±1.77 | 27.08±1.14 | 17.13±1.13 | 14.97±0.65 |
| | CORAL | 78.08±0.84 | 78.63±0.57 | 79.12±0.48 | 76.97±0.49 | 71.82±0.39 | 62.28±2.02 | 51.27±3.04 | 35.60±2.29 | 24.97±1.38 | 17.22±0.74 | 14.12±0.40 |
| | GroupDRO | 79.19±0.93 | 79.51±0.99 | 79.42±1.13 | 77.70±0.99 | 72.65±0.32 | 65.78±1.22 | 53.47±2.05 | 36.70±2.47 | 25.22±1.62 | 16.43±1.24 | 13.33±0.85 |
| | RSC | 78.61±1.87 | 79.17±0.83 | 80.45±1.41 | 78.15±0.71 | 73.73±0.60 | 66.68±0.48 | 55.65±1.38 | 40.03±0.96 | 28.95±0.91 | 19.10±0.43 | 15.05±0.19 |
| | DANN | 58.12±2.68 | 57.89±2.67 | 59.95±2.98 | 57.55±2.64 | 52.23±1.61 | 47.95±1.14 | 41.43±1.59 | 34.05±1.96 | 26.78±2.79 | 18.92±2.53 | 15.38±1.43 |
| | Mixup | 76.67±1.31 | 77.39±1.31 | 77.97±0.90 | 77.15±1.28 | 70.92±0.44 | 63.93±1.62 | 51.48±3.11 | 37.70±5.27 | 28.05±6.88 | 19.92±5.98 | 15.57±3.60 |
| | CAD | 77.01±1.47 | 76.75±0.82 | 77.70±1.38 | 75.68±1.25 | 70.33±0.79 | 63.50±0.27 | 53.23±2.41 | 37.28±2.98 | 26.03±3.42 | 16.07±2.51 | 12.63±1.32 |
| | IB-IRM | 60.86±9.09 | 59.72±9.18 | 61.82±9.88 | 59.47±9.99 | 53.52±11.48 | 46.48±11.50 | 37.85±10.50 | 27.45±8.70 | 19.85±5.30 | 14.60±2.98 | 13.12±2.08 |
| $\mathcal{D}_5$ | ERM | 79.79±1.33 | 79.63±1.02 | 80.15±0.53 | 78.72±0.77 | 73.43±1.50 | 68.43±0.70 | 56.15±0.85 | 42.32±1.47 | 30.77±0.87 | 20.75±1.11 | 15.57±0.88 |
| | VREx | 77.97±1.35 | 77.29±0.79 | 79.17±1.06 | 77.85±0.96 | 72.10±0.83 | 65.63±1.62 | 54.00±3.02 | 39.68±5.58 | 28.28±6.14 | 18.70±3.45 | 13.93±2.49 |
| | IRM | 77.63±1.90 | 78.13±2.18 | 79.25±1.71 | 76.20±0.99 | 70.77±1.21 | 63.95±0.79 | 51.25±2.07 | 36.30±3.60 | 24.55±2.48 | 16.48±1.04 | 13.22±0.66 |
| | SD | 80.32±0.62 | 80.65±0.35 | 81.35±0.62 | 80.02±0.75 | 75.08±0.88 | 67.70±0.66 | 55.67±0.16 | 40.15±0.51 | 27.33±1.30 | 17.18±1.20 | 14.28±0.39 |
| | CORAL | 75.87±0.86 | 76.04±1.39 | 77.50±1.51 | 76.07±0.91 | 70.83±1.23 | 65.27±0.80 | 54.43±0.85 | 39.07±1.63 | 27.43±1.91 | 17.88±1.29 | 14.33±1.55 |
| | GroupDRO | 78.92±0.67 | 78.51±1.29 | 79.45±1.70 | 77.80±0.92 | 71.50±1.03 | 66.13±0.97 | 54.90±0.75 | 38.95±0.96 | 27.67±0.51 | 18.10±0.90 | 14.33±1.12 |
| | RSC | 79.05±1.31 | 79.01±0.91 | 80.77±1.05 | 78.68±0.24 | 73.67±0.69 | 67.50±0.67 | 56.15±1.36 | 39.65±0.72 | 28.10±0.86 | 17.92±1.59 | 14.32±1.18 |
| | DANN | 58.12±2.68 | 57.89±2.67 | 59.95±2.98 | 57.55±2.64 | 52.23±1.61 | 47.95±1.14 | 41.43±1.59 | 34.05±1.96 | 26.78±2.79 | 18.92±2.53 | 15.38±1.43 |
| | Mixup | 75.53±0.65 | 75.69±1.45 | 76.57±0.66 | 75.07±1.42 | 70.35±0.41 | 65.13±0.71 | 54.72±0.57 | 41.70±0.99 | 30.88±3.50 | 21.57±3.34 | 16.90±1.68 |
| | CAD | 76.70±1.49 | 76.53±0.94 | 77.38±1.39 | 75.33±1.19 | 69.67±0.70 | 63.63±0.16 | 52.80±2.59 | 38.02±2.64 | 27.53±2.82 | 17.23±2.40 | 13.75±1.47 |
| | IB-IRM | 60.86±9.09 | 59.72±9.18 | 61.82±9.88 | 59.47±9.99 | 53.52±11.48 | 46.48±11.50 | 37.85±10.50 | 27.45±8.70 | 19.85±5.30 | 14.60±2.98 | 13.12±2.08 |
| $\mathcal{D}_6$ | ERM | 77.89±2.89 | 77.93±3.12 | 79.05±2.23 | 76.43±1.62 | 72.07±1.99 | 66.67±0.56 | 58.12±1.23 | 43.82±2.97 | 33.03±3.91 | 22.37±3.21 | 16.40±2.37 |
| | VREx | 74.77±5.63 | 74.09±5.25 | 76.15±4.74 | 74.23±4.65 | 70.13±3.60 | 65.03±2.47 | 55.40±2.09 | 42.83±4.53 | 31.52±4.95 | 22.07±3.47 | 16.60±3.46 |
| | IRM | 75.25±4.34 | 74.66±5.04 | 76.25±3.33 | 73.92±3.59 | 69.60±2.71 | 62.50±2.23 | 52.67±1.68 | 39.27±3.03 | 29.03±3.35 | 19.58±2.46 | 16.07±2.57 |
| | SD | 79.77±1.31 | 79.64±1.05 | 80.13±0.45 | 78.37±0.67 | 73.53±1.30 | 67.22±0.69 | 55.85±0.08 | 40.73±1.53 | 28.78±1.61 | 17.80±1.87 | 14.45±1.12 |
| | CORAL | 74.39±2.37 | 74.92±2.36 | 76.90±2.21 | 75.40±1.85 | 70.27±1.97 | 65.25±0.82 | 54.75±0.63 | 39.40±1.84 | 27.90±2.53 | 18.43±2.06 | 15.32±2.78 |
| | GroupDRO | 77.73±1.04 | 77.56±1.60 | 77.83±0.59 | 76.45±1.57 | 70.92±1.61 | 65.38±1.70 | 55.03±0.57 | 40.23±1.07 | 29.13±2.12 | 18.85±1.77 | 14.98±1.41 |
| | RSC | 78.35±1.61 | 78.76±1.13 | 80.27±1.41 | 78.02±1.18 | 72.82±0.92 | 67.35±0.79 | 58.05±1.43 | 42.70±3.65 | 31.85±5.09 | 20.45±4.28 | 15.67±2.63 |
| | DANN | 58.12±2.68 | 57.89±2.67 | 59.95±2.98 | 57.55±2.64 | 52.23±1.61 | 47.95±1.14 | 41.43±1.59 | 34.05±1.96 | 26.78±2.79 | 18.92±2.53 | 15.38±1.43 |
| | Mixup | 75.53±0.65 | 75.69±1.45 | 76.57±0.66 | 75.07±1.42 | 70.35±0.41 | 65.13±0.71 | 54.72±0.57 | 41.70±0.99 | 30.88±3.50 | 21.57±3.34 | 16.90±1.68 |
| | CAD | 75.89±0.55 | 76.09±0.33 | 76.72±0.59 | 74.65±0.52 | 69.12±0.10 | 63.53±0.29 | 53.97±2.11 | 39.40±1.66 | 28.73±1.50 | 18.23±1.08 | 14.38±0.60 |
| | IB-IRM | 60.86±9.09 | 59.72±9.18 | 61.82±9.88 | 59.47±9.99 | 53.52±11.48 | 46.48±11.50 | 37.85±10.50 | 27.45±8.70 | 19.85±5.30 | 14.60±2.98 | 13.12±2.08 |
| $\mathcal{D}_7$ | ERM | 77.25±2.57 | 77.03±2.77 | 78.60±2.02 | 76.23±1.51 | 71.62±1.77 | 66.83±0.51 | 57.75±1.74 | 44.63±1.90 | 33.95±2.92 | 23.52±2.52 | 17.47±2.01 |
| | VREx | 74.77±5.63 | 74.09±5.25 | 76.15±4.74 | 74.23±4.65 | 70.13±3.60 | 65.03±2.47 | 55.40±2.09 | 42.83±4.53 | 31.52±4.95 | 22.07±3.47 | 16.60±3.46 |
| | IRM | 71.24±7.03 | 70.50±7.11 | 71.87±6.45 | 69.55±6.38 | 65.82±5.47 | 62.60±3.67 | 50.55±3.30 | 52.25±2.19 | 40.40±1.48 | 30.17±1.75 | 16.07±2.57 |
| | SD | 74.44±5.47 | 73.67±5.67 | 76.02±4.24 | 73.70±4.41 | 68.53±3.62 | 63.70±2.56 | 54.47±1.27 | 42.37±0.33 | 31.53±0.47 | 22.37±1.40 | 16.70±1.06 |
| | CORAL | 69.49±4.40 | 69.39±4.75 | 72.10±4.52 | 70.40±4.39 | 65.70±3.65 | 60.97±3.01 | 53.35±1.90 | 41.18±0.41 | 29.53±1.98 | 19.55±1.63 | 15.70±2.09 |
| | GroupDRO | 75.36±1.85 | 75.20±1.17 | 75.57±2.06 | 73.77±2.02 | 68.02±0.99 | 62.73±1.09 | 52.58±1.23 | 41.63±1.17 | 31.65±2.57 | 23.05±3.81 | 18.37±3.39 |
| | RSC | 75.01±3.02 | 76.20±1.38 | 76.55±2.15 | 74.33±1.98 | 70.20±1.38 | 64.90±1.27 | 57.00±2.17 | 44.68±2.40 | 35.18±2.69 | 23.37±1.98 | 17.52±1.27 |
| | DANN | 58.12±2.68 | 57.89±2.67 | 59.95±2.98 | 57.55±2.64 | 52.23±1.61 | 47.95±1.14 | 41.43±1.59 | 34.05±1.96 | 26.78±2.79 | 18.92±2.53 | 15.38±1.43 |
| | Mixup | 75.53±0.65 | 75.69±1.45 | 76.57±0.66 | 75.07±1.42 | 70.35±0.41 | 65.13±0.71 | 54.72±0.57 | 41.70±0.99 | 30.88±3.50 | 21.57±3.34 | 16.90±1.68 |
| | CAD | 73.31±3.42 | 73.53±3.82 | 74.47±3.03 | 72.32±2.93 | 67.12±2.84 | 61.25±3.19 | 52.80±3.66 | 39.55±1.48 | 28.87±1.43 | 19.22±1.72 | 15.35±1.87 |
| | IB-IRM | 60.86±9.09 | 59.72±9.18 | 61.82±9.88 | 59.47±9.99 | 53.52±11.48 | 46.48±11.50 | 37.85±10.50 | 27.45±8.70 | 19.85±5.30 | 14.60±2.98 | 13.12±2.08 |

Table 15: Average accuracy of top-3 models of each algorithm at each shift degree of LowLight-CIFAR10 (EfficientNet-b0). This table shows the results on shift degrees from $\mathcal{D}_4$ to $\mathcal{D}_7$.

| | Algorithm | $\mathcal{D}_0$ | $\mathcal{D}_1$ | $\mathcal{D}_2$ | $\mathcal{D}_3$ | $\mathcal{D}_4$ | $\mathcal{D}_5$ | $\mathcal{D}_6$ | $\mathcal{D}_7$ | $\mathcal{D}_8$ | $\mathcal{D}_9$ | $\mathcal{D}_{10}$ |
|---|---|---|---|---|---|---|---|---|---|---|---|---|
| $\mathcal{D}_8$ | ERM | 70.93±3.50 | 70.77±3.08 | 72.77±4.15 | 71.67±3.58 | 66.27±3.54 | 62.37±4.06 | 54.15±4.12 | 43.23±2.60 | 35.82±1.57 | 25.37±1.12 | 18.30±1.81 |
| | VREx | 68.97±6.42 | 68.84±6.26 | 71.33±5.05 | 69.38±5.27 | 65.18±5.92 | 60.52±5.68 | 52.77±5.56 | 42.58±4.87 | 32.73±3.39 | 22.88±2.36 | 17.97±1.77 |
| | IRM | 75.37±4.35 | 74.77±5.06 | 75.95±3.27 | 73.83±3.56 | 69.02±2.67 | 62.05±2.17 | 52.40±2.00 | 40.02±2.00 | 30.82±0.84 | 21.22±0.15 | 17.62±1.05 |
| | SD | 71.21±3.67 | 71.16±4.27 | 73.62±3.02 | 71.22±2.77 | 66.27±2.29 | 62.47±1.90 | 53.35±0.48 | 41.43±0.33 | 32.18±0.15 | 21.60±1.35 | 17.08±1.34 |
| | CORAL | 69.13±1.99 | 69.19±2.45 | 71.57±2.19 | 69.47±2.40 | 64.40±3.25 | 60.10±3.31 | 52.32±2.45 | 40.78±0.82 | 31.45±0.52 | 21.50±0.15 | 18.82±0.16 |
| | GroupDRO | 70.05±9.51 | 69.77±9.34 | 70.18±10.07 | 68.47±9.80 | 63.12±8.60 | 58.15±7.53 | 50.28±5.19 | 41.03±1.84 | 33.77±1.27 | 24.68±2.86 | 19.97±2.87 |
| | RSC | 68.48±6.14 | 69.82±5.83 | 71.17±5.51 | 69.12±5.24 | 65.17±4.73 | 61.00±4.17 | 55.17±3.56 | 43.83±3.00 | 36.07±2.05 | 24.87±0.89 | 18.02±0.76 |
| | DANN | 53.93±7.08 | 53.98±6.46 | 55.35±5.67 | 52.68±5.69 | 47.30±4.87 | 43.43±4.48 | 39.20±3.20 | 33.35±2.66 | 28.40±2.10 | 19.68±1.31 | 15.72±1.02 |
| | Mixup | 74.58±0.45 | 74.81±1.14 | 76.00±0.76 | 73.50±1.34 | 68.97±1.36 | 62.45±1.91 | 52.77±0.96 | 40.28±1.79 | 32.53±2.38 | 24.73±1.21 | 19.45±1.01 |
| | CAD | 65.01±7.48 | 64.71±8.09 | 65.93±7.18 | 64.95±6.50 | 59.50±6.90 | 54.75±6.46 | 47.88±6.08 | 37.83±2.59 | 30.23±0.61 | 22.03±1.89 | 17.47±2.14 |
| | IB-IRM | 43.56±16.80 | 43.59±15.11 | 46.05±15.86 | 43.83±15.89 | 40.63±13.12 | 35.53±12.47 | 31.45±10.04 | 25.88±7.05 | 21.45±4.57 | 18.32±2.67 | 16.77±2.08 |
| $\mathcal{D}_9$ | ERM | 73.17±5.61 | 73.26±5.45 | 74.25±5.45 | 72.68±4.49 | 67.52±4.54 | 62.27±4.02 | 53.72±4.21 | 42.97±2.78 | 34.98±2.49 | 25.55±0.93 | 18.63±2.06 |
| | VREx | 68.97±6.42 | 68.84±6.26 | 71.33±5.05 | 69.38±5.27 | 65.18±5.92 | 60.52±5.68 | 52.77±5.56 | 42.58±4.87 | 32.73±3.39 | 22.88±2.36 | 17.97±1.77 |
| | IRM | 74.55±3.45 | 73.89±4.21 | 75.38±2.64 | 73.13±2.90 | 67.75±1.27 | 61.33±1.28 | 51.78±1.32 | 39.35±1.47 | 30.48±1.37 | 21.27±0.21 | 17.10±0.46 |
| | SD | 63.81±8.81 | 63.81±7.43 | 66.87±7.83 | 65.65±6.40 | 62.02±5.75 | 58.35±4.89 | 50.27±2.97 | 39.13±1.98 | 30.48±1.37 | 23.57±1.05 | 17.95±0.43 |
| | CORAL | 61.08±3.16 | 61.51±3.78 | 64.40±4.30 | 61.60±5.47 | 56.72±5.25 | 50.55±4.65 | 43.43±3.23 | 35.75±2.22 | 28.97±1.05 | 22.40±0.51 | 17.43±1.13 |
| | GroupDRO | 63.18±9.96 | 63.01±9.61 | 63.62±9.90 | 61.82±10.21 | 57.07±8.72 | 52.87±7.59 | 46.85±4.79 | 39.47±2.55 | 32.65±2.82 | 25.48±1.95 | 21.05±1.55 |
| | RSC | 71.68±6.33 | 72.82±6.68 | 73.92±5.42 | 71.62±5.41 | 66.98±4.76 | 62.22±4.00 | 53.82±1.61 | 41.82±1.79 | 33.92±1.66 | 25.60±1.92 | 19.17±1.36 |
| | DANN | 54.77±3.12 | 54.40±2.50 | 55.97±2.21 | 52.72±3.59 | 47.13±3.64 | 43.27±3.18 | 38.75±3.25 | 31.47±2.55 | 26.78±1.93 | 20.70±1.44 | 17.67±1.41 |
| | Mixup | 74.58±0.45 | 74.81±1.14 | 76.00±0.76 | 73.50±1.34 | 68.97±1.36 | 62.45±1.91 | 52.77±0.96 | 40.28±1.79 | 32.53±2.38 | 24.73±1.21 | 19.45±1.01 |
| | CAD | 59.34±2.36 | 58.89±1.73 | 60.65±1.41 | 59.23±2.16 | 53.75±1.27 | 48.73±2.16 | 42.23±2.00 | 34.80±1.70 | 29.07±1.29 | 22.47±1.35 | 18.05±1.34 |
| | IB-IRM | 30.82±18.03 | 31.67±17.47 | 33.95±19.08 | 32.02±19.04 | 30.88±17.48 | 27.78±16.55 | 26.03±13.66 | 23.25±9.39 | 21.22±4.78 | 18.52±2.40 | 17.62±1.19 |
| $\mathcal{D}_{10}$ | ERM | 75.31±2.72 | 75.00±2.98 | 77.05±1.92 | 75.15±1.34 | 69.45±1.07 | 65.27±0.85 | 55.02±3.18 | 42.40±3.42 | 34.13±3.27 | 24.28±2.04 | 19.85±0.41 |
| | VREx | 57.47±6.85 | 57.48±6.59 | 59.47±7.20 | 56.62±8.05 | 53.12±8.46 | 49.58±8.48 | 43.82±7.99 | 35.57±6.22 | 27.93±3.49 | 21.22±3.22 | 18.42±1.39 |
| | IRM | 75.37±4.35 | 74.77±5.06 | 75.95±3.27 | 73.83±3.56 | 69.02±2.67 | 62.05±2.17 | 52.40±2.00 | 40.02±2.00 | 30.82±0.84 | 21.22±0.15 | 17.62±1.05 |
| | SD | 73.05±10.83 | 72.51±11.37 | 72.73±9.75 | 71.07±10.09 | 65.00±9.76 | 58.08±7.70 | 48.30±5.11 | 35.37±2.18 | 27.70±1.36 | 21.32±0.34 | 19.02±0.32 |
| | CORAL | 70.36±2.87 | 70.21±2.85 | 72.28±2.47 | 70.52±2.37 | 63.97±3.13 | 59.07±3.77 | 49.53±3.86 | 38.30±2.86 | 29.80±1.85 | 21.25±0.22 | 18.93±0.06 |
| | GroupDRO | 68.11±8.59 | 67.84±8.37 | 68.50±9.17 | 66.73±9.00 | 61.23±7.68 | 55.95±6.60 | 48.08±4.36 | 39.35±2.67 | 32.78±2.63 | 25.00±2.48 | 21.17±1.44 |
| | RSC | 70.12±7.56 | 69.83±8.26 | 70.57±8.25 | 68.65±9.38 | 63.72±7.53 | 59.70±5.22 | 50.52±2.88 | 38.75±1.31 | 31.78±1.02 | 24.25±1.30 | 20.12±1.00 |
| | DANN | 49.88±6.34 | 50.28±5.31 | 52.67±5.08 | 48.35±6.24 | 42.88±5.04 | 38.40±4.80 | 33.92±4.26 | 27.93±2.75 | 24.05±2.56 | 19.50±2.06 | 18.22±0.78 |
| | Mixup | 72.81±2.82 | 72.47±2.96 | 73.85±2.88 | 71.25±3.80 | 66.27±4.32 | 59.88±4.95 | 49.62±4.64 | 38.33±4.10 | 31.13±3.41 | 24.15±1.95 | 19.80±0.78 |
| | CAD | 56.01±2.02 | 55.86±1.71 | 58.05±1.37 | 56.82±1.45 | 52.40±2.03 | 48.35±0.83 | 41.93±0.80 | 34.08±1.41 | 27.60±1.49 | 20.72±1.45 | 19.08±1.86 |
| | IB-IRM | 30.82±18.03 | 31.67±17.47 | 33.95±19.08 | 32.02±19.04 | 30.88±17.48 | 27.78±16.55 | 26.03±13.66 | 23.25±9.39 | 21.22±4.78 | 18.52±2.40 | 17.62±1.19 |

Table 16: Average accuracy of top-3 models of each algorithm at each shift degree of LowLight-CIFAR10 (EfficientNet-b0). This table shows the results on shift degrees from $\mathcal{D}_8$ to $\mathcal{D}_{10}$.

| | Algorithm | $\mathcal{D}_0$ | $\mathcal{D}_1$ | $\mathcal{D}_2$ | $\mathcal{D}_3$ | $\mathcal{D}_4$ | $\mathcal{D}_5$ | $\mathcal{D}_6$ | $\mathcal{D}_7$ | $\mathcal{D}_8$ | $\mathcal{D}_9$ | $\mathcal{D}_{10}$ |
|---|---|---|---|---|---|---|---|---|---|---|---|---|
| $\mathcal{D}_0$ | ERM | 83.83±0.22 | 82.92±0.92 | 14.60±1.54 | 14.98±1.76 | 14.72±0.95 | 14.97±1.61 | 13.60±2.02 | 15.23±1.95 | 13.70±1.75 | 13.47±1.72 | 13.57±0.80 |
| | VREx | 81.43±0.85 | 81.20±0.63 | 16.20±0.53 | 16.68±1.32 | 15.65±1.08 | 15.95±0.94 | 14.80±1.78 | 15.62±1.82 | 14.57±1.79 | 13.60±1.34 | 13.45±1.25 |
| | IRM | 63.35±14.01 | 62.91±13.53 | 15.78±1.55 | 17.33±0.29 | 16.97±0.26 | 17.05±0.53 | 18.07±0.97 | 16.83±1.12 | 16.45±0.12 | 15.47±1.10 | 15.37±1.10 |
| | SD | 84.37±0.31 | 83.59±1.34 | 16.82±1.97 | 17.13±1.76 | 17.03±1.27 | 16.58±0.58 | 16.07±0.69 | 15.93±0.40 | 15.07±0.47 | 14.40±0.04 | 13.77±0.23 |
| | CORAL | 84.80±0.27 | 84.18±0.68 | 16.65±0.96 | 16.93±1.30 | 15.40±0.84 | 16.75±2.16 | 15.70±1.06 | 15.62±0.88 | 14.87±0.21 | 14.20±0.39 | 14.08±0.70 |
| | GroupDRO | 84.88±0.08 | 83.73±0.26 | 15.48±0.48 | 15.43±0.32 | 16.46±0.66 | 14.85±0.60 | 14.40±0.47 | 13.15±0.90 | 13.37±0.52 | 12.82±0.02 | 12.83±0.35 |
| | RSC | 84.81±0.30 | 83.99±0.04 | 11.00±0.91 | 10.37±0.80 | 9.68±0.87 | 10.58±1.00 | 11.00±1.13 | 10.33±0.80 | 10.77±1.13 | 9.95±0.50 | 10.00±0.33 |
| | DANN | 84.09±1.06 | 82.95±0.49 | 15.77±0.78 | 16.52±0.93 | 15.60±0.83 | 15.95±0.29 | 15.58±1.10 | 14.83±1.21 | 15.10±0.48 | 14.22±0.86 | 14.33±0.31 |
| | Mixup | 84.11±0.60 | 83.72±0.60 | 14.63±1.10 | 15.05±0.32 | 15.18±0.84 | 14.80±0.96 | 13.12±0.59 | 13.53±0.66 | 12.37±0.79 | 11.77±0.53 | 11.37±0.45 |
| | CAD | 86.09±0.27 | 85.41±0.30 | 15.63±1.94 | 16.55±1.76 | 15.88±1.64 | 17.38±1.33 | 15.47±0.74 | 15.83±1.16 | 15.25±0.95 | 14.68±1.60 | 14.15±0.97 |
| | IB-IRM | 72.89±9.99 | 72.99±10.17 | 15.40±0.66 | 14.40±2.41 | 14.37±2.52 | 14.17±2.56 | 14.40±2.41 | 13.05±1.90 | 12.52±1.97 | 11.90±1.04 | 12.25±0.86 |
| $\mathcal{D}_1$ | ERM | 83.14±0.36 | 83.99±0.06 | 16.23±1.01 | 16.23±1.45 | 16.33±0.39 | 16.65±0.78 | 15.72±1.03 | 16.02±1.51 | 15.12±1.40 | 14.78±1.75 | 13.75±0.99 |
| | VREx | 81.33±0.88 | 81.53±0.26 | 16.92±1.50 | 16.83±1.53 | 15.47±0.84 | 15.92±0.91 | 15.15±1.95 | 15.23±1.65 | 14.43±1.75 | 13.10±1.32 | 13.20±1.07 |
| | IRM | 63.13±13.71 | 63.48±13.55 | 14.50±2.02 | 16.00±1.29 | 15.25±1.87 | 15.97±1.44 | 17.02±1.37 | 15.93±1.41 | 15.53±1.05 | 13.65±2.49 | 13.83±2.70 |
| | SD | 84.01±0.44 | 84.49±0.56 | 17.12±1.47 | 17.67±1.96 | 16.18±1.46 | 16.82±1.44 | 16.75±1.43 | 16.37±0.90 | 15.95±0.80 | 14.35±0.56 | 14.40±0.53 |
| | CORAL | 84.42±0.55 | 84.59±0.21 | 16.62±1.41 | 16.30±0.37 | 15.17±1.15 | 16.30±0.92 | 15.98±1.09 | 15.93±1.86 | 15.18±0.49 | 13.55±0.51 | 14.17±0.51 |
| | GroupDRO | 83.97±0.94 | 84.47±0.07 | 16.23±1.71 | 16.32±2.15 | 15.67±0.90 | 16.43±0.87 | 16.13±1.76 | 15.87±1.71 | 15.02±1.31 | 14.70±0.98 | 13.88±0.62 |
| | RSC | 84.53±0.63 | 84.29±0.35 | 10.42±1.39 | 10.40±0.78 | 9.85±0.89 | 10.22±1.31 | 10.97±1.16 | 10.50±0.56 | 10.63±1.29 | 10.18±0.16 | 9.98±0.30 |
| | DANN | 83.28±0.93 | 83.17±0.53 | 15.25±1.00 | 15.35±1.98 | 14.98±1.22 | 14.48±1.39 | 14.08±2.37 | 14.13±1.76 | 13.72±1.76 | 13.03±1.82 | 13.13±1.14 |
| | Mixup | 83.69±1.12 | 84.51±0.24 | 15.52±2.32 | 16.47±2.29 | 15.57±1.38 | 16.12±2.09 | 14.57±2.26 | 14.63±2.24 | 13.50±2.12 | 12.75±1.96 | 12.30±1.82 |
| | CAD | 85.14±0.28 | 85.75±0.16 | 16.80±1.49 | 16.80±0.84 | 16.20±1.89 | 17.40±1.85 | 15.75±1.27 | 16.13±1.55 | 16.12±1.85 | 15.03±2.34 | 14.53±1.85 |
| | IB-IRM | 72.89±9.99 | 72.99±10.17 | 15.40±0.66 | 14.92±1.86 | 14.37±2.52 | 14.17±2.56 | 14.40±2.41 | 13.05±1.90 | 12.52±1.97 | 11.90±1.04 | 12.25±0.86 |
| $\mathcal{D}_2$ | ERM | 71.15±7.59 | 71.43±9.15 | 20.68±0.83 | 20.28±0.89 | 19.73±0.94 | 19.80±0.29 | 18.50±1.17 | 18.33±0.87 | 17.93±0.69 | 15.67±1.26 | 16.12±0.09 |
| | VREx | 69.10±11.41 | 69.67±10.81 | 19.35±0.39 | 17.83±1.19 | 17.27±0.51 | 17.22±0.41 | 17.00±0.49 | 15.72±0.36 | 15.90±0.71 | 14.02±1.14 | 13.62±0.89 |
| | IRM | 57.03±16.04 | 57.44±15.62 | 17.77±0.90 | 17.62±0.56 | 18.00±1.26 | 18.03±1.79 | 17.78±0.76 | 17.27±0.82 | 17.18±0.74 | 14.97±0.65 | 14.58±1.05 |
| | SD | 72.66±8.30 | 73.11±8.01 | 21.00±0.76 | 20.70±0.70 | 20.08±0.88 | 20.38±0.94 | 18.98±0.87 | 17.03±1.65 | 15.52±0.82 | 14.83±0.10 | 13.97±0.32 |
| | CORAL | 68.62±5.20 | 68.17±5.62 | 20.33±0.26 | 19.65±0.90 | 18.47±1.68 | 19.27±0.98 | 18.70±1.53 | 17.30±1.22 | 16.20±1.71 | 14.23±1.44 | 14.03±1.98 |
| | GroupDRO | 83.58±0.43 | 82.63±0.86 | 20.57±0.74 | 19.77±0.95 | 18.37±1.17 | 18.83±1.85 | 17.73±1.11 | 17.55±0.76 | 16.08±0.76 | 14.42±0.19 | 13.68±0.21 |
| | RSC | 69.31±5.98 | 69.81±6.20 | 17.67±0.37 | 16.68±1.69 | 15.73±1.87 | 16.92±1.81 | 16.38±1.21 | 14.97±1.47 | 14.55±1.59 | 12.25±1.81 | 12.63±1.91 |
| | DANN | 61.93±21.32 | 61.86±21.16 | 21.03±0.52 | 21.38±0.12 | 19.32±0.31 | 19.52±1.57 | 17.97±1.29 | 16.77±2.12 | 15.22±1.10 | 14.50±0.37 | 13.38±0.97 |
| | Mixup | 78.97±2.43 | 79.93±2.79 | 19.32±0.14 | 18.65±0.74 | 17.27±0.44 | 18.12±0.58 | 16.52±0.65 | 16.57±1.09 | 15.32±1.19 | 13.67±1.27 | 13.50±1.25 |
| | CAD | 80.38±2.07 | 79.67±2.47 | 20.98±0.25 | 20.27±0.32 | 19.07±0.81 | 20.73±0.12 | 18.50±0.19 | 18.72±0.44 | 17.78±1.20 | 16.78±0.53 | 16.07±1.56 |
| | IB-IRM | 55.41±18.53 | 55.74±18.43 | 17.50±1.33 | 16.88±1.64 | 15.83±1.57 | 15.53±2.81 | 14.65±1.95 | 14.03±1.96 | 13.47±2.10 | 12.78±1.52 | 11.60±1.39 |
| $\mathcal{D}_3$ | ERM | 75.29±3.49 | 75.37±1.86 | 20.03±1.28 | 21.20±0.36 | 20.22±0.65 | 19.87±0.33 | 17.40±1.26 | 17.40±1.00 | 17.18±1.33 | 15.50±2.11 | 15.93±1.17 |
| | VREx | 72.66±10.40 | 71.25±12.12 | 18.28±0.59 | 19.27±0.56 | 17.35±0.58 | 17.27±0.66 | 16.73±0.89 | 15.42±1.13 | 15.35±0.47 | 13.25±0.54 | 13.87±0.41 |
| | IRM | 57.10±16.14 | 57.37±15.52 | 17.43±0.55 | 17.92±0.96 | 18.15±1.25 | 18.53±1.72 | 18.22±1.09 | 17.78±1.11 | 16.67±0.18 | 14.70±0.60 | 14.50±0.94 |
| | SD | 76.49±3.60 | 76.27±4.22 | 20.32±1.20 | 20.87±0.55 | 19.50±0.75 | 20.42±0.90 | 19.55±0.33 | 17.98±0.31 | 16.03±0.23 | 14.82±0.10 | 13.88±0.37 |
| | CORAL | 73.76±1.79 | 73.53±1.89 | 19.78±0.26 | 20.82±0.35 | 19.33±1.08 | 19.12±1.10 | 17.63±2.05 | 16.32±2.18 | 15.73±2.06 | 14.43±1.89 | 14.05±1.92 |
| | GroupDRO | 81.68±0.71 | 82.48±1.47 | 18.07±0.30 | 20.43±0.54 | 17.77±0.99 | 18.15±1.77 | 17.27±1.64 | 17.28±1.12 | 14.95±1.20 | 13.80±1.03 | 14.32±0.58 |
| | RSC | 63.69±6.17 | 63.93±6.58 | 17.32±0.86 | 17.45±0.61 | 16.65±0.73 | 17.62±0.97 | 17.35±0.29 | 16.00±0.16 | 15.78±0.18 | 13.82±1.33 | 14.08±1.69 |
| | DANN | 70.89±6.21 | 70.68±6.77 | 20.32±0.41 | 21.53±0.18 | 19.67±0.41 | 20.23±2.07 | 18.18±1.56 | 16.63±2.11 | 15.13±1.23 | 14.90±1.00 | 13.57±0.76 |
| | Mixup | 76.82±1.35 | 77.98±1.52 | 18.45±0.63 | 19.77±0.17 | 17.80±0.70 | 18.90±0.27 | 17.63±0.22 | 17.95±0.66 | 16.67±0.25 | 15.17±0.66 | 14.45±1.04 |
| | CAD | 77.40±2.59 | 76.65±2.34 | 20.38±1.06 | 20.58±0.32 | 19.47±0.66 | 20.45±0.50 | 19.45±0.29 | 17.98±0.88 | 16.93±0.90 | 16.05±0.78 | 14.90±1.00 |
| | IB-IRM | 54.35±20.78 | 54.00±21.14 | 16.92±1.75 | 17.37±1.30 | 16.52±1.52 | 16.90±2.23 | 16.13±1.60 | 15.28±1.37 | 14.50±2.37 | 13.85±1.72 | 12.90±2.40 |

Table 17: Average accuracy of top-3 models of each algorithm at each shift degree of LowLight-CIFAR10 (ResNet-50). This table shows the results on shift degrees from $\mathcal{D}_0$ to $\mathcal{D}_3$.

| | Algorithm | $\mathcal{D}_0$ | $\mathcal{D}_1$ | $\mathcal{D}_2$ | $\mathcal{D}_3$ | $\mathcal{D}_4$ | $\mathcal{D}_5$ | $\mathcal{D}_6$ | $\mathcal{D}_7$ | $\mathcal{D}_8$ | $\mathcal{D}_9$ | $\mathcal{D}_{10}$ |
|---|---|---|---|---|---|---|---|---|---|---|---|---|
| $\mathcal{D}_4$ | ERM | 75.73±4.01 | 76.44±3.37 | 20.38±1.11 | 20.53±0.87 | 20.33±0.49 | 19.87±0.33 | 18.65±1.21 | 18.28±0.84 | 18.25±0.81 | 16.77±1.69 | 16.53±0.55 |
| | VREx | 73.05±3.38 | 73.41±3.63 | 16.63±0.92 | 17.30±0.60 | 18.63±0.22 | 18.12±1.04 | 17.88±1.03 | 17.92±0.86 | 16.13±0.27 | 13.65±0.57 | 14.10±0.11 |
| | IRM | 56.55±15.38 | 56.85±14.80 | 17.65±0.76 | 17.85±0.87 | 18.48±1.34 | 18.50±1.72 | 17.88±0.81 | 17.63±0.98 | 16.90±0.37 | 15.13±0.78 | 14.87±1.43 |
| | SD | 70.95±8.17 | 71.59±8.01 | 20.42±1.54 | 20.43±0.87 | 20.47±0.35 | 19.28±0.92 | 18.67±0.92 | 16.78±1.51 | 15.37±0.68 | 15.02±0.17 | 14.13±0.55 |
| | CORAL | 72.33±5.84 | 72.04±5.81 | 18.62±2.03 | 18.98±1.84 | 19.93±0.51 | 19.22±0.87 | 18.65±0.97 | 17.55±1.16 | 16.78±0.95 | 15.12±0.79 | 14.65±1.82 |
| | GroupDRO | 74.64±9.89 | 74.94±8.93 | 19.25±1.68 | 19.75±0.98 | 19.73±0.31 | 19.12±2.04 | 17.22±2.21 | 15.92±3.22 | 15.63±3.76 | 13.70±1.88 | 13.05±2.45 |
| | RSC | 67.18±6.60 | 67.31±6.87 | 17.13±0.75 | 17.00±0.71 | 16.73±0.84 | 17.38±0.66 | 17.15±0.32 | 16.07±0.25 | 15.53±0.39 | 13.93±1.40 | 14.27±1.83 |
| | DANN | 70.89±6.21 | 70.68±6.77 | 20.32±0.41 | 21.53±0.18 | 19.67±0.41 | 20.23±2.07 | 18.18±1.56 | 16.63±2.11 | 15.13±1.23 | 14.90±1.00 | 13.57±0.76 |
| | Mixup | 75.21±5.30 | 75.15±5.52 | 18.30±0.29 | 19.35±0.22 | 19.48±0.25 | 19.52±1.00 | 18.05±0.74 | 18.63±0.96 | 16.83±1.86 | 15.67±1.64 | 14.80±1.59 |
| | CAD | 75.08±2.00 | 74.13±2.00 | 20.03±0.98 | 20.48±0.46 | 19.87±0.25 | 20.28±0.47 | 19.08±0.31 | 17.37±0.53 | 16.55±0.37 | 15.78±0.49 | 14.67±0.67 |
| | IB-IRM | 64.23±11.59 | 64.03±12.05 | 16.60±1.95 | 17.03±1.66 | 17.38±0.47 | 17.77±1.13 | 16.92±0.55 | 15.82±0.62 | 15.32±1.22 | 14.10±1.41 | 13.63±1.51 |
| $\mathcal{D}_5$ | ERM | 68.71±4.04 | 67.91±4.44 | 18.63±0.81 | 19.20±0.04 | 19.47±0.12 | 20.30±0.07 | 19.53±1.07 | 18.98±1.91 | 16.92±1.51 | 15.62±0.92 | 15.07±1.16 |
| | VREx | 68.16±7.15 | 67.51±8.39 | 18.28±0.34 | 18.28±0.33 | 18.37±0.20 | 19.53±0.67 | 18.53±0.80 | 18.78±0.93 | 16.92±1.40 | 14.73±1.83 | 15.22±2.07 |
| | IRM | 57.10±16.14 | 57.37±15.52 | 17.43±0.55 | 17.92±0.96 | 18.15±1.25 | 18.53±1.72 | 18.22±1.09 | 17.78±1.11 | 16.67±0.18 | 14.70±0.60 | 14.50±0.94 |
| | SD | 79.25±3.09 | 78.93±3.46 | 19.05±0.88 | 19.58±0.90 | 18.82±0.65 | 21.25±0.23 | 19.28±0.12 | 18.93±1.04 | 17.33±0.70 | 15.70±0.99 | 15.43±0.78 |
| | CORAL | 71.07±7.43 | 70.50±7.09 | 18.55±0.23 | 18.88±0.53 | 18.18±0.66 | 20.47±0.16 | 18.37±0.61 | 17.43±0.46 | 16.42±0.47 | 15.70±0.29 | 14.55±0.84 |
| | GroupDRO | 77.59±4.55 | 78.01±4.20 | 18.45±0.49 | 18.88±0.19 | 19.33±0.49 | 20.92±0.90 | 18.97±0.40 | 18.85±0.43 | 17.52±1.35 | 15.55±0.43 | 14.95±0.47 |
| | RSC | 63.69±6.17 | 63.93±6.58 | 17.32±0.86 | 17.45±0.61 | 16.65±0.73 | 17.62±0.97 | 17.35±0.29 | 16.00±0.16 | 15.78±0.18 | 13.82±1.33 | 14.08±1.69 |
| | DANN | 55.27±16.80 | 55.03±16.60 | 20.58±0.57 | 21.45±0.21 | 19.63±0.45 | 21.37±0.47 | 19.12±0.62 | 18.23±0.70 | 15.80±0.70 | 15.10±0.85 | 13.63±0.67 |
| | Mixup | 70.30±4.94 | 70.23±5.07 | 18.00±1.06 | 18.68±1.05 | 18.58±1.26 | 19.92±0.66 | 19.10±0.47 | 19.23±0.19 | 17.37±0.66 | 15.90±0.85 | 15.42±1.41 |
| | CAD | 79.13±0.96 | 78.26±1.07 | 19.85±1.11 | 20.02±0.52 | 19.03±1.10 | 20.78±0.08 | 19.02±0.48 | 18.78±0.57 | 16.80±0.48 | 15.88±0.24 | 14.50±0.33 |
| | IB-IRM | 50.17±25.97 | 50.22±25.79 | 17.05±1.70 | 17.30±1.36 | 16.87±1.05 | 17.85±1.04 | 17.20±0.29 | 16.33±0.17 | 16.57±0.57 | 15.13±0.78 | 14.72±1.05 |
| $\mathcal{D}_6$ | ERM | 78.19±4.32 | 77.93±4.21 | 18.32±0.83 | 19.00±0.60 | 18.45±0.69 | 19.08±0.98 | 20.30±0.61 | 20.02±1.53 | 18.12±0.59 | 16.67±0.33 | 16.05±0.59 |
| | VREx | 59.83±0.99 | 59.85±2.70 | 17.33±0.83 | 17.85±1.00 | 17.85±0.27 | 18.95±0.99 | 19.10±0.27 | 18.38±1.88 | 17.77±0.86 | 16.08±1.09 | 15.87±1.58 |
| | IRM | 57.10±16.14 | 57.37±15.52 | 17.43±0.55 | 17.92±0.96 | 18.15±1.25 | 18.53±1.72 | 18.22±1.09 | 17.78±1.11 | 16.67±0.18 | 14.70±0.60 | 14.50±0.94 |
| | SD | 68.76±7.24 | 69.38±6.98 | 16.98±1.13 | 19.13±0.25 | 18.53±0.31 | 19.53±0.97 | 20.82±0.86 | 20.43±1.65 | 18.48±0.87 | 16.58±1.37 | 16.53±1.11 |
| | CORAL | 73.74±2.08 | 73.69±2.22 | 20.05±0.58 | 19.93±0.71 | 19.13±1.08 | 19.68±0.41 | 20.00±0.37 | 18.63±1.31 | 18.00±0.97 | 16.52±1.79 | 16.75±1.88 |
| | GroupDRO | 65.62±5.68 | 66.42±5.28 | 18.12±0.45 | 18.78±0.35 | 18.75±0.32 | 19.63±1.08 | 19.48±0.65 | 18.28±0.72 | 16.92±1.48 | 15.23±1.03 | 14.02±2.18 |
| | RSC | 65.63±9.78 | 65.55±9.60 | 15.43±1.62 | 15.83±1.60 | 15.73±1.17 | 17.48±0.94 | 17.95±0.91 | 16.42±0.40 | 15.70±0.88 | 15.05±1.29 | 14.00±1.25 |
| | DANN | 55.27±16.80 | 55.03±16.60 | 20.58±0.57 | 21.45±0.21 | 19.63±0.45 | 21.37±0.47 | 19.12±0.62 | 18.23±0.70 | 15.80±0.70 | 15.10±0.85 | 13.63±0.67 |
| | Mixup | 72.94±6.81 | 73.20±7.09 | 17.23±1.30 | 18.05±0.97 | 17.38±1.55 | 18.72±1.04 | 19.32±0.35 | 18.98±0.48 | 17.57±0.59 | 15.98±0.91 | 15.12±1.50 |
| | CAD | 76.96±3.17 | 76.15±2.99 | 19.50±2.30 | 19.63±1.14 | 18.95±0.97 | 20.25±0.78 | 19.98±0.53 | 19.03±0.78 | 17.97±1.40 | 17.25±1.10 | 16.27±1.80 |
| | IB-IRM | 50.17±25.97 | 50.22±25.79 | 17.05±1.70 | 17.30±1.36 | 16.87±1.05 | 17.85±1.04 | 17.20±0.29 | 16.33±0.17 | 16.57±0.57 | 15.13±0.78 | 14.72±1.05 |
| $\mathcal{D}_7$ | ERM | 75.54±2.90 | 75.45±3.13 | 18.17±0.91 | 18.92±0.71 | 18.53±0.59 | 19.80±0.71 | 19.93±1.09 | 20.57±0.78 | 18.30±0.48 | 16.38±0.37 | 16.27±0.31 |
| | VREx | 60.56±1.77 | 59.70±2.61 | 15.73±2.86 | 16.55±1.76 | 16.73±1.64 | 18.22±1.74 | 18.93±0.49 | 19.40±0.51 | 18.03±0.56 | 16.78±0.74 | 15.65±1.75 |
| | IRM | 57.52±16.71 | 57.99±16.38 | 17.38±0.52 | 17.50±0.41 | 17.93±1.27 | 18.03±1.79 | 17.58±0.74 | 17.80±1.13 | 16.65±0.19 | 14.80±0.59 | 14.58±1.05 |
| | SD | 75.98±3.74 | 75.69±3.23 | 16.78±0.86 | 18.77±0.31 | 18.50±0.36 | 20.37±0.72 | 20.63±1.05 | 21.20±0.57 | 18.68±0.59 | 17.38±0.27 | 16.77±0.96 |
| | CORAL | 72.43±6.02 | 72.67±5.40 | 17.70±1.52 | 18.65±0.95 | 17.75±0.60 | 18.68±0.59 | 19.13±1.01 | 19.55±0.43 | 17.82±1.01 | 16.47±1.83 | 16.77±1.86 |
| | GroupDRO | 77.19±1.75 | 77.85±2.10 | 17.37±1.07 | 18.52±0.59 | 18.18±1.52 | 19.90±0.58 | 19.02±0.70 | 19.38±0.39 | 18.33±0.76 | 15.47±0.51 | 14.57±0.87 |
| | RSC | 71.41±9.63 | 71.45±9.78 | 16.22±0.60 | 16.33±0.92 | 15.97±0.88 | 17.25±1.11 | 17.80±1.07 | 16.57±0.41 | 16.55±0.43 | 15.78±0.74 | 15.35±0.78 |
| | DANN | 55.27±16.80 | 55.03±16.60 | 20.58±0.57 | 21.45±0.21 | 19.63±0.45 | 21.37±0.47 | 19.12±0.62 | 18.23±0.70 | 15.80±0.70 | 15.10±0.85 | 13.63±0.67 |
| | Mixup | 66.43±9.85 | 66.65±9.77 | 17.65±1.26 | 18.32±1.29 | 17.35±1.34 | 18.78±0.81 | 18.95±0.57 | 19.55±0.07 | 18.17±0.78 | 16.95±0.25 | 16.73±0.33 |
| | CAD | 78.31±4.24 | 78.05±4.39 | 18.87±1.90 | 19.30±0.88 | 18.00±0.35 | 20.15±0.71 | 19.52±0.88 | 19.52±0.39 | 18.62±0.96 | 17.35±1.08 | 17.05±1.57 |
| | IB-IRM | 42.05±27.48 | 42.16±27.56 | 16.73±2.05 | 17.00±1.52 | 16.25±1.28 | 17.60±1.17 | 16.95±0.11 | 16.75±0.43 | 16.52±0.59 | 14.65±0.62 | 13.80±1.04 |

Table 18: Average accuracy of top-3 models of each algorithm at each shift degree of LowLight-CIFAR10 (ResNet-50). This table shows the results on shift degrees from $\mathcal{D}_4$ to $\mathcal{D}_7$.

| | Algorithm | $\mathcal{D}_0$ | $\mathcal{D}_1$ | $\mathcal{D}_2$ | $\mathcal{D}_3$ | $\mathcal{D}_4$ | $\mathcal{D}_5$ | $\mathcal{D}_6$ | $\mathcal{D}_7$ | $\mathcal{D}_8$ | $\mathcal{D}_9$ | $\mathcal{D}_{10}$ |
|---|---|---|---|---|---|---|---|---|---|---|---|---|
| $\mathcal{D}_8$ | ERM | 75.98±2.23 | 75.15±3.41 | 18.30±0.76 | 19.60±0.86 | 19.13±1.11 | 19.80±0.39 | 19.43±0.37 | 19.48±0.87 | 19.12±0.41 | 18.57±0.68 | 17.42±0.52 |
| | VREx | 60.56±1.77 | 59.70±2.61 | 15.73±2.86 | 16.55±1.76 | 16.73±1.64 | 18.22±1.74 | 18.93±0.49 | 19.40±0.51 | 18.03±0.56 | 16.18±0.96 | 15.65±1.75 |
| | IRM | 44.99±26.43 | 45.65±26.30 | 16.00±3.32 | 16.40±2.18 | 16.45±3.28 | 17.27±2.65 | 17.53±1.10 | 17.18±0.92 | 17.20±0.73 | 15.07±0.74 | 14.92±1.10 |
| | SD | 73.58±1.90 | 74.13±2.25 | 16.60±0.63 | 18.38±0.84 | 18.20±0.78 | 19.45±1.06 | 19.82±2.09 | 20.72±1.25 | 19.17±0.10 | 17.60±0.14 | 17.53±1.17 |
| | CORAL | 81.36±3.34 | 81.50±3.33 | 19.02±0.55 | 18.68±0.84 | 17.95±0.44 | 19.32±0.24 | 19.22±0.93 | 18.78±0.95 | 18.48±0.58 | 16.57±1.78 | 16.37±2.22 |
| | GroupDRO | 74.78±9.15 | 75.56±8.89 | 17.23±0.80 | 18.60±0.59 | 18.60±1.20 | 19.03±1.05 | 18.60±1.45 | 18.23±1.09 | 18.97±0.37 | 16.78±0.93 | 16.50±0.71 |
| | RSC | 72.15±10.29 | 72.61±10.78 | 15.82±0.22 | 15.83±0.61 | 15.10±0.63 | 16.28±0.31 | 16.73±0.66 | 16.18±0.41 | 16.77±0.55 | 15.70±0.63 | 15.68±0.48 |
| | DANN | 50.20±15.53 | 51.31±14.61 | 16.98±3.08 | 17.58±2.84 | 17.08±2.29 | 18.52±2.72 | 17.80±1.74 | 16.82±1.74 | 16.68±0.18 | 15.28±0.73 | 13.85±0.56 |
| | Mixup | 67.31±8.62 | 67.48±8.63 | 17.67±1.24 | 18.18±1.48 | 17.33±1.36 | 18.67±0.97 | 18.77±0.72 | 18.67±1.18 | 18.37±0.53 | 16.27±0.93 | 15.77±1.53 |
| | CAD | 66.80±16.15 | 66.62±16.00 | 18.40±1.82 | 18.78±0.83 | 17.82±0.59 | 19.47±0.83 | 19.45±0.90 | 19.23±0.57 | 19.50±0.39 | 18.32±0.73 | 18.22±0.26 |
| | IB-IRM | 50.17±25.97 | 50.22±25.79 | 17.05±1.70 | 17.30±1.36 | 16.87±1.05 | 17.85±1.04 | 17.20±0.29 | 16.33±0.17 | 16.57±0.57 | 15.13±0.78 | 14.72±1.05 |
| $\mathcal{D}_9$ | ERM | 79.09±2.83 | 78.59±3.20 | 17.77±1.00 | 19.02±1.24 | 18.50±1.35 | 19.00±1.02 | 19.52±0.35 | 19.18±1.10 | 18.82±0.74 | 18.80±0.46 | 17.05±0.19 |
| | VREx | 66.77±10.37 | 66.40±10.73 | 16.65±1.62 | 16.75±1.53 | 16.92±1.38 | 18.08±1.90 | 18.30±1.38 | 18.97±1.08 | 17.58±1.09 | 16.50±0.59 | 15.80±1.63 |
| | IRM | 46.58±23.15 | 46.09±22.42 | 13.65±1.90 | 15.18±1.62 | 15.18±2.30 | 15.62±1.39 | 17.08±0.99 | 16.25±0.46 | 16.33±0.12 | 15.70±1.27 | 15.50±0.62 |
| | SD | 61.63±0.75 | 63.10±0.54 | 14.37±0.52 | 15.83±0.71 | 15.83±0.85 | 16.57±1.47 | 18.60±1.04 | 18.35±0.54 | 17.77±1.17 | 18.77±0.33 | 18.32±0.57 |
| | CORAL | 70.51±5.86 | 70.55±5.71 | 18.17±1.86 | 19.58±1.37 | 18.30±0.98 | 19.22±0.29 | 18.90±1.11 | 17.88±1.57 | 17.75±1.17 | 17.12±1.37 | 16.50±2.05 |
| | GroupDRO | 69.58±9.37 | 69.59±9.38 | 16.25±0.92 | 17.05±1.32 | 17.43±1.38 | 17.58±1.02 | 17.23±0.88 | 17.20±0.37 | 17.77±0.52 | 17.20±0.57 | 16.78±0.64 |
| | RSC | 71.41±9.63 | 71.45±9.78 | 16.22±0.60 | 16.33±0.92 | 15.97±0.88 | 17.25±1.11 | 17.80±1.07 | 16.57±0.41 | 16.55±0.43 | 15.78±0.74 | 15.35±0.78 |
| | DANN | 64.61±17.95 | 65.02±17.14 | 17.70±2.23 | 18.52±2.04 | 17.33±2.09 | 18.20±2.52 | 17.93±1.43 | 17.10±1.25 | 16.43±0.49 | 16.58±1.22 | 15.48±1.29 |
| | Mixup | 66.76±10.16 | 66.93±10.04 | 17.42±1.02 | 18.25±1.24 | 17.27±1.38 | 18.67±0.72 | 18.57±0.05 | 19.33±0.35 | 17.93±0.81 | 17.35±0.32 | 16.47±0.06 |
| | CAD | 66.36±15.72 | 66.43±15.80 | 17.65±0.99 | 18.37±0.27 | 18.12±0.94 | 19.32±0.62 | 19.68±0.73 | 19.08±0.65 | 19.32±0.63 | 18.55±0.40 | 17.85±0.78 |
| | IB-IRM | 50.43±26.27 | 50.35±25.94 | 15.97±0.50 | 16.27±0.42 | 15.78±0.94 | 16.70±0.94 | 16.47±1.02 | 16.70±0.94 | 15.37±1.33 | 15.87±1.45 | 15.17±0.73 |
| $\mathcal{D}_{10}$ | ERM | 72.55±7.32 | 72.05±7.16 | 16.42±3.13 | 17.93±2.32 | 17.57±1.88 | 18.50±1.66 | 19.05±0.14 | 18.48±0.84 | 18.67±0.66 | 17.60±0.28 | 17.62±0.34 |
| | VREx | 59.83±0.99 | 59.85±2.70 | 17.33±0.83 | 17.85±1.00 | 17.85±0.27 | 18.95±0.99 | 19.10±0.27 | 18.38±1.88 | 17.77±0.86 | 16.08±1.09 | 15.87±1.58 |
| | IRM | 43.40±20.38 | 43.37±20.15 | 13.82±2.11 | 15.87±1.79 | 15.05±2.24 | 16.03±1.47 | 17.32±0.99 | 16.37±0.63 | 16.67±0.39 | 15.53±1.08 | 15.83±1.07 |
| | SD | 68.13±9.91 | 69.50±9.48 | 14.95±1.23 | 16.08±0.89 | 16.03±1.05 | 16.12±1.35 | 18.23±0.97 | 18.57±1.15 | 18.35±0.62 | 18.35±0.55 | 18.78±0.38 |
| | CORAL | 81.49±3.43 | 81.30±3.18 | 17.47±1.37 | 18.18±1.10 | 17.13±1.02 | 18.25±0.66 | 18.35±1.50 | 17.98±1.49 | 17.80±1.03 | 16.55±1.83 | 17.48±1.36 |
| | GroupDRO | 68.73±9.89 | 69.32±9.55 | 16.37±1.08 | 17.40±1.66 | 17.30±1.20 | 17.93±1.47 | 17.88±1.79 | 17.70±1.02 | 18.28±0.69 | 17.17±0.60 | 17.17±0.25 |
| | RSC | 73.23±10.96 | 73.07±10.93 | 14.92±0.86 | 14.98±1.15 | 14.55±0.99 | 15.25±1.09 | 15.93±1.28 | 16.02±0.47 | 16.08±0.55 | 15.38±0.34 | 16.28±0.77 |
| | DANN | 66.21±18.60 | 66.33±17.66 | 17.97±2.60 | 18.52±2.04 | 17.20±1.91 | 17.80±1.95 | 17.55±0.90 | 16.78±0.80 | 16.28±0.45 | 16.12±1.48 | 15.58±1.21 |
| | Mixup | 60.08±13.67 | 61.42±12.98 | 16.22±2.02 | 17.02±1.98 | 15.92±2.08 | 17.27±2.48 | 17.75±1.95 | 17.97±1.93 | 17.43±0.71 | 16.43±0.24 | 17.18±0.06 |
| | CAD | 63.55±15.39 | 63.79±15.30 | 16.62±1.17 | 17.63±0.57 | 17.25±0.54 | 18.30±0.49 | 18.33±0.53 | 18.33±0.23 | 19.03±0.65 | 18.40±0.49 | 18.65±0.46 |
| | IB-IRM | 50.85±26.77 | 50.97±26.68 | 16.03±0.52 | 16.62±0.52 | 15.87±0.86 | 16.57±0.89 | 16.88±0.69 | 15.45±1.21 | 15.97±1.31 | 14.73±1.34 | 14.77±0.98 |

Table 19: Average accuracy of top-3 models of each algorithm at each shift degree of LowLight-CIFAR10 (ResNet-50). This table shows the results on shift degrees from $\mathcal{D}_8$ to $\mathcal{D}_{10}$.

## B.5 Complete results of linear probing

| | $\mathcal{D}_0$ | $\mathcal{D}_1$ | $\mathcal{D}_2$ | $\mathcal{D}_3$ | $\mathcal{D}_4$ | $\mathcal{D}_5$ | $\mathcal{D}_6$ | $\mathcal{D}_7$ | $\mathcal{D}_8$ | $\mathcal{D}_9$ | $\mathcal{D}_{10}$ |
|---|---|---|---|---|---|---|---|---|---|---|---|
| $\text{IN}_0$ | $98.4_{\pm0.1}$ | $39.8_{\pm4.2}$ | $24.8_{\pm4.7}$ | $16.7_{\pm3.2}$ | $12.4_{\pm2.1}$ | $10.9_{\pm2.0}$ | $10.3_{\pm1.5}$ | $10.1_{\pm1.4}$ | $9.8_{\pm1.1}$ | $9.6_{\pm0.8}$ | $9.5_{\pm0.6}$ |
| $\text{IN}_1$ | $97.9_{\pm0.1}$ | $97.3_{\pm0.2}$ | $90.2_{\pm0.7}$ | $62.9_{\pm2.0}$ | $33.1_{\pm1.7}$ | $20.1_{\pm1.9}$ | $15.4_{\pm1.7}$ | $13.3_{\pm1.1}$ | $11.7_{\pm1.1}$ | $10.8_{\pm0.9}$ | $10.5_{\pm1.0}$ |
| $\text{IN}_2$ | $97.6_{\pm0.2}$ | $97.6_{\pm0.4}$ | $95.8_{\pm0.4}$ | $88.2_{\pm0.5}$ | $65.4_{\pm0.8}$ | $39.9_{\pm1.8}$ | $25.6_{\pm2.6}$ | $19.1_{\pm2.8}$ | $15.7_{\pm2.1}$ | $13.5_{\pm1.3}$ | $12.2_{\pm1.0}$ |
| $\text{IN}_3$ | $97.6_{\pm0.2}$ | $97.2_{\pm0.1}$ | $95.9_{\pm0.1}$ | $92.6_{\pm0.8}$ | $81.8_{\pm0.4}$ | $57.9_{\pm1.3}$ | $36.2_{\pm1.6}$ | $23.5_{\pm1.7}$ | $17.4_{\pm1.9}$ | $14.3_{\pm1.6}$ | $12.7_{\pm1.1}$ |
| $\text{IN}_4$ | $96.6_{\pm0.3}$ | $96.6_{\pm0.5}$ | $96.1_{\pm0.6}$ | $93.5_{\pm0.6}$ | $87.4_{\pm1.0}$ | $74.4_{\pm0.6}$ | $54.4_{\pm1.4}$ | $37.3_{\pm2.0}$ | $26.5_{\pm1.7}$ | $20.4_{\pm1.1}$ | $16.9_{\pm0.9}$ |
| $\text{CLIP}_0$ | $98.1_{\pm0.1}$ | $55.4_{\pm4.1}$ | $35.9_{\pm4.4}$ | $26.4_{\pm3.3}$ | $21.1_{\pm2.7}$ | $18.1_{\pm1.4}$ | $15.8_{\pm0.4}$ | $14.1_{\pm0.4}$ | $13.0_{\pm0.4}$ | $12.3_{\pm0.5}$ | $12.1_{\pm0.2}$ |
| $\text{CLIP}_1$ | $97.9_{\pm0.2}$ | $95.9_{\pm0.2}$ | $86.3_{\pm0.7}$ | $63.0_{\pm2.1}$ | $41.2_{\pm2.4}$ | $26.2_{\pm1.2}$ | $18.3_{\pm1.0}$ | $14.5_{\pm1.0}$ | $12.8_{\pm0.8}$ | $11.8_{\pm0.8}$ | $11.4_{\pm0.7}$ |
| $\text{CLIP}_2$ | $97.4_{\pm0.2}$ | $96.2_{\pm0.2}$ | $92.7_{\pm0.5}$ | $81.0_{\pm0.6}$ | $58.3_{\pm1.7}$ | $35.9_{\pm1.7}$ | $22.9_{\pm0.6}$ | $16.9_{\pm0.4}$ | $14.0_{\pm0.3}$ | $12.8_{\pm0.6}$ | $11.8_{\pm0.4}$ |
| $\text{CLIP}_3$ | $97.0_{\pm0.3}$ | $95.7_{\pm0.4}$ | $93.0_{\pm0.7}$ | $85.9_{\pm0.6}$ | $70.6_{\pm0.2}$ | $45.9_{\pm1.4}$ | $24.0_{\pm2.1}$ | $15.0_{\pm1.6}$ | $11.7_{\pm0.8}$ | $10.8_{\pm0.5}$ | $10.3_{\pm0.2}$ |
| $\text{CLIP}_4$ | $96.1_{\pm0.4}$ | $95.7_{\pm0.5}$ | $92.9_{\pm0.1}$ | $87.1_{\pm0.7}$ | $76.9_{\pm1.0}$ | $59.3_{\pm0.5}$ | $38.9_{\pm0.6}$ | $24.6_{\pm1.0}$ | $17.0_{\pm1.0}$ | $13.6_{\pm0.6}$ | $12.2_{\pm0.4}$ |

Table 20: ResNet-50 results on NoisyMNIST.

| | $\mathcal{D}_0$ | $\mathcal{D}_1$ | $\mathcal{D}_2$ | $\mathcal{D}_3$ | $\mathcal{D}_4$ | $\mathcal{D}_5$ | $\mathcal{D}_6$ | $\mathcal{D}_7$ | $\mathcal{D}_8$ | $\mathcal{D}_9$ | $\mathcal{D}_{10}$ |
|---|---|---|---|---|---|---|---|---|---|---|---|
| $\text{IN}_0$ | $98.4_{\pm0.3}$ | $69.7_{\pm1.4}$ | $51.7_{\pm1.7}$ | $36.1_{\pm1.6}$ | $25.6_{\pm1.5}$ | $17.6_{\pm0.8}$ | $13.5_{\pm0.4}$ | $11.8_{\pm0.4}$ | $11.6_{\pm0.7}$ | $11.1_{\pm0.5}$ | $10.8_{\pm0.6}$ |
| $\text{IN}_1$ | $98.1_{\pm0.2}$ | $97.6_{\pm0.2}$ | $94.2_{\pm0.2}$ | $82.2_{\pm1.6}$ | $58.4_{\pm4.7}$ | $35.2_{\pm4.7}$ | $23.0_{\pm2.3}$ | $18.0_{\pm1.0}$ | $15.1_{\pm0.5}$ | $13.9_{\pm0.4}$ | $13.1_{\pm0.7}$ |
| $\text{IN}_2$ | $98.1_{\pm0.1}$ | $97.7_{\pm0.4}$ | $96.3_{\pm0.3}$ | $92.2_{\pm0.2}$ | $78.5_{\pm0.6}$ | $52.0_{\pm1.1}$ | $30.0_{\pm1.5}$ | $19.3_{\pm1.7}$ | $14.5_{\pm1.0}$ | $12.5_{\pm0.4}$ | $12.1_{\pm0.5}$ |
| $\text{IN}_3$ | $98.0_{\pm0.5}$ | $97.4_{\pm0.1}$ | $96.5_{\pm0.3}$ | $94.7_{\pm0.5}$ | $85.7_{\pm0.4}$ | $64.0_{\pm1.2}$ | $39.4_{\pm2.2}$ | $24.3_{\pm1.8}$ | $17.6_{\pm1.5}$ | $14.6_{\pm1.5}$ | $13.4_{\pm1.3}$ |
| $\text{IN}_4$ | $97.4_{\pm0.6}$ | $96.6_{\pm0.5}$ | $96.7_{\pm0.2}$ | $94.3_{\pm0.4}$ | $89.1_{\pm0.6}$ | $75.2_{\pm0.6}$ | $53.0_{\pm2.1}$ | $32.4_{\pm3.0}$ | $21.1_{\pm2.8}$ | $15.8_{\pm2.3}$ | $15.3_{\pm1.6}$ |
| $\text{CLIP}_0$ | $98.8_{\pm0.1}$ | $89.7_{\pm1.9}$ | $71.2_{\pm3.7}$ | $48.2_{\pm2.3}$ | $31.2_{\pm1.3}$ | $21.7_{\pm1.0}$ | $16.3_{\pm1.0}$ | $13.7_{\pm1.0}$ | $12.3_{\pm1.1}$ | $11.6_{\pm1.1}$ | $11.2_{\pm1.2}$ |
| $\text{CLIP}_1$ | $98.5_{\pm0.2}$ | $97.8_{\pm0.2}$ | $91.5_{\pm1.0}$ | $69.3_{\pm4.2}$ | $45.0_{\pm5.6}$ | $29.9_{\pm5.5}$ | $22.1_{\pm4.1}$ | $17.8_{\pm3.1}$ | $15.4_{\pm2.0}$ | $14.2_{\pm1.9}$ | $12.9_{\pm1.3}$ |
| $\text{CLIP}_2$ | $98.7_{\pm0.1}$ | $98.0_{\pm0.4}$ | $95.6_{\pm0.4}$ | $86.3_{\pm0.6}$ | $65.7_{\pm1.3}$ | $43.1_{\pm0.6}$ | $29.7_{\pm0.4}$ | $22.6_{\pm0.6}$ | $18.8_{\pm0.5}$ | $16.8_{\pm0.1}$ | $15.8_{\pm0.3}$ |
| $\text{CLIP}_3$ | $98.3_{\pm0.2}$ | $97.5_{\pm0.1}$ | $96.0_{\pm0.1}$ | $90.4_{\pm0.6}$ | $77.2_{\pm0.4}$ | $56.7_{\pm0.6}$ | $38.8_{\pm0.2}$ | $27.6_{\pm0.5}$ | $21.8_{\pm0.4}$ | $18.1_{\pm0.5}$ | $16.3_{\pm0.5}$ |
| $\text{CLIP}_4$ | $98.0_{\pm0.6}$ | $97.4_{\pm0.3}$ | $95.8_{\pm0.6}$ | $91.0_{\pm0.4}$ | $81.3_{\pm0.8}$ | $67.1_{\pm0.7}$ | $50.1_{\pm0.9}$ | $36.6_{\pm0.3}$ | $27.6_{\pm0.5}$ | $22.3_{\pm0.4}$ | $19.1_{\pm0.3}$ |

Table 21: ViT-B/32 results on NoisyMNIST.

| | $\mathcal{D}_0$ | $\mathcal{D}_1$ | $\mathcal{D}_2$ | $\mathcal{D}_3$ | $\mathcal{D}_4$ | $\mathcal{D}_5$ | $\mathcal{D}_6$ | $\mathcal{D}_7$ | $\mathcal{D}_8$ | $\mathcal{D}_9$ | $\mathcal{D}_{10}$ |
|---|---|---|---|---|---|---|---|---|---|---|---|
| $\text{IN}_0$ | $98.5_{\pm0.1}$ | $97.3_{\pm0.1}$ | $94.7_{\pm0.1}$ | $89.2_{\pm0.2}$ | $80.3_{\pm0.2}$ | $68.1_{\pm0.0}$ | $55.0_{\pm0.3}$ | $44.5_{\pm0.4}$ | $38.6_{\pm0.7}$ | $36.5_{\pm0.8}$ | $34.3_{\pm1.2}$ |
| $\text{IN}_1$ | $98.4_{\pm0.1}$ | $98.6_{\pm0.2}$ | $97.4_{\pm0.1}$ | $94.5_{\pm0.1}$ | $88.3_{\pm0.2}$ | $77.4_{\pm0.2}$ | $64.2_{\pm0.6}$ | $53.2_{\pm0.9}$ | $46.2_{\pm1.1}$ | $42.2_{\pm1.4}$ | $38.6_{\pm1.2}$ |
| $\text{IN}_2$ | $98.2_{\pm0.2}$ | $98.5_{\pm0.3}$ | $98.3_{\pm0.1}$ | $97.0_{\pm0.1}$ | $93.8_{\pm0.3}$ | $87.2_{\pm0.5}$ | $77.1_{\pm1.1}$ | $67.1_{\pm0.9}$ | $58.6_{\pm1.4}$ | $52.4_{\pm2.0}$ | $48.6_{\pm2.1}$ |
| $\text{IN}_3$ | $98.1_{\pm0.2}$ | $98.6_{\pm0.3}$ | $98.5_{\pm0.1}$ | $98.6_{\pm0.2}$ | $96.7_{\pm0.1}$ | $93.4_{\pm0.0}$ | $86.7_{\pm0.2}$ | $78.1_{\pm0.8}$ | $68.5_{\pm1.4}$ | $58.9_{\pm1.6}$ | $54.6_{\pm0.7}$ |
| $\text{IN}_4$ | $97.2_{\pm0.2}$ | $98.0_{\pm0.3}$ | $98.4_{\pm0.2}$ | $98.5_{\pm0.6}$ | $97.8_{\pm0.4}$ | $96.4_{\pm0.1}$ | $92.9_{\pm0.2}$ | $86.8_{\pm0.1}$ | $77.6_{\pm0.9}$ | $65.9_{\pm1.4}$ | $61.3_{\pm1.5}$ |
| $\text{CLIP}_0$ | $98.2_{\pm0.1}$ | $96.6_{\pm0.1}$ | $93.4_{\pm0.2}$ | $83.6_{\pm1.0}$ | $69.6_{\pm1.4}$ | $55.4_{\pm1.4}$ | $44.8_{\pm0.7}$ | $40.5_{\pm0.9}$ | $38.0_{\pm1.7}$ | $37.9_{\pm0.6}$ | $34.2_{\pm1.9}$ |
| $\text{CLIP}_1$ | $98.0_{\pm0.1}$ | $98.2_{\pm0.1}$ | $96.3_{\pm0.2}$ | $90.0_{\pm0.3}$ | $78.6_{\pm0.6}$ | $64.7_{\pm0.9}$ | $53.1_{\pm0.6}$ | $48.1_{\pm0.8}$ | $43.0_{\pm0.8}$ | $40.7_{\pm0.7}$ | $37.6_{\pm0.6}$ |
| $\text{CLIP}_2$ | $97.7_{\pm0.1}$ | $98.0_{\pm0.1}$ | $97.5_{\pm0.1}$ | $95.4_{\pm0.2}$ | $90.1_{\pm0.2}$ | $81.0_{\pm0.7}$ | $69.9_{\pm0.9}$ | $60.7_{\pm1.1}$ | $50.5_{\pm0.7}$ | $44.2_{\pm0.7}$ | $39.6_{\pm1.5}$ |
| $\text{CLIP}_3$ | $97.6_{\pm0.2}$ | $98.1_{\pm0.1}$ | $97.9_{\pm0.5}$ | $97.6_{\pm0.4}$ | $95.3_{\pm0.1}$ | $90.5_{\pm0.3}$ | $82.1_{\pm0.5}$ | $71.5_{\pm0.6}$ | $58.3_{\pm0.8}$ | $50.8_{\pm0.6}$ | $44.1_{\pm1.4}$ |
| $\text{CLIP}_4$ | $97.2_{\pm0.4}$ | $97.6_{\pm0.2}$ | $98.0_{\pm0.5}$ | $97.8_{\pm0.3}$ | $96.7_{\pm0.4}$ | $94.7_{\pm0.1}$ | $89.2_{\pm0.2}$ | $80.6_{\pm0.5}$ | $68.0_{\pm0.9}$ | $56.4_{\pm0.9}$ | $49.3_{\pm1.0}$ |

Table 22: ResNet-50 results on RotatedMNIST.

| | $\mathcal{D}_0$ | $\mathcal{D}_1$ | $\mathcal{D}_2$ | $\mathcal{D}_3$ | $\mathcal{D}_4$ | $\mathcal{D}_5$ | $\mathcal{D}_6$ | $\mathcal{D}_7$ | $\mathcal{D}_8$ | $\mathcal{D}_9$ | $\mathcal{D}_{10}$ |
|---|---|---|---|---|---|---|---|---|---|---|---|
| $\text{IN}_0$ | 98.3±0.2 | 97.7±0.0 | 96.1±0.1 | 90.3±0.3 | 80.8±0.5 | 68.4±0.7 | 55.7±0.8 | 46.7±0.1 | 41.2±0.3 | 39.8±0.6 | 38.9±0.4 |
| $\text{IN}_1$ | 98.3±0.0 | 98.6±0.1 | 97.8±0.1 | 95.0±0.2 | 89.2±0.4 | 79.5±0.5 | 67.7±0.6 | 58.2±0.9 | 51.1±0.3 | 45.9±0.9 | 45.0±0.9 |
| $\text{IN}_2$ | 98.2±0.1 | 98.7±0.3 | 98.7±0.2 | 97.4±0.1 | 94.4±0.2 | 88.3±0.5 | 78.0±1.6 | 66.6±1.9 | 57.7±1.6 | 50.8±1.5 | 48.9±0.7 |
| $\text{IN}_3$ | 97.9±0.2 | 98.4±0.3 | 98.7±0.4 | 98.6±0.1 | 96.9±0.1 | 94.1±0.3 | 87.5±0.3 | 77.7±0.2 | 67.7±0.6 | 59.7±1.0 | 56.6±0.8 |
| $\text{IN}_4$ | 97.4±0.7 | 98.2±0.5 | 99.0±0.2 | 98.5±0.3 | 98.4±0.4 | 96.7±0.1 | 92.8±0.1 | 85.4±0.2 | 76.7±0.3 | 66.6±0.1 | 62.0±0.3 |
| $\text{CLIP}_0$ | 98.8±0.1 | 97.8±0.1 | 95.1±0.1 | 87.5±0.4 | 73.9±0.7 | 58.0±0.6 | 44.5±0.5 | 37.6±0.4 | 33.3±0.1 | 32.1±0.5 | 33.3±0.2 |
| $\text{CLIP}_1$ | 98.6±0.2 | 98.9±0.1 | 97.7±0.1 | 93.3±0.3 | 84.2±0.6 | 71.0±0.7 | 57.1±1.0 | 47.3±0.9 | 39.7±0.5 | 35.3±0.4 | 36.9±0.5 |
| $\text{CLIP}_2$ | 98.6±0.2 | 98.9±0.2 | 98.4±0.2 | 96.7±0.0 | 92.2±0.2 | 82.5±0.7 | 69.1±1.5 | 57.8±1.8 | 48.4±1.3 | 41.0±1.9 | 41.1±1.0 |
| $\text{CLIP}_3$ | 98.0±0.2 | 98.6±0.1 | 98.9±0.1 | 98.0±0.2 | 96.2±0.1 | 91.9±0.2 | 82.9±0.4 | 72.3±0.3 | 60.8±0.4 | 49.8±0.6 | 47.5±1.0 |
| $\text{CLIP}_4$ | 98.1±0.5 | 98.9±0.1 | 98.7±0.2 | 98.4±0.2 | 97.8±0.4 | 95.6±0.1 | 90.2±0.2 | 81.5±0.5 | 69.7±1.2 | 56.9±1.1 | 52.8±1.7 |

Table 23: ViT-B/32 results on ROTATEDMNIST.

| | $\mathcal{D}_0$ | $\mathcal{D}_1$ | $\mathcal{D}_2$ | $\mathcal{D}_3$ | $\mathcal{D}_4$ | $\mathcal{D}_5$ | $\mathcal{D}_6$ | $\mathcal{D}_7$ | $\mathcal{D}_8$ | $\mathcal{D}_9$ | $\mathcal{D}_{10}$ |
|---|---|---|---|---|---|---|---|---|---|---|---|
| $\text{IN}_0$ | 97.7±0.1 | 92.7±0.2 | 85.5±0.5 | 75.5±1.1 | 61.7±1.3 | 47.8±1.0 | 37.1±1.0 | 29.4±1.5 | 23.9±0.5 | 20.2±0.8 | 17.2±1.1 |
| $\text{IN}_1$ | 98.1±0.5 | 96.2±0.6 | 88.2±0.8 | 78.2±0.8 | 65.9±0.9 | 52.4±0.5 | 39.7±1.1 | 32.1±1.2 | 25.9±0.6 | 20.1±0.6 | 18.1±1.0 |
| $\text{IN}_2$ | 97.1±0.6 | 96.6±0.3 | 90.7±0.4 | 81.7±0.8 | 71.7±0.3 | 59.4±1.6 | 47.8±1.3 | 37.5±1.2 | 29.0±0.9 | 23.0±1.9 | 19.3±0.7 |
| $\text{IN}_3$ | 97.4±0.5 | 96.0±0.7 | 92.6±1.4 | 85.5±0.7 | 75.5±1.5 | 65.6±1.5 | 52.8±0.6 | 41.7±0.4 | 34.1±0.8 | 26.4±1.6 | 20.8±1.0 |
| $\text{IN}_4$ | 97.7±0.2 | 96.0±0.3 | 93.1±1.8 | 85.6±2.4 | 76.6±1.8 | 67.1±1.0 | 56.6±0.9 | 47.1±2.0 | 39.2±0.7 | 31.7±0.6 | 25.2±0.8 |
| $\text{CLIP}_0$ | 93.8±0.3 | 87.7±0.3 | 77.4±0.9 | 62.5±1.5 | 46.5±1.4 | 34.5±0.4 | 26.5±1.6 | 19.9±0.9 | 16.0±1.1 | 14.5±1.1 | 12.4±1.0 |
| $\text{CLIP}_1$ | 93.4±0.9 | 90.5±0.7 | 81.4±1.0 | 67.8±0.2 | 50.0±1.2 | 33.5±1.9 | 23.2±0.7 | 15.7±0.4 | 12.2±0.4 | 10.2±0.6 | 8.7±0.5 |
| $\text{CLIP}_2$ | 93.9±0.7 | 90.9±0.6 | 84.4±0.5 | 76.2±1.1 | 64.2±1.1 | 49.3±1.9 | 34.2±1.6 | 23.9±1.2 | 18.0±1.0 | 13.2±0.8 | 10.9±1.0 |
| $\text{CLIP}_3$ | 93.0±0.8 | 90.5±1.2 | 85.5±1.8 | 77.6±1.3 | 68.6±1.0 | 56.9±1.1 | 43.8±0.5 | 31.3±1.7 | 22.7±1.2 | 17.2±0.9 | 14.0±0.4 |
| $\text{CLIP}_4$ | 92.9±1.3 | 91.0±1.6 | 84.8±1.5 | 76.8±1.8 | 71.0±1.4 | 60.6±1.8 | 52.3±1.3 | 40.7±1.1 | 32.3±1.7 | 24.4±0.7 | 19.0±0.6 |

Table 24: ResNet-50 results on NOISYIMAGENET15.

| | $\mathcal{D}_0$ | $\mathcal{D}_1$ | $\mathcal{D}_2$ | $\mathcal{D}_3$ | $\mathcal{D}_4$ | $\mathcal{D}_5$ | $\mathcal{D}_6$ | $\mathcal{D}_7$ | $\mathcal{D}_8$ | $\mathcal{D}_9$ | $\mathcal{D}_{10}$ |
|---|---|---|---|---|---|---|---|---|---|---|---|
| $\text{IN}_0$ | 98.3±0.3 | 92.9±0.2 | 92.2±0.7 | 90.9±0.3 | 89.3±0.3 | 85.9±0.8 | 82.2±0.4 | 77.9±0.5 | 71.8±0.4 | 67.7±1.5 | 61.5±0.9 |
| $\text{IN}_1$ | 98.4±0.3 | 98.4±0.3 | 92.6±0.4 | 91.5±0.3 | 89.0±0.7 | 86.6±0.6 | 82.6±0.8 | 76.7±1.4 | 72.3±1.3 | 65.7±1.0 | 59.6±1.2 |
| $\text{IN}_2$ | 98.2±0.4 | 98.3±0.3 | 96.9±0.3 | 90.7±0.2 | 89.0±0.3 | 87.1±0.6 | 81.4±0.9 | 77.8±1.2 | 71.3±0.2 | 64.2±0.9 | 57.7±2.0 |
| $\text{IN}_3$ | 98.5±0.5 | 98.7±0.6 | 97.4±0.8 | 95.9±0.8 | 89.9±0.3 | 86.7±0.9 | 83.2±0.3 | 79.8±0.9 | 73.7±0.8 | 67.8±0.8 | 60.9±0.5 |
| $\text{IN}_4$ | 98.8±0.3 | 98.5±0.8 | 97.6±0.2 | 95.5±1.2 | 92.9±1.4 | 87.2±0.5 | 84.7±0.1 | 79.2±1.3 | 75.3±0.7 | 69.1±0.8 | 62.6±0.6 |
| $\text{CLIP}_0$ | 94.4±0.1 | 92.1±0.3 | 88.8±0.6 | 81.9±0.4 | 72.7±1.1 | 62.9±1.0 | 53.7±1.0 | 45.5±1.1 | 39.7±1.7 | 32.3±1.8 | 27.6±0.7 |
| $\text{CLIP}_1$ | 94.6±0.5 | 93.7±0.4 | 90.0±0.4 | 85.5±0.6 | 76.6±1.7 | 69.7±2.6 | 60.2±2.2 | 50.9±4.0 | 42.4±3.2 | 35.0±1.8 | 28.9±2.9 |
| $\text{CLIP}_2$ | 94.5±0.8 | 93.8±0.6 | 91.8±0.6 | 87.0±0.4 | 81.4±0.9 | 75.7±1.0 | 65.4±2.3 | 55.0±1.3 | 44.5±1.7 | 38.1±1.4 | 30.3±1.5 |
| $\text{CLIP}_3$ | 94.8±1.1 | 94.6±0.7 | 90.7±0.6 | 87.4±1.2 | 82.3±0.9 | 75.7±1.0 | 69.1±0.6 | 60.8±0.6 | 51.9±1.9 | 42.8±1.3 | 35.7±0.8 |
| $\text{CLIP}_4$ | 94.4±1.3 | 94.2±1.0 | 90.4±0.5 | 86.2±1.5 | 82.1±1.4 | 77.7±1.1 | 71.5±0.2 | 64.4±1.5 | 55.2±1.1 | 46.6±1.3 | 40.7±1.8 |

Table 25: ViT-B/32 results on NOISYIMAGENET15.

| | $\mathcal{D}_0$ | $\mathcal{D}_1$ | $\mathcal{D}_2$ | $\mathcal{D}_3$ | $\mathcal{D}_4$ | $\mathcal{D}_5$ | $\mathcal{D}_6$ | $\mathcal{D}_7$ | $\mathcal{D}_8$ | $\mathcal{D}_9$ | $\mathcal{D}_{10}$ |
|---|---|---|---|---|---|---|---|---|---|---|---|
| $\text{IN}_0$ | 97.9±0.2 | 93.8±0.5 | 91.6±0.3 | 87.9±0.5 | 84.1±0.5 | 78.5±1.3 | 71.5±2.0 | 64.2±2.8 | 55.8±3.6 | 43.4±4.6 | 32.9±2.9 |
| $\text{IN}_1$ | 98.0±0.3 | 97.4±0.4 | 92.8±0.3 | 90.3±0.5 | 87.9±0.3 | 82.9±0.3 | 76.4±1.2 | 69.6±1.5 | 59.3±2.0 | 50.3±1.8 | 39.7±1.6 |
| $\text{IN}_2$ | 96.8±0.4 | 97.3±0.2 | 96.4±0.4 | 91.4±0.3 | 89.4±0.1 | 85.6±0.5 | 81.0±0.8 | 75.7±0.5 | 67.4±1.4 | 57.0±1.1 | 45.8±0.8 |
| $\text{IN}_3$ | 97.4±0.6 | 96.7±0.7 | 97.6±0.7 | 95.3±1.0 | 89.7±0.3 | 86.4±0.6 | 82.5±0.2 | 78.1±0.4 | 69.0±0.9 | 58.1±0.9 | 47.4±0.3 |
| $\text{IN}_4$ | 97.3±0.4 | 97.4±1.1 | 96.6±0.5 | 95.3±0.4 | 94.6±0.9 | 87.4±0.3 | 82.6±0.3 | 78.4±0.9 | 69.9±0.9 | 58.2±1.2 | 46.5±2.1 |
| $\text{CLIP}_0$ | 93.5±0.3 | 90.8±0.4 | 89.2±0.5 | 86.1±0.8 | 83.0±0.2 | 78.3±0.8 | 71.3±1.5 | 60.2±1.5 | 49.2±1.4 | 38.0±1.9 | 27.3±2.5 |
| $\text{CLIP}_1$ | 93.3±0.3 | 92.4±0.2 | 90.7±0.4 | 88.8±0.5 | 86.0±0.7 | 81.3±0.4 | 73.8±0.5 | 64.9±0.9 | 54.5±1.2 | 44.3±2.2 | 34.3±1.3 |
| $\text{CLIP}_2$ | 93.7±1.4 | 92.7±0.8 | 91.8±1.1 | 88.7±0.1 | 85.7±0.9 | 81.7±0.7 | 74.1±0.4 | 63.6±1.6 | 52.5±1.2 | 42.1±0.8 | 31.1±2.1 |
| $\text{CLIP}_3$ | 93.2±0.3 | 92.2±0.9 | 92.3±0.3 | 91.0±2.0 | 86.4±0.7 | 83.0±0.4 | 75.8±0.3 | 66.3±1.2 | 55.4±1.9 | 44.7±1.0 | 33.1±1.8 |
| $\text{CLIP}_4$ | 92.2±0.5 | 93.4±0.9 | 91.5±0.2 | 91.3±1.4 | 89.4±1.0 | 83.5±0.2 | 77.3±0.9 | 68.4±0.3 | 58.0±0.6 | 46.7±0.5 | 34.2±1.1 |

Table 26: ResNet-50 results on LR-IMAGENET15.

| | $\mathcal{D}_0$ | $\mathcal{D}_1$ | $\mathcal{D}_2$ | $\mathcal{D}_3$ | $\mathcal{D}_4$ | $\mathcal{D}_5$ | $\mathcal{D}_6$ | $\mathcal{D}_7$ | $\mathcal{D}_8$ | $\mathcal{D}_9$ | $\mathcal{D}_{10}$ |
|---|---|---|---|---|---|---|---|---|---|---|---|
| $\text{IN}_0$ | 98.5±0.2 | 94.2±0.1 | 93.6±0.2 | 92.2±0.1 | 90.3±0.2 | 88.4±0.3 | 85.6±0.1 | 81.2±0.8 | 77.0±1.1 | 70.9±1.3 | 59.8±1.4 |
| $\text{IN}_1$ | 98.1±0.4 | 98.6±0.3 | 93.7±0.1 | 92.4±0.1 | 91.1±0.3 | 89.2±0.2 | 86.5±0.2 | 83.4±0.3 | 79.5±0.2 | 73.6±0.3 | 62.7±0.5 |
| $\text{IN}_2$ | 98.2±0.8 | 98.3±0.4 | 97.9±0.3 | 92.8±0.2 | 91.0±0.2 | 89.2±0.2 | 86.9±0.2 | 84.3±0.2 | 80.5±0.0 | 74.6±0.3 | 63.6±0.7 |
| $\text{IN}_3$ | 98.5±1.3 | 98.3±0.8 | 98.4±0.0 | 98.2±0.2 | 91.3±0.4 | 90.0±0.5 | 87.4±0.3 | 83.9±0.0 | 80.4±0.4 | 74.4±0.3 | 63.6±0.5 |
| $\text{IN}_4$ | 98.9±0.3 | 98.3±0.5 | 98.3±0.7 | 98.3±0.1 | 97.6±0.6 | 90.6±0.1 | 88.1±0.3 | 85.1±0.4 | 81.1±0.5 | 75.2±0.3 | 66.2±0.7 |
| $\text{CLIP}_0$ | 94.5±0.3 | 93.3±0.1 | 92.1±0.4 | 91.0±0.1 | 89.9±0.4 | 88.3±0.3 | 85.7±0.3 | 80.0±0.7 | 73.6±1.0 | 62.7±1.2 | 54.1±1.9 |
| $\text{CLIP}_1$ | 94.1±0.7 | 94.8±0.5 | 92.4±0.1 | 91.6±0.1 | 90.3±0.1 | 88.6±0.8 | 86.2±1.8 | 79.3±1.6 | 73.0±2.1 | 63.2±1.4 | 54.0±0.8 |
| $\text{CLIP}_2$ | 94.8±1.1 | 94.2±1.0 | 93.9±1.0 | 92.1±0.1 | 91.0±0.2 | 89.9±0.3 | 86.6±0.3 | 79.8±0.3 | 74.0±0.3 | 64.6±0.5 | 55.2±0.5 |
| $\text{CLIP}_3$ | 94.0±0.7 | 94.4±0.9 | 94.9±0.3 | 94.1±0.8 | 90.8±0.1 | 89.3±0.4 | 86.0±1.2 | 80.0±0.9 | 73.7±1.4 | 64.4±1.1 | 55.1±2.6 |
| $\text{CLIP}_4$ | 94.9±0.4 | 93.6±1.0 | 94.3±1.2 | 92.9±0.2 | 92.1±0.6 | 90.0±0.2 | 86.4±0.3 | 81.0±0.7 | 75.6±0.6 | 64.6±0.3 | 56.2±0.3 |

Table 27: ViT-B/32 results on LR-IMAGENET15.

