# OpenReview forum: "Robustness May be More Brittle than We Think under Different Degrees of Distribution Shifts"
_ICLR.cc/2024/Conference — Submitted to ICLR 2024_

### Official Review · Reviewer_8Ku5 · 2023-10-27

**Soundness:** 1 poor
**Presentation:** 3 good
**Contribution:** 3 good
**Rating:** 5
**Confidence:** 4

**Summary:**

This paper argues that methods that improve OOD robustness may only improve robustness under small shifts, and in fact that methods that seem to be less effective when tested under small distribution shifts are actually more effective under large shifts. They demonstrate these claims using image classification experiments. They show that image classifiers pretrained with clip are more brittle to the type of distribution shifts they consider.

**Strengths:**

I was really delighted to read the message of this paper, as I think it's a very important one that needs to be considered by everyone who works on robustness. The framing is very strong, and I agreed with the message,  considered it very important to the robustness community, and thought it was well presented in this paper. I deeply appreciate the presentation of an apparent abrupt phase transition where the degree of shift becomes dramatically more challenging; perhaps it is this point where true "OOD" should be marked.

The clip result could be interesting, and I would encourage them to expand on it and explore with more experiments. It might fit better into a separate paper.

**Weaknesses:**

Unfortunately, the experiments are extremely weak. They consider a single task---image classification---and test with only a few types of distribution shift---rotation and gaussian noise for MNIST, compression and gaussian noise for Imagenet. I don't consider these settings to be representative, and in fact I don't even believe that models good on one type of shift will be good at another type of shift in general, let alone degree.

I don't think that the framing here is complete. It is well-understood that "OOD" performance is often strictly correlated with iid performance (see https://arxiv.org/abs/2206.13089) but that certain natural shifts can be inversely correlated with iid performance (https://arxiv.org/abs/2209.00613). It is likely that this sort of inverse correlation in performance is associated with the difficulty of a shift (section 5.1.1 in https://arxiv.org/abs/2310.03646). Therefore, there is an ongoing discussion about the difference between different types of shifts and how much we can generalize from findings on small shifts.

A minor thing I'm seeing is also a lack of good visualizations for some of the conclusions. I'd like to see a visualization specifically focused on the point that some models that are very good at small distribution shifts are bad at large distribution shifts, for example a plot that directly shows the correlation in performance for each model on these two settings.

Section 4.3: Why not just use the variation across different distribution shift methods and correlate the performance on strong shifts with the performance on weak shifts? You aren’t really measuring robustness to strong shift, you’re just training a model on a multimodal distribution.

Section 5: I really think that it’s not fair to measure the robustness of clip against distribution shifts that don’t really exist. I think that it is likely to help with robustness when it comes to novel tasks and out of distribution combinations (compositionally).

Overall, I don't think that this paper is ready to be published, but I appreciate the goal a lot and would be very receptive to it with expanded experiments.

**Questions:**

Figure 3: Top three for each data point or top three over a specific data point? Is this the top three for in-distribution specifically? These factors change the expected outcome, because the degree of random variation is likely to differ for different degrees of shift.

5.2: So this is basically just that training on more data creates overfitting to the training distribution? Would you describe it that way? I believe this is an area of discussion in distribution shift literature, but I don't know the specific citations that are best for it.

---

> ### Author Response · Authors · 2023-11-23
> **Response to the review (1/2)**
>
> We thank the reviewer’s valuable time and constructive feedback. Regarding the concerns and questions you have raised, we provide detailed explanations below.
>
> **1. “Unfortunately, the experiments are extremely weak. They consider a single task---image classification---and test with only a few types of distribution shift---rotation and gaussian noise for MNIST, compression and gaussian noise for Imagenet. I don't consider these settings to be representative ...”**
>
> We updated our paper to include experiments on an additional dataset that is much more realistic compared to NoisyMNIST. The dataset is based on CIFAR10, and the considered distribution shift is a combination of shifts in brightness and shot-noise intensity. Since photos captured in darker environments tend to exhibit more intense shot noises, this dataset simulates realistic photographic effects for photos captured in low-light conditions. Examples of this dataset can be found in Figure 8 (Appendix A.1).  The results on this dataset are shown in the updated Figure 3. The results on LowLightCIFAR10 are largely consistent with NoisyMNIST.
>
> In addition, we also furnished the experiments in Sec 4.3 and Sec. 5. While the datasets used in this study are synthetic, the main point of our paper is to demonstrate that even under these simplest forms of distribution shifts, the models are nonetheless brittle to changes in the shift degrees. For more complicated problems, we are fairly confident that similar conclusions would also hold.
>
> **2. “I don't think that the framing here is complete. It is well-understood that "OOD" performance is often strictly correlated with iid performance … but that certain natural shifts can be inversely correlated with iid performance … It is likely that this sort of inverse correlation in performance is associated with the difficulty of a shift.”**
>
> Thank you for your comment. Indeed, the framing here can be extended to include some discussion on the correlation of ID and OOD performance. During this rebuttal period, we have significantly furnished the experiment results of this paper (please also check the appendix, if possible, since we have made a lot of updates there too). We believe the current empirical evidence is sufficient to support the main arguments of this paper. While a better framing would definitely improve the paper, we do not think it is a deciding factor for the decision of this paper. Nevertheless, we very much appreciate your suggestion, and we will consider extending the framing (as you kindly suggested) after the rebuttal.
>
> **3. “A minor thing I'm seeing is also a lack of good visualizations for some of the conclusions. I'd like to see a visualization specifically focused on the point that some models that are very good at small distribution shifts are bad at large distribution shifts, for example, a plot that directly shows the correlation in performance for each model on these two settings.”**
>
> Thank you very much for your constructive advice. We agree with your point that a visualization specifically focused on the point that some models that are very good at small distribution shifts are bad at large distribution shifts would be great. We will add the suggested visualization in the next version of this paper.
>
> **4. “Section 4.3: Why not just use the variation across different distribution shift methods and correlate the performance on strong shifts with the performance on weak shifts? You aren’t really measuring robustness to strong shift, you’re just training a model on a multimodal distribution.”**
>
> You are absolutely right. Technically speaking, we are indeed training on a multimodal distribution. We will revise the paper to better reflect this. Please also note that training on a multimodal distribution and then evaluating on the milder shifts do have real-world implications. This suggests that only learning from the hardest cases (with the easiest cases) is sometimes not enough for the model to generalize reasonably well across the degrees in between. Therefore, when collecting data for real-world applications, it could be very important to cover all potential degrees.

---

> ### Author Response · Authors · 2023-11-23
> **Response to the review (2/2)**
>
> **5. “Section 5: I really think that it’s not fair to measure the robustness of clip against distribution shifts that don’t really exist. I think that it is likely to help with robustness when it comes to novel tasks and out of distribution combinations (compositionally).”**
>
> We are currently running the corresponding experiments (in Section 5) on LowLightCIFAR10 which exhibits a much more realistic distribution shifts than the other datasets we considered. The distribution shift in this dataset is a combination of two primitive types of distribution shifts which are shifts in brightness and shot-noise intensity. Since photos captured in darker environments tend to exhibit more intense shot noises, this dataset simulates realistic photographic effects in photos captured under low-light conditions and hence is much more realistic than the other datasets. Examples of this dataset can be found in Figure 8 (Appendix A.1). The results of the experiments in the previous sections on this dataset (called LowLightCIFAR10) are shown in the updated Figure 3 and Figure 5. The new results are largely consistent with the previous ones. Please check the updated paper for more details.
>
> **6. “Figure 3: Top three for each data point or top three over a specific data point? Is this the top three for in-distribution specifically? These factors change the expected outcome, because the degree of random variation is likely to differ for different degrees of shift.”**
>
> It is the top three for each degree and then evaluated on all the degrees. They are not for in-distribution specifically.
>
> **7. “5.2: So this is basically just that training on more data creates overfitting to the training distribution? Would you describe it that way? I believe this is an area of discussion in distribution shift literature, but I don't know the specific citations that are best for it.”**
>
> Yes, to put it in extremely concise terms, this is the way that we would describe it.

---

### Official Review · Reviewer_tBMy · 2023-10-31

**Soundness:** 2 fair
**Presentation:** 4 excellent
**Contribution:** 2 fair
**Rating:** 5
**Confidence:** 4

**Summary:**

The paper studies robustness of domain generalization (DG) methods under different levels of distribution shift. The observations include that the robustness can be brittle, and methods that are more robust for smaller amounts of shift may not be as competitive in the presence of larger amounts of shift. It is also shown that large-scale pre-trained models can be sensitive to even very small distribution shifts of novel downstream tasks.

**Strengths:**

* The questions asked in the paper regarding robustness are interesting and practically valuable. One would intuitively expect that methods that perform better for larger domain shifts also perform better for smaller domain shifts (or vice versa). This paper challenges the intuition via empirical evaluation on the selected datasets, which can be very valuable for future research in the area.
* A large number of methods are evaluated and HPO is conducted to evaluate them more fairly.
* The paper is well-written and easy to read.

**Weaknesses:**

* Most of the investigation is done on variations of MNIST that are likely to have only limited implications for real-world computer vision. For more reliable implications it would be recommended to use at least CIFAR-C or TinyImageNet-C that also include various levels of distribution shifts. Ideally a study would be conducted also on real-world data in addition to synthetic ones, even though those may be hard to find. To balance compute costs, a smaller number of methods could be evaluated.
* A small number of noise types is studied so it would be good to cover a larger variety.
* In Figure 3 left for ResNet-50 it could be argued that the behaviour seen generally is what one would expect based on intuition i.e. contrary to the message of the paper and meaning that models that perform better at stronger distribution shifts are also generally better at smaller distribution shifts. This would suggest that perhaps the brittleness may not be as common for larger models, i.e. the ones that are of primary interest.

**Questions:**

* Is the behaviour similar also for more realistic datasets such as CIFAR-C?
* Is the described behaviour present also when using other types of noise (i.e. if we try a large variety of noises, do we still see the pattern on average)?
* How exactly is the HPO conducted? Using 20 trials is appropriate in some cases, but it depends on how many hyperparameters there are and sometimes many more need to be sampled.

---

> ### Author Response · Authors · 2023-11-22
> **Response to the review**
>
> We thank the reviewer’s valuable time and constructive feedback. Regarding the concerns and questions you have raised, we provide detailed explanations below.
>
> **1. “Most of the investigation is done on variations of MNIST that are likely to have only limited implications for real-world computer vision. For more reliable implications it would be recommended to use at least CIFAR-C or TinyImageNet-C ...”**
>
> Thank you very much for your great advice. Due to the limited time frame, we were unable to experiment with CIFAR-10-C or TinyImageNet-C, but we did conduct further experiments on CIFAR-10 under a much more realistic shift. The distribution shift is a combination of two primitive types of distribution shifts which are shifts in brightness and shot-noise intensity. Since photos captured in darker environments tend to exhibit more intense shot noises, this dataset simulates realistic photographic effects in photos captured under low-light conditions and hence is much more realistic than the other datasets. Examples of this dataset can be found in Figure 8 (Appendix A.1). The results on this dataset (called LowLightCIFAR10) are shown in the updated Figure 3 and Figure 5. The new results are largely consistent with the previous ones.
>
> Regarding your concern that the results in this paper are likely to have only limited implications for real-world computer vision, we would like to argue that the implications may be more general than you might think. Our reasons are as follows:
>
> - The distribution shifts considered in this paper are very simple. We intend to show that the models are more brittle than we expect *even to these simple distribution shifts*. For more complicated shifts involving multiple types of inter-correlated shifts, the same general conclusions would likely follow.
>
> - Performance under synthetic shifts are correlated with performance under real-world (or natural) shifts. While [1] suggests that this correlation is weak, [2] delves deeper into the issue and provides evidence that synthetic corruptions sometimes do correlate well with corruptions that appear in the wild.
>
> - In this work, we focus on the *failures* of models under the distribution shifts as we vary their degrees. While robustness under synthetic shifts may not always imply robustness under natural shifts, failure under synthetic shifts almost certainly implies failure under natural shifts.
>
> **2. “A small number of noise types is studied so it would be good to cover a larger variety.”**
>
> As you kindly suggested, we conducted additional experiments on a very different kind of noise, impulse noise. Examples of this dataset (called ImpulseNoiseMNIST) are shown in Figure 11 (Appendix A.1) of the updated paper. The experiment results are given in Figure 14 (Appendix B.2). In addition, the shot noise in LowLightCIFAR10 dataset is another type of noise. The results on these datasets are consistent with each other and have been partially discussed in our response to your previous comment.
>
> In particular, the new results on LowLightCIFAR10 (in Figure 5) and on ImpulseNoiseMNIST (in Figure 14, right) demonstrate an even more interesting pattern than we have previously shown. When the shift in the training data is not very strong (e.g., $D_0$ and $D_5$), the generalization pattern looks just like that of NoisyMNIST---robustness against milder shifts are not affected. On the other hand, when the shift in the training data is very strong ($D_0$ and $D_{10}$), the pattern looks like that of RotatedMNIST---robustness against milder shifts is significantly weakened. In addition, for LowLightCIFAR10, even the performance on the clean data is affected.
>
> **3. “How exactly is the HPO conducted? Using 20 trials is appropriate in some cases, but it depends on how many hyperparameters there are and sometimes many more need to be sampled.”**
>
> Some DG algorithms have several additional hyperparameters (2 and 3 are common; one has 6) in addition to shared ones such as learning rate and batch size. If all the hyperparameters were allowed to vary, the search space would be enormous. To reduce the size of the search space, we used the same fixed learning rate, batch size, weight decay, and dropout for all algorithms. These hyperparameters are set to appropriate values such that ERM performs reasonably well with them. As for the other hyperparameters, the HPO process is conducted following [3]. We believe that 20 trials are appropriate for most (if not all) algorithms in this reduced hyperparameter search space.
>
> ***
>
> **References**
>
> [1] Taori, Rohan, et al. "Measuring robustness to natural distribution shifts in image classification." NeurIPS. 2020.
>
> [2] Hendrycks, Dan, et al. "The many faces of robustness: A critical analysis of out-of-distribution generalization." ICCV. 2021.
>
> [3] Gulrajani, Ishaan, and David Lopez-Paz. "In Search of Lost Domain Generalization." ICLR. 2021.

---

### Official Review · Reviewer_tbnw · 2023-10-31

**Soundness:** 2 fair
**Presentation:** 3 good
**Contribution:** 2 fair
**Rating:** 5
**Confidence:** 5

**Summary:**

This paper introduces a setup for a relevant analysis of out-of-distribution (OOD) performance. The authors propose an analysis of different degrees of distribution shift, for a more realistic understanding of the generalization capabilities (Fig. 1 is a clear illustration of the tackled issue). This enables a better estimation of real-world performance.

The authors introduce 4 datasets: NoisyMNIST, RotatedMNIST, NoisyImageNet15, and LR-ImageNet15. For each dataset, they have a clean set ($D_0$) and they continuously alter this set, simulating covariate shits, by adding Gaussian noise, rotating the image, or applying image degradation techniques. Through this process, they obtain additional sets: $D_1$, $D_2$, $D_3$, etc., which should capture continuously increasing degrees of covariate shift.

Three main assumptions are analyzed:
- robustness may not even extrapolate to slightly higher degrees of distribution shift
- robustness at higher degrees does not always guarantee robustness at lower degrees
- pre-trained representations are sensitive to novel downstream distribution shifts

**Strengths:**

**S1** The tackled problem is of tremendous importance for the field.

**S2** The proposed direction of analysis has great potential of shading light over the estimated real-world performance of models when dealing with covariate shifts.

*S2.1* many relevant domain generalization algorithms have been considered  (20 different initializations and hyperparameters for each)

*S2.2* analysis over both randomly initialized and pre-trained models, as well as over CNN and Transformer-based architectures.

**Weaknesses:**

**W1** The performance of many domain generalization approaches improves with the number of training domains, but, in the current paper, the training domains of DG methods are limited to two (also valid for ERM, and the behavior can be observed in Fig. 6)
Although a comprehensive list of domain generalization methods is considered (Appendix A.2), results are reported only on a subset of the methods, with part of the analysis being performed solely on ERM.

**W2** The analysis is performed over purely synthetic distribution shifts, considering a single type of shift.

*W2.1* natural shifts do not follow this 'single type' shift -- previous works, like [1], introduce benchmarks that address the natural distribution shifts while studying the continuous degradation under various degrees of shift

*W2.2* four datasets are proposed (NoisyMNIST, RotatedMNIST, NoisyImageNet15, and LR-ImageNet15), but they are used inconsistently for the analysis (e.g. Sec.4.2 - only NoisyMNIST, Sec.4.3 NoisyMNIST+RotatedMNIST)

**W3** The paper does not benefit from a rigorous analysis of the results

*W3.1* regarding the conclusion: "robustness at higher degrees does not always guarantee robustness at lower degrees" (Sec. 4.2) - see questions **Q1**, **Q2**, **Q3** and **Q4**

*W3.2* regarding the conclusion: "robustness may not even extrapolate to slightly higher degrees" (Sec. 4.3) - see questions **Q5**, **Q6** and **Q7**

**W4** The analysis of pre-trained models from Sec. 5 is shallow

*W4.1* as Gaussian noise may be less common in the pre-training data, while rotations may be more common, it is hard to establish what is in-distribution(ID) and what is out-of-distribution(OOD)

*W4.2* Regarding the assumption about CLIP from Sec. 5.2 - following the DFR paper [2], the embedding space of heavily pre-trained models is expected to contain both relevant and spurious features. The balance of spurious vs relevant correlations present in the training set used for linear probing may dictate the generalization capabilities.

*W4.3* analysis on NoisyImageNet15 and LR-ImageNet15 shows that "ImageNet pre-trained models are generally more robust than CLIP models" -- the conclusion can be strongly influenced by the fact that ImageNet pre-trained models have already seen classes present in the two sets (you can also see that CLIP based models have significant lower performances even for the training sets, which is not true in general, when the downstream task is not based on ImageNet data)

*W4.4* it seems that only ERM is used during linear probing or training of RI (randomly initialized) models  (see question **Q8**)

*W4.5* RI performance is not reported for NoisyImageNet15 & LR-ImageNet15  (see question **Q9**)

**W5** RotatedMNIST - the authors consider a large range of rotation angles, some of which may induce significant confusion between class labels (e.g. numbers 6 and 9)
The continuous covariate shift is less clear than in the case of the NoisyMNIST dataset. (see **Q7**)


[1] Dragoi et al.  "AnoShift: A distribution shift benchmark for unsupervised anomaly detection" - NeurIPS 2022
[2] Izmailov et al. "On feature learning in the presence of spurious correlations" - NeurIPS 2022

**Questions:**

**Q1** Sec.4.2., Fig. 3 (left) -- we only observe the results of the best 3 models at each degree.
 - the models selected for $D_0$ or $D_1$ display poor generalization capabilities (it is very likely that ERM models were the top performing ones)
- meanwhile, the models selected for $D_{10}$ seem to be pretty robust (which is natural, as you select top-performing models on $D_{10}$, while the models were trained on $D_0$ and $D_1$)
- models selected for $D_{10}$ have significantly lower gaps between $D_4$ and $D_5$, than the ones selected for $D_0$ or $D_1$, indicating that the presence of this gap may not be a general behavior

Can you provide detailed (per domain generalization (DG) method) numerical results? You can also present ERM, average over all methods, and top-performing DG method.

**Q2** Sec. 4.2, Fig. 3 (right) -- only part of the considered DG methods (as presented in Appendix A.2) are mentioned.
On what considerations did you choose the methods for which you reported the results?

**Q3** According to Fig. 2, it seems very natural to observe a performance decrease between $D_4$, $D_5$, and $D_6$, as this is the point where the noise starts to interfere with the content of the image.
Have you considered analyzing the performance decrease in correlation with the dataset distances (e.g. similar to the OTDD analysis from [3] or some perceptual similarity metrics [4])? The dataset distances can be computed considering either RGB or embedding spaces of pre-trainedd models.
Although you have built the NoisyMNIST with a gradual noise increase, it is hard to establish how this noise is actually perceived by the model. (distance between $D_0$ and $D_1$ may be different than the distance between $D_4$ and $D_5$)

**Q4** Sec.4.2, Fig. 3 (left) -- there is a significant difference between the generalization capabilities of the 4-layer CNN and ResNet-50.
ResNet-50 seems more prone to overfitting in the considered scenario, indicating that the model may not be appropriate for the considered data and the results of the analysis are less relevant.
Have you considered any in-between model capacity?

**Q5** Sec. 4.3, Fig. 5 - results are presented only for ERM, ignoring DG methods.
Have you performed this analysis considering also the DG methods?

**Q6** Sec. 4.3, Fig. 5, NoisyMNIST, 4-layer CNN - there is a significant performance gap between $D_0$ and $D_8$, for the model trained on $D_0$ and $D_8$. This is not observed in any other setup. What is the intuition behind this result?

**Q7** Sec. 4.3, Fig. 5 - there is a significant decrese on milder shifts only for RotatedMNIST, models trained on $D_0$ and $D_8$ and partially for models trained on $D_0$ and $D_4$.
As ERM results are reported and we observe no drop on NoisyMNIST, it seems that RotatedMNIST is not actually capturing a continous covariate shift.
Can you perform a comparative analysis of the two sets maybe based on dataset distances (similar to **Q3**)?

**Q8** The analysis in Sec.5 is based solely on ERM? Have you considered using any DG method here?

**Q9** RI models are not reported for NoisyImageNet15 and LR-ImageNet15. Have you performed those experiments? Is there any limitation regarding those experiments?

[3] Alvarez-Melis and Fusi "Geometric dataset distances via optimal transport" -NeurIPS 2020
[4] Zhang et al. "The Unreasonable Effectiveness of Deep Features as a Perceptual Metric" - CVPR 2018

**Minor comments**:
- citations look different than provided template, you should check them
- Table 1 - it would be useful to add averages of over all the considered methods
- Fig. 4 - a legend for the heatmap would be useful
- Fig. 5 - maybe use '0&2' instead of '0/2' to denote models trained on $D_0$ and $D_2$

---

> ### Author Response · Authors · 2023-11-23
> **Response to the review (1/2)**
>
> We thank the reviewer’s valuable time and constructive feedback. Regarding the concerns/questions you raised, we provide detailed explanations below.
>
> **W1.1. “The performance of many domain generalization approaches improves with the number of training domains, but, in the current paper, the training domains of DG methods are limited to two.”**
>
> Thank you for pointing this out. We have added the results of DG methods (and ERM) trained on more than two domains in Figure 12 and Figure 13 (Appendix B.1). The results show that more training domains do help. As more training domains are added, the gaps between the best-performing models gradually decrease. Nevertheless, the gaps seem to be closing at a slow rate. The discrepancy between the best-performing models is still largely present. In addition, the advantage of DG methods over ERM becomes less pronounced as the number of training domains increases.
>
> **W1.2. “Although a comprehensive list of domain generalization methods is considered (Appendix A.2), results are reported only on a subset of the methods, with part of the analysis being performed solely on ERM.”**
>
> More results on this have been provided in the updated paper. For details about this part of the update, please refer to our responses below to your Q1 and Q2.
>
> **W2.1. “natural shifts do not follow this 'single type' shift -- previous works, like [1], introduce benchmarks that address the natural distribution shifts while studying the continuous degradation under various degrees of shift”**
>
> It is true that many natural shifts do not follow the 'single type' shift considered in our paper. Our main point, however, is to demonstrate even under these very simple shifts, the models are brittle to changes in the shift degrees. For more complicated shifts involving multiple types of shifts, it is almost certain that similar conclusions would follow.
>
> To verify this, we experimented with a new distribution shift on CIFAR10. The distribution shift is a combination of two primitive types of distribution shifts which are shifts in brightness and shot-noise intensity. Since photos captured in darker environments tend to exhibit more intense shot noises, this dataset simulates realistic photographic effects in photos captured under low-light conditions and hence is much more realistic than the other datasets. Examples of this dataset can be found in Figure 8 (Appendix A.1). The results on this dataset (called LowLightCIFAR10) are shown in the updated Figure 3 and Figure 5. The new results are largely consistent with the previous ones. Due to time constraint, the new results in Figure 3 are based on only the 11 algorithms listed in the figure.
>
> Thank you for letting us know the great work of Dragoi et al [1]. We will add a brief discussion about it in the related work.
>
> **W2.2. “four datasets are proposed (NoisyMNIST, RotatedMNIST, NoisyImageNet15, and LR-ImageNet15), but they are used inconsistently for the analysis (e.g. Sec.4.2 - only NoisyMNIST, Sec.4.3 NoisyMNIST+RotatedMNIST)”**
>
> This inconsistency is largely due to computational cost. Experiments in Sec. 4.2 require much more computation than the other experiments. We have tried our best to fill in the gaps between Sec. 4.2, Sec. 4.3, and Sec. 5. Please kindly check the paper for this update.
>
> **Q1. “Can you provide detailed (per domain generalization (DG) method) numerical results? You can also present ERM, average over all methods, and top-performing DG method.”**
>
> As you kindly suggested, we have provided the detailed numerical results in Appendix B.3.
>
> **Q2. “Sec. 4.2, Fig. 3 (right) -- only part of the considered DG methods (as presented in Appendix A.2) are mentioned. On what considerations did you choose the methods for which you reported the results?”**
>
> The methods are chosen on account of their academic influence, overall performance, and distinctiveness. For example, methods such as IRM and GroupDRO are well known to the research community and are common baselines in DG research. To balance informativeness and conciseness, we omitted methods that perform much poorer than the other methods, due to difficulty in hyperparameter tuning (e.g., Transfer has 6 hyperparameters in addition to learning rate, batch size, etc.), or optimization (e.g., gradient-based methods). Finally, to cover a wide variety of methods while preserving clarity, some conditional variants (e.g., CDANN, CondCAD) of the base methods (DANN, CAD) are also omitted from the main figures and tables.
>
> **Q3. “Have you considered analyzing the performance decrease in correlation with the dataset distances (e.g. similar to the OTDD analysis from [3] or some perceptual similarity metrics [4])?”**
>
> We deeply appreciate your great advice. Unfortunately, we are unable to conduct the analysis in time due to the limited time frame and the relatively low priority (we think) compared to your other great suggestions. We will include this analysis in the next version of this paper.

---

> ### Author Response · Authors · 2023-11-23
> **Response to the review (2/2)**
>
> **Q4. “Sec.4.2, Fig. 3 (left) … ResNet-50 seems more prone to overfitting in the considered scenario, indicating that the model may not be appropriate for the considered data and the results of the analysis are less relevant. Have you considered any in-between model capacity?”**
>
> First, we would like to gently point out that there is no clear evidence showing that ResNet-50 has overfitted to NoisyMNIST. In fact, most of the top models attained near-perfect (> 98%) validation accuracy in the training domains.
>
> Following your advice, we further experimented with EfficientNet-b0. It has 5.3M learnable parameters whose capacity is between that of Simple CNN (0.37M) and ResNet-50 (25M). The results have been shown in Figure 3 of the updated paper. Comparing the results from the three networks, it seems that smaller networks have stronger inductive bias and thus can maintain better performance under the distribution shifts.
>
> In a sense, we could say that larger networks are more prone to overfitting to the training *domains* (note that this is distinct from the conventional definition of overfitting). However, this does not render the results inappropriate or make the analysis irrelevant. On the contrary, this gives us deeper insights into the relation between model capacity and generalization. For instance, the results suggest that more training domains might be needed for larger models (even if sufficient data are provided for each domain; thus no conventional overfitting) to be as robust as smaller models.
>
> **Q5. “Sec. 4.3, Fig. 5 - results are presented only for ERM, ignoring DG methods. Have you performed this analysis considering also the DG methods?”**
>
> Yes, we have. Please see Figure 15-17 (Appendix B.3) of the updated paper for the detailed results. In brief, we find that most DG algorithms can improve the generalization from stronger shifts to milder shifts in the case of RotatedMNIST, although only to a limited extent. On the other two datasets, only some of the DG algorithms are able to improve the generalization performance by a very small margin.
>
> **Q6. “Sec. 4.3, Fig. 5, NoisyMNIST, 4-layer CNN - there is a significant performance gap between $D_0$ and $D_8$, for the model trained on $D_0$ and $D_8$. This is not observed in any other setup. What is the intuition behind this result?”**
>
> The intuition is that, as the noise gets stronger, the digits become more difficult to recognize (even for humans), and therefore the upper bound of the attainable accuracy becomes lower. We believe this is the main reason for the performance gap.
>
> **Q7. “Sec. 4.3, Fig. 5 … As ERM results are reported and we observe no drop on NoisyMNIST, it seems that RotatedMNIST is not actually capturing a continous covariate shift. Can you perform a comparative analysis of the two sets maybe based on dataset distances (similar to Q3)?”**
>
> We indeed considered a large range of rotation angles; however, we do not think it would induce significant confusion between class labels (e.g., numbers 6 and 9). The rotation angle between $D_0$ and $D_8$ is 80 degrees. Even if we assume the original handwritten digits could tilt within a 30-degree range, which is a large range, it is very unlikely that there is room for confusion. In the worst case, for example, to confuse a 6 that is first tilted clockwise by 30 degrees and then further rotated clockwise by 80 degrees, with a 9 that is tilted counterclockwise by 30 degrees, the sum 30+80+30 needs to be at least 180 degrees. Clearly, this is far from the real case.
>
> Again, we would love to perform a comparative analysis of the two sets as you kindly suggested, but the time frame is too short for us to do so. We will include this analysis in the next version of this paper.
>
> **Q8. “The analysis in Sec.5 is based solely on ERM? Have you considered using any DG method here?”**
>
> We have considered using DG methods here in the case of NoisyMNIST. The results are given in Figure 13 (Appendix B.1). Some of the DG algorithms can further improve upon ERM, however, the improvements become smaller when more training domains are added. Please also note the main point of the analysis in Sec. 5 is to highlight the brittleness of large-scale pre-trained representations, so the results of DG algorithms are not quite relevant to Sec. 5.
>
> **Q9. “RI models are not reported for NoisyImageNet15 and LR-ImageNet15. Have you performed those experiments? Is there any limitation regarding those experiments?”**
>
> Yes, we have performed those experiments. We omitted them because the results are not informative due to severe overfitting to the only 15 ImageNet classes in those datasets.
>
> **Minor comments.**
>
> Thank you for the detailed comments. We will fix the problems accordingly.

---

### Official Review · Reviewer_V5DH · 2023-10-31

**Soundness:** 2 fair
**Presentation:** 3 good
**Contribution:** 2 fair
**Rating:** 3
**Confidence:** 4

**Summary:**

The paper performs several studies to show that robustness at a certain degree of distribution shift is not indicative of its robustness at another degree of shift. This includes studying if models trained a low shifts can generalize to higher ones and vice versa. It also looks at the sensitivity of large pre-trained models like CLIP to different degrees of shifts.

**Strengths:**

The paper was well written and easy to follow. The experimental setups were clear.

**Weaknesses:**

1. The studies conducted to support the argument that evaluations should be done over degrees of shifts is not very convincing.
    - Would such controlled analysis be indicative of the performance of the model in the real world? Smoothly changing shifts implies that we should create synthetic shifts as it is expensive to create manually. Furthermore, only type one shift is changing at a time. In reality, shifts are correlated e.g., low light and iso noise or motion blur tends to occur together.
    - The paper seems to assume that distribution shifts are smoothly changing. What about semantic/categorical shifts? e.g. background changes like in NICO, changing shapes etc.
    - There are different types of shifts, but are these shifts comparable? It seems like in Fig 5, its comparing the trends from two types different shifts, why do we expect the trends to be the same?
    - Generalization to higher or lower shifts (sec 4.2, 4.3). Why do we assume that generalization will occur? Training on a certain level of shift makes that shift ID, a different level of shift then becomes OOD to the model. Isnt this the same problem as training on clean data and testing on OOD data?
2. It is not clear what the take home message is.
    - The results seems to say that the OOD performance of the model depends on its inductive biases and it is hard to predict a model's performance OOD, which is quite generic.
    - The overall goal seems to be more thorough evaluations (via fine grained degrees of shifts) for models to be useful in the real world. Real world datasets provides such evaluations but some cases it is impossible to have smoothly changing shifts e.g. smoothly changing animal occlusions in iwildcam. Furthermore, synthetic benchmarks like common corruptions already proposes evaluating on different degrees of shifts for each corruption (see 2), although as mentioned, they do not provide analysis at each level of corruption.
    - Furthermore is it easy to anticipate the degree of shift at test-time such that such analysis would be useful? It seems more natural to anticipate the type of shift than its extent e.g., adverse weather, than how adverse the weather will be.
3. "Similar problems can also arise when only the aggregate performance across multiple degrees is examined (Hendrycks & Dietterich, 2019)"
    - The common corruptions benchmark measures performance over 5 degrees of shift and the paper measures it over 8/10, how do we know what is the ideal number of degrees to evaluate on?
4. The statement "robustness may be more brittle ..." sounds contradictory as robustness is the opposite of brittle. It is our models that are brittle.

**Questions:**

See the points raised in weaknesses for the questions.

A suggestion for having smoothly changing yet realistic shift is to perform evaluations on videos. The camera's viewpoint is smoothly changing, so are shifts in the real world.

---

> ### Author Response · Authors · 2023-11-22
> **Response to the review (1/3)**
>
> We thank the reviewer’s valuable time and constructive feedback. Regarding the concerns and questions you have raised, we provide detailed explanations below.
>
> **1.1. “Would such controlled analysis be indicative of the performance of the model in the real world? Smoothly changing shifts implies that we should create synthetic shifts as it is expensive to create manually. Furthermore, only type one shift is changing at a time. In reality, shifts are correlated e.g., low light and iso noise or motion blur tends to occur together.”**
>
> While we cannot be sure that the analysis in this paper would be indicative of the performance of the model in the real world, we are fairly confident that similar conclusions can still be drawn for the following reasons:
>
> - The distribution shifts considered in this paper are very simple. We intend to show that the models are more brittle than we expect *even to these simple distribution shifts*. For more complicated shifts involving multiple types of inter-correlated shifts, the same general conclusions would likely follow.
>
> - Performance under synthetic shifts are correlated with performance under real-world (or natural) shifts. While [1] suggests that this correlation is weak, [2] delves deeper into the issue and provides evidence that synthetic corruptions sometimes do correlate well with corruptions that appear in the wild.
>
> - In this work, we focus on the *failures* of models under the distribution shifts as we vary their degrees. While robustness under synthetic shifts may not always imply robustness under natural shifts, failure under synthetic shifts almost certainly implies failure under natural shifts.
>
> - In practice, properly constructed synthetic shifts are better than nothing, especially for high-stake applications such as self-driving. The key here is to make sure that the synthetic shifts are as faithful to the real shifts in consideration as possible. The more faithful they are, the more reliable and indicative the evaluation would be. In the meantime, we can also proactively collect more real data targeted at the specific distribution shifts in the studied problems (if it is possible).
>
> Finally, inspired by your constructive feedback, we experimented with a new distribution shift on CIFAR10. The distribution shift is a combination of two primitive types of distribution shifts which are shifts in brightness and shot-noise intensity. Since photos captured in darker environments tend to exhibit more intense shot noises, this dataset simulates realistic photographic effects in photos captured under low-light conditions and hence is much more realistic than the other datasets. Examples of this dataset can be found in Figure 8 (Appendix A.1). The results on this dataset (called LowLightCIFAR10) are shown in the updated Figure 3 and Figure 5. The new results are largely consistent with the previous ones. We will discuss more about the results on this new dataset in our response to some of your other comments below.
>
> **1.2. “The paper seems to assume that distribution shifts are smoothly changing. What about semantic/categorical shifts? e.g. background changes like in NICO, changing shapes etc.”**
>
> From a theoretical perspective, we can always smoothly change a data distribution $p(x)$ into $p’(x)$ by properly parameterizing the distributions, even for categorical shifts. In reality, this is slightly more nuanced but still generally possible. For example, one can smoothly transition from “indoor” to “outdoor” (these are commonly regarded as categorical shifts) by walking out of the “door”. The real question, therefore, is which transitions are of real-world significance. For example, transitions from autumn to winter are meaningful, whereas direct transitions from autumn to spring (skipping winter) are not.
>
> It is possible that in some problems we do not care about the transitions due to their low real-world significance, so the corresponding evaluations are not needed. When we do care about the transitions, however, our paper suggests that those evaluations could be very informative and useful.

---

> ### Author Response · Authors · 2023-11-22
> **Response to the review (2/3)**
>
> **1.3. “There are different types of shifts, but are these shifts comparable? It seems like in Fig 5, its comparing the trends from two types different shifts, why do we expect the trends to be the same?”**
>
> Comparing the different types of shifts is not our intention, nor do we expect the trends to be the same *or* different. For Fig. 5, we just would like to see if training under stronger shifts helps with performance under milder shifts (of the same type), and more importantly, how much it helps if it does.
>
> We experimented with two datasets to investigate this. As two opposite trends are shown in Fig. 5, the result is largely task-dependent: one demonstrates almost perfect generalization to milder shifts, while the other shows limited generalization. Admittedly, the limited generalization is not very surprising because, as you and another reviewer have pointed out, the model is trained on a multimodal distribution, so it is not expected to generalize in the middle. Nevertheless, this theory cannot explain why almost perfect generalization is observed in the former case. The fact that both phenomena, not just one of them, are observed is actually quite interesting. Furthermore, it supports our view that evaluating across different degrees of shifts should be encouraged as it may be difficult to predict how the model performs in the middle.
>
> **1.4. “Generalization to higher or lower shifts (sec 4.2, 4.3). Why do we assume that generalization will occur? Training on a certain level of shift makes that shift ID, a different level of shift then becomes OOD to the model. Isnt this the same problem as training on clean data and testing on OOD data?”**
>
> In Sec. 4.2, it is *not* the same problem as training on clean data and testing on OOD data. The difference is that we allow model selection using another set of OOD data different from the test data. This kind of model selection is commonly practiced in large OOD benchmarks [3, 4]. More specifically, we trained the models on $D_0$ and $D_1$. Then, for example, we select the best-performing models on $D_4$. If the selected models have good performance on $D_4$ (which they do), then they have generalized from $D_0$ and $D_1$ to $D_4$. Intuitively, we would expect these models to also generalize better than other models in the test domains, at least in nearing ones such as $D_5$. However, as we shown in Sec. 4.2 with a very simple example (NoisyMNIST), this is not generally true.
>
> In Sec. 4.3, it is the same problem as training on clean data and testing on OOD data, but with a caveat. The caveat is that we look at generalization from stronger shifts to milder shifts. The models are trained with clean data ($D_0$) together with strongly shifted data (e.g., $D_8$, $D_{10}$). Intuitively, if a model has seen and overcome bad situations, then it should also be able to overcome better situations. This is indeed true in the case of NoisyMNIST (as we have shown in Sec. 4.3). In RotatedMNIST, however, we find that while training on strongly shifted data helps, the help is limited.
>
> The new results on LowLightCIFAR10 (in Figure 5) demonstrate an even more interesting pattern. When the shift in the training data is not very strong (e.g., $D_0$ and $D_5$), the generalization pattern looks just like that of NoisyMNIST---robustness against milder shifts is not affected. On the other hand, when the shift in the training data is very strong ($D_0$ and $D_{10}$), the pattern looks like that of RotatedMNIST---robustness against milder shifts is significantly weakened. In addition, even the performance on the clean data is affected.
>
> **2.1. “The results seems to say that the OOD performance of the model depends on its inductive biases and it is hard to predict a model's performance OOD, which is quite generic.”**
>
> With all due respect, the message we tried to convey is more than that. It is not just hard, but even harder than it seems. To the best of our knowledge, this has not been clearly articulated or systematically examined before in the literature. That said, we do very much appreciate your comment. We will make this clearer in future revisions. Some of the more specific points have been discussed in our response to your previous questions.

---

> ### Author Response · Authors · 2023-11-22
> **Response to the review (3/3)**
>
> **2.2. “The overall goal seems to be more thorough evaluations ... Real world datasets provides such evaluations but some cases it is impossible to have smoothly changing shifts e.g. smoothly changing animal occlusions in iwildcam. Furthermore, synthetic benchmarks like common corruptions already proposes evaluating on different degrees of shifts for each corruption ...”**
>
> It is true that we encourage more thorough evaluations. We also agree that it is impossible to observe smoothly changing shifts in some real-world scenarios. The real advice that we would like to convey is to conduct more thorough evaluations *whenever they are feasible*. This means that we should not be restricted to only using readily available data. In fact, we can proactively collect and/or synthesize more data for this purpose. For example, smoothly changing animal occlusions in iWildCam is actually possible if we maintain a series of shots each time an animal passes by. In addition, we can simulate occlusions by editing the images post-hoc. After all, while synthetic shifts are not as good as real shifts, they are at least better than nothing.
>
> Regarding your second argument, we would like to point out that previous synthetic benchmarks not only did not provide analysis at each level of corruption, but also did not articulate the importance of evaluating models on multiple degrees of shifts, let alone encouraging such practice to be followed in other tasks.
>
> **2.3. “Furthermore is it easy to anticipate the degree of shift at test-time such that such analysis would be useful? It seems more natural to anticipate the type of shift than its extent e.g., adverse weather, than how adverse the weather will be.”**
>
> Thank you for the great question. Indeed, anticipating how adverse the weather will exactly be is sometimes difficult. Fortunately, we do not need to be precise, we just need to ensure that almost every possible extent is approximately covered. In general, the more fine-grained it is, the better. Of course, the return of further increasing the granularity would diminish at some point. So, the real problem is to find the best trade-off between evaluation cost and informativeness. For high-stake applications, e.g., self-driving, the cost of conducting such evaluations is not really a concern. In conclusion, whether it is easy to anticipate the degree of shift at test-time usually does not have much to do with whether such analysis would be useful and whether it should be done.
>
> **3. “The common corruptions benchmark measures performance over 5 degrees of shift and the paper measures it over 8/10, how do we know what is the ideal number of degrees to evaluate on?”**
>
> For our answer to this question, please first kindly refer to our response to your previous question (2.3). Operationally speaking, we can start with only 3 or 4 degrees, if the analysis done on some of the degrees is largely consistent with the evaluation results on the rest of the degrees, then we can stop. If not, we need to increase the number of degrees until the results are consistent or the trade-off between return and cost is unacceptable.
>
> **4. “The statement ‘robustness may be more brittle …' sounds contradictory as robustness is the opposite of brittle. It is our models that are brittle.”**
>
> We agree that “models are brittle” would be technically more precise and perhaps a better choice. By “robustness may be brittle”, we aim to stress that the seemingly robustness of a model against a certain degree of shifts can easily break when the degree changes. We would love to consider revising the statement; however, we are not allowed to change the title of this submission. Finally, and perhaps most importantly, this statement does not allow much room for misinterpretation and should not cause much trouble for readers to understand the main ideas of this paper.
>
> **5. “A suggestion for having smoothly changing yet realistic shift is to perform evaluations on videos.”**
>
> Thank you for your great suggestion! Please also understand that our work is the first of its kind, and with the newly added results we believe the current empirical evidence is sufficient to deliver the main points within this conference paper. We will definitely try video datasets as you kindly suggested in an extended version of this paper (e.g., a journal version).
>
> ***
>
> **References:**
>
> [1] Taori, Rohan, et al. "Measuring robustness to natural distribution shifts in image classification." NeurIPS. 2022.
>
> [2] Hendrycks, Dan, et al. "The many faces of robustness: A critical analysis of out-of-distribution generalization." ICCV. 2021.
>
> [3] Gulrajani, Ishaan, and David Lopez-Paz. "In Search of Lost Domain Generalization." ICLR. 2021.
>
> [4] Koh, Pang Wei, et al. "Wilds: A benchmark of in-the-wild distribution shifts." International Conference on Machine Learning. PMLR, 2021.

---

### Author Response · Authors · 2023-11-23
**Summary of the updates made in our paper**

Dear ACs and Reviewers,

Thank you again for your valuable time and contribution to the reviewing process of our paper. The main concern of most reviewers rests on insufficient experiments. During the rebuttal period, we have added many new experiment results to the paper. Most of the added experiments are kindly suggested by the reviewers. For your convenience, below is a list of the main updates we made to the paper:

1. We experimented with a new dataset, LowLightCIFAR10, which is much more realistic than the MNIST variants. The new results are added to Figure 3 and Figure 5 of the paper.
2. We included the results of the DG algorithms trained on more than two domains in Appendix B.1.
3. We added the results of the DG algorithms for the experiments in Sec. 4.3. You can find the results in Appendix B.3.
4. We furnished the numerical details of the experiments in Sec. 4.2. The statistics can be found in Appendix B.4 (Table 2-19).
5. Finally, we also experimented with a different type of noise on MNIST. The results can be found in Appendix B.2.

Please note that some of the relatively minor experiments suggested by the reviewers are still ongoing. Discussions and further details regarding the experiments will be added to the paper later.

Best Regards,
Authors

---

### Meta-Review · Area_Chair_7UFC · 2023-12-12

**Metareview:**

This paper studies different degrees of distribution shifts and notes that generalization depends on the degree of distribution shift indicating that the models are not robust to larger shifts. There are also some explorations for large image-language models. All of the reviewers on this paper were negative, while they note the paper is well written, the takeaways were unclear mostly from the experiments considered in that the datasets focus on MNIST, the small number of domains, the consideration of whats fair to pass to CLIP. The large number of experiments were appreciated, but revision based on the reviewers comments and a simpler take home message would make this paper better.

**Justification For Why Not Higher Score:**

There were lots of questions about the experiments which were the main focus of this paper along with the take home message for those experiments.

**Justification For Why Not Lower Score:**

N/A

---

### Decision · Program_Chairs · 2024-01-16

Reject